# Grokking Beyond the Euclidean Norm of Model Parameters

**Pascal Jr Tikeng Notsawo** [1 2 3]   **Guillaume Dumas** [1 2 3]   **Guillaume Rabusseau** [1 2 4]

## Abstract

Grokking refers to a delayed generalization following overfitting when optimizing artificial neural networks with gradient-based methods. In this work, we demonstrate that grokking can be induced by regularization, either explicit or implicit. More precisely, we show that when there exists a model with a property $P$ (e.g., sparse or low-rank weights) that generalizes on the problem of interest, gradient descent with a small but non-zero regularization of $P$ (e.g., $\ell_1$ or nuclear norm regularization) results in grokking. This extends previous work showing that small non-zero weight decay induces grokking. Moreover, our analysis shows that over-parameterization by adding depth makes it possible to grok or ungrok without explicitly using regularization, which is impossible in shallow cases. We further show that the $\ell_2$ norm is not a reliable proxy for generalization when the model is regularized toward a different property $P$, as the $\ell_2$ norm grows in many cases where no weight decay is used, but the model generalizes anyway. We also show that grokking can be amplified solely through data selection, with any other hyperparameter fixed.

## 1. Introduction

The optimization of machine learning models today relies entirely on gradient descent (GD). The reasons behind the ability of such a procedure to converge towards generalizing solutions are still not fully understood, particularly in over-parameterized regimes. Power et al. (2022) recently observed an even more surprising feature of this optimization procedure, *grokking*: optimization first converges to a solution that memorizes the training data, but after a sufficiently long training time, it suddenly converges to a solution that generalizes.

Previous work has shown that grokking can be observed by using a large-scale initialization and a small (but non-zero) weight decay (Liu et al., 2023a; Lyu et al., 2023). Moreover, some works have shown that the $\ell_2$ norm of the model weights can be used during optimization as a progression measure for generalization since it generally decreases during the transition from memorization to generalization (Liu et al., 2023a; Thilak et al., 2022; Varma et al., 2023). All these theories have left open the question of whether we always need an $\ell_2$ regularization to observe delayed generalization or whether the $\ell_2$ norm of the parameter is always a good predictor of grokking. This paper attempts to answer these questions. We show that the dynamic of grokking goes beyond the $\ell_2$ norm, that is: *If there exists a model with a property $P$ (e.g., sparse or low-rank weights) that fits the data, then GD with a small non-zero regularization of $P$ (e.g., $\ell_1$ or nuclear norm regularization) will also result in grokking, provided the number of training samples is large enough and the model is complex enough. Additionally, the regularization of $P$ can be implicit (e.g., model overparameterization, the choice of training samples). Moreover, the $\ell_2$ norm of the parameters is no longer guaranteed to decrease with generalization when it is not the property sought.*

We first establish our main theorem (Theorem 2.1), which theoretically characterizes the relation between grokking and regularization. This theorem is a cornerstone of our argument, demonstrating that the generalization delay scales like $1/(\alpha\beta)$, where $\alpha$ is the gradient descent step size and $\beta$ is the regularization strength of an arbitrary and appropriately chosen regularizer that enforces an inductive bias toward generalization. This theoretical characterization extends previous observations, which have been mainly focused on $\ell_2$ regularization, providing a more general framework for understanding grokking dynamics.

Building upon this theoretical foundation, we validate its implications both theoretically and empirically across various settings: sparsity (Theorem 3.1) and low-rankness (Theorem 3.4). For sparsity, we focus on a linear teacher-student setup and show that recovery of sparse vectors using gradient descent and Lasso exhibits a grokking phenomenon, which is impossible using only $\ell_2$ regularization, regardless

[1]Université de Montréal, Montréal, Quebec, Canada [2]Mila, Quebec AI Institute, Montréal, Quebec, Canada [3]CHU Sainte-Justine Research Center, Montréal, Quebec, Canada [4]CIFAR AI Chair. Correspondence to: Pascal Jr Tikeng Notsawo <pascal.tikeng@mila.quebec>.

*Proceedings of the 42$^{nd}$ International Conference on Machine Learning*, Vancouver, Canada. PMLR 267, 2025. Copyright 2025 by the author(s).

of the initialization scale, as advocated by previous art (Lyu et al., 2023; Liu et al., 2023b). Moreover, with a deeper over-parameterized model, there is no need for explicit $\ell_1$ regularization, as gradient descent is implicitly biased toward such sparse solutions. Similarly, we focus on matrix factorization for the low-rank structure and demonstrate that nuclear norm regularization (denoted $\ell_*$) is necessary for generalization in the shallow case. This complements prior work demonstrating that deeper linear networks can factorize low-rank matrices without explicit regularization (Arora et al., 2018; 2019).

These findings hold beyond shallow and/or linear networks. We show that $\ell_1$ or $\ell_*$ can replace $\ell_2$ in a more general setting and induce grokking. We demonstrate this on a non-linear teacher-student setup, on the algorithmic data setup where grokking was first observed (Power et al., 2022), and on image classification tasks. In settings where $\ell_2$ regularization is not used, the $\ell_2$ norm of the model parameters tends to grow during training and after generalization, yet optimization still produces a generalizable solution. This directly challenges the previously held belief that the $\ell_2$ norm of parameters is always a good indicator of grokking.

Our contributions can be summarized as follows[1]:

(i) We show that grokking can be induced by the interplay between the sparse/low-rank structure of the solution and the $\ell_1/\ell_*$ regularization used during training, extending previous results on $\ell_2$ regularization (Lyu et al., 2023). Our theoretical results extend beyond these specific regularizations, as we characterize the relationship between grokking time, regularization strength, and learning rate in a general setting.

(ii) We show that regularization is necessary to observe grokking on sparse or low-rank solutions. Moreover, we empirically show that in deep linear networks, the sparse/low-rank structure of the data is enough to have generalization without explicit regularization. Adding depth makes it possible to grok or ungrok simply from the implicit regularization of gradient descent.

(iii) Leveraging the notion of coherence, we show that grokking can be amplified through data selection.

(iv) We show that $\ell_1/\ell_*$ can replace $\ell_2$ in a more general setting and induce grokking. Moreover, in such a scenario, and in the shallow sparse/low-rank scenario mentioned above, the grokking phenomenon can not be explained by the $\ell_2$ norm.

(v) We also show that other forms of domain-specific regularizers strongly impact the grokking delay.

[1]Code to reproduce our experiments: https://github.com/Tikquuss/grokking_beyong_l2_norm.

This paper is organized as follows. We begin by presenting our main theoretical result on grokking time and regularization in Section 2. Sections 3 extend our theoretical framework to concrete sparse recovery and matrix factorization settings, establishing additional theoretical results and providing empirical validation of the two-phase dynamics predicted by our main theorem. Our findings are extended to general non-linear models and other domain-specific regularizations in Section 4. Finally, we discuss and conclude our work in Section 5.

Throughout this paper, $\Omega(\cdot)$, $\mathcal{O}(\cdot)$ and $\Theta(\cdot)$ follow their standard asymptotic definitions.

## 2. Learning Dynamics: Early and Later Bias

In this section, we will illustrate our hypothesis concerning the induction of grokking by regularization. Lyu et al. (2023) proved that enlarging the initialization scale and/or reducing the weight decay delays the transition from memorization to generalization, a conclusion already drawn experimentally by Liu et al. (2023a). As we will see below, replacing the weight decay with various other regularizations similarly impacts the generalization delay.

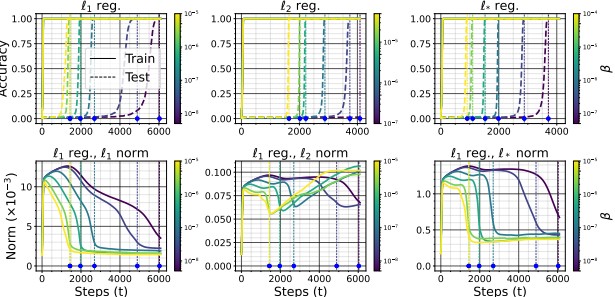

Figure 1: (*Top*) Training and test accuracy of a MLP trained on modular addition with $\ell_1$ (left), $\ell_2$ (middle), and $\ell_*$ (right) regularization for different values of the regularization strength $\beta$. Smaller values of $\beta$ delay generalization. (*Bottom*) In the case of $\ell_1$ (top-left), we show the evolution of the $\ell_1$ (left), $\ell_2$ (middle), and $\ell_*$ (right) norm during training. The $\ell_2$ norm increases despite generalization.

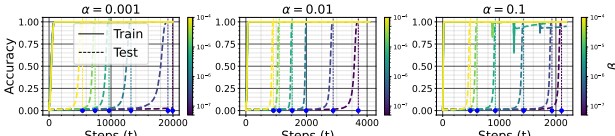

Figure 2: Training and test accuracy of a MLP trained on modular addition with $\ell_*$ regularization for different values of the learning rate $\alpha$ and the $\ell_*$ regularization strength $\beta$. When $\alpha$ increases, the generalization delay decreases.

### 2.1. Motivating Experiment

Consider a binary mathematical operator $\circ$ on $\mathcal{S} = \mathbb{Z}/p\mathbb{Z}$ for some prime integer $p$. We want to predict $y^*(\mathbf{x}) =$

$x_1 \circ x_2$ given $\mathbf{x} = (x_1, x_2) \in \mathcal{S}^2$. The dataset $\mathcal{D} = \{(\mathbf{x}, y^*(\mathbf{x})) \mid \mathbf{x} \in \mathcal{S}^2\}$ is randomly partitioned into two disjoint and non-empty sets $\mathcal{D}_{\text{train}}$ and $\mathcal{D}_{\text{val}}$, the training and the validation dataset respectively. Let us consider as a model a multilayer perceptron (MLP) for which the logits are given by $\mathbf{y}(\mathbf{x}) = \mathbf{b}^{(2)} + \mathbf{W}^{(2)}\phi\left(\mathbf{b}^{(1)} + \mathbf{W}^{(1)}\left(\mathbf{E}_{\langle x_1 \rangle} \circ \mathbf{E}_{\langle x_2 \rangle}\right)\right)$, where $\langle x \rangle$ stands for the index of the token corresponding to $x \in \mathcal{S}$ and $\phi(z) = \max(z, 0)$ is the activation function. $\mathbf{E} \in \mathbb{R}^{p \times d_1}$ is the embedding matrix for all the symbols in $\mathcal{S}$. The learnable parameters are $\mathbf{E}$, $\mathbf{W}^{(1)} \in \mathbb{R}^{d_2 \times d_1}$, $\mathbf{b}^{(1)} \in \mathbb{R}^{d_2}$, $\mathbf{W}^{(2)} \in \mathbb{R}^{p \times d_2}$, and $\mathbf{b}^{(2)} \in \mathbb{R}^p$. We train this model on addition modulo $p = 97$ with $r_{\text{train}} := |\mathcal{D}_{\text{train}}|/|\mathcal{D}| = 40\%$.

We can see in Figures 1 and 2 that $\ell_1$ and $\ell_*$ have the same effect on grokking as $\ell_2$. For all these regularization techniques, the smaller $\alpha\beta$ is, the longer the delay between memorization and generalization. Figure 1 (*bottom*) also shows that in the absence of *weight decay*, the $\ell_2$ norm of the model parameters is not monotonic after memorization, and even increases without harming generalization performance.

## 2.2. Theoretical Insights

We now provide high-level theoretical insights on how the dynamics of regularized gradient descent can induce grokking. Let $g : \mathbb{R}^p \to [0, \infty)$ be a differentiable function, $h : \mathbb{R}^p \to [0, \infty)$ be a subdifferentiable function and $\beta > 0$. Typically, $g$ is the loss function of an overparameterized neural network on the training data, while $h$ serves as the regularizer. Our goal is to minimize the composite objective $f := g + \beta h$ using subgradient descent with a learning rate $\alpha > 0$. The update rule for this problem is given by

$$\mathbf{x}^{(t+1)} = \mathbf{x}^{(t)} - \alpha\left(G(\mathbf{x}^{(t)}) + \beta H(\mathbf{x}^{(t)})\right) \quad \forall t \geq 0 \quad (1)$$

where $G(\mathbf{x}) = \nabla g(\mathbf{x})$ is the gradient of $g$ at $\mathbf{x}$ and $H(\mathbf{x}) \in \partial h(\mathbf{x})$ is any subgradient of $h$ at $\mathbf{x}$. We let $f^* := \inf_{\mathbf{x} \in \mathbb{R}^p} f(\mathbf{x})$ and $\Theta_f := \arg\min_{\mathbf{x} \in \mathbb{R}^p} f(\mathbf{x}) \subset \mathbb{R}^p$. Similarly, we define $g^*$ and $\Theta_g$, and assume $g^* = 0$ without loss of generality. Lastly, we define $\text{dist}(\mathbf{x}, \Theta_f) := \inf_{\mathbf{y} \in \Theta_f} \|\mathbf{x} - \mathbf{y}\|_2$. Intuitively, the training dynamics can be decomposed into two phases under certain conditions on $\alpha$ and $\beta$. The iterates $\mathbf{x}^{(t)}$ initially move toward a solution close to the initialization $\mathbf{x}^{(0)}$ that minimizes $g$ (for instance, the kernel solution associated with $g$). Later in training, the influence of $H(\mathbf{x})$ dominates the update, driving $f(\mathbf{x}^{(t)}) \approx f^*$ and $h(\mathbf{x}^{(t)}) \approx h_g^* := \inf_{\mathbf{x} \in \Theta_g} h(\mathbf{x})$ within an additional $\Theta(1/\alpha\beta)$ training steps.

We define the Chatterjee–Łojasiewicz (CL) constant for $g$ at $\mathbf{x} \in \mathbb{R}^p$ with radius $r > 0$ as (Chatterjee, 2022):

$$\chi(g, \mathbf{x}, r) := \inf_{\mathbf{y} \in B(\mathbf{x}, r), g(\mathbf{y}) \neq 0} \|\nabla g(\mathbf{y})\|_2^2 / g(\mathbf{y}) \quad (2)$$

where $B(\mathbf{x}, r) := \{\mathbf{y} \in \mathbb{R}^p \mid \|\mathbf{x} - \mathbf{y}\|_2 \leq r\}$. The function $g$ is said to satisfy the $r$-CL inequality at $\mathbf{x}$ or to be $r$-CL

at $\mathbf{x}$ if and only if $4g(\mathbf{x}) < r^2\chi(g, \mathbf{x}, r)$. The CL inequality is a strengthening of the classical Polyak-Łojasiewicz (PL) inequality (Chatterjee, 2022), which has been shown to hold for wide overparameterized neural networks in a neighborhood of their initialization (Liu et al., 2021). The advantage here is that we will require the CL inequality to be satisfied only at initialization for the theorem to hold, unlike standard results under PL, which require the function to satisfy the PL inequality over its entire domain (Karimi et al., 2020).

**Theorem 2.1.** *Take any $\mathbf{x}^{(0)} \in \mathbb{R}^p$ with $g^{(0)} := g(\mathbf{x}^{(0)}) \neq 0$. Assume that $g$ is $r$-CL at $\mathbf{x}^{(0)}$ for some $r > 0$, i.e. $4g^{(0)} < r^2\chi$, $\chi := \chi(g, \mathbf{x}^{(0)}, r)$. There exist $\beta_{\max} > 0$, $\alpha_{\max} > 0$ and two constants $C, C' > 0$ such that for all $\alpha \in (0, \alpha_{\max})$ and $\beta \in (0, \beta_{\max})$, by defining the subgradient descent update (1) with $\alpha$ and $\beta$ starting at $\mathbf{x}^{(0)}$, the following hold:*

- *For any $\epsilon = \Omega(\beta^C)$, there exists a step $t_1 \geq \max\left\{0, -\log\left(\epsilon/g^{(0)}\right)/\log\left(1 - \Theta(\alpha \cdot \chi)\right)\right\}$ such that $g(\mathbf{x}^{(t_1)}) \leq \epsilon$, $\|G(\mathbf{x}^{(t_1)})\|_2^2 = \mathcal{O}(\epsilon)$ and $\mathbf{x}^{(t)} \in B(\mathbf{x}^{(0)}, r) \quad \forall t \leq t_1$.*

- *For any $\eta > 0$, $\min_{t_1 \leq t \leq t_2}\left(f(\mathbf{x}^{(t)}) - f^*\right) \leq 0.5\left(\eta + C'\alpha\beta\right)\beta$ if and only if $t_2 > t_1 + \Delta t(\eta, t_1)$, with $\Delta t(\eta, t_1) := \frac{\text{dist}^2(\mathbf{x}^{(t_1)}, \Theta_f)}{\alpha\beta\eta}$. Moreover, assuming $\Theta_f \cap \Theta_g \neq \emptyset$, $\min_{t_1 \leq t \leq t_2}\left(h(\mathbf{x}^{(t)}) - h_g^*\right) = 0.5\left(\eta + C'\alpha\beta\right)$ if and only if $t_2 > t_1 + \Delta t(\eta, t_1)$.*

*Proof Sketch.* We show that, as long as $\beta\|H(\mathbf{x}^{(k)})\|_2 \leq \|G(\mathbf{x}^{(k)})\|_2 \quad \forall 0 \leq k < t$, for a certain $t \geq 1$, the following hold : $\mathbf{x}^{(k)} \in B(\mathbf{x}^{(0)}, r)$, $g(\mathbf{x}^{(k)}) \leq (1 - \delta)^k g(\mathbf{x}^{(0)})$ and $\|\nabla g(\mathbf{x}^{(k)})\|_2^2 \leq 2Lg(\mathbf{x}^{(k)})$ for all $0 \leq k \leq t$, for a certain $\delta \in (0, 1)$ and $L > 0$. Using this result, we find a constant $C > 0$ such that for any precision $\epsilon = \Omega(\beta^C)$, we can adjust hyperparameters to ensure that $g$ reaches this precision before $\beta\|H(\cdot)\|_2 \gg \|G(\cdot)\|_2$. We then bound $f(\mathbf{x}^{(t)}) - f^*$ and $h(\mathbf{x}^{(t)}) - h_g^*$ by quantities that depend on $t, \alpha, \beta$. Using these bounds, we extract the delay $\Delta t(\eta, t_1)$ that ensures the desired precision $\eta$ on $f(\mathbf{x}^{(t)}) - f^*$ and $h(\mathbf{x}^{(t)}) - h_g^*$ after any step $t_1$. A more formal version of this Theorem and the corresponding proof can be found in Section C.1 of the Appendix. $\square$

Theorem 2.1 highlights a dichotomy between memorization and regularization and formalizes a two-phase dynamic in the training process, under the CL assumption on the loss function $g$ at initialization. The first phase, described in the first bullet point of the theorem, shows that for a sufficiently small regularization strength $\beta$, the iterates $\mathbf{x}^{(t)}$ remain close to initialization and quickly minimize $g$ up to any precision $\epsilon = \Omega(\beta^C)$. This corresponds to memorization of the training data, as the model fits the loss $g$ while staying in a local region where regularization does not dominate. By choosing $\beta$ too large, we may never minimize $g$, hence the lower bound $\beta^C$ on $\epsilon$, which vanishes as $\beta \to 0$.

In the second phase, captured in the second bullet point, when $g$ and its gradient are already small, the regularization term $\beta h(\mathbf{x})$ gradually drives the iterates toward low values of $f$ and $h$, until reaching $f(\mathbf{x}^{(t)}) \approx f^*$ and $h(\mathbf{x}^{(t)}) \approx h_g^*$ up to an error of order $\mathcal{O}(\alpha\beta^2)$ and $\mathcal{O}(\alpha\beta)$ respectively, within $\Delta t = \Theta(1/\alpha\beta)$ additional steps. If this late-phase bias induced by $h$ is aligned with generalization, then the grokking time is proportional to $1/\alpha\beta$. This conclusion is consistent with previous findings for $h(\mathbf{x}) = \|\mathbf{x}\|_2^2$, where grokking time scales as $1/\beta$ (Lyu et al., 2023). The learning rate $\alpha$ does not appear in that result because Lyu et al. (2023) analyze the continuous-time gradient flow setting.

We also establish that under some assumptions on $h$ and $g$, if $g$ is $r$-CL at $\mathbf{x} \in \mathbb{R}^p$, then for all $\varepsilon \in (0,1)$, there exists $\beta_{\max} = \beta_{\max}(\varepsilon, \mathbf{x}, r) > 0$ such that for all $\beta < \beta_{\max}$, the function $f = g + \beta h$ is also $\varepsilon r$-CL at $\mathbf{x}$, i.e., $4f(\mathbf{x}) < \varepsilon^2 r^2 \cdot \chi(f, \mathbf{x}, \varepsilon r)$. As a consequence, the first point of Theorem 2.1 applies to $f - f^*$ when $\beta \leq \beta_{\max}$. More specifically, we have $\mathbf{x}^{(t)} \in B(\mathbf{x}^{(0)}, \varepsilon r)$ for all $t$, and as $t \to \infty$, $\mathbf{x}^{(t)}$ converges to a point $\mathbf{x}^* \in B(\mathbf{x}^{(0)}, \varepsilon r)$ where $f(\mathbf{x}^*) = f^*$. Moreover, for each $t \geq 0$, $\|\mathbf{x}^{(t)} - \mathbf{x}^*\|_2^2 \leq (1-\delta)^t \varepsilon^2 r^2$ and $f(\mathbf{x}^{(t)}) - f^* \leq (1-\delta)^t \left( f(\mathbf{x}^{(0)}) - f^* \right)$ with $\delta = \min\{1, \Theta\left(\chi(f, \mathbf{x}^{(0)}, \varepsilon r) \cdot \alpha\right)\}$. The proof of this fact can be found in Section C.3 of the Appendix. The result required that $\forall \mathbf{x} \in \mathbb{R}^p, 0 \in \partial h(\mathbf{x}) \implies h(\mathbf{x}) = 0$. A function $h : \mathbb{R}^p \to \mathbb{R}$ satisfies this if, for example, it is convex and there exists $\mathbf{x}^* \in \mathbb{R}^p$ such that $h(\mathbf{x}^*) = \inf_{\mathbf{x}} h(\mathbf{x}) > -\infty$. This is the case of $\ell_{1/2/*}$ norm.

## 3. Beyond Weight Decay

In this section, we will illustrate examples in which late phase bias associated with generalization is not driven by $\ell_2$, namely, sparse recovery and low rank matrix factorization. Indeed, we will see that the use of weight decay alone causes an abrupt transition in the generalization error, as predicted by previous works, but that this transition does not correspond to generalization. We will also show that for a fixed training data size, the depth of the model used or an appropriate choice of training samples can significantly reduce the grokking delay.

We consider the operator $\mathcal{F}_{\mathbf{a}}(\mathbf{M}) = \mathbf{M}\mathbf{a} \in \mathbb{R}^N$ that take $N$ measurement vectors $\{\mathbf{M}_i \in \mathbb{R}^n\}_i$ and return the measures $\{\mathbf{M}_i^\top \mathbf{a}\}_i$ of $\mathbf{a} \in \mathbb{R}^n$. For a matrix $\mathbf{A} \in \mathbb{R}^{m \times n}$, the operator $\text{vec}(\mathbf{A}) \in \mathbb{R}^{mn}$ stacks the column of $\mathbf{A}$ in a vector; $\sigma_{\max/\min/i}(\mathbf{A})$ is the maximum/minimum/$i^{\text{th}}$ singular values of $\mathbf{A}$; $\|\mathbf{A}\|_* := \sum_i \sigma_i(\mathbf{A})$ and $\|\mathbf{A}\|_{2\to 2} := \sigma_{\max}(\mathbf{A})$.

### 3.1. Sparse Recovery

Consider a sparse vector $\mathbf{a}^* \in \mathbb{R}^n$, i.e. $s = \|\mathbf{a}^*\|_0 \ll n$, where $\|\mathbf{a}^*\|_0$ is the number of non-zero components of $\mathbf{a}^*$. Given $\mathbf{y}^* = \mathbf{X}\mathbf{a}^* + \boldsymbol{\xi}$ with $\mathbf{X} \in \mathbb{R}^{N \times n}$ a design

matrix and $\boldsymbol{\xi} \in \mathbb{R}^N$ a noise vector, we want to minimize $f(\mathbf{a}) = g(\mathbf{a}) + \beta h(\mathbf{a})$ using gradient descent with a learning rate $\alpha > 0$, where $\mathbf{g}(\mathbf{a}) = \frac{1}{2}\|\mathbf{X}\mathbf{a} - \mathbf{y}^*\|_2^2$ and $h(\mathbf{a}) = \|\mathbf{a}\|_1$. Like in Theorem 2.1, the update $\mathbf{a}^{(t)}$ first moves near the least square solution $\hat{\mathbf{a}} := \left(\mathbf{X}^\top \mathbf{X}\right)^\dagger \mathbf{X}^\top \mathbf{y}^*$ leading to memorization, with $g(\hat{\mathbf{a}}) \leq \frac{1}{2}\|\boldsymbol{\xi}\|_2^2$. Later in training, $\partial h(\mathbf{a})$ dominates the update, leading to $\|\mathbf{a}^{(t)}\|_1 - \|\mathbf{a}^*\|_1 = \mathcal{O}(\alpha\beta)$ in the order of $1/(\alpha\beta)$ more training steps. When $\alpha\beta$ is small, this causes a non-trivial delay between memorization and generalization (i.e., grokking), for a suitable $\mathbf{X}$.

**Theorem 3.1.** *Assume the learning rate, regularization coefficient and the noise term satisfy* $0 < \alpha\sigma_{\max}(\mathbf{X}^\top \mathbf{X}) < 2$, $0 < \beta\sqrt{n} < \sigma_{\max}(\mathbf{X}^\top \mathbf{X})$ *and* $\|\mathbf{X}^\top \boldsymbol{\xi}\|_2 \leq \sqrt{C\alpha}\beta$, $C > 0$. *Let* $\rho_2 := \sigma_{\max}\left(\mathbb{I}_n - \alpha\mathbf{X}^\top \mathbf{X}\right)$. *There exist* $t_1 < \infty$ *and* $C' > 0$ *such that* $\|\mathbf{a}^{(t)} - \hat{\mathbf{a}}\|_2 \leq \frac{2\alpha\beta n^{1/2}}{1-\rho_2} \ \forall t \geq t_1$, *and* $\min_{t_1 \leq t \leq t_2}\left(\|\mathbf{a}^{(t)}\|_1 - \|\mathbf{a}^*\|_1\right) \leq \frac{\eta + (C+C')\alpha\beta}{2} \iff t_2 \geq t_1 + \Delta t(\eta, t_1)$ *for any* $\eta > 0$, $\Delta t(\eta, t_1) := \frac{\|\mathbf{a}^{(t_1)} - \mathbf{a}^*\|_2^2}{\alpha\beta\eta}$.

*Proof.* See the proof in Section C.4 of the Appendix. $\square$

Now let illustrate why $\mathbf{X}\mathbf{a}^{(t)} = \mathbf{y}^*$ (memorization) and $\|\mathbf{a}^{(t)}\|_1 = \|\mathbf{a}^*\|_1$ are enough to conclude $\mathbf{a}^{(t)} = \mathbf{a}^*$ (generalization) when $N$ is large enough with a specific class of design matrices $\mathbf{X}$ commonly used in practice (Foucart & Rauhut, 2013). For that, let us consider the problem in a more general context, where we have a measurement matrix $\mathbf{M} \in \mathbb{R}^{N \times n}$ and a list of noisy measurements $\mathbf{y}^* = \mathcal{F}_{\mathbf{b}^*}(\mathbf{M}) + \boldsymbol{\xi} \in \mathbb{R}^N$ of an unknown signal $\mathbf{b}^* \in \mathbb{R}^n$ assumed sparse in a know basis $\Phi \in \mathbb{R}^{n \times n}$, i.e. $\mathbf{b}^* = \sum_{i=1}^n \mathbf{a}_i^* \Phi_{:,i} = \Phi \mathbf{a}^*$ with $\|\mathbf{a}^*\|_0 \ll n$. The aim is to find $\mathbf{a}^*$ by minimizing $\|\mathbf{a}\|_0$ subject to $\|\mathcal{F}_{\Phi\mathbf{a}}(\mathbf{M}) - \mathbf{y}^*\|_2 \leq \epsilon$, where $\epsilon$ an upper bound on the size of the error term $\boldsymbol{\xi}$, $\|\boldsymbol{\xi}\|_2 \leq \epsilon$. Since this is NP-hard (Natarajan, 1995; Donoho, 2006a), $\|\mathbf{a}\|_0$ is often relaxed to its tightest convex relaxation, $\|\mathbf{a}\|_1$ (Foucart & Rauhut, 2013; Chandrasekaran et al., 2012), leading to the convex problem of minimizing $\|\mathbf{a}\|_1$ subject to $\|\mathbf{X}\mathbf{a} - \mathbf{y}^*\|_2 \leq \epsilon$, with $\mathbf{X} = \mathbf{M}\Phi$. This is equivalent to the problem of minimizing $f(\mathbf{a})$ presented above, for a suitable choice of $\beta$ (Rockafellar, 1970; Boyd & Vandenberghe, 2004; Candes et al., 2006).

While classical sparse recovery assumes that the true signal is exactly sparse and the measurements are noise-free, in practice, it is common to design the measurement matrix $\mathbf{M}$ for stable and robust recovery. Stability refers to the ability of a reconstruction method to recover an approximation $\mathbf{a}$ of a signal $\mathbf{a}^*$ whose energy is concentrated on a few components (i.e., approximately sparse), with reconstruction error controlled by the sparsity defect $\inf_{\|\mathbf{a}\|_0 \leq s}\|\mathbf{a}^* - \mathbf{a}\|$, the distance from $\mathbf{a}^*$ to the set of exactly $s$-sparse vectors. Robustness, on the other hand, means that the reconstruction is resilient to measurement noise, i.e., small perturbations in the observed measurements $\mathbf{y}^* = \mathcal{F}_{\mathbf{b}^*}(\mathbf{M}) + \boldsymbol{\xi}$ lead to

small deviations in the reconstructed signal $\mathbf{a}$, with the error scaling with the noise level $\epsilon \geq \|\boldsymbol{\xi}\|_2$.

**Definition 3.2** (Robust Null Space Property). A matrix $\mathbf{A} \in \mathbb{R}^{m \times n}$ is said to satisfy the robust null space property with constant $\rho \in (0,1)$ and $\tau > 0$ relative to a set $S \subset [n]$ if $\|\mathbf{u}_S\|_1 < \rho\|\mathbf{u}_{[n]\setminus S}\|_1 + \tau\|\mathbf{A}\mathbf{u}\|_2$ for all $\mathbf{u} \in \mathbb{R}^n$; where $\mathbf{u}_S = [\mathbf{u}_i]_{i\in S} \in \mathbb{R}^{|S|}$. It is said to satisfy the robust null space property of order $s \in \mathbb{N}^*$ if it satisfies the robust null space property relative to any set $S \subset [n]$ with $|S| \leq s$.

**Theorem 3.3.** *Assume the matrix $\mathbf{X} \in \mathbb{R}^{N \times n}$ satisfies the robust null space property with constant $\rho \in (0,1)$ and $\tau > 0$ relative to the support of $\mathbf{a}^*$. Then, under the same condition as in Theorem 3.1 on $\alpha$, $\beta$ and $\boldsymbol{\xi}$, there exist $C_1, C_2, C_3 > 0$ such that $\min_{t_1 \leq t \leq t_2} \|\mathbf{a}^{(t)} - \mathbf{a}^*\|_1 \leq C_1\eta + C_2\alpha\beta + C_3\|\boldsymbol{\xi}\|_2$ if and only if $t_2 \geq t_1 + \Delta t(\eta, t_1), \forall \eta > 0$.*

*Proof.* Proof in Section C.5 of the Appendix. $\square$

It is worth noting that we should instead quantify the training error $\frac{1}{2}\sum_{i=1}^{N}(y(\mathbf{X}_i) - y^*(\mathbf{X}_i))^2 = g(\mathbf{a})$ and the generalization error $\mathcal{E}(\mathbf{a}) = \mathbb{E}_{\mathbf{x},\varepsilon}(y(\mathbf{x}) - y^*_\varepsilon(\mathbf{x}))^2$, as is common in the context of grokking, with $y(\mathbf{x}) = \mathbf{x}^\top\mathbf{a}$ and $y^*_\varepsilon(\mathbf{x}) = \mathbf{x}^\top\mathbf{a}^* + \varepsilon$ in our case. Assuming $\mathbb{E}\varepsilon = 0$, and using $\Sigma = \mathbb{E}[\mathbf{x}\mathbf{x}^\top]$, we get $\mathcal{E}(\mathbf{a}) = (\mathbf{a} - \mathbf{a}^*)^\top \Sigma (\mathbf{a} - \mathbf{a}^*) + \mathbb{E}\varepsilon^2$. Random matrices with independent entries, notably Gaussian and Bernoulli matrices, allow for the lowest restricted isometry constant and, thus, better recovery of sparse vectors (Rauhut, 2010). Under this independent and identically distributed (iid) assumption, we have $\Sigma = \sigma^2\mathbf{I}_n$ for a certain $\sigma > 0$, which implies $\mathcal{E}(\mathbf{a}) = \sigma^2\|\mathbf{a} - \mathbf{a}^*\|_2^2 + \mathbb{E}\varepsilon^2$. So $\|\mathbf{a} - \mathbf{a}^*\|_2^2$ captures well the notion of test (or validation) error commonly used in the context of grokking, while $\mathbb{E}\varepsilon^2$ captures the irreducible part of that error.

**Memorization** When the initialization $\mathbf{a}^{(1)}$ is close to $\hat{\mathbf{a}}$, it takes less time to memorize since $t_1$ decreases with $\|\mathbf{a}^{(1)} - \hat{\mathbf{a}}\|_\infty$. Another alternative for reducing the term $\frac{\alpha\beta}{1-\rho_2}$ and guaranteeing perfect memorization earlier is to reduce $\beta$. But this increases the generalization delay $\Delta t$. Low signal-to-noise ratio $\|\mathbf{a}^*\|_2^2/\mathbb{E}\|\boldsymbol{\xi}\|_2^2$ and over-regularization (large $\beta$) also make perfect memorization difficult.

**Generalization** After memorization, when $\|\mathbf{a}^{(t)}\|_1 - \|\mathbf{a}^*\|_1$ becomes too small, we have $\|\mathbf{a}^{(t)} - \mathbf{a}^*\|_1 \approx 0$ (Figure 3) since for the problem of interest, the sparse solution $\mathbf{a}^*$ is the minimum $\ell_1$ solution to $\|\mathbf{X}\mathbf{a} - \mathbf{y}^*\|_2 \leq \epsilon$ under the sparsity constraint $s = \|\mathbf{a}^*\|_0 \ll n$ and certain assumptions on $\mathbf{X}$ (as illustrated above). The additional number of steps it takes to reach this minimum $\ell_1$-norm solution is inversely proportional to $\alpha\beta$, so that the smaller $\alpha\beta$ is, the longer it takes to recover $\mathbf{a}^*$ (grokking), and the smaller is the error $\|\mathbf{a}^{(t)} - \mathbf{a}^*\|_1$ when $t \to \infty$ (Figure 4).

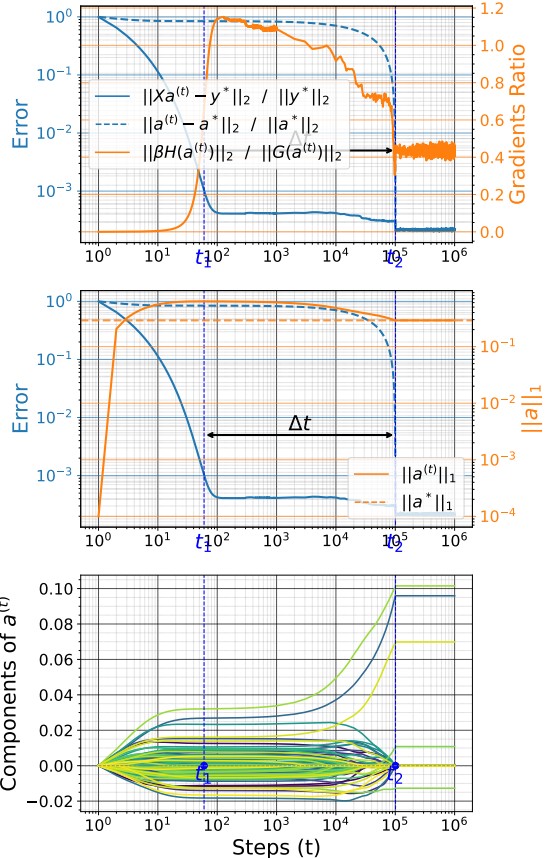

Figure 3: Relative errors, gradient ratio, norm of $\|\mathbf{a}^{(t)}\|_1$, and components of $\mathbf{a}^{(t)}$ as a function of $t$. $G(\mathbf{a}^{(t)})$ dominates $\beta H(\mathbf{a}^{(t)})$ until memorization at $t_1$, $g(\mathbf{a}^{(t_1)}) \approx 0$. From memorization $\beta H(\mathbf{a}^{(t)})$ dominates and make $\|\mathbf{a}^{(t)}\|_1$ converge to $\|\mathbf{a}^*\|_1$ at $t_2 = t_1 + \Delta t$, and so $\mathbf{a}^{(t_2)} = \mathbf{a}^*$.

**Validation Experiments** Using $(n, s, N, \alpha, \beta) = (10^2, 5, 30, 10^{-1}, 10^{-5})$, we observe a grokking-like pattern, where the training error $\|\mathbf{X}\mathbf{a}^{(t)} - \mathbf{y}^*\|_2$ first decreases to $10^{-6}$, then after a long training time, the recovery error $\|\mathbf{a}^{(t)} - \mathbf{a}^*\|_2$ decreases and matches the training error (Figure 3). The generalization results from the interplay between the sparsity level $s = \|\mathbf{a}^*\|_0$, the number of measures $N$, the $\ell_1$ regularization $\beta$ and the learning rate $\alpha$. Small $s$ requires small $N$ for generalization. Generalization occurs mainly for small (but non-zero) $\beta$ (Figure 4). Large $\beta$ pushes the recovery error to plateau at a suboptimal value early in training (causing complete memorization) or oscillations (causing no memorization). However, small values require longer training time to plateau and generally do so at a lower value of recovery error (grokking). We provided more experiments in Section H.1 of the Appendix.

**Other Iterative Method for $\ell_1$ Minimization** The above results hold for the projected subgradient method, for which the update writes $\mathbf{a}^{(t+1)} = \Pi\left(\mathbf{a}^{(t)} - \alpha\beta H(\mathbf{a}^{(t)})\right)$ with

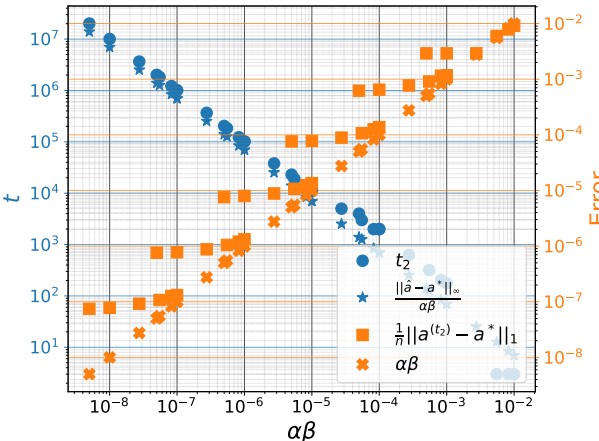

Figure 4: Generalization step $t_2$ (smaller $t$ such that $\|\mathbf{a}^{(t)} - \mathbf{a}^*\|_2 / \|\mathbf{a}^*\|_2 \leq 10^{-4}$) and recovery error $\|\mathbf{a}^{(t_2)} - \mathbf{a}^*\|_1$ as a function of $\alpha\beta$. We can see that $t_2 \propto \|\hat{\mathbf{a}} - \mathbf{a}^*\|_\infty / \alpha\beta$ and $\|\mathbf{a}^{(t_2)} - \mathbf{a}^*\|_1 \propto \alpha\beta$, i.e. small $\alpha\beta$ require longer time to converge, but do so at a lower generalization error.

$\Pi$ the projection on the set $\{\mathbf{a} \mid \mathbf{X}\mathbf{a} = \mathbf{y}^*\}$; and for the proximal gradient descent method; for which the update writes $\mathbf{a}^{(t+1)} = S_{\alpha\beta}\left(\mathbf{a}^{(t)} - \alpha G(\mathbf{a}^{(t)})\right)$ with $S_\gamma(\mathbf{a}) = \text{sign}(\mathbf{a}) \odot \max(|\mathbf{a}| - \gamma, 0)$ the soft-thresholding operator. Note the dichotomy between these two methods. Each uses one of the terms of $f(\mathbf{a})$ as the objective to be minimized and the other as a constraint. After every gradient descent update with the gradient of the objective, the iterate is projected onto the feasible set of the constraint. Still, they all exhibit a grokking behavior (more details in Section D.1 of the Appendix). One training step is enough for the projected subgradient to get zero training error. This further shows that generalization here is primarily driven by $h$.

## 3.2. Matrix Factorization

In this section, we study grokking for matrix factorization and extend the results of the previous section on sparse recovery. Given a low rank matrix $\mathbf{A}^* \in \mathbb{R}^{n_1 \times n_2}$ of rank $r$ and a measurement matrix $\mathbf{X} \in \mathbb{R}^{N \times n_1 n_2}$; we aim to minimize $\text{rank}(\mathbf{A})$ s.t. $\|\mathcal{F}_\mathbf{a}(\mathbf{X}) - \mathbf{y}^*\|_2 \leq \epsilon$ for $\mathbf{A} \in \mathbb{R}^{n_1 \times n_2}$ and $\mathbf{a} = \text{vec}(\mathbf{A}) \in \mathbb{R}^{n_1 n_2}$; where $\mathbf{y}^* = \mathcal{F}_{\text{vec}(\mathbf{A}^*)}(\mathbf{X}) + \boldsymbol{\xi}$ are the measures and $\boldsymbol{\xi} \in \mathbb{R}^N$ the error term with $\|\boldsymbol{\xi}\|_2 \leq \epsilon$. This is NP-hard (Vandenberghe & Boyd, 1996; Fazel et al., 2004). The usual convex approach for matrix factorization is to minimize $\|\mathbf{A}\|_*$ s.t. $\|\mathcal{F}_\mathbf{a}(\mathbf{X}) - \mathbf{y}^*\|_2 \leq \epsilon$ since the trace norm is the tightest convex relaxation of the rank (Fazel et al., 2001; Recht et al., 2010; Candes & Recht, 2012).

Let $n = n_1 n_2$ and $\mathbf{A}^* = \mathbf{U}^* \mathbf{\Sigma}^* \mathbf{V}^{*\top}$ be the full SVD of $\mathbf{A}^*$. We are dealing with a compressed sensing problem with the signal vector $\mathbf{a}^* = \text{vec}(\mathbf{A}^*) = \Phi \text{vec}(\mathbf{\Sigma}^*)$; where $\text{vec}(\mathbf{\Sigma}^*) \in \mathbb{R}^n$ is sparse since $\|\mathbf{\Sigma}^*\|_0 = r \leq \min(n_1, n_2) \ll n$; and $\Phi = \mathbf{V}^* \otimes \mathbf{U}^* \in \mathbb{R}^{n \times n}$ (Kro-

necker product) has orthonormal column since $\Phi^\top \Phi = \left(\mathbf{V}^{*\top} \mathbf{V}^*\right) \otimes \left(\mathbf{U}^{*\top} \mathbf{U}^*\right) = \mathbb{I}_n$. This framework encompasses several matrix factorization problems. Matrix sensing seeks the matrix $\mathbf{A}^*$ from $N$ measurement matrices $\{\mathbf{X}_i \in \mathbb{R}^{n_1 \times n_2}\}_{i \in [N]}$ and measures $\mathbf{y}^* = \left(\text{tr}(\mathbf{X}_i^\top \mathbf{A}^*)\right)_{i \in [N]}$. In this case $\mathbf{X} = [\text{vec}(\mathbf{X}_i)]_{i \in [N]} \in \mathbb{R}^{N \times n}$ since $y_i^* = \mathcal{F}_{\mathbf{a}^*}(\text{vec}(\mathbf{X}_i))$. For a matrix completion task, we have $N$ measurement vectors[2] $\left(\mathbf{X}_i^{(1)}, \mathbf{X}_i^{(2)}\right) \in \mathbb{R}^{n_1} \times \mathbb{R}^{n_2}$ and measures $y_i^* = \mathbf{X}_i^{(1)\top} \mathbf{A}^* \mathbf{X}_i^{(2)} = \mathcal{F}_{\mathbf{a}^*}\left(\mathbf{X}_i^{(2)} \otimes \mathbf{X}_i^{(1)}\right)$, that is $\mathbf{y}^* = \mathcal{F}_{\mathbf{a}^*}(\mathbf{X})$ with $\mathbf{X} = \mathbf{X}^{(2)} \bullet \mathbf{X}^{(1)} \in \mathbb{R}^{N \times n}$ (face-splitting product).

We now analyze grokking when minimizing $f(\mathbf{A}) = \frac{1}{2}\|\mathbf{X}\text{vec}(\mathbf{A}) - \mathbf{y}^*\|_2^2 + \beta\|\mathbf{A}\|_*$ via subgradient descent with learning rate $\alpha$. Like in sparse recovery, under some condition on $\alpha$ and $\beta$, the training dynamics can be decomposed in two phases; the memorization phase where update $\mathbf{A}^{(t)}$ first moves near the least square solution $\text{vec}(\hat{\mathbf{A}}) := \left(\mathbf{X}^\top \mathbf{X}\right)^\dagger \mathbf{X}^\top \mathbf{y}^*$, and a generalization phase where $\mathbf{A}^{(t)}$ converge to the minimum $\ell_*$ solution.

**Theorem 3.4.** *Assume the learning rate, the regularization coefficient and the noise satisfy $0 < \alpha\sigma_{\max}(\mathbf{X}^\top \mathbf{X}) < 2$, $0 < \beta\sqrt{\min(n_1, n_2)} < \sigma_{\max}(\mathbf{X}^\top \mathbf{X})$ and $\|\mathbf{X}^\top \boldsymbol{\xi}\|_2 \leq \sqrt{C}\alpha\beta$, $C > 0$. Let $\rho_2 := \sigma_{\max}\left(\mathbb{I}_n - \alpha\mathbf{X}^\top \mathbf{X}\right)$. There exist $t_1 < \infty$ and $C' > 0$ such that $\|\text{vec}(\mathbf{A}^{(t)} - \hat{\mathbf{A}})\|_2 \leq \frac{2\alpha\beta n^{1/2}}{1-\rho_2}$ $\forall t \geq t_1$, and $\min_{t_1 \leq t \leq t_2}\left(\|\mathbf{A}^{(t)}\|_* - \|\mathbf{A}^*\|_*\right) \leq \frac{\eta + (C+C')\alpha\beta}{2} \iff t_2 \geq t_1 + \Delta t(\eta, t_1)$ for any $\eta > 0$, with $\Delta t(\eta, t_1) := \frac{\|\mathbf{A}^{(t_1)} - \mathbf{A}^*\|_F^2}{\alpha\beta\eta}$.*

*Proof.* See the proof in Section C.6 of the Appendix. $\square$

For a suitable $\mathbf{X}$, $\mathbf{X}\text{vec}(\mathbf{A}^{(t)}) = \mathbf{y}^*$ (memorization) and $\|\mathbf{A}^{(t)}\|_* = \|\mathbf{A}^*\|_*$ are enough to conclude $\mathbf{A}^{(t)} = \mathbf{A}^*$.

**Theorem 3.5.** *Assume the linear measurement map $\mathcal{F}_.(\mathbf{X})$ satisfies the robust rank null space property of order $r$ with constants $\rho \in (0, 1)$ and $\tau > 0$, i.e for all $\mathbf{A} \in \mathbb{R}^{n_1 \times n_2}$, $\sum_{\ell=1}^r \sigma_\ell(\mathbf{A}) \leq \rho \sum_{\ell=r+1}^{\min\{n_1, n_2\}} \sigma_\ell(\mathbf{A}) + \tau\|\mathcal{F}_{\text{vec}(\mathbf{A})}(\mathbf{X})\|_2$. Then, under the same condition as in Theorem 3.4 on $\alpha$, $\beta$ and $\boldsymbol{\xi}$, there exist $C_1, C_2, C_3 > 0$ such that $\min_{t_1 \leq t \leq t_2}\|\mathbf{A}^{(t)} - \mathbf{A}^*\|_* \leq C_1\eta + C_2\alpha\beta + C_3\|\boldsymbol{\xi}\|_2$ if and only if $t_2 \geq t_1 + \Delta t(\eta, t_1)$, for any $\eta > 0$.*

*Proof.* Section C.7 of the Appendix. $\square$

Despite the similarity between this result and the one obtained on sparse recovery, this similarity is trivial only in the early phases of training. In fact, in sparse recovery problems, we take into account only the iterate

---

[2]For standard matrix completion, each $\mathbf{X}_i^{(1)}$ (resp. $\mathbf{X}_i^{(2)}$) represents a row $\ell \in [n_1]$ (resp. a column $c \in [n_2]$) of $\mathbf{A}^*$ (one-hot encoded in dimensions $n_1$ and $n_2$ respectively), giving $y_i^* = \mathbf{A}_{\ell c}^*$.

$\mathbf{a}^{(t)}$, whereas, in matrix factorization, we take into account not only the singular value matrix $\Sigma^{(t)}$ of the iterate $\mathbf{A}^{(t)}$, but also its matrix of singular vectors $\mathbf{U}^{(t)}$ (left) and $\mathbf{V}^{(t)}$ (right). We used the Wedin's $\sin\Theta$ bound (Wedin, 1972) to quantify the variations of singular vectors after the memorization phase, and the result shows that they change by at most $\mathcal{O}\left(\alpha\beta/(\gamma-1)\right)$ at each iteration, with $\gamma = \sigma_{\min}\left(\mathbf{A}^{(t_1)}\right)/\sigma_{\max}(\mathbf{A}^*)$. Also, if we take a matrix factorization problem and optimize it with only $\ell_1$, there is no grokking unless the matrix is extremely sparse so that the notion of sparsity prevails over the notion of rank, which shows the point of studying $\ell_*$ separately. Finally, neural networks trained with $\ell_1$, $\ell_2$, and $\ell_*$ have very different properties.

**Generalization** When $G(\mathbf{A})$ become negligible compare to $\beta H(\mathbf{A})$, the singular values start involving approximately as $\sigma_i\left(\mathbf{A}^{(t+1)} - \mathbf{A}^*\right) = |\sigma_i\left(\mathbf{A}^{(t)} - \mathbf{A}^*\right) - \alpha\beta|$ up to and error of order $\alpha\beta/(\gamma-1)$. This leads to a generalization through a multiscale singular value decay phenomenon (Figure 5). The small singular value after memorization converges to 0, followed by the next smaller one until $\|\mathbf{A}^{(t)}\|_* \approx \|\mathbf{A}^*\|_*$. This process requires $\Theta\left(1/\alpha\beta\right)$ steps.

**Validation Experiments** We set $(n_1, n_2, r, N, \alpha, \beta) = (10, 10, 2, 70, 10^{-1}, 10^{-4})$, and optimize the noiseless matrix completion problem using subgradient descent. We observe a grokking-like pattern, where the training error $\|\mathbf{X}\operatorname{vec}\mathbf{A}^{(t)} - \mathbf{y}^*\|_2$ first decreases to $10^{-4}$, then after a long training time, the recovery error $\|\mathbf{A}^{(t)} - \mathbf{A}^*\|_F$ decreases and matches the training error (Figures 5 and 6).

**Other Iterative Method for $\ell_*$ Minimization** The above results hold for the projected subgradient method, for which the update writes $\mathbf{A}^{(t+1)} = \Pi\left(\mathbf{A}^{(t)} - \alpha\beta H(\mathbf{A}^{(t)})\right)$ with $\Pi$ the projection on the set $\{\mathbf{A}, \mathbf{X}\operatorname{vec}\mathbf{A} = \mathbf{y}^*\}$; and for the proximal gradient descent method; for which the update writes $\mathbf{A}^{(t+1)} = S_{\alpha\beta}\left(\mathbf{A}^{(t)} - \alpha G(\mathbf{A}^{(t)})\right)$ with $S_\gamma(\mathbf{A}) = \mathbf{U}\max(\Sigma - \gamma, 0)\mathbf{V}^\top$ the soft-thresholding operator for $\mathbf{A} = \mathbf{U}\Sigma\mathbf{V}^\top$ under SVD, where $\max(\Sigma - \gamma, 0)_{ij} = \delta_{ij}\max(\Sigma_{ij} - \gamma, 0)$. More details in Section D.2.

### 3.3. Grokking Without Understanding

Contrary to prior studies (Lyu et al., 2023; Liu et al., 2023a), we find that in the over-parameterized regime ($N < n$), large-scale initialization and non-zero weight decay do not necessarily lead to grokking (Figure 7). In fact, by using only the $\ell_2$ regularization under large-scale initialization, there is an abrupt transition in the generalization error during training, driven by changes in the $\ell_2$-norm of the model parameters. This transition, however, does not result in convergence to an optimal solution and can arise even in cases where the problem has no optimal solution due to

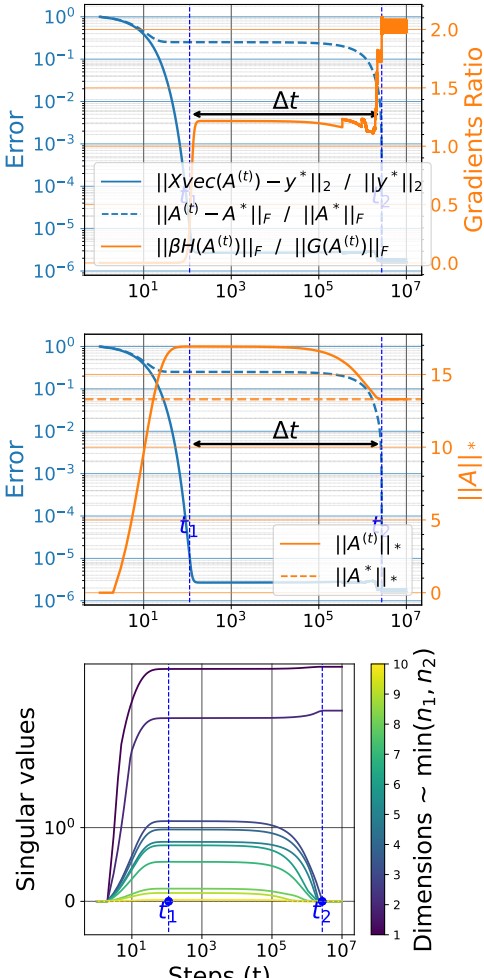

Figure 5: Relative errors, gradient ratio, the norm $\|\mathbf{A}^{(t)}\|_*$, and evolution of singular values. $G(\mathbf{A}^{(t)})$ dominates $\beta H(\mathbf{A}^{(t)})$ until memorization ($t \leq t_1$). From memorization $\beta H(\mathbf{A}^{(t)})$ dominates and make $\|\mathbf{A}^{(t)}\|_*$ converge to $\|\mathbf{A}^*\|_*$ at $t_2 = t_1 + \Delta t$, and so $\mathbf{A}^{(t_2)} = \mathbf{A}^*$.

insufficient training samples (such as sparse recovery or matrix completion using a number of samples far below the theoretical limit required for optimal recovery). We call this phenomenon "grokking without understanding" like Levi et al. (2024) who illustrated it in the case of linear classification, and we attribute it to the fact that the assumptions underlying prior theoretical predictions, particularly *Assumption 3.2* of Lyu et al. (2023), are violated in our setting (more details in the Section E of the Appendix).

For the problems above (sparse recovery and matrix factorization), by replacing $h(\mathbf{a})$ in $f(\mathbf{a}) = g(\mathbf{a}) + \beta h(\mathbf{a})$ by $h(\mathbf{a}) = \frac{1}{2}\|\mathbf{a}\|_2^2$, the model converge to the least square solution $\hat{\mathbf{a}} := \left(\mathbf{X}^\top\mathbf{X} + \beta\mathbb{I}_n\right)^\dagger \mathbf{X}^\top\mathbf{y}^*$, but this solution can not give rise to generalization when $N < n$.

**Theorem 3.6.** *Define* $\rho_2 := \left\|\mathbb{I}_n - \alpha\left(\mathbf{X}^\top\mathbf{X} + \beta\mathbb{I}_n\right)\right\|_{2\to2}$.

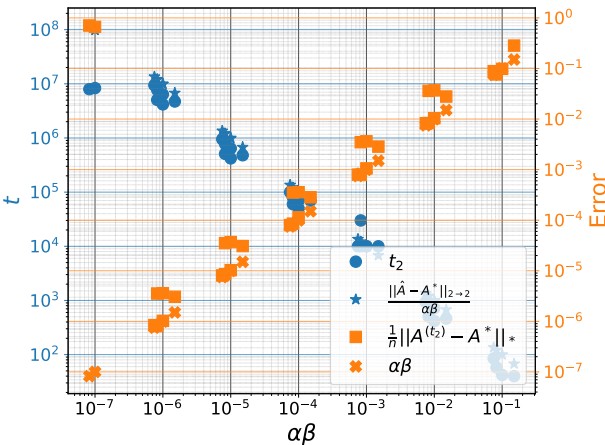

Figure 6: Generalization step $t_2$ (smaller $t$ such that $\|\mathbf{A}^{(t)} - \mathbf{A}^*\|_F/\|\mathbf{A}^*\|_F \leq 10^{-4}$) and recovery error $\|\mathbf{A}^{(t_2)} - \mathbf{A}^*\|_2$ as a function of $\alpha\beta$. We can see that $t_2 \propto \|\hat{\mathbf{A}} - \mathbf{A}^*\|_{2\to2}/\alpha\beta$ and $\|\mathbf{A}^{(t_2)} - \mathbf{A}^*\|_F \propto \alpha\beta$, i.e. small $\alpha\beta$ require longer time to converge, but do so at a lower generalization error. The outlier for very small $\alpha\beta$ is due to insufficient training.

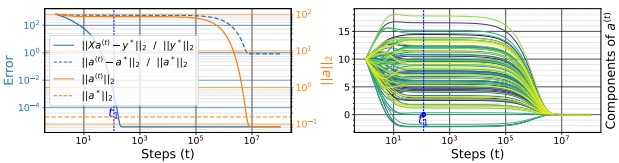

Figure 7: Relative errors and the the norm $\|\mathbf{a}^{(t)}\|_2$ for large initialization scale $\|\mathbf{a}^{(1)}\|_2 = 10$ and small weights decay $\beta = 10^{-5}$. Weight decay alone just reduces the components of the iterate $\mathbf{a}^{(t)}$, not $\|\mathbf{a}^{(t)}\|_0$.

*Assume the learning rate satisfies $0 < \alpha < \frac{2}{\sigma_{\max}(\mathbf{X}^\top\mathbf{X})+\beta}$. Then $\|\mathbf{a}^{(t)} - \hat{\mathbf{a}}\|_2 \leq \rho_2^{t-1}\|\mathbf{a}^{(1)} - \hat{\mathbf{a}}\|_2 \; \forall t \geq 1$. On the other hand, for $N < n$, $\|\hat{\mathbf{a}} - \mathbf{a}^*\|_2^2 \geq \|(\mathbb{I}_n - \mathbf{X}^\top(\mathbf{X}\mathbf{X}^\top)^\dagger\mathbf{X})\mathbf{a}^*\|_2^2$. In particular, if $\mathbf{a}^*$ has a nonzero component orthogonal to the column space of $\mathbf{X}$, then $\hat{\mathbf{a}}$ cannot perfectly generalize to $\mathbf{a}^*$.*

*Proof.* Section C.8 of the Appendix. $\square$

Even when $\ell_1$ is present, if the weight decay $\beta$ is choose such that $\|\hat{\mathbf{a}}\|_\infty \ll \alpha\beta$, then $\mathbf{a}^{(t)}$ will get stuck near $\hat{\mathbf{a}}$, and there will be no generalization. So, a bad choice of $\beta$ can be detrimental to generalization (it is better not to use $\ell_2$ on that problem unless the initialization scale is nontrivial). Since the minimum $\ell_2$ norm solution only gives memorization, the $\ell_2$ norm cannot be used as an indicator of grokking.

### 3.4. Amplifying Grokking through Data Selection

The analysis of recovery guarantees for matrix factorization hinges on local coherence of the target matrix $\mathbf{A}^* = \mathbf{U}^*\Sigma^*\mathbf{V}^{*\top}$ (compact SVD). The local coherence measures

$\mu_i = \frac{n_1}{r}\|\mathbf{U}^{*\top}\mathbf{e}_i^{(n_1)}\|^2$ and $\nu_j = \frac{n_2}{r}\|\mathbf{V}^{*\top}\mathbf{e}_j^{(n_2)}\|^2$ (for $(i,j) \in [n_1] \times [n_2]$) quantify how strongly individual rows and columns align with the top singular vectors, where $\mathbf{e}_i^{(n_i)}$ be the $i^{th}$ vector of the canonical basis of $\mathbb{R}^{n_i}$. These quantities, also known as leverage scores, indicate the "influence" of each row $i$ or column $j$ on the low-rank structure. A row/column with a high leverage score projects strongly onto the span of the singular vectors, meaning that a relatively small number of its entries capture much of the matrix's structure. Uniformly low coherence ($\mu_i$ and $\nu_i$ close to 1) implies that the matrix's information is well-distributed across rows and columns, thereby reducing the number of samples needed for exact recovery. The significance of local coherence extends to sampling strategies and recovery bounds. For matrix completion, given $N \leq n_1 n_2$ and $\tau \in [0,1]$, we select the first $\tau N$ entries $(i,j)$ with the highest values of $\mu_i + \nu_j$, and the remaining $(1-\tau)N$ entries uniformly among the rest. As $\tau \to 1$, performance improves, and the number of examples required to generalize decreases exponentially, as does the time it takes the models to do so (Figure 8).

Unlike matrix factorization, large values of coherence are detrimental to generalization for compressed sensing. The incoherence between the measurement vectors (rows of $\mathbf{M}$) and the sparse basis (columns of $\Phi$) is crucial for successfully recovering $\mathbf{b}^* = \Phi\mathbf{a}^*$ from the measures $\mathbf{y}^* = \mathbf{M}\mathbf{b}^* + \xi$. If

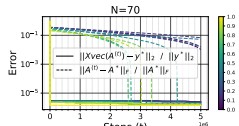

Figure 8: Training and recovery error as a function of data size and coherence $\tau$.

$\mathbf{M}$ is incoherent with $\Phi$, each measurement captures a distinct "view" of $\mathbf{b}^*$, reducing redundancy. This diversity of information allows for the successful reconstruction of $\mathbf{a}^*$ even with fewer measurements (e.g., below the Nyquist rate for signals). We also observed that higher coherence increases the number of samples required for recovery and delays generalization, while lower coherence results in faster generalization and better recovery. While grokking has been studied extensively, the impact of data selection on grokking remains largely unexplored, making this one of the first works to address this critical aspect. We provide more details in Section G of the Appendix.

### 3.5. Implicit Bias of the Depth

We explore the role of overparameterization in sparse recovery using a linear network parameterized as $\mathbf{a} = \odot_{k=1}^{L}\mathbf{A}_k$, where the depth $L \geq 2$ introduces overparameterization without altering the linearity of the function class. Our findings reveal that depth can replace $\ell_1$-regularization for generalization when initializa-

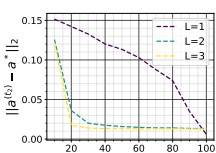

Figure 9: Recovery error as a function of depth and data size.

tion is small, as the gradient updates introduce a preconditioning effect that promotes sparsity and enables signal recovery. Unlike the shallow case ($L = 1$), where sparsity cannot be enforced without $\ell_1$-regularization, depth provides an implicit mechanism that biases updates toward sparsity, resulting in better generalization with fewer measurements (Figure 9). Additionally, depth reduces the generalization delay, with abrupt phase transitions and a staircase-like loss curve during training. The generalization error decreases with increasing depth $L$, showing that depth effectively compensates for fewer measurements and facilitates recovery, albeit at the cost of longer training times. For $L \geq 2$, large-scale initialization combined with small non-zero $\ell_2$-regularization results in grokking, unlike the shallow case where the phenomenon of "grokking without understanding" is observed. More details in Section F.1 of the Appendix.

We discuss deep matrix factorization in Section F.2 of the Appendix, as it is already well-known that overparametrization by adding depth makes it possible to recover low-rank matrices without any explicit regularization (Gunasekar et al., 2017; Arora et al., 2019; Gidel et al., 2019; Gissin et al., 2019; Razin & Cohen, 2020; Li et al., 2020).

# 4. General Setting

In this section, we show that $\ell_1$, $\ell_*$, and other domain-specific regularizers can replace $\ell_2$ in a more general setting and induce or control grokking.

**Non linear Teacher-Student**   We consider a teacher $\mathbf{y}^*(\mathbf{x}) = \mathbf{B}^*\phi(\mathbf{A}^*\mathbf{x})$ from $\mathbb{R}^d$ to $\mathbb{R}^c$ with $r$ hidden neurons; where $\phi(z) = \max(z,0)$. We independently sample $N$ inputs output pair $\mathcal{D}_{\text{train}} = \{(\mathbf{x}_i, \mathbf{y}^*(\mathbf{x}_i))\}_{i=1}^N$ and optimize the parameters $\theta = \{\mathbf{A}, \mathbf{B}\}$ of a student $\mathbf{y}_\theta(\mathbf{x}) = \mathbf{B}\phi(\mathbf{A}\mathbf{x})$ on them with the square loss function $g(\theta) = \frac{1}{2N}\sum_{i=1}^N \|\mathbf{y}_\theta(\mathbf{x}_i) - \mathbf{y}^*(\mathbf{x}_i)\|_2^2$ and different regularizer $h(\theta)$, $\ell_p$ for $p \in \{1, 2, *\}$. For all of these regularizer, the smaller is $\alpha\beta$, the longer is the delay between memorization and generalization (see Figures 10 for the training curve with $\ell_1$, and 31, 32, 33 for more results with $\ell_{*/2}$).

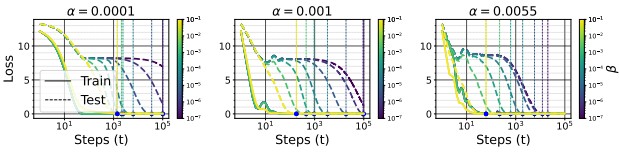

Figure 10: Training and test error two layers `ReLU` teacher-student with $\ell_1$ regularization, for different values of the learning rate $\alpha$ and the $\ell_1$ coefficient $\beta$.

**Domain Specific Regularization**   Physics-Informed Neural Networks (Raissi et al., 2019) leverage prior knowledge from differential equations by incorporating their residuals into the loss function, ensuring that solutions remain

consistent with physical laws. Sobolev training (Czarnecki et al., 2017) generalizes this idea by incorporating not only input-output pairs but also derivatives of the target function. We optimizer the student from the previous paragraph by adding on the objective function the first order Sobolev penalty $\frac{\beta}{N}\sum_{i=1}^N \left\|\frac{\partial \mathbf{y}_\theta}{\partial \mathbf{x}}(\mathbf{x}_i) - \frac{\partial \mathbf{y}^*}{\partial \mathbf{x}}(\mathbf{x}_i)\right\|_{\text{F}}^2$, where the hyperparameter $\beta$ ensures that the model not only fits the data but also respects known smoothness constraints or differential structure, which is crucial in physics-based applications. Large $\alpha\beta$ values lead to faster grokking ( Figures 11 and 34).

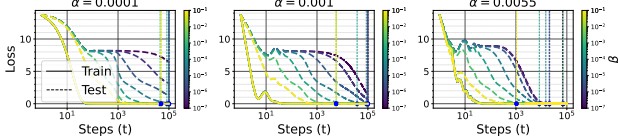

Figure 11: Training and test error two layers `ReLU` teacher-student with Sobolev training, for different values of the learning rate $\alpha$ and the Sobolev coefficient $\beta$.

**Classification**   On the algorithmic dataset (Power et al., 2022), $\ell_1$ and $\ell_*$ have the same effect on grokking as $\ell_2$, i.e., smaller regularization coefficient (and learning rate) delay generalization (see Figure 2 for $\ell_*$, and 30 for $\ell_1$ and $\ell_2$). We observe a similar phenomenon on a two-layer `ReLU` MLP trained on MNIST (Section H.3.4).

# 5. Discussion and Conclusion

This work extends the understanding of grokking, showing that the transition from memorization to generalization can be induced not just by $\ell_2$ regularization but also by sparsity or low-rank structure regularization (e.g., $\ell_1$ and nuclear norm regularization) or domain-specific regularization. These findings are particularly relevant in practice, where large-scale initialization is not always feasible, yet grokking still occurs. Sparse and low-rank-based models with good generalization performance are very useful in machine learning today, not only because they consume less memory during training and inference but also because they are more interpretable than their dense and full-rank counterparts. The sparsity and low-rank assumptions are also central to techniques such as Low-Rank Adaptation of larger deep learning models (Hu et al., 2021) and model pruning (Han et al., 2015) in resource-efficient deep learning; sparse autoencoders in mechanistic interpretability (Cunningham et al., 2023; Bricken et al., 2023) for AI safety (Bereska & Gavves, 2024); to name a few.

Our results highlight that in deep models, gradient descent implicitly drives the model towards solutions with sparse or low-rank properties, effectively mitigating overfitting (Arora et al., 2018). We also study the impact of data selection on grokking and demonstrate that it can be mitigated solely through data selection.

## Acknowledgements

We thank the members of Guillaume Rabusseau's group at Mila, for the insightful discussions during the development of this project. Pascal Tikeng is grateful to Mahta Ramezanian-Panahi, Sékou-Oumar Kaba, and Mohammad Pezeshki for helpful conversations in the early stages of this work. The authors acknowledge the material support of NVIDIA in the form of computational resources. Pascal Tikeng Notsawo acknowledges the support from the Canada Excellence Research Chairs (CERC) program. Guillaume Rabusseau acknowledges the support of the CIFAR AI Chair program. Guillaume Dumas was supported by the Institute for Data Valorization, Montreal and the Canada First Research Excellence Fund (IVADO; CF00137433), the Fonds de Recherche du Québec (FRQ; 285289), the Natural Sciences and Engineering Research Council of Canada (NSERC; DGECR-2023-00089), and the Canadian Institute for Health Research (CIHR 192031; SCALE).

## Impact Statement

This paper aims to advance the field of Machine Learning by improving the understanding of grokking in neural networks. While the focus is on theoretical contributions, our work has practical implications for optimizing model training. We acknowledge the potential ethical considerations of AI technologies, but do not feel any specific issues need to be highlighted here at this time.

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

# A. Related Works

## A.1. Grokking

**Universality**  Liu et al. (2023a) took the first step in demonstrating the universality of grokking by inducing it on non-algorithmic data, notably on image classification, sentiment analysis, and molecule property prediction. On the same line, Gromov (2023); vZunkovivc & Ilievski (2024); Levi et al. (2024) show that grokking can occur for (analytically solvable toy) models. Barak et al. (2022) observed grokking on the binary sparse parity problem, Charton (2024) on greatest common divisor (with Transformer), Doshi et al. (2024b) on modular polynomials, Lyu et al. (2023) on matrix completion and sparse linear predictors, Beck et al. (2024) and Levi et al. (2024) on binary logistic classification, Xu et al. (2023) on XOR-cluster training data. Miller et al. (2024b) show that grokking occurs beyond neural networks—in models like Gaussian process (regression and classification), linear regression, and Bayesian neural networks—driven by a trade-off between model complexity and error, not just feature learning or weight decay. Wang et al. (2024) and Abramov et al. (2025) show that grokking enables Transformers to develop reasoning abilities, emerging only after extended training, whether on synthetic comparison/composition tasks or real-world multi-hop reasoning augmented with inferred facts. Li (2024) observe that both binary and ternary Transformer exhibit grokking on modular addition under weight decay regularization, and do not do so without weight decay. Mallinar et al. (2024) show that grokking occurs beyond neural networks and gradient descent, as kernel-based Recursive Feature Machines trained with Average Gradient Outer Product also exhibit delayed generalization through emergent structured features. Kumar et al. (2024) find that mice show grokking-like behavior, i.e., even after behavioral performance plateaus, neural representations in sensory cortex continue evolving, improving generalization through margin maximization. Mustafa & Burkholz (2024) investigate grokking-like phenomena in Graph Neural Networks, particularly focusing on tasks involving heterophilic and homophilic graphs.

**Interpretability and Explainability**  Since the concept of "grokking" emerged, numerous theories have been proposed to explain the phenomenon. Liu et al. (2023b) present representation learning as the main underlying factor of the existence of generalizing solutions, supporting Power et al. (2022)'s preliminary observations. Exploring the direction that separates generalization from memorization solutions (i.e., progressions measures for generalization), recent studies shed light on various factors, such as neuron activity (Nanda et al., 2023), weight norm of the model parameters (Liu et al., 2023a), sparsity (Merrill et al., 2023), time scales of pattern formation (Davies et al., 2023), Fourier gap (Barak et al., 2022), last layer norm (Thilak et al., 2022), fast vs low-frequency components (Zhou et al., 2024), mutual information ratio of neural representation (Song et al., 2024), phase transition (Clauw et al., 2024; Rubin et al., 2024), rank minimization (Yunis et al., 2024), rise and fall in model complexity (DeMoss et al., 2024; Humayun et al., 2024), etc. Also, weight decay and weight norm decrease (Liu et al., 2023a; Varma et al., 2023), a gradual process facilitated by optimization (Nanda et al., 2023; Merrill et al., 2023; Barak et al., 2022; Davies et al., 2023; Notsawo et al., 2023), the instability induced by the Adam optimizer (Thilak et al., 2022), are among the factors that has been use to explain why the transition from memorization to generalization. According to Varma et al. (2023), grokking arises from a generalizing circuit being favored over a memorizing circuit. Miller et al. (2024a) introduce a method to quantify the sharpness of the grokking transition in neural networks. Doshi et al. (2024a) show that in the presence of label corruption, grokking can still occur, with networks first generalizing before later unlearning memorized noise, highlighting a separation between generalization and memorization phases. Zhu et al. (2024) show that grokking in language models emerges when training data exceeds a critical size, with larger models requiring more data to generalize beyond memorization. Huang et al. (2024) propose a unified framework explaining grokking, double descent, and emergent abilities through the lens of competition between memorization and generalization circuits. Chughtai et al. (2023) and Stander et al. (2024) reverse engineer (small) neural networks learning finite group composition via mathematical representation theory. Golechha (2024) show that $\ell_2$ norm alone can not explain grokking, as it occurs even outside the typical "goldilocks zone"(Liu et al., 2023a), and propose alternative measures like activation sparsity and weight entropy that better track generalization dynamics. Prieto et al. (2025) explain grokking as a result of Softmax Collapse—numerical instability from floating-point errors that halts learning after memorization.

**Predictability**  To the question of whether grokking can be predicted, Notsawo et al. (2023) answered in the affirmative, proposing as a predictable measure the spectral signature of the training loss in the early optimization phases of the model training. Hu et al. (2024) models neural network training as a Hidden Markov Model (HMM) over weight statistics to uncover discrete training phases. Applied to grokking tasks, this reveals distinct latent states corresponding to memorization and generalization, with "detour" states explaining delayed generalization. The approach offers a structured way to analyze and predict grokking by linking it to transitions between these latent training states. Murty et al. (2023) explore how vanilla Transformers eventually grok hierarchical rules after extended training, and find that weight norm and attention sparsity

are inconsistent and often unreliable for forecasting generalization. They introduce a "tree-structuredness" (Murty et al., 2022) metric that significantly outperforms weight norm in predicting whether and when a model will grok. Miller et al. (2024b) argue that grokking may be possible in any model where the solution search is guided by complexity and error. This notion of complexity is related to what we refer to as regularization in our work. Fan et al. (2024) show that deeper MLPs on MNIST grok more often and in stages, and that weight norm poorly predicts generalization compared to internal feature rank dynamics.

**Stochasticity**   Our work also contradicts the hypothesis put forward when grokking was first observed, namely that grokking may be due to stochasticity or an anomaly in the optimization (Power et al., 2022; Thilak et al., 2022). For sparse recovery and matrix factorization, the optimization algorithms we use are all deterministic (up to initialization).

**Feature Learning**   Kumar et al. (2023) show that grokking can emerge without weight decay, as a result of a delayed transition from lazy (fixed-feature) to rich (feature-learning) dynamics during training. In contrast, Lyu et al. (2023) explain grokking through implicit bias, showing that large initialization and small weight decay induce a shift from kernel-like to margin-maximizing behavior, highlighting different but complementary mechanisms behind grokking. Xu et al. (2023) provide a theoretical example of grokking in ReLU networks by analyzing the feature learning process under GD, and show that test generalization can emerge well after perfect (and initially non-generalizing) memorization of noisy XOR-cluster training data. Morwani et al. (2024) demonstrate that neural networks trained on tasks like modular addition and sparse parity naturally converge to solutions based on Fourier features and group-theoretic representations (Nanda et al., 2023; Chughtai et al., 2023). This behavior arises from the principle of margin maximization, even without explicit regularization, providing a theoretical explanation for grokking as a delayed transition from memorization to structured feature learning.

**Large Initialization and $\ell_2$ Regularization**   A direct work related to ours on grokking is Lyu et al. (2023), which analyzes the emergence of grokking under $\ell_2$ regularization and large-scale initialization in homogeneous models. Their results show that under such conditions, grokking can occur as a delayed generalization phenomenon. However, their analysis is limited to $\ell_2$ weight decay and does not generalize to other regularization schemes. In contrast, our work extends beyond $\ell_2$, demonstrating that grokking also arises under $\ell_1$ and nuclear norm ($\ell_*$) regularization, even without large-scale initialization. In the theoretical settings we study—such as sparse recovery and low-rank matrix factorization—$\ell_2$ regularization alone fails to induce grokking. Instead, it produces a sharp drop in generalization error during training that does not correspond to true generalization. We refer to this failure mode as grokking without understanding, which we attribute to the violation of key assumptions in Lyu et al. (2023) (notably Assumption 3.2). Our results further show that grokking may even be necessary in practice: for instance, when targeting low-rank solutions via $\ell_*$ regularization, it is preferable to use the smallest feasible regularization strength and train well beyond the point of overfitting to achieve generalization. Levi et al. (2024) work on classification settings and show that the sharp increase in generalization accuracy may not imply a transition from "memorization" to "understanding" but can be an artifact of the accuracy measure. This aligns with the "grokking without understanding" problem we observe in sparse recovery and low-rank matrix factorization.

**Accelerating Grokking**   Park et al. (2024) accelerate grokking by using data augmentation for commutative operations and transfer learning across model components aligned with the Kolmogorov-Arnold representation. Lee et al. (2024) accelerate grokking by amplifying slow gradient components, reducing training time across tasks. Xu et al. (2025) accelerate grokking by initializing larger models with embeddings from smaller trained models, eliminating delayed generalization. Minegishi et al. (2025) show that grokking can be accelerated by identifying "grokking tickets"—sparse subnetworks that emerge after generalization—and retraining them, which leads to much faster generalization than the original dense model. (Prieto et al., 2025) accelerate grokking using a stable softmax variant and a gradient projection method to prevent collapse and enable faster generalization without regularization.

**Sparsity and Low-Rankness**   Barak et al. (2022) observed grokking on the binary sparse parity problem, and Merrill et al. (2023) shows that two subnetworks compete during training on such a task: a dense (memorization) subnetwork and a sparse (generalization) subnetwork. Since a very sparse network that generalizes the sparse parity data can be built (Merrill et al., 2023), we conjecture that it is this sparsity that gives the models trained on this task their grokking nature, as well as the algorithmic dataset (Power et al., 2022), but leave further investigation for future work. To the best of our knowledge, we are the first to formally study grokking in the context of sparse recovery and low-rank matrix factorization (the shallow case). Lyu et al. (2023) show that low-rank matrix completion problems exhibit grokking with large initialization. But we

prove that even on such a simple model, we do not need way decay and large initialization to observe grokking, but just $\ell_{1/*}$ regularization.

### A.2. Model Distillation, Sparse Dictionary Learning, Sparse Auto-Encoder, and Coreset Selection

Note that the parameterization $\mathbf{b} = \Phi\mathbf{a}$ gives $\mathcal{F}_{\mathbf{a}}(\mathbf{x}) = \mathbf{a}^\top \Phi^\top \mathbf{x}$, i.e. $\mathcal{F}_{\mathbf{a}}(\mathbf{X}) = \mathbf{X}\Phi\mathbf{a}$; which is a two-layer linear neural network with the weights of the first layer given by $\Phi^\top \in \mathbb{R}^{m \times n}$ (fixed) and those of the second layer given by $\mathbf{a} \in \mathbb{R}^m$, to be optimized. Compressed sensing thus corresponds somehow to a model distillation (teacher-student), where we aimed to find a sparse version of model parameters using fewer data points. This could be strengthened by assuming that we have only the measures $\mathbf{y}^*$ of the sparse representation $\mathbf{a}^*$ of the signal and the measurement matrix $\mathbf{M}$, but not the basis in which it is sparse, and the problem is to find both $\Phi$ and $\mathbf{a}^*$, i.e. minimize $\|\mathbf{a}\|_1$ subject to $\|\mathcal{F}_{\Phi\mathbf{a}}(\mathbf{M}) - \mathbf{y}^*\|_2 \leq \epsilon$ and $\Phi \in \mathcal{C}$. This is the case for matrix factorization, where the sparse basis (singular vector space) is jointly optimized along with the sought sparse coordinates $\mathbf{a}^*$ (singular values). Here, $\mathcal{C}$ can be the set of orthonormal matrix ($\Phi^\top \Phi = \mathbb{I}_n$), unit column norm matrix ($\Phi_{:,i}^\top \Phi_{:,i} = 1$), etc. It can be interesting to see if $\ell_1$ allows us to recover the parameters with fewer samples than the most used $\ell_2$ in deep learning. Looking for a sparse $\mathbf{a}$ mean, we assume that only a few components of the representation $\mathbf{h}(\mathbf{x}) = \Phi^\top \mathbf{x} \in \mathbb{R}^n$ contribute to the output $\mathbf{y}(\mathbf{x}) = \mathbf{a}^\top \mathbf{h}(\mathbf{x})$. This kind of method (*sparse auto-encoder*) is used a lot in mechanistic interpretability today to separate and interpret the features a pre-trained model has learned (Cunningham et al., 2023; Bricken et al., 2023), a promising direction for AI safety (Bereska & Gavves, 2024). Compressed sensing theory gives us an idea of how to select data to extract important features with the fewest possible examples: we need to select the samples that are most "incoherent" with the feature extractor (assumed fixed).

This formulation is also related to:

- The coreset selection problem (Tsang et al., 2005; Huggins et al., 2017; Lucic et al., 2018), which consists of selecting a small subset comprised of most informative training samples such that training on this subset can achieve comparable or even better performance with that on the full dataset (Zhou et al., 2022).

- The sparse dictionary learning problem (Olshausen & Field, 1996; Aharon et al., 2006; Mairal et al., 2010; 2014), where we have a collection $\mathbf{B} \in \mathbb{R}^{d \times N}$ of $N$ point in $\mathbb{R}^d$ and aim to write each of them as a combination of a few numbers of atoms of a dictionary $\Phi \in \mathbb{R}^{d \times n}$, that is $\mathbf{B}_{:,i} = \Phi\mathbf{A}_{:,i}\forall i \in [N]$, with $\|\mathbf{A}_{:,i}\|_0$ small. Let $\mathcal{C} = \{\mathbf{C} \in \mathbb{R}^{d \times n}, \|\mathbf{C}_{:,i}\|_2 \leq 1 \ \forall i \in [n]\}$. We want $\hat{\Phi} = \arg\min_{\Phi \in \mathcal{C}} \frac{1}{N} \sum_{i=1}^N l(\mathbf{B}_{:,i}, \Phi)$ with $l(\mathbf{b}, \Phi) = \min_{\mathbf{a} \in \mathbb{R}^n} \|\mathbf{b} - \Phi\mathbf{a}\|_2^2 + \lambda\|\mathbf{a}\|_1$. That is

$$\hat{\Phi}, \hat{\mathbf{A}} = \underset{\Phi \in \mathcal{C}, \mathbf{A} \in \mathbb{R}^{n \times N}}{\arg\min} \frac{1}{N}\|\mathbf{B} - \Phi\mathbf{A}\|_F^2 + \lambda \sum_{i=1}^N \|\mathbf{A}_{:,i}\|_1$$

The generalization error is $\mathcal{E}(\Phi) = \mathbb{E}_{\mathbf{b}} l(\mathbf{b}, \Phi)$. The sparse coding problem $l(\mathbf{b}, \Phi) = \min_{\mathbf{a} \in \mathbb{R}^n} \|\mathbf{b} - \Phi\mathbf{a}\|_2^2 + \lambda\|\mathbf{a}\|_1$ is just the problem we study above.

We leave these points as a future direction for this work.

### A.3. Finite Field

**Compressed Sensing**  In compressed sensing and sparse recovery in $\mathbb{F}_p = \mathbb{Z}/p\mathbb{Z}$ (for a prime integer $p$), incoherence between the measurement matrix $\mathbf{M}$ and the sparse basis $\Phi$ remains important but requires modular arithmetic. Recovery of a sparse signal $\mathbf{b}^*$ from the measurements $\mathbf{y}^* = \mathbf{M}\Phi\mathbf{b}^*$ is possible, provided that $\mathbf{M}$ and $\Phi$ are chosen to ensure low coherence in the modular structure. However, recovery algorithms are more complex due to the finite nature of the field and may involve different mathematical tools (e.g., algebraic techniques from coding theory) instead of standard optimization methods.

**Matrix Factorization**  We can see $\mathbf{A}^* \in \mathbb{F}_p^{n_1 \times n_2}$ as representing a binary operation over $\mathbb{F}_p$, and the aim is to learn this operation from a proportion of sample[3]. This is the vanilla setup where grokking was first observed (Power et al., 2022). Large-scale initialization can not explain the grokking phenomenon in such a setup since it is often observed with standard initialization. Many works also use mechanistic interpretability to find the algorithm deep neural networks use to generalize

---

[3]For example, $\mathbf{A}^* = \mathbf{u}\mathbf{u}^\top$ with $\mathbf{u} = [0, \ldots, p - 1]$ for the multiplication operation.

in such a setup and propose a progression measure for generalization (Nanda et al., 2023; Gromov, 2023). Still, none explain what makes such data particular. Liu et al. (2023b) propose the representation learning hypothesis but uses a loss function that encourages such representation to emerge for standard deep neural networks (MLP and Transformer). Mohamadi et al. (2024) explain grokking in modular addition as a transition from a kernel regime, where generalization fails, to a rich regime that captures the task's global algebraic structure—making grokking especially pronounced in modular arithmetic. Investigating how our work can effectively answer why grokking is naturally observed in such a setup is a future direction for this work.

## B. Notations

Our notation is standard.

- We use the notation iid for "independently and identically distributed". For $\mathbf{A} \in \mathbb{R}^{m \times n}$, the notation $\mathbf{A} \overset{iid}{\sim} \mathcal{N}\left(\mu, \sigma^2\right)$ means the entries of $\mathbf{A}$ are independently sampled from the normal distribution with mean $\mu$ and standard deviation $\sigma$. This notation is valid for any other distribution used in this paper.

- For two functions $\phi, \psi : \mathbb{R}_{\geq 0} \to \mathbb{R}$, we write $\phi(z) = \mathcal{O}(\psi(z))$ if there exist constants $C > 0$ and $z_0 \in \mathbb{R}_{\geq 0}$ such that $|\phi(z)| \leq C \psi(z)$ for all $z \geq z_0$; $\phi(z) = \Omega(\psi(z))$ if $\phi(z) \geq C \psi(z)$ for all $z \geq z_0$; and $\phi(z) = \Theta(\psi(z))$ if both $\phi(z) = \mathcal{O}(\psi(z))$ and $\phi(z) = \Omega(\psi(z))$.

- We let $\mathbf{e}_k^{(n)} = [\mathbb{I}_n]_{:,k}$ be the $k^{th}$ vector of the canonical basis of $\mathbb{R}^n$, $\mathbf{e}_{kl}^{(n)} = \delta_{kl} \forall l$. The subscript $(n)$ will be omitted when the context is clear.

- $\sigma_{\max / \min}(\mathbf{A})$ is the maximum (resp. minimum) singular value of a matrix $\mathbf{A}$, with $\lambda_{\max / \min}(\mathbf{A})$ the corresponding eigenvalue

- For a vector $\mathbf{x} \in \mathbb{R}^n$, $\|\mathbf{x}\|_0 = |\{i \in [n], \mathbf{x}_i \neq 0\}|$, $\|\mathbf{x}\|_p = \left(\sum_{i=1}^n |\mathbf{x}_i|^p\right)^{\frac{1}{p}} \forall p \in (0, \infty)$ and $\|\mathbf{x}\|_\infty = \max_{i \in [n]} |\mathbf{x}_i|$.

- For a matrix $\mathbf{A} \in \mathbb{R}^{m \times n}$, the schatten $p$-norm of $\mathbf{A}$ is $\|\mathbf{A}\|_p = \left(\sum_i \sigma_i(\mathbf{A})^p\right)^{1/p}$, where $\{\sigma_i(\mathbf{A})\}_i$ is the set of singular value of $\mathbf{A}$. For $p = 1$, this gives the trace/nuclear norm $\|\mathbf{A}\|_* = \sum_i \sigma_i(\mathbf{A}) = \operatorname{tr}\left(\sqrt{\mathbf{A}^\top \mathbf{A}}\right)$. The induced $p \to q$ norm of $\mathbf{A}$ is $\|\mathbf{A}\|_{p \to q} = \sup_{\mathbf{x} \neq 0} \frac{\|\mathbf{A}\mathbf{x}\|_q}{\|\mathbf{x}\|_p} = \sup_{\|\mathbf{x}\|_p = 1} \|\mathbf{A}\mathbf{x}\|_q$. We have $\|\mathbf{A}\|_{1 \to 1} = \max_{j \in [n]} \sum_{i=1}^m |\mathbf{A}_{ij}|$ (maximum absolute column sum), $\|\mathbf{A}\|_{2 \to 2} = \|\mathbf{A}\|_2 = \sigma_{\max}(\mathbf{A})$ (operator norm, spectral norm, induced 2-norm) and $\|\mathbf{A}\|_{\infty \to \infty} = \max_{i \in [m]} \sum_{j=1}^n |\mathbf{A}_{ij}|$ (maximum absolute row sum).

- $\odot$ is Hadamard product. For $\mathbf{A} \in \mathbb{R}^{m \times n}$ and $\mathbf{B} \in \mathbb{R}^{m \times n}$, $(\mathbf{A} \odot \mathbf{B})_{i,j} = \mathbf{A}_{i,j} \mathbf{B}_{i,j}$ ($0 \leq i < m, 0 \leq j < p$).

- $\otimes$ is the Kronecker product. For $\mathbf{A} \in \mathbb{R}^{m \times n}$ and $\mathbf{B} \in \mathbb{R}^{p \times q}$, $(\mathbf{A} \otimes \mathbf{B})_{pr+v,qs+w} = \mathbf{A}_{rs} \mathbf{B}_{vw}$ ($0 \leq r < m, 0 \leq v < p$, $0 \leq s < n$ and $0 \leq w < q$).

- For $\mathbf{A} \in \mathbb{R}^{m \times n}$ and $\mathbf{B} \in \mathbb{R}^{p \times n}$, the Khatri-Rao product $\mathbf{A} \star \mathbf{B} \in \mathbb{R}^{mp \times n}$ contains in each column $i \in [n]$ the matrix $\mathbf{A}_{:,i} \otimes \mathbf{B}_{:,i}$. We have the formula $\mathbf{A} \star \mathbf{B} = (\mathbf{A} \otimes 1_p) \odot (1_m \otimes \mathbf{B})$.

- For $\mathbf{A} \in \mathbb{R}^{m \times n}$ and $\mathbf{B} \in \mathbb{R}^{m \times p}$, the face-splitting product $\mathbf{A} \bullet \mathbf{B} \in \mathbb{R}^{m \times np}$ contains in each row $i \in [m]$ the matrix $\mathbf{A}_{i,:} \otimes \mathbf{B}_{i,:}$. It can be seen as the row-wise Khatri-Rao product, and we have $(\mathbf{A} \bullet \mathbf{B}) = (\mathbf{A}^\top \star \mathbf{B}^\top)^\top = \left(\mathbf{A} \otimes 1_p^\top\right) \odot (1_n^\top \otimes \mathbf{B})$.

## C. Proofs

### C.1. Proof of Theorem 2.1

Let $g : \mathbb{R}^p \to [0, \infty)$ be a differentiable function, $h : \mathbb{R}^p \to [0, \infty)$ be a subdifferentiable (often convex) function and $\beta > 0$. We want to minimize $f := g + \beta h$ using gradient descent with a learning rate $\alpha > 0$. The subgradient update rule for this problem is given by

$$\mathbf{x}^{(t+1)} = \mathbf{x}^{(t)} - \alpha F(\mathbf{x}^{(t)}) = \mathbf{x}^{(t)} - \alpha \left(G(\mathbf{x}^{(t)}) + \beta H(\mathbf{x}^{(t)})\right) \ \forall t \geq 0 \tag{3}$$

where $F(\mathbf{x}) \in \partial f(\mathbf{x}) = G(\mathbf{x}) + \beta \partial h(\mathbf{x})$, with $G(\mathbf{x}) = \nabla g(\mathbf{x})$ the gradient of $g$ at $\mathbf{x}$ and $H(\mathbf{x}) \in \partial h(\mathbf{x})$ any subgradient of $h(\mathbf{x})$ at $\mathbf{x}$. We want to show that, under certain conditions on $\alpha$ and $\beta$, this procedure will first converge to a solution $\hat{\mathbf{x}}$ that minimize $g$ after $t_1 < \infty$ training steps, then take additional $\Delta t = \Theta(\frac{1}{\alpha\beta})$ training steps to converge to a solution $\mathbf{x}^*$ that minimizes both $f$ and $h$. Subdifferentiability, convexity, and smoothness are defined below for completeness.

**Definition C.1** (Subdifferentiability). A function $\varphi : \mathbb{R}^p \to \mathbb{R}$ is said to be subdifferentiable at $\mathbf{x} \in \mathbb{R}^p$ if and only if there exists $\mathbf{z} \in \mathbb{R}^p$, $\varphi(\mathbf{y}) \geq \varphi(\mathbf{x}) + (\mathbf{y} - \mathbf{x})^\top \mathbf{z} \; \forall \mathbf{y} \in \mathbb{R}^p$. The set of all such $\mathbf{z}$ at is called the subdifferential of $\varphi$ at $\mathbf{x}$ and is denoted $\partial\varphi(\mathbf{x})$. When $\partial\varphi(\mathbf{x})$ is the singleton set, we say that $\varphi$ is differentiable at $\mathbf{x} : \partial\varphi(\mathbf{x}) = \{\nabla\varphi(\mathbf{x})\}$.

**Definition C.2** (Convexity). A function $\varphi : \mathbb{R}^p \to \mathbb{R}$ is said to be convex if and only if $\varphi(t\mathbf{x} + (1 - t)\mathbf{y}) \leq t\varphi(\mathbf{x}) + (1 - t)\varphi(\mathbf{y}) \quad \forall \mathbf{x}, \mathbf{y} \in \mathbb{R}^p, \forall t \in (0, 1)$. If $\varphi$ is subdifferentiable, this implies $\varphi(\mathbf{y}) \geq \varphi(\mathbf{x}) + (\mathbf{y} - \mathbf{x})^\top \mathbf{z} \, \forall \mathbf{x}, \mathbf{y} \in \mathbb{R}^p, \forall \mathbf{z} \in \partial\varphi(\mathbf{x})$. If $\varphi$ is twice-differentiable, this implies $\lambda_{\min}(\nabla^2\varphi(\mathbf{x})) \geq 0 \, \forall \mathbf{x} \in \mathbb{R}^p$.

**Definition C.3** (Smoothness). Let $\varphi : \mathbb{R}^p \to \mathbb{R}$ be a differentiable function and $L > 0$. We say that $\varphi$ is $L$-smooth if and only if $\nabla\varphi$ is $L$-Lipschitz continuous, i.e. $\|\nabla\varphi(\mathbf{y}) - \nabla\varphi(\mathbf{x})\|_2 \leq L\|\mathbf{y} - \mathbf{x}\|_2 \, \forall \mathbf{x}, \mathbf{y} \in \mathbb{R}^p$.

**Lemma C.4.** *If a function $\varphi : \mathbb{R}^p \to \mathbb{R}$ is $L$-smooth, then*

- $\varphi(\mathbf{y}) \leq \varphi(\mathbf{x}) + (\mathbf{y} - \mathbf{x})^\top \nabla\varphi(\mathbf{x}) + \frac{L}{2}\|\mathbf{y} - \mathbf{x}\|_2^2 \, \forall \mathbf{x}, \mathbf{y} \in \mathbb{R}^p$. *The converse is false in general, and true for a convex $\varphi$.*

- $\|\nabla\varphi(\mathbf{x})\|_2^2 \leq 2L\varphi(\mathbf{x}) \, \forall \mathbf{x} \in \mathbb{R}^p$ *for a non-negative $\varphi$. The converse is false.*

- $\lambda(\nabla^2\varphi(\mathbf{x})) \in [-L, L] \, \forall \mathbf{x} \in \mathbb{R}^p$ *for a twice-differentiable $\varphi$. The converse is true.*

- $\nabla\varphi$ *is $1/L$-cocoercive for a convex $\varphi$, i.e. $(\nabla\varphi(\mathbf{y}) - \nabla\varphi(\mathbf{x}))^\top (\mathbf{x} - \mathbf{y}) \geq \frac{1}{L}\|\nabla\varphi(\mathbf{y}) - \nabla\varphi(\mathbf{x})\|_2^2 \, \forall \mathbf{x}, \mathbf{y} \in \mathbb{R}^p$ (Baillon–Haddad inequality). The converse is true.*

We use the following notations. For a non empty set $\Theta \subset \mathbb{R}^p$ and a vector $\mathbf{x} \in \mathbb{R}^p$, we let $\mathrm{dist}(\mathbf{x}, \Theta) := \inf_{\mathbf{y} \in \Theta} \|\mathbf{x} - \mathbf{y}\|_2$. We define $f^* := \inf_{\mathbf{x} \in \mathbb{R}^p} f(\mathbf{x})$ and $\Theta_f := \arg\min_{\mathbf{x} \in \mathbb{R}^p} f(\mathbf{x})$. Similarly we define $g^*$ and $\Theta_g$, $h^*$ and $\Theta_h$. We finally set $h_g^* := \inf_{\mathbf{x} \in \Theta_g} h(\mathbf{x})$, i.e. $h_g^* = \inf_{\mathbf{x} \in \mathbb{R}^p} h(\mathbf{x})$ s.t. $\mathbf{x} \in \Theta_g$, and $\Theta_h^g := \arg\min_{\mathbf{x} \in \Theta_g} h(\mathbf{x})$.

In general, $f^* \neq g^* + \beta h^*$. In fact, we have $f^* \geq g^* + \beta h^*$. This implies

$$
\begin{aligned}
f^* \geq g^* + \beta h^* &\Longleftrightarrow -f^* \leq -g^* - \beta h^* \\
&\Longrightarrow 0 \leq f(\mathbf{x}) - f^* \leq f(\mathbf{x}) - g^* - \beta h^* = g(\mathbf{x}) - g^* + \beta(h(\mathbf{x}) - h^*) \\
&\Longrightarrow |f(\mathbf{x}) - f^*| \leq |g(\mathbf{x}) - g^* + \beta(h(\mathbf{x}) - h^*)| \leq |g(\mathbf{x}) - g^*| + \beta|h(\mathbf{x}) - h^*|
\end{aligned}
\tag{4}
$$

So $g(\mathbf{x}) \to g^*$ and $h(\mathbf{x}) \to h^*$ implies $f(\mathbf{x}) \to f^*$. But in general, it will not be possible to jointly have $g(\mathbf{x}^{(t)}) \to g^*$ and $h(\mathbf{x}^{(t)}) \to h^*$ by gradient descent. We will show that we first have $g(\mathbf{x}^{(t)}) \to g^*$, then $h(\mathbf{x}^{(t)}) \to h_g^*$.

**Assumption C.5.** $\Theta_f \cap \Theta_g \neq \emptyset$.

This assumption assumes too little noise compared to the signal. Even if initially $g(\mathbf{x}^{(t)}) \to g^*$ (i.e., memorization of training data with associated noise), as soon as generalization occurs, $|g(\mathbf{x}^{(t)}) - g^*|$ becomes proportional to the noise in the data. But under this assumption, when memorization is achieved, we can not have $f(\mathbf{x}^{(t)}) \to f^*$ without $h(\mathbf{x}^{(t)}) \to h_g^*$.

**Lemma C.6.** *If assumption C.5 holds, then $f^* = g^* + \beta h_g^*$; and hence $\beta(h(\mathbf{x}) - h_g^*) \leq g(\mathbf{x}) - g^* + \beta(h(\mathbf{x}) - h_g^*) \leq f(\mathbf{x}) - f^* \quad \forall \mathbf{x} \in \mathbb{R}^p$.*

*Proof.* Assume $\Theta_f \cap \Theta_g \neq \emptyset$. Then there exists $\mathbf{x}_f^* \in \Theta_f$ such that $g(\mathbf{x}_f^*) = g^*$. This implies $\mathbf{x}_f^* \in \Theta_g$. Under this assumption, $f^* = g(\mathbf{x}_f^*) + \beta h(\mathbf{x}_f^*) = g^* + \beta h(\mathbf{x}_f^*)$. Since $\mathbf{x}_f^* \in \Theta_g$, $h(\mathbf{x}_f^*) \geq h_g^*$, so $f^* \geq g^* + \beta h_g^*$. Conversely, we have $g^* + \beta h_g^* = g(\mathbf{x}) + \beta h(\mathbf{x}) = f(\mathbf{x}) \geq f^*$ for any $\mathbf{x} \in \Theta_h^g \subset \Theta_g$. Therefore, $f^* = g^* + \beta h_g^*$. This implies $f(\mathbf{x}) - f^* = g(\mathbf{x}) - g^* + \beta(h(\mathbf{x}) - h_g^*) \geq \beta(h(\mathbf{x}) - h_g^*)$ since $g(\mathbf{x}) - g^* \geq 0$. $\qquad\square$

**Assumption C.7.** Since $\Theta_g = \arg\min_{\mathbf{x} \in \mathbb{R}^p}(g(\mathbf{x}) - g^*)$, we assume without loss of generality $g^* = 0$.

If $g$ is the loss function of an overparameterized neural network, then for $\beta = 0$, $f = g$ converges to a solution $\mathbf{x}^*$ close to the initialization $\mathbf{x}^{(0)}$ such that $g(\mathbf{x}^*) = 0$, under certain conditions on $\alpha$ and $g$ (see, for example, Theorems C.25 and C.15). We will extend this result to the case $\beta > 0$.

**Assumption C.8.** We assume there exists $\mathbf{x} \in \mathbb{R}^p$ such that $g(\mathbf{x}) = g^*$.

We define the Chatterjee–Łojasiewicz (CL) constant of a subdifferentiable function $\varphi : \mathbb{R}^p \to [0, \infty)$ at $\mathbf{x} \in \mathbb{R}^p$ with radius $r > 0$, and with respect to another function $\phi : \mathbb{R}^p \to [0, \infty)$ as:

$$\chi(\varphi, \mathbf{x}, r, \phi) := \begin{cases} \infty & \text{if } \phi(\mathbf{y}) = 0 \; \forall \mathbf{y} \in B(\mathbf{x}, r) \\ \inf_{\substack{\mathbf{y} \in B(\mathbf{x}, r) \\ \phi(\mathbf{y}) \neq 0}} \frac{\inf_{\mathbf{z} \in \partial \varphi(\mathbf{y})} \|\mathbf{z}\|_2^2}{\phi(\mathbf{y})} & \text{otherwise.} \end{cases}$$

$$= \begin{cases} \infty & \text{if } \phi(\mathbf{y}) = 0 \; \forall \mathbf{y} \in B(\mathbf{x}, r) \\ \inf_{\substack{\mathbf{y} \in B(\mathbf{x}, r) \\ \phi(\mathbf{y}) \neq 0}} \frac{\|\nabla \varphi(\mathbf{y})\|_2^2}{\phi(\mathbf{y})} & \text{otherwise.} \end{cases} \quad \text{if } \varphi \text{ is differentiable} \tag{5}$$

and

$$\chi(\varphi, \mathbf{x}, r) := \chi(\varphi, \mathbf{x}, r, \varphi) \tag{6}$$

where $B(\mathbf{x}, r) := \{\mathbf{y} \in \mathbb{R}^p \mid \|\mathbf{x} - \mathbf{y}\|_2 \leq r\}$ denote the closed Euclidean ball of radius $r$ centered at $\mathbf{x}$.

**Definition C.9** (Chatterjee–Łojasiewicz inequality). For $r > 0$, a nonnegative subdifferentiable function $\varphi : \mathbb{R}^p \to [0, \infty)$ is said to satisfy the $r$-CL inequality at $\mathbf{x} \in \mathbb{R}^p$ if and only if $4\varphi(\mathbf{x}) < r^2 \chi(\varphi, \mathbf{x}, r)$.

**Theorem C.10.** *Take any $\mathbf{x}^{(0)} \in \mathbb{R}^p$ with $g^{(0)} := g(\mathbf{x}^{(0)}) \neq 0$. Assume that $g$ satisfy the $r$-CL inequality at $\mathbf{x}^{(0)}$ for some $r > 0$, i.e. $4g^{(0)} < r^2 \chi(g, \mathbf{x}^{(0)}, r)$. There exist $\beta_{\max} > 0$, $\alpha_{\max} > 0$ and three constants $C, C', C'' > 0$ such that for all $\alpha \in (0, \alpha_{\max})$ and $\beta \in (0, \beta_{\max})$, by defining the subgradient descent update (3) with $\alpha$ and $\beta$ starting at $\mathbf{x}^{(0)}$, the following hold:*

- *Fo any $\epsilon_g = \Omega(\beta^C)$, there exists a step $t_1 \geq \max\left\{0, -\log\left(\epsilon_g / g^{(0)}\right) / \log\left(1 - \Theta(\chi(g, \mathbf{x}^{(0)}, r) \cdot \alpha)\right)\right\}$ such that $g(\mathbf{x}^{(t_1)}) \leq \epsilon_g$, $\|\nabla g(\mathbf{x}^{(t_1)})\|_2^2 \leq C'' g(\mathbf{x}^{(t_1)}) \leq C'' \epsilon_g$, and $\{\mathbf{x}^{(t)}\}_{0 \leq t \leq t_1} \subset B(\mathbf{x}^{(0)}, r)$.*

- *For any $\eta > 0$, $\min_{t_1 \leq t \leq t_2}\left(f(\mathbf{x}^{(t)}) - f^*\right) \leq \frac{(\eta + C' \alpha \beta)\beta}{2}$ if and only if $t_2 > t_1 + \Delta t(\eta, t_1)$, with $\Delta t(\eta, t_1) := \frac{\text{dist}^2(\mathbf{x}^{(t_1)}, \Theta_f)}{\alpha \beta \eta}$. Moreover, assuming Assumptions C.5 holds, we have $f(\mathbf{x}^{(t)}) - f^* = g(\mathbf{x}^{(t)}) - g^* + \beta(h(\mathbf{x}^{(t)}) - h_g^*)$ and so $\min_{t_1 \leq t \leq t_2}\left(h(\mathbf{x}^{(t)}) - h_g^*\right) = \frac{\eta + C'\alpha\beta}{2}$ if and only if $t_2 > t_1 + \Delta t(\eta, t_1)$.*

*Proof.* Set $\chi := \chi(g, \mathbf{x}^{(0)}, r)$. Since $4g^{(0)} < r^2 \chi$, there exist $\epsilon \in (0, 1)$ and $\gamma \in (0, \epsilon)$ such that $4g^{(0)} < \left(\frac{1-\epsilon}{1+\gamma}\right)^2 r^2 \chi$ (see Lemma C.11). Let $L_1 := \sup_{\mathbf{x} \in B(\mathbf{x}^{(0)}, r)} \|\nabla g(\mathbf{x})\|_\infty < \infty$, $L_2 := \sup_{\mathbf{x} \in B(\mathbf{x}^{(0)}, 2r)} \|\text{vec}\left(\nabla^2 g(\mathbf{x})\right)\|_\infty < \infty$, $L := \sqrt{p} L_2$, and $L_h := \sup_{\mathbf{x} \in B(\mathbf{x}^{(0)}, r)} \sup_{H \in \partial h(\mathbf{x})} \|H\|_\infty < \infty$. We choose any $\beta \in (0, \beta_{\max})$ with $\beta_{\max} > 0$ and $\beta_{\max} \sup_{H \in \partial h(\mathbf{x}^{(0)})} \|H\|_2 \leq \gamma \|G(\mathbf{x}^{(0)})\|_2$, and any step size $\alpha \in (0, \alpha_{\max})$ such that

$$\alpha_{\max} = \min\left\{\frac{r}{(L_1 + \beta_{\max} L_h)\sqrt{p}}, \frac{2(\epsilon - \gamma)}{L_2 p}, \frac{1}{(1 + \gamma)L}\right\} \tag{7}$$

Let $\delta = (1 - \epsilon) \cdot \chi \cdot \alpha \in (0, 1)$ (Lemma C.20) and $\tau = 1 - (1 + \gamma)\alpha L \in (0, 1)$. From Theorem C.14, when $\gamma \|\nabla g(\mathbf{x}^{(0)})\|_2 / \beta L_h \geq \tau^{-t}$, we have $\mathbf{x}^{(k)} \subset B(\mathbf{x}^{(0)}, r)$, $g(\mathbf{x}^{(k)}) \leq (1 - \delta)^k g^{(0)}$ and $\|\nabla g(\mathbf{x}^{(k)})\|_2^2 \leq 2Lg(\mathbf{x}^{(k)}) \leq 2L(1 - \delta)^k g^{(0)}$ for all $0 \leq k \leq t$. Setting $(1 - \delta)^t g^{(0)} \leq \epsilon_g$ gives

$$t := t(\epsilon_g) \geq \max\left\{0, \frac{\log(\epsilon_g / g^{(0)})}{\log(1 - \delta)}\right\}$$

$$= \begin{cases} 0 & \text{if } \epsilon_g > g^{(0)} \\ \frac{\log(\epsilon_g / g^{(0)})}{\log\left(1 - (1-\epsilon)\cdot \chi(g, \mathbf{x}^{(0)}, r) \cdot \alpha\right)} \approx \frac{-\log(\epsilon_g / g^{(0)})}{(1-\epsilon)\cdot\chi(g, \mathbf{x}^{(0)}, r)\cdot\alpha} = \frac{-1}{\Theta(\alpha \cdot \chi(g, \mathbf{x}^{(0)}, r))} \log\left(\frac{\epsilon_g}{g^{(0)}}\right) & \text{otherwise} \end{cases} \tag{8}$$

From $\gamma \|\nabla g(\mathbf{x}^{(0)})\|_2 / \beta L_h \geq \tau^{-t}$, we have $t \leq \log(\beta L_h / \|\nabla g(\mathbf{x}^{(0)})\|_2) / \log(\tau)$. So for $t(\epsilon_g)$ to be well define, we need $\frac{\log(\epsilon_g / g^{(0)})}{\log(1 - \delta)} \leq \frac{\log(\beta L_h / \|\nabla g(\mathbf{x}^{(0)})\|_2)}{\log(\tau)}$ when $\epsilon_g \leq g^{(0)}$. This is equivalent to $\epsilon_g > \zeta g^{(0)} = D\beta^C$, with

$$\zeta = \left(\frac{\beta L_h}{\gamma \|\nabla g(\mathbf{x}^{(0)})\|_2}\right)^{\frac{\log(1-\delta)}{\log \tau}} \in (0, 1) \text{ since } \beta L_h < \gamma \|\nabla g(\mathbf{x}^{(0)})\|_2$$

$$C = \frac{\log(1 - \delta)}{\log \tau} > 0 \text{ since } 1 - \delta \in (0, 1) \text{ and } \tau \in (0, 1) \tag{9}$$

$$D = g^{(0)} \left(\frac{L_h}{\gamma \|\nabla g(\mathbf{x}^{(0)})\|_2}\right)^C$$

Let $F(\mathbf{x}) \in \partial f(\mathbf{x}) = \nabla g(\mathbf{x}) + \beta \partial h(\mathbf{x})$. From Lemma C.12, we have

$$\min_{t_1 \leq t \leq t_2} \left( f(\mathbf{x}^{(t)}) - f^* \right) \leq \frac{\mathrm{dist}^2(\mathbf{x}^{(t_1)}, \Theta_f) + (t_2 - t_1)\alpha^2 \max_{t_1 \leq t \leq t_2} \|F(\mathbf{x}^{(t)})\|_2^2}{2(t_2 - t_1)\alpha} \xrightarrow[t_2 \to \infty]{} \frac{\alpha}{2} \max_{t_1 \leq t} \|F(\mathbf{x}^{(t)})\|_2^2 \quad (10)$$

Since $\|F(\mathbf{x}^{(t)})\|_2^2 = \mathcal{O}(\beta^2)$ for all $t \geq t_1$ when $\|G(\mathbf{x}^{(t)})\|_2 \ll \beta H(\mathbf{x}^{(t)})$ (the second phase of training is driven by $\beta H(\mathbf{x}^{(t)})$, see Theorem C.16), we obtain from Theorem C.13 that there exists $C' > 0$,

$$\min_{t_1 \leq t \leq t_2} \left( f(\mathbf{x}^{(t)}) - f^* \right) \leq \frac{(\eta + C'\alpha\beta)\,\beta}{2} \iff t_2 \geq t_1 + \frac{\mathrm{dist}^2(\mathbf{x}^{(t_1)}, \Theta_f)}{\alpha\beta\eta} \quad (11)$$

And if $\Theta_f \cap \Theta_g \neq \emptyset$,

$$\min_{t_1 \leq t \leq t_2} \left( h(\mathbf{x}^{(t)}) - h_g^* \right) \leq \frac{\eta + C'\alpha\beta}{2} \iff t_2 \geq t_1 + \frac{\mathrm{dist}^2(\mathbf{x}^{(t_1)}, \Theta_f)}{\alpha\beta\eta} \quad (12)$$

$\square$

**Lemma C.11.** *For all $a, b \in \mathbb{R}$ with $0 < a < b$. (1) There exists $\epsilon \in (0, 1)$ such that $a < (1 - \epsilon)^2 b$. (2) There exist $\epsilon \in (0, 1)$ and $\gamma \in (0, \epsilon)$ such that $a < \left( \frac{1-\epsilon}{1+\gamma} \right)^2 b$. (3) There exist $\epsilon \in (0, 1)$ and $\gamma \in (0, 1 - \epsilon)$ such that $a < \left( \frac{1-(\epsilon+\gamma)}{1+\gamma} \right)^2 b$.*

*Proof.* Set $\rho = \frac{a}{b} \in (0, 1)$ and $\tau = \sqrt{\rho} \in (0, 1)$. (1) Take any $\epsilon \in (0, 1 - \tau) \iff \rho < (1 - \epsilon)^2 < 1$. (2) Pick $\gamma = \frac{1-\tau}{2\tau} \in \left( 0, \frac{1-\tau}{\tau} \right)$ and $\epsilon \in \left( 0, \frac{1+\tau}{2} \right)$. (3) Pick $\gamma = \frac{1-\tau}{2(1+\tau)} \in \left( 0, \frac{1-\tau}{1+\tau} \right)$ and $\epsilon \in \left( 0, \frac{1-\tau}{2} \right)$. $\square$

**Lemma C.12.** *For all $t_1$ and $t_2$ with $t_1 \leq t_2$, we have*

$$\min_{t_1 \leq t \leq t_2} \left( f(\mathbf{x}^{(t)}) - f^* \right) \leq \frac{\mathrm{dist}^2(\mathbf{x}^{(t_1)}, \Theta_f) + (t_2 - t_1)\alpha^2 \max_{t_1 \leq t \leq t_2} \|F(\mathbf{x}^{(t)})\|_2^2}{2(t_2 - t_1)\alpha} \xrightarrow[t_2 \to \infty]{} \alpha \max_{t_1 \leq t} \|F(\mathbf{x}^{(t)})\|_2^2 \quad (13)$$

*and*

$$\min_{t_1 \leq t \leq t_2} \left[ \sup_{\mathbf{x} \in \Theta_g} \left( \mathbf{x}^{(t)} - \mathbf{x} \right)^\top G(\mathbf{x}^{(t)}) + \beta \left( h(\mathbf{x}^{(t)}) - h_g^* \right) \right] \leq \min_{t_1 \leq t \leq t_2} \left[ \left( g(\mathbf{x}^{(t)}) - g^* \right) + \beta \left( h(\mathbf{x}^{(t)}) - h_g^* \right) \right]$$

$$\leq \frac{\mathrm{dist}^2(\mathbf{x}^{(t_1)}, \Theta_g) + (t_2 - t_1)\alpha^2 \max_{t_1 \leq t \leq t_2} \|F(\mathbf{x}^{(t)})\|_2^2}{2(t_2 - t_1)\alpha} \quad (14)$$

*Moreover, if Assumptions C.5 holds, then*

$$\beta \min_{t_1 \leq t \leq t_2} \left( h(\mathbf{x}^{(t)}) - h_g^* \right) \leq \min_{t_1 \leq t \leq t_2} \left( f(\mathbf{x}^{(t)}) - f^* \right) \leq \frac{\mathrm{dist}^2(\mathbf{x}^{(t_1)}, \Theta_f) + (t_2 - t_1)\alpha^2 \max_{t_1 \leq t \leq t_2} \|F(\mathbf{x}^{(t)})\|_2^2}{2(t_2 - t_1)\alpha} \quad (15)$$

*Proof.* By the definition of the subgradient $F(\mathbf{x}^{(t_2)})$ of $f$ at $\mathbf{x}^{(t_2)}$, we have $\left( \mathbf{x} - \mathbf{x}^{(t_2)} \right)^\top F(\mathbf{x}^{(t_2)}) \leq f(\mathbf{x}) - f(\mathbf{x}^{(t_2)})$ $\forall \mathbf{x} \iff - \left( \mathbf{x}^{(t_2)} - \mathbf{x} \right)^\top F(\mathbf{x}^{(t_2)}) \leq -(f(\mathbf{x}^{(t_2)}) - f(\mathbf{x}))$ $\forall \mathbf{x}$. So, for all $\mathbf{x}$, we have

$$\begin{aligned}
\|\mathbf{x}^{(t_2+1)} - \mathbf{x}\|_2^2 &= \|\mathbf{x}^{(t_2)} - \alpha F(\mathbf{x}^{(t_2)}) - \mathbf{x}\|_2^2 \\
&= \|\mathbf{x}^{(t_2)} - \mathbf{x}\|_2^2 - 2\alpha \left( \mathbf{x}^{(t_2)} - \mathbf{x} \right)^\top F(\mathbf{x}^{(t_2)}) + \alpha^2 \|F(\mathbf{x}^{(t_2)})\|_2^2 \\
&\leq \|\mathbf{x}^{(t_2)} - \mathbf{x}\|_2^2 - 2\alpha \left( f(\mathbf{x}^{(t_2)}) - f(\mathbf{x}) \right) + \alpha^2 \|F(\mathbf{x}^{(t_2)})\|_2^2 \\
&\leq \|\mathbf{x}^{(t_1)} - \mathbf{x}\|_2^2 - 2\alpha \sum_{t=t_1}^{t_2} \left( f(\mathbf{x}^{(t)}) - f(\mathbf{x}) \right) + \alpha^2 \sum_{t=t_1}^{t_2} \|F(\mathbf{x}^{(t)})\|_2^2
\end{aligned} \quad (16)$$

This implies

$$0 \leq \inf_{\mathbf{x} \in \Theta_f} \|\mathbf{x}^{(t_2+1)} - \mathbf{x}\|_2^2 \leq \inf_{\mathbf{x} \in \Theta_f} \|\mathbf{x}^{(t_1)} - \mathbf{x}\|_2^2 - 2\alpha \sup_{\mathbf{x} \in \Theta_f} \sum_{t=t_1}^{t_2} \left( f(\mathbf{x}^{(t)}) - f(\mathbf{x}) \right) + \alpha^2 \sum_{t=t_1}^{t_2} \|F(\mathbf{x}^{(t)})\|_2^2$$

$$\implies 2\alpha \sup_{\mathbf{x} \in \Theta_f} \sum_{t=t_1}^{t_2} \left( f(\mathbf{x}^{(t)}) - f(\mathbf{x}) \right) \leq \mathrm{dist}^2(\mathbf{x}^{(t_1)}, \Theta_f) + \alpha^2 \sum_{t=t_1}^{t_2} \|F(\mathbf{x}^{(t)})\|_2^2 \tag{17}$$

$$\implies 2\alpha(t_2 - t_1) \sup_{\mathbf{x} \in \Theta_f} \min_{t_1 \leq t \leq t_2} \left( f(\mathbf{x}^{(t)}) - f(\mathbf{x}) \right) \leq \mathrm{dist}^2(\mathbf{x}^{(t_1)}, \Theta_f) + (t_2 - t_1)\alpha^2 \max_{t_1 \leq t \leq t_2} \|F(\mathbf{x}^{(t)})\|_2^2$$

$$\iff \min_{t_1 \leq t \leq t_2} \left( f(\mathbf{x}^{(t)}) - \inf_{\mathbf{x} \in \Theta_f} f(\mathbf{x}) \right) \leq \frac{\mathrm{dist}^2(\mathbf{x}^{(t_1)}, \Theta_f) + (t_2 - t_1)\alpha^2 \max_{t_1 \leq t \leq t_2} \|F(\mathbf{x}^{(t)})\|_2^2}{2(t_2 - t_1)\alpha}$$

This proves the first inequality. We also have, for all $\mathbf{x}$, $-\left(\mathbf{x}^{(t_2)} - \mathbf{x}\right)^\top H(\mathbf{x}^{(t_2)}) \leq -(h(\mathbf{x}^{(t_2)}) - h(\mathbf{x}))$ and $-\left(\mathbf{x}^{(t_2)} - \mathbf{x}\right)^\top G(\mathbf{x}^{(t_2)}) \leq -(g(\mathbf{x}^{(t_2)}) - g(\mathbf{x}))$. So

$$\|\mathbf{x}^{(t_2+1)} - \mathbf{x}\|_2^2 = \|\mathbf{x}^{(t_2)} - \alpha F(\mathbf{x}^{(t_2)}) - \mathbf{x}\|_2^2$$

$$= \|\mathbf{x}^{(t_2)} - \mathbf{x}\|_2^2 - 2\alpha \left(\mathbf{x}^{(t_2)} - \mathbf{x}\right)^\top F(\mathbf{x}^{(t_2)}) + \alpha^2 \|F(\mathbf{x}^{(t_2)})\|_2^2$$

$$= \|\mathbf{x}^{(t_2)} - \mathbf{x}\|_2^2 - 2\alpha \left(\mathbf{x}^{(t_2)} - \mathbf{x}\right)^\top G(\mathbf{x}^{(t_2)}) - 2\alpha\beta \left(\mathbf{x}^{(t_2)} - \mathbf{x}\right)^\top H(\mathbf{x}^{(t_2)}) + \alpha^2 \|F(\mathbf{x}^{(t_2)})\|_2^2 \tag{18}$$

$$\leq \|\mathbf{x}^{(t_2)} - \mathbf{x}\|_2^2 - 2\alpha \left(g(\mathbf{x}^{(t_2)}) - g(\mathbf{x})\right) - 2\alpha\beta \left(h(\mathbf{x}^{(t_2)}) - h(\mathbf{x})\right) + \alpha^2 \|F(\mathbf{x}^{(t_2)})\|_2^2$$

$$\leq \|\mathbf{x}^{(t_1)} - \mathbf{x}\|_2^2 - 2\alpha \sum_{t=t_1}^{t_2} \left(g(\mathbf{x}^{(t)}) - g(\mathbf{x})\right) - 2\alpha\beta \sum_{t=t_1}^{t_2} \left(h(\mathbf{x}^{(t)}) - h(\mathbf{x})\right) + \alpha^2 \sum_{t=t_1}^{t_2} \|F(\mathbf{x}^{(t)})\|_2^2$$

This implies

$$2\alpha \left[ \sup_{\mathbf{x} \in \Theta_g} \sum_{t=t_1}^{t_2} \left(g(\mathbf{x}^{(t)}) - g(\mathbf{x})\right) + \beta \sup_{\mathbf{x} \in \Theta_g} \sum_{t=t_1}^{t_2} \left(h(\mathbf{x}^{(t)}) - h(\mathbf{x})\right) \right] \leq \mathrm{dist}^2(\mathbf{x}^{(t_1)}, \Theta_g) + \alpha^2 \sum_{t=t_1}^{t_2} \|F(\mathbf{x}^{(t)})\|_2^2$$

$$\iff \min_{t_1 \leq t \leq t_2} \left[ \left(g(\mathbf{x}^{(t)}) - g^*\right) + \beta \left(h(\mathbf{x}^{(t)}) - h_g^*\right) \right] \leq \frac{\mathrm{dist}^2(\mathbf{x}^{(t_1)}, \Theta_g) + (t_2 - t_1)\alpha^2 \max_{t_1 \leq t \leq t_2} \|F(\mathbf{x}^{(t)})\|_2^2}{2(t_2 - t_1)\alpha} \tag{19}$$

If $\Theta_f \cap \Theta_g$, then $f(\mathbf{x}) - f^* \geq \beta(h(\mathbf{x}) - h_g^*)$ (Lemma C.6), and so

$$\beta \min_{t_1 \leq t \leq t_2} \left(h(\mathbf{x}^{(t)}) - h_g^*\right) \leq \min_{t_1 \leq t \leq t_2} \left(f(\mathbf{x}^{(t)}) - f^*\right) = \min_{t_1 \leq t \leq t_2} \left(f(\mathbf{x}^{(t)}) - \inf_{\mathbf{x} \in \Theta_f} f(\mathbf{x})\right) \tag{20}$$

$\square$

**Theorem C.13.** *Let* $t_1 > 0$. *Define* $R := \mathrm{dist}(\mathbf{x}^{(t_1)}, \Theta_f)$ *and* $L := \max_{t_1 \leq t} \|F(\mathbf{x}^{(t)})\|_2^2$. *Assume there exists a constant* $C > 0$, $L \leq C\beta^2$. *Then, for any* $\eta > 0$, $\min_{t_1 \leq t \leq t_2} \left(f(\mathbf{x}^{(t)}) - f^*\right) \leq \frac{(\eta + C\alpha\beta)\beta}{2}$ *if and only if* $t_2 \geq t_1 + \frac{R^2}{\alpha\beta\eta}$. *Moreover, if Assumptions C.5 holds, then* $\min_{t_1 \leq t \leq t_2} \left(h(\mathbf{x}^{(t)}) - h_g^*\right) \leq \frac{\eta + C\alpha\beta}{2}$ *if and only if* $t_2 \geq t_1 + \frac{R^2}{\alpha\beta\eta}$.

*Proof.* Using $\max_{t_1 \leq t \leq t_2} \|F(\mathbf{x}^{(t)})\|_2^2 \leq \max_{t_1 \leq t} \|F(\mathbf{x}^{(t)})\|_2^2 = L \leq C\beta^2$, we derive the following from Lemma C.12 :

$$\min_{t_1 \leq t \leq t_2} \left(f(\mathbf{x}^{(t)}) - f^*\right) \leq \frac{R^2 + C(t_2 - t_1)\alpha^2\beta^2}{2(t_2 - t_1)\alpha} \leq \frac{(\eta + C\alpha\beta)\beta}{2} \iff \frac{R^2}{2\alpha(t_2 - t_1)} \leq \frac{\eta\beta}{2} \iff t_2 - t_1 \geq \frac{R^2}{\alpha\beta\eta} \tag{21}$$

and

$$\min_{t_1 \leq t \leq t_2} \left(h(\mathbf{x}^{(t)}) - h_g^*\right) \leq \frac{1}{\beta} \frac{R^2 + C(t_2 - t_1)\alpha^2\beta^2}{2(t_2 - t_1)\alpha} \leq \frac{\eta + C\alpha\beta}{2} \iff \frac{R^2}{2\beta\alpha(t_2 - t_1)} \leq \frac{\eta}{2} \iff t_2 - t_1 \geq \frac{R^2}{\alpha\beta\eta} \tag{22}$$

$\square$

**Theorem C.14** (Time of Memorization of fixed-step gradient descent under CL condition and a subdifferentiable regularization). *Let $g : \mathbb{R}^p \to [0, \infty)$ be a nonnegative $C^2$ function and $h : \mathbb{R}^p \to [0, \infty)$ be a nonnegative subdifferentiable function. Take any $\mathbf{x}^{(0)} \in \mathbb{R}^p$ with $g(\mathbf{x}^{(0)}) \neq 0$. Assume that $g$ is $r$-CL at $\mathbf{x}^{(0)} \in \mathbb{R}^p$ for some $r > 0$. Choose $\epsilon \in (0, 1)$ and $\gamma \in (0, \epsilon)$ such that $4g(\mathbf{x}^{(0)}) < \left(\frac{1-\epsilon}{1+\gamma}\right)^2 r^2 \chi(g, \mathbf{x}^{(0)}, r)$, which is possible since $4g(\mathbf{x}^{(0)}) < r^2 \chi(g, \mathbf{x}^{(0)}, r)$ (Lemma C.11). Let $L_1 := \sup_{\mathbf{x} \in B(\mathbf{x}^{(0)}, r)} \|\nabla g(\mathbf{x})\|_\infty < \infty$, $L_2 := \sup_{\mathbf{x} \in B(\mathbf{x}^{(0)}, 2r)} \| \operatorname{vec}\left(\nabla^2 g(\mathbf{x})\right) \|_\infty < \infty$ and $L_h := \sup_{\mathbf{x} \in B(\mathbf{x}^{(0)}, r)} \sup_{H \in \partial h(\mathbf{x})} \|H\|_\infty < \infty$. Define $G(\mathbf{x}) := \nabla g(\mathbf{x})$ and choose $\beta \leq \beta_{\max}$ with $\beta_{\max} > 0$ and $\beta_{\max} \sup_{H \in \partial h(\mathbf{x}^{(0)})} \|H\|_2 \leq \gamma \|G(\mathbf{x}^{(0)})\|_2$. Choose any step size $\alpha > 0$ such that*

$$\alpha < \min\left\{\frac{r}{(L_1 + \beta_{\max} L_h)\sqrt{p}}, \frac{2(\epsilon - \gamma)}{L_2 p}, \frac{1}{(1+\gamma)L}\right\} \text{ with } L = \sqrt{p}L_2 \tag{23}$$

*Set $\chi := \chi(g, \mathbf{x}^{(0)}, r)$ and define $\delta := \min\{1, (1 - \epsilon)\chi\alpha\}$. Iteratively define $\mathbf{x}^{(k+1)} = \mathbf{x}^{(k)} - \alpha\left(G(\mathbf{x}^{(k)}) + \beta H(\mathbf{x}^{(k)})\right) \forall H(\mathbf{x}^{(k)}) \in \partial h(\mathbf{x}^{(k)})$ for each $k \geq 0$. Let $\tau := 1 - (1 + \gamma)\alpha L \in (0, 1)$ and assume $\gamma\|G(\mathbf{x}^{(0)})\|_2 \geq \frac{\beta L_h}{\tau^k}$ for some $k > 0$. Then*

$$\gamma\|G(\mathbf{x}^{(j)})\|_2 \geq \frac{\beta L_h}{\tau^{k-j}} \text{ and } \beta \sup_{H \in \partial h(\mathbf{x}^{(j)})} \|H\|_2 \leq \gamma\|G(\mathbf{x}^{(j)})\|_2 \quad \forall\, 0 \leq j \leq k \tag{24}$$

*and*

$$\|G(\mathbf{x}^{(k+1)})\|_2 \geq \tau\beta L_h \text{ with } \tau\beta L_h < \beta L_h \tag{25}$$

*As a consequence,*

- $\mathbf{x}^{(1)}, \ldots, \mathbf{x}^{(k)} \in B(\mathbf{x}^{(0)}, r)$

- $g(\mathbf{x}^{(j)}) \leq (1 - \delta)^j g(\mathbf{x}^{(0)})$ *and* $\|\nabla g(\mathbf{x}^{(j)})\|_2^2 \leq 2Lg(\mathbf{x}^{(j)}) \leq 2L(1-\delta)^j g(\mathbf{x}^{(0)})$ *for all* $0 \leq j \leq k$

- $\|\mathbf{x}^{(k)} - \mathbf{x}^{(j)}\|_2^2 \leq (1 - \delta)^j r^2$ *for all* $0 \leq j < k$

*Proof.* $\gamma\|G(\mathbf{x}^{(0)})\|_2 \geq \frac{\beta L_h}{\tau^k} \geq \beta L_h \geq \beta H(\mathbf{x}^{(0)})$ since $\mathbf{x}^{(0)} \in B(\mathbf{x}^{(0)}, r)$. So $\mathbf{x}^{(1)} \in B(\mathbf{x}^{(0)}, r)$ (Lemma C.17), and hence $\sup_{H \in \partial h(\mathbf{x}^{(1)})} \|H\|_2 \leq L_h$. Since $g$ is $C^2$, $\nabla^2 g(\mathbf{x})$ is symmetric for all $\mathbf{x} \in \mathbb{R}^p$, and we thus have $\sigma_{\max}(\nabla^2 g(\mathbf{x})) \leq \sqrt{p}\max_{ij}\left|\left[\nabla^2 g(\mathbf{x})\right]_{ij}\right| \leq \sqrt{p}L_2 = L$ for all $\mathbf{x} \in B(\mathbf{x}^{(0)}, r) \subset B(\mathbf{x}^{(0)}, 2r)$. So $g$ is $L$-smooth on $B(\mathbf{x}^{(0)}, r)$ by Lemma C.4, i.e.

$$\begin{aligned}
\|G(\mathbf{x}^{(1)}) - G(\mathbf{x}^{(0)})\|_2 &\leq L\|\mathbf{x}^{(1)} - \mathbf{x}^{(0)}\|_2 \\
&= \alpha L\|G(\mathbf{x}^{(0)}) + \beta H(\mathbf{x}^{(0)})\|_2 \\
&\leq \alpha L\left(\|G(\mathbf{x}^{(0)})\|_2 + \beta\|H(\mathbf{x}^{(0)})\|_2\right) \text{ (Triangle inequality)} \\
&\leq (1 + \gamma)\alpha L\|G(\mathbf{x}^{(0)})\|_2 \text{ since } \beta\|H(\mathbf{x}^{(0)})\|_2 \leq \gamma\|G(\mathbf{x}^{(0)})\|_2
\end{aligned} \tag{26}$$

So

$$\begin{aligned}
\|G(\mathbf{x}^{(1)})\|_2 &\geq \|G(\mathbf{x}^{(0)})\|_2 - \|G(\mathbf{x}^{(1)}) - G(\mathbf{x}^{(0)})\|_2 \text{ (Triangle inequality)} \\
&\geq \|G(\mathbf{x}^{(0)})\|_2 - (1 + \gamma)\alpha L\|G(\mathbf{x}^{(0)})\|_2 \text{ (Equation (26))} \\
&= \tau\|G(\mathbf{x}^{(0)})\|_2 \text{ since } \tau = 1 - (1 + \gamma)\alpha L \\
&\geq \frac{\beta L_h}{\gamma\tau^{k-1}} \text{ since } \gamma\|G(\mathbf{x}^{(0)})\|_2 \geq \frac{\beta L_h}{\tau^k}
\end{aligned} \tag{27}$$

We thus have $\gamma\|G(\mathbf{x}^{(1)})\|_2 \geq \beta L_h \geq \beta H(\mathbf{x}^{(1)})$. So $\mathbf{x}^{(2)} \in B(\mathbf{x}^{(0)}, r)$ (Lemma C.17), and hence $\sup_{H \in \partial h(\mathbf{x}^{(2)})} \|H\|_2 \leq L_h$. And so on, we prove that $\|G(\mathbf{x}^{(j)})\|_2 \geq \frac{\beta L_h}{\gamma\tau^{k-j}}$ and $\beta \sup_{H \in \partial h(\mathbf{x}^{(j)})} \|H\|_2 \leq \gamma\|G(\mathbf{x}^{(j)})\|_2 \,\forall\, j \leq k$. We have $\|G(\mathbf{x}^{(k+1)})\|_2 \geq \tau\beta L_h$ with $\tau\beta L_h < \beta L_h$. So either $\|G(\mathbf{x}^{(k+1)})\|_2 \geq \beta L_h$ (and we can proceed to the next iteration), or $\beta L_h \geq \|G(\mathbf{x}^{(k+1)})\|_2 \geq \tau\beta L_h$ (we do not have control after $k$). The last part of the Theorem follows from Theorem C.16. $\square$

The following Theorem is an extension of the result obtained by Chatterjee (2022), where we add an additional bound on the gradient. We extend it below to handle the case $\beta \neq 0$ in Theorem C.16.

**Theorem C.15** (Convergence of fixed-step gradient descent under CL condition). *Let $g : \mathbb{R}^p \to [0, \infty)$ be a nonnegative $C^2$ function. Take any $\mathbf{x}^{(0)} \in \mathbb{R}^p$ with $g(\mathbf{x}^{(0)}) \neq 0$. Assume that $g$ is $r$-CL at $\mathbf{x}^{(0)} \in \mathbb{R}^p$ for some $r > 0$, i.e. $4g(\mathbf{x}^{(0)}) < r^2\chi(g, \mathbf{x}^{(0)}, r)$. Choose $\epsilon \in (0, 1)$ such that $4g(\mathbf{x}^{(0)}) < (1 - \epsilon)^2 r^2 \chi(g, \mathbf{x}^{(0)}, r)$ which is possible since $4g(\mathbf{x}^{(0)}) < r^2\chi(g, \mathbf{x}^{(0)}, r)$ (Lemma C.11). Let $L_1 := \sup_{\mathbf{x} \in B(\mathbf{x}^{(0)}, r)} \|\nabla g(\mathbf{x})\|_\infty < \infty$ and $L_2 := \sup_{\mathbf{x} \in B(\mathbf{x}^{(0)}, 2r)} \|\operatorname{vec}(\nabla^2 g(\mathbf{x}))\|_\infty < \infty$. Choose any step size $\alpha > 0$ such that $\alpha < \min\left\{\frac{r}{L_1\sqrt{p}}, \frac{2\epsilon}{L_2 p}\right\}$ and iteratively define $\mathbf{x}^{(k+1)} = \mathbf{x}^{(k)} - \alpha\nabla g(\mathbf{x}^{(k)})$ for each $k \geq 0$. Then,*

- *$\mathbf{x}^{(k)} \in B(\mathbf{x}^{(0)}, r)$ for all $k$, and as $k \to \infty$, $\mathbf{x}^{(k)}$ converges to a point $\mathbf{x}^* \in B(\mathbf{x}^{(0)}, r)$ where $g(\mathbf{x}^*) = 0$ and $\|\nabla g(\mathbf{x}^*)\|_2 = 0$.*

- *Moreover, for each $k \geq 0$, $\|\mathbf{x}^{(k)} - \mathbf{x}^*\|_2^2 \leq (1 - \delta)^k r^2$, $g(\mathbf{x}^{(k)}) \leq (1 - \delta)^k g(\mathbf{x}^{(0)})$ and $\|\nabla g(\mathbf{x}^{(k)})\|_2^2 \leq 2Lg(\mathbf{x}^{(k)}) \leq 2L(1 - \delta)^k g(\mathbf{x}^{(0)})$ with $\delta := \min\{1, (1 - \epsilon) \cdot \chi(g, \mathbf{x}^{(0)}, r) \cdot \alpha\}$ and $L = \sqrt{p}L_2$.*

*Proof.* This is a special case of Theorem C.16. The proof sketch is the following. Since $\|\mathbf{x}^{(k)} - \mathbf{x}^{(j)}\|_2 \leq \sum_{l=j}^{k-1} \|\mathbf{x}^{(l+1)} - \mathbf{x}^{(l)}\|_2 = \sum_{l=j}^{k-1} \alpha\|\nabla g(\mathbf{x}^{(l)})\|_2$, Lemma C.17 use the fact that, if for some $k \geq 1$, $\mathbf{x}^{(1)}, \ldots, \mathbf{x}^{(k-1)} \in B(\mathbf{x}^{(0)}, r)$ (this is need to prove Lemma C.17 by induction), then $\sum_{l=j}^{k-1} \alpha\|\nabla g(\mathbf{x}^{(l)})\|_2 \leq (1 - \delta)^{j/2}\sqrt{\frac{4g(\mathbf{x}^{(0)})}{\chi(g, \mathbf{x}^{(0)}, r) \cdot (1-\epsilon)^2}}$ for all $0 \leq j \leq k - 1$ (Lemma C.22); so that $\|\mathbf{x}^{(k)} - \mathbf{x}^{(0)}\|_2 < r$ since $4g(\mathbf{x}^{(0)}) < (1 - \epsilon)^2 r^2 \cdot \chi(g, \mathbf{x}^{(0)}, r)$, i.e $\mathbf{x}^{(k)} \in B(\mathbf{x}^{(0)}, r)$. Lemma C.22 on its turn uses $\alpha\|\nabla g(\mathbf{x}^{(l)})\|_2^2 \leq \frac{g(\mathbf{x}^{(l)}) - g(\mathbf{x}^{(l+1)})}{1 - \epsilon}$ (Lemma C.21) and $g(\mathbf{x}^{(j+1)}) \leq (1 - \delta)\,g(\mathbf{x}^{(j)})$ (Lemma C.20). Both Lemmas C.21 and C.20 use $|R_j| \leq \epsilon\alpha\|\nabla g(\mathbf{x}^{(j)})\|_2^2$ (Lemma C.19) with $R_j = g(\mathbf{x}^{(j+1)}) - g(\mathbf{x}^{(j)}) + \alpha\|\nabla g(\mathbf{x}^{(j)})\|_2^2$. $\square$

**Theorem C.16** (Convergence of fixed-step gradient descent under CL condition and a subdifferentiable regularization). *Let $g : \mathbb{R}^p \to [0, \infty)$ be a nonnegative $C^2$ function and $h : \mathbb{R}^p \to [0, \infty)$ be a nonnegative subdifferentiable function. Take any $\mathbf{x}^{(0)} \in \mathbb{R}^p$ with $g(\mathbf{x}^{(0)}) \neq 0$. Assume that $g$ is $r$-CL at $\mathbf{x}^{(0)} \in \mathbb{R}^p$ for some $r > 0$, i.e.*

$$4g(\mathbf{x}^{(0)}) < r^2\chi(g, \mathbf{x}^{(0)}, r) \tag{28}$$

*Choose $\epsilon \in (0, 1)$ and :*

*(1) $\gamma \in (0, \epsilon)$ such that*

$$4g(\mathbf{x}^{(0)}) < \left(\frac{1 - \epsilon}{1 + \gamma}\right)^2 r^2\chi(g, \mathbf{x}^{(0)}, r) \tag{29}$$

*(2) $\gamma \in (0, 1 - \epsilon)$ such that*

$$4g(\mathbf{x}^{(0)}) < \left(\frac{1 - (\epsilon + \gamma)}{1 + \gamma}\right)^2 r^2\chi(g, \mathbf{x}^{(0)}, r) \tag{30}$$

*The two choices are possible since Equation (28) holds (Lemma C.11). Let*

- *$L_1 := \sup_{\mathbf{x} \in B(\mathbf{x}^{(0)}, r)} \|\nabla g(\mathbf{x})\|_\infty < \infty$ be a uniform upper bound on the magnitudes of the first-order derivatives of $g$ in $B(\mathbf{x}^{(0)}, r)$*

- *$L_2 := \sup_{\mathbf{x} \in B(\mathbf{x}^{(0)}, 2r)} \|\operatorname{vec}(\nabla^2 g(\mathbf{x}))\|_\infty < \infty$ be a uniform upper bound on the magnitudes of the second-order derivatives of $g$ in $B(\mathbf{x}^{(0)}, 2r)$*

- *and $L_h := \sup_{\mathbf{x} \in B(\mathbf{x}^{(0)}, r)} \sup_{H \in \partial h(\mathbf{x})} \|H\|_\infty < \infty$ be a uniform upper bound on the magnitudes of the first-order subderivatives of $h$ in $B(\mathbf{x}^{(0)}, r)$*

*Define $G(\mathbf{x}) := \nabla g(\mathbf{x})$ and choose $\beta \leq \beta_{\max}$ with $\beta_{\max} > 0$ and $\beta_{\max} \sup_{H \in \partial h(\mathbf{x}^{(0)})} \|H\|_2 \leq \gamma\|G(\mathbf{x}^{(0)})\|_2$. Choose any step size $\alpha \in (0, \alpha_{\max})$ with*

$$\alpha_{\max} := \begin{cases} \min\left\{\frac{r}{(L_1 + \beta_{\max}L_h)\sqrt{p}}, \frac{2(\epsilon - \gamma)}{L_2 p}\right\} & (1) \\[2mm] \min\left\{\frac{r}{(L_1 + \beta_{\max}L_h)\sqrt{p}}, \frac{2\epsilon}{L_2 p}\right\} & (2) \end{cases} \tag{31}$$

*Set $\chi := \chi(g, \mathbf{x}^{(0)}, r)$ and define*

$$\delta := \begin{cases} \min\{1, (1-\epsilon)\chi\alpha\} & \text{(1)} \\ \min\{1, (1-(\epsilon+\gamma))\chi\alpha\} & \text{(2)} \end{cases} \tag{32}$$

*Iteratively define $\mathbf{x}^{(k+1)} = \mathbf{x}^{(k)} - \alpha\left(G(\mathbf{x}^{(k)}) + \beta H(\mathbf{x}^{(k)})\right) \ \forall \ H(\mathbf{x}^{(k)}) \in \partial h(\mathbf{x}^{(k)})$ for each $k \geq 0$. For all $k$ such that $\beta \sup_{H \in \partial h(\mathbf{x}^{(j)})} \|H\|_2 \leq \gamma \|G(\mathbf{x}^{(j)})\|_2 \quad \forall \ 0 \leq j < k$, the following hold :*

- $\mathbf{x}^{(1)}, \ldots, \mathbf{x}^{(k)} \in B(\mathbf{x}^{(0)}, r)$

- $g(\mathbf{x}^{(j)}) \leq (1-\delta)^j g(\mathbf{x}^{(0)})$ *for all $0 \leq j \leq k$*

- $\|\nabla g(\mathbf{x}^{(j)})\|_2^2 \leq 2Lg(\mathbf{x}^{(j)}) \leq 2L(1-\delta)^j g(\mathbf{x}^{(0)})$ *for all $0 \leq j \leq k$, with $L = \sqrt{p}L_2$.*

- $\|\mathbf{x}^{(k)} - \mathbf{x}^{(j)}\| \leq (1-\delta)^{j/2} r$ *for all $0 \leq j < k$*

- *The sequence $\{\mathbf{x}^{(j)}\}_{0 \leq j < k}$ is Cauchy.*

*Moreover, if $\beta \sup_{H \in \partial h(\mathbf{x}^{(j)})} \|H\|_2 \leq \gamma \|G(\mathbf{x}^{(j)})\|_2 \quad \forall \ j \geq 0$ (e.g $\beta = 0$), then $\{\mathbf{x}^{(k)}\}_{k \geq 0}$ converges to a limit $\mathbf{x}^* \in B(\mathbf{x}^{(0)}, r)$ where $g(\mathbf{x}^*) = 0$ and $\|\nabla g(\mathbf{x}^*)\|_2 = 0$, and we have $\|\mathbf{x}^{(k)} - \mathbf{x}^*\|_2 \leq r(1-\delta)^{k/2}$ for all $k \geq 0$.*

*Proof.* Let $k \geq 1$ such that $\beta \sup_{H \in \partial h(\mathbf{x}^{(j)})} \|H\|_2 \leq \gamma \|G(\mathbf{x}^{(j)})\|_2 \quad \forall \ 0 \leq j < k$. We prove in Lemma C.17 that $\mathbf{x}^{(1)}, \ldots, \mathbf{x}^{(k)} \in B(\mathbf{x}^{(0)}, r)$. We also prove in Lemma C.20 that $g(\mathbf{x}^{(j)}) \leq (1-\delta)^j g(\mathbf{x}^{(0)})$ for all $0 \leq j \leq k$. Since $g$ is $C^2$, $\nabla^2 g(\mathbf{x})$ is symmetric for all $\mathbf{x} \in \mathbb{R}^p$. So we have for all $\mathbf{x} \in B(\mathbf{x}^{(0)}, r) \subset B(\mathbf{x}^{(0)}, 2r)$, $\lambda_{\max}(\nabla^2 g(\mathbf{x})) = \sigma_{\max}(\nabla^2 g(\mathbf{x})) \leq \sqrt{p} \max_{ij} \left|\left[\nabla^2 g(\mathbf{x})\right]_{ij}\right| \leq \sqrt{p}L_2$. So $g$ is $L$-smooth on $B(\mathbf{x}^{(0)}, r)$ by Lemma C.4. This implies $\|\nabla g(\mathbf{x})\|_2^2 \leq 2Lg(\mathbf{x})$ for all $\mathbf{x} \in B(\mathbf{x}^{(0)}, r)$ by the same lemma. So $\|\nabla g(\mathbf{x}^{(j)})\|_2^2 \leq 2Lg(\mathbf{x}^{(j)}) \leq 2L(1-\delta)^j g(\mathbf{x}^{(0)})$ for all $0 \leq j \leq k$.

For all $j \leq k-1$, we have

$$\begin{aligned}
\|\mathbf{x}^{(k)} - \mathbf{x}^{(j)}\|_2 &\leq \sum_{l=j}^{k-1} \|\mathbf{x}^{(l+1)} - \mathbf{x}^{(l)}\|_2 \\
&= \sum_{l=j}^{k-1} \alpha \|G(\mathbf{x}^{(l)}) + \beta H(\mathbf{x}^{(l)})\|_2 \\
&\leq \begin{cases} (1-\delta)^{j/2}\sqrt{\frac{4(1+\gamma)^2 g(\mathbf{x}^{(0)})}{\chi(1-\epsilon)^2}} & \text{if } \alpha \leq 2(\epsilon - \gamma)/L_2 p \\ (1-\delta)^{j/2}\sqrt{\frac{4(1+\gamma)^2 g(\mathbf{x}^{(0)})}{\chi(1-(\epsilon+\gamma))^2}} & \text{if } \alpha \leq 2\epsilon/L_2 p \end{cases} \quad \text{(Lemma C.22)} \\
&< r(1-\delta)^{j/2} \text{ (Equations (29) and (30))}
\end{aligned} \tag{33}$$

with

$$\delta := \begin{cases} (1-\epsilon)\chi\alpha & \text{if } \alpha \leq 2(\epsilon-\gamma)/L_2 p \\ (1-(\epsilon+\gamma))\chi\alpha & \text{if } \alpha \leq 2\epsilon/L_2 p \end{cases} \in (0,1] \text{ (Lemma C.20)} \tag{34}$$

For all $\varepsilon > 0$, if we pick any

$$K(\varepsilon) \geq \begin{cases} 1 & \text{if } \delta = 1 \\ \max\left\{1, \frac{2\log(\varepsilon/r)}{\log(1-\delta)}\right\} & \text{otherwise} \end{cases} = \begin{cases} 1 & \text{if } \delta = 1 \text{ or } \epsilon \leq r \\ \frac{2\log(\varepsilon/r)}{\log(1-\delta)} & \text{otherwise} \end{cases} \tag{35}$$

then we have $\|\mathbf{x}^{(k)} - \mathbf{x}^{(j)}\| \leq \varepsilon$ for all $j, k > K(\varepsilon)$. So the sequence $\{\mathbf{x}^{(j)}\}_{0 \leq j < k}$ is Cauchy.

If $\beta \sup_{H \in \partial h(\mathbf{x}^{(j)})} \|H\|_2 \leq \gamma \|G(\mathbf{x}^{(j)})\|_2 \quad \forall \ j \geq 0$ (e.g $\beta = 0$), then $\{\mathbf{x}^{(k)}\}_{k \geq 0}$ is Cauchy, and as a Cauchy sequence in the closed subset $B(\mathbf{x}^{(0)}, r)$ of $\mathbb{R}^p$, it converges to a limit $\mathbf{x}^* \in B(\mathbf{x}^{(0)}, r)$ that also satisfy $\|\mathbf{x}^* - \mathbf{x}^{(j)}\|_2 \leq r(1-\delta)^{j/2}$ for all $j \geq 1$. Finally, taking $k \to \infty$ in Lemma C.20, we get $g(\mathbf{x}^{(j)}) \leq (1-\delta)^j g(\mathbf{x}^{(0)})$ for all $j \geq 0$. This achieves the proof of the Theorem. $\square$

**Lemma C.17** (Iterates stay inside the ball). *For all $k \geq 0$ such that $\beta \sup_{H \in \partial h(\mathbf{x}^{(j)})} \|H\|_2 \leq \gamma \|G(\mathbf{x}^{(j)})\|_2 \ \forall \ 0 \leq j < k$, we have $\mathbf{x}^{(1)}, \ldots, \mathbf{x}^{(k)} \in B(\mathbf{x}^{(0)}, r)$.*

*Proof.* We prove that $\mathbf{x}^{(k)} \in B(\mathbf{x}^{(0)}, r)$ for all $k \geq 0$ such that $\beta \sup_{H \in \partial h(\mathbf{x}^{(j)})} \|H\|_2 \leq \gamma \|G(\mathbf{x}^{(j)})\|_2 \ \forall \ 0 \leq j < k$ by induction on $k$. The base case $k = 0$ is obvious since $\beta \sup_{H \in \partial h(\mathbf{x}^{(0)})} \|H\|_2 \leq \beta_{\max} \sup_{H \in \partial h(\mathbf{x}^{(0)})} \|H\|_2 \leq \gamma \|G(\mathbf{x}^{(0)})\|_2$. Let $k \geq 1$. Assume $\mathbf{x}^{(j)} \in B(\mathbf{x}^{(0)}, r)$ and $\beta \sup_{H \in \partial h(\mathbf{x}^{(j)})} \|H\|_2 \leq \gamma \|G(\mathbf{x}^{(j)})\|_2^2$ for all $0 \leq j \leq k - 1$. Then $\mathbf{x}^{(k)} \in B(\mathbf{x}^{(0)}, r)$ since

$$
\|\mathbf{x}^{(k)} - \mathbf{x}^{(0)}\|_2 \leq \sum_{l=0}^{k-1} \alpha \|G(\mathbf{x}^{(l)}) + \beta H(\mathbf{x}^{(l)})\|_2
$$

$$
\leq \begin{cases} \sqrt{\frac{4(1+\gamma)^2 g(\mathbf{x}^{(0)})}{\chi(1-\epsilon)^2}} & \text{if } \alpha \leq 2(\epsilon - \gamma)/L_2 p \\ \sqrt{\frac{4(1+\gamma)^2 g(\mathbf{x}^{(0)})}{\chi(1-(\epsilon+\gamma))^2}} & \text{if } \alpha \leq 2\epsilon/L_2 p \end{cases} \quad \text{(Lemma C.22 with } j = 0\text{)} \tag{36}
$$

$$
< \begin{cases} r & \text{if } \alpha \leq 2(\epsilon - \gamma)/L_2 p \text{ since } 4g(\mathbf{x}^{(0)}) < \left(\frac{1-\epsilon}{1+\gamma}\right)^2 r^2 \chi \text{ (Equation (29))} \\ r & \text{if } \alpha \leq 2\epsilon/L_2 p \text{ since } 4g(\mathbf{x}^{(0)}) < \left(\frac{1-(\epsilon+\gamma)}{1+\gamma}\right)^2 r^2 \chi \text{ (Equation (30))} \end{cases}
$$

$\square$

**Lemma C.18** (Second-order Taylor formula with integral remainder). *Let $g : \mathbb{R}^p \to \mathbb{R}$ be a $C^2$ function. For every base point $\mathbf{x} \in \mathbb{R}^p$ and step $\mathbf{h} \in \mathbb{R}^p$, $g(\mathbf{x} + \mathbf{h}) = g(\mathbf{x}) + \nabla g(\mathbf{x})^\top \mathbf{h} + \int_0^1 (1-t) \mathbf{h}^\top \nabla^2 g(\mathbf{x} + t\mathbf{h}) \mathbf{h} \, dt$.*

**Lemma C.19** (Second-order remainder control). *For each $j \geq 0$ set $R_j := g(\mathbf{x}^{(j+1)}) - g(\mathbf{x}^{(j)}) + \alpha \|G(\mathbf{x}^{(j)})\|_2^2$. Suppose that $\mathbf{x}^{(j)} \in B(\mathbf{x}^{(0)}, r)$ and $\beta \sup_{H \in \partial h(\mathbf{x}^{(j)})} \|H\|_2 \leq \gamma \|G(\mathbf{x}^{(j)})\|_2$ for all $0 \leq j \leq k - 1$, for some $k \geq 1$. Then for all $0 \leq j \leq k - 1$,*

$$
|R_j| \leq \begin{cases} \epsilon \alpha \|\nabla g(\mathbf{x}^{(j)})\|_2^2 & \text{if } \alpha \leq 2(\epsilon - \gamma)/L_2 p \\ (\epsilon + \gamma) \alpha \|\nabla g(\mathbf{x}^{(j)})\|_2^2 & \text{if } \alpha \leq 2\epsilon/L_2 p \end{cases} \tag{37}
$$

*Proof.* Let $S_j := g(\mathbf{x}^{(j+1)}) - g(\mathbf{x}^{(j)}) + \alpha G(\mathbf{x}^{(j)})^\top (G(\mathbf{x}^{(j)}) + \beta H(\mathbf{x}^{(j)})) = R_j + \alpha \beta G(\mathbf{x}^{(j)})^\top H(\mathbf{x}^{(j)})$. Fix $j \leq k - 1$ and let $\mathbf{x}(t) := \mathbf{x}^{(j)} - t\alpha \left( G(\mathbf{x}^{(j)}) + \beta H(\mathbf{x}^{(j)}) \right)$ with $t \in [0, 1]$. Apply the second-order Taylor formula with integral remainder (Lemma C.18):

$$
g(\mathbf{x}^{(j+1)}) = g\left(\mathbf{x}^{(j)} - \alpha(G(\mathbf{x}^{(j)}) + \beta H(\mathbf{x}^{(j)}))\right)
$$
$$
= g(\mathbf{x}^{(j)}) - \alpha G(\mathbf{x}^{(j)})^\top (G(\mathbf{x}^{(j)}) + \beta H(\mathbf{x}^{(j)})) + \alpha^2 \int_0^1 \nabla g(\mathbf{x}^{(j)})^\top \nabla^2 g\left(\mathbf{x}(t)\right) \nabla g(\mathbf{x}^{(j)})(1-t) dt \tag{38}
$$

Hence

$$
S_j = \alpha^2 \int_0^1 \nabla g(\mathbf{x}^{(j)})^\top \nabla^2 g(\mathbf{x}(t)) \nabla g(\mathbf{x}^{(j)})(1-t) dt \tag{39}
$$

Along the segment $\mathbf{x}(t)$ with $t \in [0, 1]$ we have $\mathbf{x}(0) = \mathbf{x}^{(j)}$, $\mathbf{x}(1) = \mathbf{x}^{(j+1)}$. So all points $\mathbf{x}(t)$ lie in the line segment joining $\mathbf{x}^{(j)}$ and $\mathbf{x}^{(j+1)}$. Because $\|\mathbf{x}^{(j+1)} - \mathbf{x}^{(j)}\|_2 \leq \sqrt{p}\|\mathbf{x}^{(j+1)} - \mathbf{x}^{(j)}\|_\infty = \alpha\sqrt{p}\|G(\mathbf{x}^{(j)}) + \beta H(\mathbf{x}^{(j)})\|_\infty = \alpha\sqrt{p}(L_1 + \beta L_h) \leq r$, that segment stays inside $B(\mathbf{x}^{(0)}, 2r)$; thus every entry of $\nabla^2 g(\mathbf{x}(t))$ is bounded by $L_2$. Therefore, for every $t$,

$$
\left| \nabla g(\mathbf{x}^{(j)})^\top \nabla^2 g(\mathbf{x}(t)) \nabla g(\mathbf{x}^{(j)}) \right| = \left| \sum_{i=1}^p \sum_{i'=1}^p [\nabla g(\mathbf{x}^{(j)})]_i [\nabla^2 g(\mathbf{x}(t)) \nabla g(\mathbf{x}^{(j)})]_{i,i'} [\nabla g(\mathbf{x}^{(j)})]_{i'} \right|
$$
$$
\leq L_2 \sum_{i=1}^p \sum_{i'=1}^p \left| [\nabla g(\mathbf{x}^{(j)})]_i \right| \left| g(\mathbf{x}^{(j)})]_{i'} \right| = L_2 \|\nabla g(\mathbf{x}^{(j)})\|_1^2 \leq L_2 p \|\nabla g(\mathbf{x}^{(j)})\|_2^2 \tag{40}
$$

So

$$
|S_j| \leq \alpha^2 \int_0^1 \left| \nabla g(\mathbf{x}^{(j)})^\top \nabla^2 g(\mathbf{x}(t)) \nabla g(\mathbf{x}^{(j)}) \right| |1 - t| \, dt
$$
$$
\leq \alpha^2 L_2 p \|\nabla g(\mathbf{x}^{(j)})\|_2^2 \int_0^1 (1-t) dt \tag{41}
$$
$$
= \frac{\alpha^2}{2} L_2 p \|\nabla g(\mathbf{x}^{(j)})\|_2^2 \leq \begin{cases} (\epsilon - \gamma)\alpha \|G(\mathbf{x}^{(j)})\|_2^2 & \text{if } \alpha \leq 2(\epsilon - \gamma)/L_2 p \\ \epsilon \alpha \|G(\mathbf{x}^{(j)})\|_2^2 & \text{if } \alpha \leq 2\epsilon/L_2 p \end{cases}
$$

This implies

$$
\begin{aligned}
|R_j| = |S_j - \alpha\beta G(\mathbf{x}^{(j)})^\top H(\mathbf{x}^{(j)})| &\leq |S_j| + \alpha\beta\|G(\mathbf{x}^{(j)})\|_2\|H(\mathbf{x}^{(j)})\|_2 \\
&\leq |S_j| + \alpha\gamma\|G(\mathbf{x}^{(j)})\|_2^2 \leq \begin{cases} (\epsilon - \gamma)\alpha\|G(\mathbf{x}^{(j)})\|_2^2 + \alpha\gamma\|G(\mathbf{x}^{(j)})\|_2^2 & \text{if } \alpha \leq 2(\epsilon - \gamma)/L_2 p \\ \epsilon\alpha\|G(\mathbf{x}^{(j)})\|_2^2 + \alpha\gamma\|G(\mathbf{x}^{(j)})\|_2^2 & \text{if } \alpha \leq 2\epsilon/L_2 p \end{cases}
\end{aligned}
\tag{42}
$$

$\square$

**Lemma C.20** (Geometric decay of the objective). *Let*

$$
\delta := \begin{cases} (1 - \epsilon)\chi\alpha & \text{if } \alpha \leq 2(\epsilon - \gamma)/L_2 p \\ (1 - (\epsilon + \gamma))\chi\alpha & \text{if } \alpha \leq 2\epsilon/L_2 p \end{cases}
\tag{43}
$$

*We have $\delta \leq 1$, and if $\mathbf{x}^{(j)} \in B(\mathbf{x}^{(0)}, r)$ and $\beta\sup_{H \in \partial h(\mathbf{x}^{(j)})}\|H\|_2 \leq \gamma\|G(\mathbf{x}^{(j)})\|_2$ for all $0 \leq j \leq k - 1$, for some $k \geq 1$; then*

$$
g(\mathbf{x}^{(j)}) \leq (1 - \delta)^j g(\mathbf{x}^{(0)}) \; \forall 0 \leq j \leq k
\tag{44}
$$

*Proof.* By the definition of $\chi := \chi(g, \mathbf{x}^{(0)}, r)$, we have $\chi g(\mathbf{x}) \leq \|\nabla g(\mathbf{x})\|_2^2$ for all $\mathbf{x} \in B(\mathbf{x}^{(0)}, r)$. Since $\mathbf{x}^{(j)} \in B(\mathbf{x}^{(0)}, r)$ for all $0 \leq j \leq k - 1$, $\|\nabla g(\mathbf{x}^{(j)})\|_2^2 \geq \chi g(\mathbf{x}^{(j)})$ for all $0 \leq j \leq k - 1$. So, for all $0 \leq j \leq k - 1$,

$$
\begin{aligned}
g(\mathbf{x}^{(j+1)}) &= g(\mathbf{x}^{(j)}) - \alpha\|\nabla g(\mathbf{x}^{(j)})\|_2^2 + R_j \text{ by the definition of } R_j \\
&\leq \begin{cases} g(\mathbf{x}^{(j)}) - \alpha\|\nabla g(\mathbf{x}^{(j)})\|_2^2 + \epsilon\alpha\|\nabla g(\mathbf{x}^{(j)})\|_2^2 & \text{if } \alpha \leq 2(\epsilon - \gamma)/L_2 p \\ g(\mathbf{x}^{(j)}) - \alpha\|\nabla g(\mathbf{x}^{(j)})\|_2^2 + (\epsilon + \gamma)\alpha\|\nabla g(\mathbf{x}^{(j)})\|_2^2 & \text{if } \alpha \leq 2\epsilon/L_2 p \end{cases} \quad \text{(Lemma C.19)} \\
&= \begin{cases} g(\mathbf{x}^{(j)}) - (1 - \epsilon)\alpha\|\nabla g(\mathbf{x}^{(j)})\|_2^2 & \text{if } \alpha \leq 2(\epsilon - \gamma)/L_2 p \\ g(\mathbf{x}^{(j)}) - (1 - (\epsilon + \gamma))\alpha\|\nabla g(\mathbf{x}^{(j)})\|_2^2 & \text{if } \alpha \leq 2\epsilon/L_2 p \end{cases} \\
&\leq \begin{cases} g(\mathbf{x}^{(j)}) - (1 - \epsilon)\alpha\chi g(\mathbf{x}^{(j)}) & \text{if } \alpha \leq 2(\epsilon - \gamma)/L_2 p \\ g(\mathbf{x}^{(j)}) - (1 - (\epsilon + \gamma))\alpha\chi g(\mathbf{x}^{(j)}) & \text{if } \alpha \leq 2\epsilon/L_2 p \end{cases} \\
&= (1 - \delta)g(\mathbf{x}^{(j)})
\end{aligned}
\tag{45}
$$

Moreover, taking $j = 0$ gives $g(\mathbf{x}^{(1)}) \leq (1 - \delta)g(\mathbf{x}^{(0)}) \implies 1 - \delta \geq g(\mathbf{x}^{(1)})/g(\mathbf{x}^{(0)}) \geq 0 \implies \delta \leq 1$. $\square$

**Lemma C.21** (One-step descent bound). *Suppose that $\mathbf{x}^{(j)} \in B(\mathbf{x}^{(0)}, r)$ and $\beta\sup_{H \in \partial h(\mathbf{x}^{(j)})}\|H\|_2 \leq \gamma\|G(\mathbf{x}^{(j)})\|_2$ for all $0 \leq j \leq k - 1$, for some $k \geq 1$. Then for all $0 \leq j \leq k - 1$,*

$$
g(\mathbf{x}^{(j)}) - g(\mathbf{x}^{(j+1)}) \geq \begin{cases} (1 - \epsilon)\alpha\|\nabla g(\mathbf{x}^{(j)})\|_2^2 & \text{if } \alpha \leq 2(\epsilon - \gamma)/L_2 p \\ (1 - (\epsilon + \gamma))\alpha\|\nabla g(\mathbf{x}^{(j)})\|_2^2 & \text{if } \alpha \leq 2\epsilon/L_2 p \end{cases} = \frac{\delta}{\chi}\|\nabla g(\mathbf{x}^{(j)})\|_2^2
\tag{46}
$$

*Proof.* For all $0 \leq j \leq k - 1$,

$$
\begin{aligned}
g(\mathbf{x}^{(j)}) - g(\mathbf{x}^{(j+1)}) &= \alpha\|\nabla g(\mathbf{x}^{(j)})\|_2^2 - R_j \text{ by the definition of } R_j \\
&\geq \begin{cases} \alpha\|\nabla g(\mathbf{x}^{(j)})\|_2^2 - \epsilon\alpha\|\nabla g(\mathbf{x}^{(j)})\|_2^2 & \text{if } \alpha \leq 2(\epsilon - \gamma)/L_2 p \\ \alpha\|\nabla g(\mathbf{x}^{(j)})\|_2^2 - (\epsilon + \gamma)\alpha\|\nabla g(\mathbf{x}^{(j)})\|_2^2 & \text{if } \alpha \leq 2\epsilon/L_2 p \end{cases} \quad \text{(Lemma C.19)}
\end{aligned}
\tag{47}
$$

$\square$

**Lemma C.22** (Bounding cumulative step lengths). *Suppose that $\mathbf{x}^{(j)} \in B(\mathbf{x}^{(0)}, r)$ and $\beta\sup_{H \in \partial h(\mathbf{x}^{(j)})}\|H\|_2 \leq \gamma\|G(\mathbf{x}^{(j)})\|_2$ for all $0 \leq j \leq k - 1$, for some $k \geq 1$. Then for all $0 \leq j \leq k - 1$,*

$$
\sum_{l=j}^{k-1} \alpha\|G(\mathbf{x}^{(l)}) + \beta H(\mathbf{x}^{(l)})\|_2 = \begin{cases} (1 - \delta)^{j/2}\sqrt{\frac{4(1+\gamma)^2 g(\mathbf{x}^{(0)})}{\chi(1-\epsilon)^2}} & \text{if } \alpha \leq 2(\epsilon - \gamma)/L_2 p \\ (1 - \delta)^{j/2}\sqrt{\frac{4(1+\gamma)^2 g(\mathbf{x}^{(0)})}{\chi(1-(\epsilon+\gamma))^2}} & \text{if } \alpha \leq 2\epsilon/L_2 p \end{cases}
\tag{48}
$$

*Proof.* We have

$$
\sum_{l=j}^{k-1} \alpha \| G(\mathbf{x}^{(l)}) + \beta H(\mathbf{x}^{(l)}) \|_2
$$

$$
\leq \sum_{l=j}^{k-1} \alpha \left( \| G(\mathbf{x}^{(l)}) \|_2 + \beta \| H(\mathbf{x}^{(l)}) \|_2 \right)
$$

$$
\leq \sum_{l=j}^{k-1} \alpha(1+\gamma) \| G(\mathbf{x}^{(l)}) \|_2 \text{ since } \beta \| H(\mathbf{x}^{(l)}) \|_2 \leq \gamma \| G(\mathbf{x}^{(l)}) \|_2
$$

$$
= \sum_{l=j}^{k-1} \sqrt{\alpha^2 (1+\gamma)^2 \| \nabla g(\mathbf{x}^{(l)}) \|_2^2}
$$

$$
\leq \sum_{l=j}^{k-1} \sqrt{\frac{\alpha^2 (1+\gamma)^2 \chi}{\delta} \left( g(\mathbf{x}^{(l)}) - g(\mathbf{x}^{(l+1)}) \right)} \text{ (Lemma C.21)}
$$

$$
= \sqrt{\frac{\alpha^2 (1+\gamma)^2 \chi}{\delta}} \sum_{l=j}^{k-1} \left( \sqrt{g(\mathbf{x}^{(l)})} - \sqrt{g(\mathbf{x}^{(l+1)})} \right)^{\frac{1}{2}} \left( \sqrt{g(\mathbf{x}^{(l)})} + \sqrt{g(\mathbf{x}^{(l+1)})} \right)^{\frac{1}{2}}
$$

$$
\leq \sqrt{\frac{\alpha^2 (1+\gamma)^2 \chi}{\delta}} \left( \sum_{l=j}^{k-1} \left( \sqrt{g(\mathbf{x}^{(l)})} - \sqrt{g(\mathbf{x}^{(l+1)})} \right) \sum_{l=j}^{k-1} \left( \sqrt{g(\mathbf{x}^{(l)})} + \sqrt{g(\mathbf{x}^{(l+1)})} \right) \right)^{\frac{1}{2}} \text{ (Cauchy–Schwarz)}
$$

$$
= \sqrt{\frac{\alpha^2 (1+\gamma)^2 \chi}{\delta}} \left( \left( \sqrt{g(\mathbf{x}^{(j)})} - \sqrt{g(\mathbf{x}^{(k)})} \right) \sum_{l=j}^{k-1} \left( \sqrt{g(\mathbf{x}^{(l)})} + \sqrt{g(\mathbf{x}^{(l+1)})} \right) \right)^{\frac{1}{2}} \text{ (Telescoping the first sum)}
$$

$$
\leq \sqrt{\frac{\alpha^2 (1+\gamma)^2 \chi}{\delta}} \left( \sqrt{g(\mathbf{x}^{(j)})} \sum_{l=j}^{k-1} \left( 2\sqrt{g(\mathbf{x}^{(l)})} \right) \right)^{\frac{1}{2}} \text{ since } g(\mathbf{x}^{(l+1)}) \leq g(\mathbf{x}^{(l)}) \, \forall l \leq k-1 \text{ (Lemma C.21)}
$$

$$
\leq \sqrt{\frac{2\alpha^2 (1+\gamma)^2 \chi}{\delta}} \left( g(\mathbf{x}^{(0)}) \sqrt{(1-\delta)^j} \sum_{l=j}^{k-1} \sqrt{(1-\delta)^l} \right)^{\frac{1}{2}} \text{ since } g(\mathbf{x}^{(j)}) \leq (1-\delta)^j g(\mathbf{x}^{(0)}) \, \forall j \leq k \text{ (C.20)}
$$

$$
= (1-\delta)^{j/4} \sqrt{\frac{2\alpha^2 (1+\gamma)^2 \chi g(\mathbf{x}^{(0)})}{\delta}} \left( \sum_{l=j}^{k-1} (1-\delta)^{l/2} \right)^{\frac{1}{2}}
$$

$$
= (1-\delta)^{j/4} \sqrt{\frac{2\alpha^2 (1+\gamma)^2 \chi g(\mathbf{x}^{(0)})}{\delta}} \left( (1-\delta)^{j/2} \sum_{l=0}^{k-j-1} (1-\delta)^{l/2} \right)^{\frac{1}{2}}
$$

$$
\leq (1-\delta)^{j/4+j/4} \sqrt{\frac{2\alpha^2 (1+\gamma)^2 \chi g(\mathbf{x}^{(0)})}{\delta}} \left( \sum_{l=0}^{k-j-1} (1-\delta/2)^l \right)^{\frac{1}{2}} \text{ since } \sqrt{1-\delta} \leq 1-\delta/2
$$

$$
= (1-\delta)^{j/2} \sqrt{\frac{2\alpha^2 (1+\gamma)^2 \chi g(\mathbf{x}^{(0)})}{\delta}} \left[ \frac{2}{\delta} \left( 1 - (1-\delta/2)^{k-j} \right) \right]^{\frac{1}{2}}
$$

$$
\leq (1-\delta)^{j/2} \sqrt{\frac{2\alpha^2 (1+\gamma)^2 \chi g(\mathbf{x}^{(0)})}{\delta}} \sqrt{\frac{2}{\delta}}
$$

$$
= \begin{cases} (1-\delta)^{j/2} \sqrt{\frac{4(1+\gamma)^2 g(\mathbf{x}^{(0)})}{\chi(1-\epsilon)^2}} & \text{for } \delta = (1-\epsilon)\chi\alpha \\ (1-\delta)^{j/2} \sqrt{\frac{4(1+\gamma)^2 g(\mathbf{x}^{(0)})}{\chi(1-(\epsilon+\gamma))^2}} & \text{for } \delta = (1-(\epsilon+\gamma))\chi\alpha \end{cases}
$$

$$\tag{49}$$

$\square$

## C.2. Main Theorem under Polyak-Łojasiewicz Inequality

In this section, we prove the geometric convergence for $g$ under the classical Polyak-Łojasiewicz (PL) inequality.

**Definition C.23** (Generalized Polyak-Łojasiewicz inequality). Let $\varphi : \mathbb{R}^p \to \mathbb{R}$ be a sub-differentiable function such that $\arg\min_{\mathbf{x} \in \mathbb{R}^p} \varphi(\mathbf{x}) \neq \emptyset$ and $\varphi^* := \min_{\mathbf{x} \in \mathbb{R}^p} \varphi(\mathbf{x}) > -\infty$. For $\mu > 0$, $\varphi$ is said to satisfy the $\mu$-PL inequality if and only if

$$\varphi(\mathbf{x}) - \varphi^* \leq \frac{1}{2\mu} \inf_{\mathbf{z} \in \partial\varphi(\mathbf{x})} \|\mathbf{z}\|_2^2 \quad \forall \mathbf{x} \in \mathbb{R}^p \tag{50}$$

The PL inequality is typically stated for differentiable functions. This version allows $\varphi$ to be subdifferentiable, which is a generalization, and thus is not always equivalent to properties known for the smooth case. The CL inequality can be seen as a strengthening of the PL inequality and is related to the classical Kurdyka–Łojasiewicz inequality (Chatterjee, 2022).

**Theorem C.24.** Let $\varphi : \mathbb{R}^p \to [0, \infty)$ be a subdifferentiable function such that $\arg\min_{\mathbf{x} \in \mathbb{R}^p} \varphi(\mathbf{x}) \neq \emptyset$. Assume $\varphi^* := \min_{\mathbf{x} \in \mathbb{R}^p} \varphi(\mathbf{x}) = 0$. If $\varphi$ is $\mu$-PL for some $\mu > 0$, then for all $\mathbf{x} \in \mathbb{R}^p$, $\varphi$ is r-CL at $\mathbf{x}$ whatever $r^2 \geq 2\varphi(\mathbf{x})/\mu$.

**Theorem C.25** (Convergence of fixed-step gradient descent under PL condition). Let $g : \mathbb{R}^p \to [0, \infty)$ be a nonnegative $C^2$, L-smooth and $\mu$-PL function such that $\Theta_g := \arg\min_{\mathbf{x} \in \mathbb{R}^p} g(\mathbf{x}) \neq \emptyset$, and $h : \mathbb{R}^p \to [0, \infty)$ be a nonnegative subdifferentiable function. Let $g^* := \min_{\mathbf{x} \in \mathbb{R}^p} g(\mathbf{x})$. Take any $\mathbf{x}^{(0)} \in \mathbb{R}^p$, choose any step size $\alpha \in (0, 2/L)$ and any $\beta \geq 0$. Iteratively define $\mathbf{x}^{(t+1)} = \mathbf{x}^{(t)} - \alpha \left( G(\mathbf{x}^{(t)}) + \beta H(\mathbf{x}^{(t)}) \right)$ for each $t \geq 0$, with $G(\cdot) = \nabla g(\cdot)$ and $H(\cdot) \in \partial h(\cdot)$. For any $\gamma \in [0, 1)$ such that $\kappa(\gamma) := (2 - \alpha L)(1 - \gamma) - \alpha L \gamma^2 \in [0, 2/\mu\alpha]$, and for any $t \geq 0$, we have :

$$\beta\|H(\mathbf{x}^{(t)})\|_2 \leq \gamma\|G(\mathbf{x}^{(t)})\|_2 \quad \Longrightarrow \quad g(\mathbf{x}^{(t+1)}) - g^* \leq (1 - \mu \cdot \alpha \cdot \kappa(\gamma)) \left(g(\mathbf{x}^{(t)}) - g^*\right) \tag{51}$$

$$\beta\|H(\mathbf{x}^{(k)})\|_2 \leq \gamma\|G(\mathbf{x}^{(k)})\|_2 \quad \forall k < t \quad \Longrightarrow \quad \left|g(\mathbf{x}^{(k)}) - g^*\right| \leq |1 - \mu \cdot \alpha \cdot \kappa(\gamma)|^k \left(g(\mathbf{x}^0) - g^*\right) \quad \forall k \leq t \tag{52}$$

As a consequence, if Equation (52) holds for some $\gamma \in [0, 1)$ with $\kappa(\gamma) \in [0, 2/\mu\alpha]$ and for all $t \geq 0$ (e.g. $\beta = 0$), then:

$$
\begin{aligned}
g(\mathbf{x}^{(k)}) - g^* &\leq (1 - \mu \cdot \alpha \cdot \kappa(\gamma))^k \left(g(\mathbf{x}^{(0)}) - g^*\right) \; \forall k \geq 0 \\
&= (1 - \mu \cdot \alpha \cdot (2 - \alpha L))^k \left(g(\mathbf{x}^{(0)}) - g^*\right) \; \forall k \geq 0, \text{ for } \gamma = 0 \\
&= \left(\frac{\mu}{L}\right)^k \left(g(\mathbf{x}^{(0)}) - g^*\right) \; \forall k \geq 0, \text{ for } \gamma = 0 \text{ and } \alpha = 1/L
\end{aligned}
\tag{53}
$$

*Proof.* Fix $t \geq 0$, and assume $\beta\|H(\mathbf{x}^{(t)})\|_2 \leq \gamma\|G(\mathbf{x}^{(t)})\|_2$. Since $g$ has an $L$-Lipschitz continuous gradient, we have

$$
\begin{aligned}
g(\mathbf{x}^{(t+1)}) &\leq g(\mathbf{x}^{(t)}) + (\mathbf{x}^{(t+1)} - \mathbf{x}^{(t)})^\top G(\mathbf{x}^{(t)}) + \frac{L}{2}\|\mathbf{x}^{(t+1)} - \mathbf{x}^{(t)}\|_2^2 \text{ (Lemma C.4)} \\
&= g(\mathbf{x}^{(t)}) - \alpha(G(\mathbf{x}^{(t)}) + \beta H(\mathbf{x}^{(t)}))^\top G(\mathbf{x}^{(t)}) + \frac{L\alpha^2}{2}\|G(\mathbf{x}^{(t)}) + \beta H(\mathbf{x}^{(t)})\|_2^2 \\
&= g(\mathbf{x}^{(t)}) - \frac{\alpha}{2}(2 - \alpha L)\|G(\mathbf{x}^{(t)})\|_2^2 - \frac{\alpha}{2}(2 - \alpha L)\beta H(\mathbf{x}^{(t)})^\top G(\mathbf{x}^{(t)}) + \frac{\alpha^2 L}{2}\|\beta H(\mathbf{x}^{(t)})\|_2^2 \\
&\leq g(\mathbf{x}^{(t)}) - \frac{\alpha}{2}(2 - \alpha L)\|G(\mathbf{x}^{(t)})\|_2^2 + \frac{\alpha}{2}|2 - \alpha L|\beta\left|H(\mathbf{x}^{(t)})^\top G(\mathbf{x}^{(t)})\right| + \frac{\alpha^2 L}{2}\|\beta H(\mathbf{x}^{(t)})\|_2^2 \\
&\leq g(\mathbf{x}^{(t)}) - \frac{\alpha}{2}(2 - \alpha L)\|G(\mathbf{x}^{(t)})\|_2^2 + \frac{\alpha}{2}(2 - \alpha L)\beta\|H(\mathbf{x}^{(t)})\|_2\|G(\mathbf{x}^{(t)})\|_2 + \frac{\alpha^2 L}{2}\|\beta H(\mathbf{x}^{(t)})\|_2^2 \\
&\leq g(\mathbf{x}^{(t)}) - \frac{\alpha}{2}(2 - \alpha L)\|G(\mathbf{x}^{(t)})\|_2^2 + \frac{\alpha}{2}(2 - \alpha L)\gamma\|G(\mathbf{x}^{(t)})\|_2^2 + \frac{\alpha}{2}\alpha L\gamma^2\|G(\mathbf{x}^{(t)})\|_2^2 \\
&= g(\mathbf{x}^{(t)}) - \frac{\alpha\kappa}{2}\|G(\mathbf{x}^{(t)})\|_2^2 \text{ with } \kappa = (2 - \alpha L) - (2 - \alpha L)\gamma - \alpha L\gamma^2 \geq 0 \\
&\leq g(\mathbf{x}^{(t)}) - \alpha\mu\kappa \cdot \left(g(\mathbf{x}^{(t)}) - g^*\right) \text{ since } \kappa \geq 0 \text{ and } g \text{ is } \mu\text{-PL, i.e. } \|G(\mathbf{x}^{(t)})\|_2^2 \geq 2\mu(g(\mathbf{x}^{(t)}) - g^*) \\
&= (1 - \alpha\mu\kappa)\left(g(\mathbf{x}^{(t)}) - g^*\right) + g^*
\end{aligned}
\tag{54}
$$

As a consequence, if we have $\beta\|H(\mathbf{x}^{(k)})\|_2 \leq \gamma\|G(\mathbf{x}^{(k)})\|_2$ for all $k < t$, then $g(\mathbf{x}^{(1)}) - g^* \leq (1 - \alpha\mu\kappa)\left(g(\mathbf{x}^0) - g^*\right)$, $|g(\mathbf{x}^{(2)}) - g^*| \leq |1 - \alpha\mu\kappa||g(\mathbf{x}^1) - g^*| \leq |1 - \alpha\mu\kappa|^2(g(\mathbf{x}^0) - g^*), \cdots, |g(\mathbf{x}^{(t)}) - g^*| \leq |1 - \alpha\mu\kappa||g(\mathbf{x}^{t-1}) - g^*| \leq |1 - \alpha\mu\kappa|^t(g(\mathbf{x}^0) - g^*)$. Note that $\frac{d}{d\alpha}(1 - \mu\alpha(2 - \alpha L)) = 0 \iff \alpha = 1/L$. $\square$

One recovers the standard convergence result under PL inequality (Karimi et al., 2020) from this Theorem with $\beta = 0$ (Equation (53)). Note that the smoothness of $g$ along the trajectory is only a consequence of the result under CL, whereas under PL, it is assumed in advance. More importantly, the CL is only required at initialization, whereas PL is required on the entire domain of $g$.

Let $\tau = \alpha L \in (0, 2)$ and $\kappa(\gamma) = (2-\tau) - (2-\tau)\gamma - \tau\gamma^2$. We want $(\alpha, \gamma) \in (0, 2/L) \times [0, 1]$ such that $0 \leq \kappa(\gamma) \leq 2/\mu\alpha$. We have $0 \leq \kappa(\gamma) \iff \tau\gamma^2 + (2-\tau)\gamma - (2-\tau) \leq 0 \iff 0 \leq \gamma \leq \frac{\sqrt{(2-\tau)(2+3\tau)}-(2-\tau)}{2\tau} \leq 1$. Since $\kappa$ is a downward-opening parabola in $\gamma$, its maximum value in the interval $[0, 1]$ occurs at $\gamma = 0$ (since the vertex is at a negative $\gamma$ value). Therefore, we only need to satisfy the inequality $\kappa(\gamma) \leq 2/\mu\alpha$ at $\gamma = 0$, that is $2 - \alpha L \leq \frac{2}{\mu\alpha} \iff \mu L\alpha^2 - 2\mu\alpha + 2 \leq 0$. The solutions for $\alpha$ depend on the discriminant $\Delta = (-2\mu)^2 - 4(\mu L)(2) = 4\mu(\mu - 2L)$ of this quadratic.

If $\mu < 2L$, $\Delta < 0$, and the quadratic function $\mu L\alpha^2 - 2\mu\alpha + 2$ is always positive. So the set of solutions is empty.

If $\mu = 2L$, $\Delta = 0$, and the quadratic inequality for $\alpha$ is satisfied only at its root $\alpha = \frac{2\mu}{2\mu L} = \frac{1}{L}$. For this single value of $\alpha$, the condition on $\gamma$ is: $0 \leq \gamma \leq \frac{\sqrt{(2-1)(2+3)}-(2-1)}{2(1)} = \frac{\sqrt{5}-1}{2}$. So, the solution set is a line segment, $\left\{ \left(\frac{1}{L}, \gamma\right) \mid \gamma \in \left[0, \frac{\sqrt{5}-1}{2}\right]\right\}$.

If $\mu > 2L$, $\Delta > 0$, and the quadratic inequality $\mu L\alpha^2 - 2\mu\alpha + 2 \leq 0$ is satisfied for $\alpha$ between the two real roots $\alpha \in \left[\frac{\mu - \sqrt{\mu^2 - 2\mu L}}{\mu L}, \frac{\mu + \sqrt{\mu^2 - 2\mu L}}{\mu L}\right] \subset (0, 2/L)$. For any $\alpha$ chosen from this interval, the corresponding values for $\gamma$ are given by $0 \leq \gamma \leq \frac{\sqrt{(2-\alpha L)(2+3\alpha L)}-(2-\alpha L)}{2\alpha L}$.

## C.3. Preservation of the Chatterjee–Łojasiewicz Inequality under Perturbation

In Section C.1, we study how $g$ behaves when a regularization $h$ is added to the optimization objective $f = g + \beta h$. We also study the behavior of $f$ and the regularization term $h$ when $g$ and its gradient are already small. We establish below that, under some assumptions on $h$ and $g$, if $g$ is $r$-CL at $\mathbf{x} \in \mathbb{R}^p$, then for all $\varepsilon \in (0, 1)$, there exists $\beta_{\max} = \beta_{\max}(\varepsilon, \mathbf{x}, r) > 0$ such that for all $\beta < \beta_{\max}$, the function $f = g + \beta h$ is also $\varepsilon r$-CL at $\mathbf{x}$. This means that Theorem C.15 also applies to $f - f^*$. In other words, we also have (undefined terms are those of Theorem C.15):

- $\mathbf{x}^{(k)} \in B(\mathbf{x}^{(0)}, \varepsilon r)$ for all $k$, and as $k \to \infty$, $\mathbf{x}^{(k)}$ converges to a point $\mathbf{x}^* \in B(\mathbf{x}^{(0)}, \varepsilon r)$ where $f(\mathbf{x}^*) = f^*$.

- For each $k \geq 0$, $\|\mathbf{x}^{(k)} - \mathbf{x}^*\|_2^2 \leq (1 - \delta)^k \varepsilon^2 r^2$ and $f(\mathbf{x}^{(k)}) - f^* \leq (1 - \delta)^k \left(f(\mathbf{x}^{(0)}) - f^*\right)$ with $\delta := \min\{1, (1 - \epsilon) \cdot \chi(f, \mathbf{x}^{(0)}, \varepsilon r) \cdot \alpha\}$.

We start by introducing a property less restrictive than convexity, which we call the Zero-Stationary Property (ZSP).

**Definition C.26** (Zero-Stationary Property). A subdifferentiable function $\varphi : \mathbb{R}^p \to \mathbb{R}$ is said to satisfy the zero-stationary property (ZSP) or to be ZSP on $U \subset \mathbb{R}^p$ if and only if for all $\mathbf{x} \in U$, $0 \in \partial\varphi(\mathbf{x}) \implies \varphi(\mathbf{x}) = 0$.

For a ZSP function, if a point is stationary (in the sense that 0 is a subgradient), its value must be zero. In the context of non-negative functions, this implies that only points with the optimal value (zero) can be stationary. This is linked to Fermat's or interior extremum theorem (which identifies stationary points via derivatives), but adds a zero constraint on the function's value. The $\ell_p$ norm ($p > 0$) and nuclear norm $\ell_*$ are ZSP on $\mathbb{R}^p$. Convex functions are not ZSP in general. Take for example $\varphi(\mathbf{x}) = 1$ or $\varphi(\mathbf{x}) = (\mathbf{x} - 1)^2 - 1$. But if $\varphi$ is convex and $\varphi^* := \inf_{\mathbf{x} \in \mathbb{R}^p} \varphi(\mathbf{x})$ is in the image $Im(\varphi)$ of $\varphi$, then $\varphi$ is ZSP (Lemma C.27). The ZSP property does not imply convexity. The ZSP is local because it only constrains the function's value at points where the subgradient contains zero. It does not constrain the function's curvature or behavior away from these stationary points. Therefore, for a ZSP function to be convex, it must satisfy one of the standard conditions that define or characterize convexity (first or second-order condition, Jensen's inequality, convexity of the Epigraph, etc).

**Lemma C.27.** Let $\varphi : \mathbb{R}^p \to \mathbb{R}$ be a subdifferentiable function such that $-\infty < \varphi^* := \inf_{\mathbf{x} \in \mathbb{R}^p} \varphi(\mathbf{x}) \in Im(\varphi)$. If $\varphi$ is convex, then $\varphi - \varphi^*$ is ZSP. The converse is false.

*Proof.* Let $\varphi$ be convex and assume $-\infty < \varphi^* = \inf_{\mathbf{x} \in \mathbb{R}^p} \varphi(\mathbf{x}) \in Im(\varphi)$. Define $\tilde{\varphi}(\mathbf{x}) := \varphi(\mathbf{x}) - \varphi^*$. Let $\mathbf{x} \in \mathbb{R}^p$ such that $0 \in \partial\tilde{\varphi}(\mathbf{x}) = \partial\varphi(\mathbf{x})$. By the definition of convexity, we know that $\varphi(\mathbf{y}) \geq \varphi(\mathbf{x}) + (\mathbf{y} - \mathbf{x})^\top \mathbf{z}$ for all $\mathbf{y}$ and all $\mathbf{z} \in \partial\varphi(\mathbf{x})$. Since $0 \in \partial\varphi(\mathbf{x})$, we choose $\mathbf{z} = 0$, and get $\varphi(\mathbf{y}) \geq \varphi(\mathbf{x})$ for all $\mathbf{y}$. This means that $\mathbf{x}$ is a global minimum of $\varphi$. Since $\varphi^* \in Im(\varphi)$, we conclude $\varphi(\mathbf{x}) = \varphi^*$, i.e. $\tilde{\varphi}(\mathbf{x}) = 0$. This proves that $\tilde{\varphi}$ is ZSP.

Take $\varphi(x) = \frac{x^2}{1+x^2}$ for $x \in \mathbb{R}$, with $\varphi^* = 0$. Since $\varphi'(x) = \frac{2x}{(1+x^2)^2}$, we have $0 \in \partial\varphi(x) \iff x = 0 \iff \varphi(x) = 0$. So $\varphi - \varphi^*$ is ZSP. But $\varphi$ is neither globally convex nor globally concave on $\mathbb{R}$ because $\varphi''(x) = \frac{2(1-3x^2)}{(1+x^2)^3}$ changes sign. $\qquad\square$

**Theorem C.28** (Chatterjee–Łojasiewicz stability under ZSP Perturbation). *Let $g : \mathbb{R}^p \to [0, \infty)$ be a differentiable function, and let $h : \mathbb{R}^p \to [0, \infty)$ be a subdifferentiable function. For all $\mathbf{x} \in \mathbb{R}^p$ and $r > 0$, if $g$ satisfies the $r$-CL inequality at $\mathbf{x}$, i.e $4g(\mathbf{x}) < r^2 \cdot \chi(g, \mathbf{x}, r)$, then:*

- *For all $\varepsilon \in (0, 1)$, letting $r' := \varepsilon r$, the function $f := g + \beta h$ satisfies the $r'$-CL inequality at $\mathbf{x}$ with respect to $g$, i.e., $4f(\mathbf{x}) < r'^2 \cdot \chi(f, \mathbf{x}, r', g)$, provided that $\beta \le \beta_{\max}(\varepsilon, \mathbf{x}, r) = \min\left\{ \frac{\sqrt{\underline{g}\cdot\chi(g,\mathbf{x},r)}}{4L}, \frac{\sqrt{\Delta} - B}{2C} \right\}$. Moreover, if $g(\mathbf{y}) = 0 \implies f(\mathbf{y}) = 0$, then $f = g + \beta h$ satisfies the $r'$-CL inequality at $\mathbf{x}$ provided that $\beta$ satisfies this constraint, i.e., $4f(\mathbf{x}) < r'^2 \cdot \chi(f, \mathbf{x}, r')$.*

- *If $h$ is ZSP, then for all $\varepsilon \in (0, 1)$ such that $4h(\mathbf{x}) > r'^2 \chi(h, \mathbf{x}, r')$ with $r' := \varepsilon r$, the function $f := g + \beta h$ satisfies the $\varepsilon r$-CL inequality at $\mathbf{x}$, i.e., $4f(\mathbf{x}) < \varepsilon^2 r^2 \cdot \chi(f, \mathbf{x}, \varepsilon r)$, provided that $\beta \le \beta_{\max}(\varepsilon, \mathbf{x}, r)$ with*

$$\beta_{\max}(\varepsilon, \mathbf{x}, r) = \min\left\{ \frac{\sqrt{\underline{g}\cdot\chi(g,\mathbf{x},r)}}{4L}, \frac{\sqrt{\Delta} - B}{2C}, \frac{4g(\mathbf{x})}{r'^2 \cdot \chi(h, \mathbf{x}, r') - 4h(\mathbf{x})} \right\} \tag{55}$$

- *If $h$ is ZSP, then for all $\varepsilon \in (0, 1)$ such that $0 \notin g\left(B(\mathbf{x}, \varepsilon r)\right)$, the function $f := g + \beta h$ satisfies the $\varepsilon r$-CL inequality at $\mathbf{x}$, i.e., $4f(\mathbf{x}) < \varepsilon^2 r^2 \cdot \chi(f, \mathbf{x}, \varepsilon r)$, provided that $\beta \le \beta_{\max}(\varepsilon, \mathbf{x}, r)$ with*

$$\beta_{\max}(\varepsilon, \mathbf{x}, r) = \min\left\{ \frac{\sqrt{\underline{g}\cdot\chi(g,\mathbf{x},r)}}{4L}, \frac{\sqrt{\Delta} - B}{2C}, \frac{\sqrt{\Delta'} - \overline{g}\chi(h,\mathbf{x},\varepsilon r)}{2\overline{h}\chi(h,\mathbf{x},\varepsilon r)} \right\} \tag{56}$$

*where*

$$r' := \varepsilon r, \qquad \underline{g} := \inf_{\substack{\mathbf{y}\in B(\mathbf{x},r') \\ g(\mathbf{y})\neq 0}} g(\mathbf{y}); \qquad \overline{g} := \sup_{\mathbf{y}\in B(\mathbf{x},r')} g(\mathbf{y}), \qquad \overline{h} := \sup_{\mathbf{y}\in B(\mathbf{x},r')} h(\mathbf{y})$$

$$L := \sup_{\mathbf{y}\in B(\mathbf{x},r)} \sup_{\mathbf{z}\in\partial h(\mathbf{y})} \|\mathbf{z}\|_2 \in (0,\infty), \qquad \Delta' := \left(\overline{g}\chi(h,\mathbf{x},r')\right)^2 + 2\underline{g}\chi(g,\mathbf{x},r) \tag{57}$$

$$A := 4g(\mathbf{x}) \cdot \overline{g}, \qquad B := 4g(\mathbf{x}) \cdot \overline{h} + 4h(\mathbf{x}) \cdot \overline{g}, \qquad C := 4h(\mathbf{x}) \cdot \overline{h}, \qquad D := \frac{1}{2}r'^2 \cdot \chi(g,\mathbf{x},r) \cdot \underline{g}$$

$$\Delta := B^2 - 4C(A - D) = 16(g(\mathbf{x})\overline{h} - h(\mathbf{x})\overline{g})^2 + 8h(\mathbf{x})\overline{h} \cdot r'^2 \cdot \chi(g,\mathbf{x},r) \cdot \underline{g} \ge 0$$

*Proof.* Fix any $\varepsilon \in (0, 1)$ and define $r' := \varepsilon r$. We have $B(\mathbf{x}, r') \subset B(\mathbf{x}, r)$ since $r' < r$. Let $\mathbf{y} \in B(\mathbf{x}, r')$ with $f(\mathbf{y}) \neq 0$. Write $\nabla f(\mathbf{y}) = \nabla g(\mathbf{y}) + \beta\mathbf{z_y}$ for a certain $\mathbf{z_y} \in \partial h(\mathbf{y})$.

1. If $g(\mathbf{y}) \neq 0$[4], then since $\mathbf{y} \in B(\mathbf{x}, r)$ and $g(\mathbf{y}) \neq 0$, we have $\|\nabla g(\mathbf{y})\|_2^2 \ge g(\mathbf{y}) \cdot \chi(g, \mathbf{x}, r)$ from the CL constant for $g$. We also have $\|\mathbf{z_y}\|_2 \le L$ since $L = \sup_{\mathbf{y}'\in B(\mathbf{x},r)} \sup_{\mathbf{z}\in\partial h(\mathbf{y}')} \|\mathbf{z}\|_2$. Using Cauchy–Schwarz:

$$\langle \nabla g(\mathbf{y}), \mathbf{z_y} \rangle \ge -\|\nabla g(\mathbf{y})\|_2 \cdot \|\mathbf{z_y}\|_2 \ge -L\|\nabla g(\mathbf{y})\|_2 \tag{58}$$

So

$$\begin{aligned}
\|\nabla f(\mathbf{y})\|_2^2 &= \|\nabla g(\mathbf{y})\|_2^2 + 2\beta\langle\nabla g(\mathbf{y}), \mathbf{z_y}\rangle + \beta^2\|\mathbf{z_y}\|_2^2 \\
&\ge \|\nabla g(\mathbf{y})\|_2^2 + 2\beta\langle\nabla g(\mathbf{y}), \mathbf{z_y}\rangle \\
&\ge \|\nabla g(\mathbf{y})\|_2^2 - 2\beta L\|\nabla g(\mathbf{y})\|_2 \\
&= \|\nabla g(\mathbf{y})\|_2 \left(\|\nabla g(\mathbf{y})\|_2 - 2\beta L\right) \\
&\ge \sqrt{g(\mathbf{y})\cdot\chi(g,\mathbf{x},r)}\left(\sqrt{g(\mathbf{y})\cdot\chi(g,\mathbf{x},r)} - 2\beta L\right) \\
&= \left(\sqrt{g(\mathbf{y})\cdot\chi(g,\mathbf{x},r)} - \beta L\right)^2 - \beta^2 L^2
\end{aligned} \tag{59}$$

---

[4] $g(\mathbf{y}) \neq 0 \ \vee \ h(\mathbf{y}) \neq 0 \implies f(\mathbf{y}) \neq 0$, but the converse is false.

Since $\sqrt{\underline{g} \cdot \chi(g, \mathbf{x}, r)} \geq 4\beta L$ by the choice of $\beta$, we have $\sqrt{g(\mathbf{y}) \cdot \chi(g, \mathbf{x}, r)} \geq 4\beta L$. So $\sqrt{g(\mathbf{y}) \cdot \chi(g, \mathbf{x}, r)} - 2\beta L \geq 2\beta L > 0$ and $\sqrt{g(\mathbf{y}) \cdot \chi(g, \mathbf{x}, r)} - 2\beta L \geq \frac{1}{2}\sqrt{g(\mathbf{y}) \cdot \chi(g, \mathbf{x}, r)}$. This gives $\|\nabla f(\mathbf{y})\|_2^2 \geq \frac{1}{2}g(\mathbf{y}) \cdot \chi(g, \mathbf{x}, r)$.

Since $f(\mathbf{y}) = g(\mathbf{y}) + \beta h(\mathbf{y}) \leq \overline{g} + \beta \overline{h}$, we get $\frac{\|\nabla f(\mathbf{y})\|_2^2}{f(\mathbf{y})} \geq \frac{1}{2}\chi(g, \mathbf{x}, r) \cdot \frac{g(\mathbf{y})}{f(\mathbf{y})} \geq \frac{1}{2}\chi(g, \mathbf{x}, r) \cdot \theta(\beta)$ with

$$\theta(\beta) := \inf_{\substack{\mathbf{y} \in B(\mathbf{x}, r') \\ f(\mathbf{y}) \neq 0, \, g(\mathbf{y}) \neq 0}} \frac{g(\mathbf{y})}{f(\mathbf{y})} \geq \frac{\underline{g}}{\overline{g} + \beta \overline{h}} \tag{60}$$

Plugging $\chi(f, \mathbf{x}, r') \geq \frac{1}{2}\chi(g, \mathbf{x}, r) \cdot \theta(\beta)$ into the CL inequality:

$$4f(\mathbf{x}) = 4g(\mathbf{x}) + 4\beta h(\mathbf{x}) < \frac{1}{2}r'^2 \cdot \chi(g, \mathbf{x}, r) \cdot \frac{\underline{g}}{\overline{g} + \beta \overline{h}} \leq r'^2 \cdot \chi(f, \mathbf{x}, r') \tag{61}$$

Multiply both sides by $\overline{g} + \beta \overline{h}$, we obtain the inequality $(4g(\mathbf{x}) + 4\beta h(\mathbf{x}))(\overline{g} + \beta \overline{h}) < \frac{1}{2}r'^2 \cdot \chi(g, \mathbf{x}, r) \cdot \underline{g}$. Expand and rearrange, we get $C\beta^2 + B\beta + (A - D) < 0$ where $A, B, C, D$ are defined above. The discriminant of this quadratic is:

$$\begin{aligned}
\Delta &= B^2 - 4C(A - D) \\
&= 16(g(\mathbf{x})\overline{h} + h(\mathbf{x})\overline{g})^2 - 4 \cdot (4h(\mathbf{x})\overline{h}) \cdot \left(4g(\mathbf{x})\overline{g} - \frac{1}{2}r'^2 \cdot \chi(g, \mathbf{x}, r) \cdot \underline{g}\right) \\
&= 16g(\mathbf{x})^2\overline{h}^2 + 32g(\mathbf{x})h(\mathbf{x})\overline{h}\overline{g} + 16h(\mathbf{x})^2\overline{g}^2 - 64g(\mathbf{x})h(\mathbf{x})\overline{h}\overline{g} + 8h(\mathbf{x})\overline{h}r'^2 \cdot \chi(g, \mathbf{x}, r) \cdot \underline{g} \\
&= 16g(\mathbf{x})^2\overline{h}^2 - 32g(\mathbf{x})h(\mathbf{x})\overline{h}\overline{g} + 16h(\mathbf{x})^2\overline{g}^2 + 8h(\mathbf{x})\overline{h}r'^2 \cdot \chi(g, \mathbf{x}, r) \cdot \underline{g} \\
&= 16(g(\mathbf{x})\overline{h} - h(\mathbf{x})\overline{g})^2 + 8h(\mathbf{x})\overline{h}r'^2 \cdot \chi(g, \mathbf{x}, r) \cdot \underline{g} \geq 0
\end{aligned} \tag{62}$$

Therefore, the inequality admits a real solution, and the optimal range of $\beta$ is given by $\beta < \frac{-B + \sqrt{\Delta}}{2C}$. So $f$ satisfies the $r'$-CL inequality at $\mathbf{x}$ with respect to $g$, i.e., $4f(\mathbf{x}) < r'^2 \cdot \chi(f, \mathbf{x}, r', g)$, provided that $\beta < \min\left\{\frac{\sqrt{\underline{g} \cdot \chi(g, \mathbf{x}, r)}}{4L}, \frac{\sqrt{\Delta} - B}{2C}\right\}$. Moreover, if $g(\mathbf{y}) = 0 \implies f(\mathbf{y}) = 0$, then $f = g + \beta h$ satisfies the $r'$-CL inequality at $\mathbf{x}$ provided that $\beta$ satisfies this constraint, i.e., $4f(\mathbf{x}) < r'^2 \cdot \chi(f, \mathbf{x}, r')$.

2. If $g(\mathbf{y}) = 0$, then $f(\mathbf{y}) = \beta h(\mathbf{y}) \neq 0$ and $\mathbf{z}_y \neq 0$ (since $0 \in \partial h(\mathbf{y}) \implies h(\mathbf{y}) = 0$). So $\frac{\|\nabla f(\mathbf{y})\|_2^2}{f(\mathbf{y})} \geq \beta\frac{\|\mathbf{z}_y\|_2^2}{h(\mathbf{y})} \geq \beta\chi(h, \mathbf{x}, r')$. Plugging $\chi(f, \mathbf{x}, r') \geq \beta\chi(h, \mathbf{x}, r')$ into the CL inequality:

$$\begin{aligned}
4f(\mathbf{x}) = 4g(\mathbf{x}) + 4\beta h(\mathbf{x}) &< r'^2\beta \cdot \chi(h, \mathbf{x}, r') \leq r'^2 \cdot \chi(f, \mathbf{x}, r') \\
&\iff 4g(\mathbf{x}) < (r'^2 \cdot \chi(h, \mathbf{x}, r') - 4h(\mathbf{x}))\beta \text{ with } r'^2 \cdot \chi(h, \mathbf{x}, r') - 4h(\mathbf{x}) > 0 \\
&\iff \frac{4g(\mathbf{x})}{r'^2 \cdot \chi(h, \mathbf{x}, r') - 4h(\mathbf{x})} \begin{cases} < \beta & \text{if } 4h(\mathbf{x}) < r'^2\chi(h, \mathbf{x}, r') \\ > \beta & \text{if } 4h(\mathbf{x}) > r'^2\chi(h, \mathbf{x}, r') \end{cases}
\end{aligned} \tag{63}$$

The equation $4h(\mathbf{x}) < r'^2\chi(h, \mathbf{x}, r')$ means $h$ is $r'$-CL at $\mathbf{x}$. There exists $\mathbf{x}$ such that the $\ell_p$ norm $(p > 0)$ and nuclear norm $\ell_*$ satisfy this condition.

Assume $4h(\mathbf{x}) < r'^2\chi(h, \mathbf{x}, r')$. Since $\beta < \min\left\{\frac{\sqrt{\underline{g} \cdot \chi(g, \mathbf{x}, r)}}{4L}, \frac{\sqrt{\Delta} - B}{2C}\right\}$ (Equation (56)), the choice $\beta > \frac{4g(\mathbf{x})}{r'^2 \cdot \chi(h, \mathbf{x}, r') - 4h(\mathbf{x})}$ is possible if and only if

$$\frac{4g(\mathbf{x})}{r'^2 \cdot \chi(h, \mathbf{x}, r') - 4h(\mathbf{x})} < \min\left\{\frac{\sqrt{\underline{g} \cdot \chi(g, \mathbf{x}, r)}}{4L}, \frac{\sqrt{\Delta} - B}{2C}\right\} \tag{64}$$

Ensuring that this constraint is satisfied can be complicated. To avoid this, we will make sure that only the first case above is considered. We have

$$\frac{\|\nabla f(\mathbf{y})\|_2^2}{f(\mathbf{y})} \geq \begin{cases} \frac{1}{2}\frac{\chi(g, \mathbf{x}, r) \cdot \underline{g}}{\overline{g} + \beta \overline{h}} & \text{if } g(\mathbf{y}) \neq 0 \\ \beta\chi(h, \mathbf{x}, r') & \text{otherwise} \end{cases} \tag{65}$$

So if we impose $\frac{1}{2}\frac{\chi(g,\mathbf{x},r)\cdot g}{\overline{g}+\beta\overline{h}} \geq \beta\chi(h,\mathbf{x},r')$, we have $\frac{\|\nabla f(\mathbf{y})\|_2^2}{f(\mathbf{y})} \geq \frac{1}{2}\frac{\chi(g,\mathbf{x},r)\cdot g}{\overline{g}+\beta\overline{h}}$, and thus $\beta < \frac{-B+\sqrt{\Delta}}{2C}$. The constraint is equivalent to

$$
\begin{aligned}
&\overline{h}\chi(h,\mathbf{x},r')\cdot\beta^2 + \overline{g}\chi(h,\mathbf{x},r')\cdot\beta - \frac{1}{2}\underline{g}\chi(g,\mathbf{x},r) \leq 0 \\
&\iff \frac{-\sqrt{\Delta'}-\overline{g}\chi(h,\mathbf{x},r')}{2\overline{h}\chi(h,\mathbf{x},r')} \leq \beta \leq \frac{\sqrt{\Delta'}-\overline{g}\chi(h,\mathbf{x},r')}{2\overline{h}\chi(h,\mathbf{x},r')} \text{ with } \Delta' = \left(\overline{g}\chi(h,\mathbf{x},r')\right)^2 + 2\underline{g}\chi(g,\mathbf{x},r)
\end{aligned}
\tag{66}
$$

$\square$

## C.4. Proof of Theorem 3.1

Let $\mathbf{y}(\mathbf{a}) = \mathbf{X}\mathbf{a}$. We have $\mathbf{y}^* = \mathbf{X}\mathbf{a}^* + \boldsymbol{\xi}$, and want to minimize $f(\mathbf{a}) = g(\mathbf{a}) + \beta h(\mathbf{a})$ using gradient descent with learning rate $\alpha > 0$, where $h(\mathbf{a}) := \|\mathbf{a}\|_1$ and

$$
g(\mathbf{a}) := \frac{1}{2}\|\mathbf{y}(\mathbf{a}) - \mathbf{y}^*\|_2^2 = \frac{1}{2}\mathbf{a}^\top\mathbf{X}^\top\mathbf{X}\mathbf{a} - \left(\mathbf{X}^\top\mathbf{X}\mathbf{a}^* + \mathbf{X}^\top\boldsymbol{\xi}\right)^\top\mathbf{a} + \frac{1}{2}\|\mathbf{X}\mathbf{a}^* + \boldsymbol{\xi}\|_2^2 \tag{67}
$$

We write $F(\mathbf{a}) := G(\mathbf{a}) + \beta H(\mathbf{a})$ with $G(\mathbf{a}) := \nabla_{\mathbf{a}}g(\mathbf{a}) = \mathbf{X}^\top\mathbf{X}\mathbf{a} - \left(\mathbf{X}^\top\mathbf{X}\mathbf{a}^* + \mathbf{X}^\top\boldsymbol{\xi}\right)$ and $H(\mathbf{a}) \in \partial\|\mathbf{a}\|_1$ any subgradient of $\|\mathbf{a}\|_1$, that is $H(\mathbf{a})_i = \text{sign}(\mathbf{a}_i)$ for $\mathbf{a}_i \neq 0$, and any value in $[-1,+1]$ for $\mathbf{a}_i = 0$ [5]. Suppose we start at some $\mathbf{a}^{(1)}$. The subgradient update rule is

$$
\mathbf{a}^{(t+1)} = \mathbf{a}^{(t)} - \alpha F(\mathbf{a}^{(t)}) = \left(\mathbb{I}_n - \alpha\mathbf{X}^\top\mathbf{X}\right)\mathbf{a}^{(t)} + \alpha\left(\mathbf{X}^\top\mathbf{X}\mathbf{a}^* + \mathbf{X}^\top\boldsymbol{\xi}\right) - \beta\alpha H(\mathbf{a}^{(t)}) \quad \forall t > 1 \tag{68}
$$

To explain grokking in such a setting, we will look at the landscape of the iterate $\mathbf{a}^{(t)}$. Let $\mathbf{X} = \mathbf{U}\Sigma^{\frac{1}{2}}\mathbf{V}^\top$ under the SVD decomposition, with $\Sigma = \text{diag}(\sigma_k)_{k\in[r]}$, where $r = \text{rank}(\mathbf{X})$ and $\sigma_{\max} = \sigma_1 \geq \cdots\sigma_k \geq \sigma_{k+1}\cdots \geq \sigma_{\min} = \sigma_r > \sigma_{r+1} = \cdots = 0$. We assume by default the SVD to be compact, i.e., $\mathbf{U} \in \mathbb{R}^{N\times r}$ and $\mathbf{V} \in \mathbb{R}^{n\times r}$ have orthonormal columns, but we will make precision when we want it full, i.e., they also orthonormal rows, with that time $\mathbf{U} \in \mathbb{R}^{N\times N}$ and $\mathbf{V} \in \mathbb{R}^{n\times n}$. Using $\tilde{\Sigma} = \mathbb{I} - \alpha\Sigma$, the dynamics rewrites

$$
\mathbf{a}^{(t+1)} = \mathbf{V}\tilde{\Sigma}\mathbf{V}^\top\mathbf{a}^{(t)} + \alpha\left(\mathbf{V}\Sigma\mathbf{V}^\top\mathbf{a}^* + \mathbf{V}\Sigma^{\frac{1}{2}}\mathbf{U}^\top\boldsymbol{\xi}\right) - \alpha\beta H(\mathbf{a}^{(t)})
$$

We assume the step size $\alpha$ satisfies $0 < \alpha < \frac{2}{\sigma_{\max}}$. In fact, for the dynamical system to converge, we need $\lambda\left(\mathbb{I}_n - \alpha\mathbf{X}^\top\mathbf{X}\right) = 1 - \alpha\sigma\left(\mathbb{I}_n - \alpha\mathbf{X}^\top\mathbf{X}\right) \subset (-1,1)$, that is $0 < \alpha\sigma_k < 2 \quad \forall k \in [n]$. For all $p > 0$, let define $\rho_p := \left\|\mathbb{I}_n - \alpha\mathbf{X}^\top\mathbf{X}\right\|_{p\to p}$, so that $\rho_2 = \|\mathbb{I}_n - \alpha\Sigma\|_{2\to 2} = \max_{k\in[r]}|1 - \alpha\sigma_k| \in (0,1]$. We will show that for $\beta$ small enough, the update first moves near the least square solution of the problem, $\hat{\mathbf{a}} = \left(\mathbf{X}^\top\mathbf{X}\right)^\dagger\mathbf{X}^\top\mathbf{y}^* = \mathbf{V}\left(\mathbf{V}^\top\mathbf{a}^* + \Sigma^{-\frac{1}{2}}\mathbf{U}^\top\boldsymbol{\xi}\right)$ with $g(\hat{\mathbf{a}}) = \frac{1}{2}\boldsymbol{\xi}^\top\left(\mathbb{I}_N - \mathbf{U}\mathbf{U}^\top\right)\boldsymbol{\xi} \leq \frac{1}{2}\|\boldsymbol{\xi}\|_2^2$. Later in training, $H(\mathbf{a})$ dominates the update, leading to $\|\mathbf{a}^{(t)}\|_1 \approx \|\mathbf{a}^*\|_1$.

**Theorem C.29.** *Assume the learning rate, the regularization coefficent and the noise satisfy $0 < \alpha < \alpha_{\max} := \frac{2}{\sigma_{\max}(\mathbf{X}^\top\mathbf{X})}$, $0 < \beta < \frac{\sigma_{\max}(\mathbf{X}^\top\mathbf{X})}{\sqrt{n}}$ and $\|\mathbf{X}^\top\boldsymbol{\xi}\|_2 \leq \sqrt{C}\alpha\beta$, $C > 0$. Let $\rho_2 := \sigma_{\max}\left(\mathbb{I}_n - \alpha\mathbf{X}^\top\mathbf{X}\right)$. There exist $t_1 < \infty$ and a constant $C' > 0$ such that:*

$$
\|\mathbf{a}^{(t)} - \hat{\mathbf{a}}\|_2 \leq \frac{2\alpha\beta n^{1/2}}{1 - \rho_2} \text{ and } g(\mathbf{a}^{(t)}) \leq g(\hat{\mathbf{a}}) + \frac{2n\alpha^2\beta^2\sigma_{\max}^2(\mathbf{X})}{(1 - \rho_2)^2} \forall t \geq t_1, \ 2g(\hat{\mathbf{a}}) = \boldsymbol{\xi}^\top(\mathbb{I}_N - \mathbf{U}\mathbf{U}^\top)\boldsymbol{\xi} \leq \|\boldsymbol{\xi}\|_2^2
$$

$$
\forall\eta > 0, \quad \min_{t_1\leq t\leq t_2}\left(f(\mathbf{a}^{(t)}) - f(\mathbf{a}^*)\right) \leq \frac{(\eta + C'\alpha\beta)\beta}{2} \iff t_2 \geq t_1 + \Delta t(\eta,t_1), \quad \Delta t(\eta,t_1) := \frac{\|\mathbf{a}^{(t_1)} - \mathbf{a}^*\|_2^2}{\alpha\beta\eta} \tag{69}
$$

$$
\forall\eta > 0, \quad \min_{t_1\leq t\leq t_2}\left(\|\mathbf{a}^{(t)}\|_1 - \|\mathbf{a}^*\|_1\right) \leq \frac{\eta + (C + C')\alpha\beta}{2} \iff t_2 \geq t_1 + \Delta t(\eta,t_1)
$$

*Proof.* First, we observe that if $\beta$ is too high, the subgradient term $H(\mathbf{a})$ dominates early, and there is no convergence, i.e., no memorization nor generalization. In fact, if $\beta > \frac{\sigma_{\max}}{\sqrt{n}}$ then the $\ell_1$-term dominates the updates, causing the sequence $\mathbf{a}^{(t)}$ to exhibit oscillatory behavior without convergence to a minimizer of $f(\mathbf{a}) = g(\mathbf{a}) + \beta\|\mathbf{a}\|_1$ (Lemma C.33).

---
[5]For the experiments, we used $H(\mathbf{a}) = \text{sign}(\mathbf{a})$.

Let first evaluate $g(\hat{\mathbf{a}})$ and $\|\hat{\mathbf{a}} - \mathbf{a}^*\|_2^2$. We have

$$\hat{\mathbf{a}} = \left(\mathbf{X}^\top \mathbf{X}\right)^\dagger \mathbf{X}^\top \mathbf{y}^* = \mathbf{V}\Sigma^{-1}\mathbf{V}^\top \mathbf{V}\Sigma^{\frac{1}{2}}\mathbf{U}^\top \left(\mathbf{U}\Sigma^{\frac{1}{2}}\mathbf{V}^\top \mathbf{a}^* + \boldsymbol{\xi}\right) = \mathbf{V}\mathbf{V}^\top \mathbf{a}^* + \mathbf{V}\Sigma^{-\frac{1}{2}}\mathbf{U}^\top \boldsymbol{\xi} \tag{70}$$

So

$$\mathbf{X}\hat{\mathbf{a}} - \mathbf{y}^* = \mathbf{U}\Sigma^{\frac{1}{2}}\mathbf{V}^\top \mathbf{V}\mathbf{V}^\top \mathbf{a}^* + \mathbf{U}\Sigma^{\frac{1}{2}}\mathbf{V}^\top \mathbf{V}\Sigma^{-\frac{1}{2}}\mathbf{U}^\top \boldsymbol{\xi} - \mathbf{U}\Sigma^{\frac{1}{2}}\mathbf{V}^\top \mathbf{a}^* - \boldsymbol{\xi} = (\mathbb{I}_N - \mathbf{U}\mathbf{U}^\top)\boldsymbol{\xi} \tag{71}$$

and

$$\|\mathbf{X}\hat{\mathbf{a}} - \mathbf{y}^*\|_2^2 = \boldsymbol{\xi}^\top(\mathbb{I}_N - \mathbf{U}\mathbf{U}^\top)(\mathbb{I}_N - \mathbf{U}\mathbf{U}^\top)\boldsymbol{\xi} = \boldsymbol{\xi}^\top(\mathbb{I}_N - \mathbf{U}\mathbf{U}^\top)\boldsymbol{\xi} \le \|\boldsymbol{\xi}\|_2^2 \tag{72}$$

i.e. $g(\hat{\mathbf{a}}) - \frac{1}{2}\|\boldsymbol{\xi}\|_2^2 = -\frac{1}{2}\boldsymbol{\xi}^\top \mathbf{U}\mathbf{U}^\top \boldsymbol{\xi} = -\frac{1}{2}\|\mathbf{U}^\top \boldsymbol{\xi}\|_2^2$. This implies, assuming $\mathbb{E}[\boldsymbol{\xi}] = 0$ and $\mathrm{Cov}(\boldsymbol{\xi}) = \sigma_\xi^2 \mathbb{I}_N$,

$$\begin{aligned}
2\mathbb{E}_{\boldsymbol{\xi}} g(\hat{\mathbf{a}}) = \mathbb{E}_{\boldsymbol{\xi}}\|\mathbf{X}\hat{\mathbf{a}} - \mathbf{y}^*\|_2^2 = \mathbb{E}_{\boldsymbol{\xi}}\|(\mathbb{I}_N - \mathbf{U}\mathbf{U}^\top)\mathbf{y}^*\|_2^2 &= \mathbb{E}_{\boldsymbol{\xi}}\left[\boldsymbol{\xi}^\top(\mathbb{I}_N - \mathbf{U}\mathbf{U}^\top)\boldsymbol{\xi}\right] \\
&= \mathrm{tr}\left((\mathbb{I}_N - \mathbf{U}\mathbf{U}^\top)\mathrm{Cov}(\boldsymbol{\xi})\right) + (\mathbb{E}\boldsymbol{\xi})^\top(\mathbb{I}_N - \mathbf{U}\mathbf{U}^\top)(\mathbb{E}\boldsymbol{\xi}) \\
&= \sigma_\xi^2 \mathrm{tr}\left(\mathbb{I}_N - \mathbf{U}\mathbf{U}^\top\right) = \sigma_\xi^2(N - r)
\end{aligned} \tag{73}$$

Since

$$\hat{\mathbf{a}} - \mathbf{a}^* = \left(\mathbf{X}^\top \mathbf{X}\right)^\dagger \mathbf{X}^\top \mathbf{y}^* - \mathbf{a}^* = \mathbf{V}\left(\mathbf{V}^\top \mathbf{a}^* + \Sigma^{-\frac{1}{2}}\mathbf{U}^\top \boldsymbol{\xi}\right) - \mathbf{a}^* = (\mathbb{I}_n - \mathbf{V}\mathbf{V}^\top)\mathbf{a}^* + \mathbf{V}\Sigma^{-\frac{1}{2}}\mathbf{U}^\top \boldsymbol{\xi} \tag{74}$$

we have

$$\|\hat{\mathbf{a}} - \mathbf{a}^*\|_2^2 = \mathbf{a}^{*\top}\left(\mathbb{I}_n - \mathbf{V}\mathbf{V}^\top\right)\mathbf{a}^* + \boldsymbol{\xi}^\top \mathbf{U}\Sigma^{-1}\mathbf{U}^\top \boldsymbol{\xi} \tag{75}$$

Equations (72) and (75) show that the least square memorizes but does not generalize (for $N < n$). In particular, if $\mathbf{a}^*$ has a nonzero component orthogonal to the column space $\mathrm{Col}(\mathbf{V})$ of $\mathbf{V}$, then $\hat{\mathbf{a}}$ cannot perfectly generalize.

From now on, we fixed $p > 0$ such that $\rho_p < 1$, e.g. $p = 2$. Recall $G(\mathbf{a}) = \mathbf{X}^\top(\mathbf{y} - \mathbf{y}^*) = \left(\mathbf{X}^\top \mathbf{X}\right)\mathbf{a} - \left(\mathbf{X}^\top \mathbf{X}\mathbf{a}^* + \mathbf{X}^\top \boldsymbol{\xi}\right)$. Starting from the update rule $\mathbf{a}^{(t+1)} = \mathbf{a}^{(t)} - \alpha\left(G(\mathbf{a}^{(t)}) + \beta H(\mathbf{a}^{(t)})\right)$, we have $\mathbf{a}^{(t+1)} - \hat{\mathbf{a}} = \left(\mathbf{a}^{(t)} - \hat{\mathbf{a}}\right) - \alpha\left(G(\mathbf{a}^{(t)}) + \beta H(\mathbf{a}^{(t)})\right)$. Since $G(\hat{\mathbf{a}}) = 0$ and $G$ is linear, $G(\mathbf{a}^{(t)}) = \mathbf{X}^\top \mathbf{X}(\mathbf{a}^{(t)} - \hat{\mathbf{a}})$. Substituting this back,

$$\mathbf{a}^{(t+1)} - \hat{\mathbf{a}} = \left(\mathbf{a}^{(t)} - \hat{\mathbf{a}}\right) - \alpha\left(G(\mathbf{a}^{(t)}) + \beta H(\mathbf{a}^{(t)})\right)\left(\mathbb{I}_n - \alpha\mathbf{X}^\top \mathbf{X}\right)\left(\mathbf{a}^{(t)} - \hat{\mathbf{a}}\right) - \alpha\beta H(\mathbf{a}^{(t)}) \tag{76}$$

Taking the norm; applying triangle inequality and using[6] $\|H(\mathbf{a}^{(t)})\|_p \le n^{1/p}$ give

$$\|\mathbf{a}^{(t+1)} - \hat{\mathbf{a}}\|_p \le \rho_p \|\mathbf{a}^{(t)} - \hat{\mathbf{a}}\|_p + \alpha\beta n^{1/p} \tag{77}$$

Repeatedly applying the recurrence,

$$\begin{aligned}
\|\mathbf{a}^{(t)} - \hat{\mathbf{a}}\|_p &\le \rho_p^t \|\mathbf{a}^{(1)} - \hat{\mathbf{a}}\|_p + \alpha\beta n^{1/p}\left(1 + \rho_p + \cdots + \rho_p^{t-1}\right) \\
&= \rho_p^t \|\mathbf{a}^{(1)} - \hat{\mathbf{a}}\|_p + \alpha\beta n^{1/p}\frac{1 - \rho_p^t}{1 - \rho_p} \quad \text{for } \rho_p \ne 1 \\
&\le \rho_p^t \|\mathbf{a}^{(1)} - \hat{\mathbf{a}}\|_p + \frac{\alpha\beta n^{1/p}}{1 - \rho_p} \text{ for } \rho_p < 1
\end{aligned} \tag{78}$$

Define

$$t_1 := \left\lceil -\frac{\ln\left(1 + \frac{(1-\rho_p)\|\mathbf{a}^{(1)} - \hat{\mathbf{a}}\|_p}{\alpha\beta n^{1/p}}\right)}{\ln(\rho_p)} \right\rceil \tag{79}$$

The definition of $t_1$ ensures that for $t \ge t_1$,

$$\rho^t \|\mathbf{a}^{(1)} - \hat{\mathbf{a}}\|_p \le \alpha\beta n^{1/p}\frac{1 - \rho_p^t}{1 - \rho_p} \tag{80}$$

---

[6]Let $\mathbf{u} \in \partial\|\mathbf{a}\|_1$. Then $|\mathbf{u}_i| \le 1$ for all $i \in [n]$. So $\|\mathbf{u}\|_p = \left(\sum_{i=1}^n |\mathbf{u}_i|^p\right)^{1/p} \le n^{1/p}$.

Thus, using Equation (78), we have for all $t \geq t_1$,

$$\|\mathbf{a}^{(t)} - \hat{\mathbf{a}}\|_p \leq 2\alpha\beta n^{1/p}\frac{1 - \rho_p^t}{1 - \rho_p} \tag{81}$$

Using this, we derive

$$\begin{aligned}
\|\mathbf{X}\mathbf{a}^{(t)} - \mathbf{y}^*\|_p &= \|\mathbf{X}(\mathbf{a}^{(t)} - \hat{\mathbf{a}}) + (\mathbf{X}\hat{\mathbf{a}} - \mathbf{y}^*)\|_p \\
&\leq \|\mathbf{X}\|_{p \to p}\|\mathbf{a}^{(t)} - \hat{\mathbf{a}}\|_p + \|\mathbf{X}\hat{\mathbf{a}} - \mathbf{y}^*\|_p \\
&\leq 2\alpha\beta n^{1/p}\frac{1 - \rho_p^t}{1 - \rho_p}\|\mathbf{X}\|_{p \to p} + \|\mathbf{X}\hat{\mathbf{a}} - \mathbf{y}^*\|_p
\end{aligned} \tag{82}$$

and

$$\begin{aligned}
\|G(\mathbf{a}^{(t)})\|_p &= \|\mathbf{X}^\top\mathbf{X}(\mathbf{a}^{(t)} - \hat{\mathbf{a}})\|_p \\
&\leq \|\mathbf{X}^\top\mathbf{X}\|_{p \to p}\|\mathbf{a}^{(t)} - \hat{\mathbf{a}}\|_p \\
&\leq 2\alpha\beta n^{1/p}\|\mathbf{X}^\top\mathbf{X}\|_{p \to p}\frac{1 - \rho_p^t}{1 - \rho_p} \\
&\leq \frac{2\alpha\beta n^{1/p}}{1 - \rho_p}\|\mathbf{X}^\top\mathbf{X}\|_{p \to p} \\
&\propto 2\sqrt{n}\beta\sigma_{\max}(\mathbf{X}^\top\mathbf{X}) \text{ for } p = 2 \text{ since } 1 - \rho_2 \propto \alpha
\end{aligned} \tag{83}$$

and

$$\begin{aligned}
\|F(\mathbf{a}^{(t)})\|_2 &= \|G(\mathbf{a}^{(t)}) + \beta H(\mathbf{a}^{(t)})\|_2 \\
&\leq \|G(\mathbf{a}^{(t)})\|_2 + \beta\|H(\mathbf{a}^{(t)})\|_2 \\
&\propto \frac{2\alpha\beta\sqrt{n}}{1 - \rho_2}\|\mathbf{X}^\top\mathbf{X}\|_{2 \to 2} + \beta\sqrt{n} \\
&= \sqrt{n}\left(2\sigma_{\max}\left(\mathbf{X}^\top\mathbf{X}\right) + 1\right)\beta
\end{aligned} \tag{84}$$

Also, since $\mathbf{X}(\mathbf{a}^{(t)} - \hat{\mathbf{a}}) \in \text{Col}(\mathbf{X}) = \text{Col}(\mathbf{U})$ and $\mathbf{X}\hat{\mathbf{a}} - \mathbf{y}^* = (\mathbb{I}_N - \mathbf{U}\mathbf{U}^\top)\boldsymbol{\xi} \in \text{Col}(\mathbf{U})^\perp$, we have for $t \geq t_1$,

$$\begin{aligned}
g(\mathbf{a}^{(t)}) - g(\hat{\mathbf{a}}) &= \frac{1}{2}\|\mathbf{X}\mathbf{a}^{(t)} - \mathbf{y}^*\|_2^2 - \frac{1}{2}\|\mathbf{X}\hat{\mathbf{a}} - \mathbf{y}^*\|_2^2 \\
&= \frac{1}{2}\|\mathbf{X}(\mathbf{a}^{(t)} - \hat{\mathbf{a}}) + (\mathbf{X}\hat{\mathbf{a}} - \mathbf{y}^*)\|_2^2 - \frac{1}{2}\|\mathbf{X}\hat{\mathbf{a}} - \mathbf{y}^*\|_2^2 \\
&= \frac{1}{2}\|\mathbf{X}(\mathbf{a}^{(t)} - \hat{\mathbf{a}})\|_2^2 + \frac{1}{2}\|\mathbf{X}\hat{\mathbf{a}} - \mathbf{y}^*\|_2^2 - \frac{1}{2}\|\mathbf{X}\hat{\mathbf{a}} - \mathbf{y}^*\|_2^2 \\
&= \frac{1}{2}\|\mathbf{X}(\mathbf{a}^{(t)} - \hat{\mathbf{a}})\|_2^2
\end{aligned} \tag{85}$$

So

$$g(\mathbf{a}^{(t)}) - g(\hat{\mathbf{a}}) \leq \frac{1}{2}\|\mathbf{X}\|_{2 \to 2}^2\|\mathbf{a}^{(t)} - \hat{\mathbf{a}}\|_2^2 \leq \frac{2n\alpha^2\beta^2\sigma_{\max}^2(\mathbf{X})}{(1 - \rho_2)^2} \propto 2n\beta^2\sigma_{\max}(\mathbf{X}^\top\mathbf{X}) \tag{86}$$

All this shows that after $t_1$, the iterate $\mathbf{a}^{(t)}$, its error $g(\mathbf{a}^{(t)})$, and the associated gradient $G(\mathbf{a}^{(t)})$ behave respectively like $\hat{\mathbf{a}}$, $g(\hat{\mathbf{a}})$, and $G(\hat{\mathbf{a}}) = 0$ up to an error of order $\mathcal{O}(\beta)$. Note that we assume $0 < \beta\sqrt{n} \ll \sigma_{\max}(\mathbf{X}^\top\mathbf{X})$. So, the gradient of $g$ can be made much smaller than the subgradient term after $t_1$ by choosing $\beta$ sufficiently small. After time $t_1$, the contribution of the gradient $G$ to the update of $\mathbf{a}_i^{(t)}$ is dominated by the $\ell_1$–regularization term up to an error of order $\mathcal{O}(\beta)$. Specifically for all $t \gg t_1$, the update rule approximates $\mathbf{a}_i^{(t+1)} \approx \mathbf{a}_i^{(t)} - \alpha\beta H(\mathbf{a}_i^{(t)})$. If $|\mathbf{a}_i^{(t_1)}| > |\mathbf{a}_i^*|$, then $H(\mathbf{a}_i^{(t)}) = H(\mathbf{a}_i^{(t)} - \mathbf{a}_i^*)$ (Lemma C.32), and so $\mathbf{a}_i^{(t+1)} - \mathbf{a}_i^* = \mathbf{a}_i^{(t)} - \mathbf{a}_i^* - \alpha\beta H(\mathbf{a}_i^{(t)} - \mathbf{a}_i^*)$ for all $t \geq t_1$. By Lemma C.31, this lead to $|\mathbf{a}_i^{(t)} - \mathbf{a}_i^*| \leq \alpha\beta$ for (and only for) $t \geq t_1 + \left\lfloor \frac{|\mathbf{a}_i^{(t_1)} - \mathbf{a}_i^*|}{\alpha\beta} \right\rfloor$. This suggest that we may have $\|\mathbf{a}^{(t)} - \mathbf{a}^*\|_\infty = \mathcal{O}(\alpha\beta)$

for (and only for) $t \geq t_1 + \left\lfloor \frac{\|\mathbf{a}^{(t_1)} - \mathbf{a}^*\|_\infty}{\alpha\beta} \right\rfloor$. In fact, considering the canonical subgradient $H(\mathbf{a}) = \text{sign}(\mathbf{a})$, we have $\mathbf{a}^{t+1} - \mathbf{a}^* = \mathbf{a}^{(t)} - \mathbf{a}^* - \alpha G(\mathbf{a}^{(t)}) - \alpha\beta \, \text{sign}(\mathbf{a}^{(t)}) = C_{\alpha\beta}(\mathbf{a}^{(t)} - \mathbf{a}^* - \alpha G(\mathbf{a}^{(t)}))$ with

$$C_\gamma(z) = \begin{cases} z - \gamma & \text{if } z > \gamma \\ z + \gamma & \text{if } z < \gamma \\ \in [z - \gamma, z + \gamma] & \text{if } -\gamma \leq z \leq \gamma \end{cases} \in z \pm \gamma$$

a coordinate shrink operator, since $|C_\gamma(z)| = \begin{cases} |z| - \gamma & \text{if } |z| > \gamma \\ \in [0, \gamma] & \text{if } |z| \leq \gamma \end{cases} \leq \max(|z| - \gamma, \gamma)$. So the goal of $H(\mathbf{a})$ is to reduce the components of $\mathbf{a}^{(t)} - \mathbf{a}^*$ at each iteration until they are all in $[-\alpha\beta, \alpha\beta]$. We now prove this intuition below using Lemma C.12, which proves the second part of the theorem.

We have $g(\mathbf{a}^{(t)}) = \frac{1}{2}\|\mathbf{X}\mathbf{a}^{(t)} - \mathbf{y}^*\|_2^2$, $h(\mathbf{a}^{(t)}) = \|\mathbf{a}^{(t)}\|_1$, $f(\mathbf{a}^{(t)}) = g(\mathbf{a}^{(t)}) + \beta h(\mathbf{a}^{(t)})$ and $f(\mathbf{a}^*) = \beta\|\mathbf{a}^*\|_1 + \frac{1}{2}\|\boldsymbol{\xi}\|_2^2$. We also have $\Theta_f = \{\mathbf{a}^*\}$ and $\Theta_g = \{\mathbf{a} \mid \mathbf{X}(\mathbf{a} - \mathbf{a}^*) = \boldsymbol{\xi}\}$. Applying Lemma C.12, we get

$$\min_{t_1 \leq t \leq t_2} \left( f(\mathbf{a}^{(t)}) - f(\mathbf{a}^*) \right) \leq \frac{\|\mathbf{a}^{(t_1)} - \mathbf{a}^*\|_2^2 + (t_2 - t_1)\alpha^2 \max_{t_1 \leq t \leq t_2} \|F(\mathbf{a}^{(t)})\|_2^2}{2\alpha(t_2 - t_1)} \quad \forall t_2 \geq t_1 \tag{87}$$

Using $\|F(\mathbf{a}^{(t)})\|_2 = \mathcal{O}(\beta) \; \forall t \geq t_1$ (Equation (84)) we get from Theorem C.13 that there exists $C' > 0$,

$$\min_{t_1 \leq t \leq t_2} \left( f(\mathbf{a}^{(t)}) - f(\mathbf{a}^*) \right) \leq \frac{(\eta + C'\alpha\beta)\beta}{2} \iff t_2 \geq t_1 + \frac{\|\mathbf{a}^{(t_1)} - \mathbf{a}^*\|_2^2}{\alpha\beta\eta} \tag{88}$$

And it $\Theta_f \cap \Theta_g \neq \emptyset$, i.e. $\boldsymbol{\xi} = 0$,

$$\min_{t_1 \leq t \leq t_2} \left( \|\mathbf{a}^{(t)}\|_1 - \|\mathbf{a}^*\|_1 \right) \leq \frac{\eta + C'\alpha\beta}{2} \iff t_2 \geq t_1 + \frac{\|\mathbf{a}^{(t_1)} - \mathbf{a}^*\|_2^2}{\alpha\beta\eta} \tag{89}$$

We now prove this result in a general case $\boldsymbol{\xi} \neq 0$. Let $R(\mathbf{a}) := (\mathbf{a} - \mathbf{a}^*)^\top G(\mathbf{a})$. We have from Lemma C.12,

$$\min_{t_1 \leq t \leq t_2} \left[ R(\mathbf{a}^{(t)}) + \beta \left( \|\mathbf{a}^{(t)}\|_1 - \|\mathbf{a}^*\|_1 \right) \right] \leq \min_{t_1 \leq t \leq t_2} \left[ \left( g(\mathbf{a}^{(t)}) - g(\mathbf{a}^*) \right) + \beta \left( \|\mathbf{a}^{(t)}\|_1 - \|\mathbf{a}^*\|_1 \right) \right]$$
$$\leq \min_{t_1 \leq t \leq t_2} \left( f(\mathbf{a}^{(t)}) - f(\mathbf{a}^*) \right) \quad \forall t_2 \geq t_1 \tag{90}$$

So

$$\beta \min_{t_1 \leq t \leq t_2} \left( \|\mathbf{a}^{(t)}\|_1 - \|\mathbf{a}^*\|_1 \right) \leq \min_{t_1 \leq t \leq t_2} \left( f(\mathbf{a}^{(t)}) - f(\mathbf{a}^*) \right) - \left( \min_{t_1 \leq t \leq t_2} g(\mathbf{a}^{(t)}) - \frac{1}{2}\|\boldsymbol{\xi}\|_2^2 \right) \quad \forall t_2 \geq t_1 \tag{91}$$

Since

$$g(\mathbf{a}^{(t)}) - \frac{1}{2}\|\boldsymbol{\xi}\|_2^2 = g(\hat{\mathbf{a}}) + \frac{1}{2}\|\mathbf{X}(\mathbf{a}^{(t)} - \hat{\mathbf{a}})\|_2^2 - \frac{1}{2}\|\boldsymbol{\xi}\|_2^2 = \frac{1}{2}\|\mathbf{X}(\mathbf{a}^{(t)} - \hat{\mathbf{a}})\|_2^2 - \frac{1}{2}\|\mathbf{U}^\top\boldsymbol{\xi}\|_2^2 \geq -\frac{1}{2}\|\mathbf{U}^\top\boldsymbol{\xi}\|_2^2 \tag{92}$$

We obtain

$$\beta \min_{t_1 \leq t \leq t_2} \left( \|\mathbf{a}^{(t)}\|_1 - \|\mathbf{a}^*\|_1 \right) \leq \min_{t_1 \leq t \leq t_2} \left( f(\mathbf{a}^{(t)}) - f(\mathbf{a}^*) \right) + \frac{1}{2}\|\mathbf{X}^\top\boldsymbol{\xi}\|_2^2 \quad \forall t_2 \geq t_1$$
$$\leq \frac{(\eta + C'\alpha\beta)\beta}{2} + \frac{C\alpha\beta^2}{2} \iff t_2 \geq t_1 + \Delta t(\eta, t_1) \text{ since } \|\mathbf{X}^\top\boldsymbol{\xi}\|_2 \leq \sqrt{C}\alpha\beta \tag{93}$$
$$= \frac{(\eta + (C + C')\alpha\beta)\beta}{2} \iff t_2 \geq t_1 + \Delta t(\eta, t_1)$$

$\square$

**Lemma C.30.** *Given $\alpha > 0$ and $a^{(1)} \in \mathbb{R}$, let $a^{(t+1)} = a^{(t)} - \alpha H(a^{(t)})$ for all $t \geq 1$, where $H(a) \in \partial|a|$.*

1. *A point $a$ is stationary for this dynamical system if and only if $|a| \leq \alpha$.*

2. *We have $|a^{(t)}| \leq \alpha$ if and only if $t > \lfloor \frac{|a^{(1)}|}{\alpha} \rfloor$.*

3. *In particular, for $h(a) = \text{sign}(a) \, \forall a \in \mathbb{R}$, if $a^{(1)}/\alpha \in \mathbb{Z}$, then $a^{(t)} = 0$ for all $t > \lfloor \frac{|a^{(1)}|}{\alpha} \rfloor$.*

*Proof.* Let first consider the simple case $H(a) = \text{sign}(a)$, so that $a^{(t+1)} = a^{(t)} - \alpha \text{sign}(a^{(t)})$.

- If $a^{(t)} \in \{0, \alpha, -\alpha\}$, then $a^{(t+\Delta)} = 0$ for all $\Delta > 0$.

- If $a^{(t)} \in (0, \alpha)$, then $a^{(t+1)} = a^{(t)} - \alpha \in (-\alpha, 0)$, and $a^{(t+2)} = a^{(t+1)} + \alpha = a^{(t)} \in (0, \alpha)$, and so on.

- If $a^{(t)} \in (-\alpha, 0)$, then $a^{(t+1)} = a^{(t)} + \alpha \in (0, \alpha)$, and $a^{(t+2)} = a^{(t+1)} - \alpha = a^{(t)} \in (-\alpha, 0)$, and so on.

- If $a^{(t)} > \alpha$ (resp. $a^{(t)} < -\alpha$), it will be decreased (resp. increase) by $\alpha$ until $a^{(t)} \in (0, \alpha]$ (resp. $a^{(t)} \in [-\alpha, 0)$), and we get back to the previous cases. In that case, $|a^{(t+1)}| = |a^{(t)}| - \alpha = |a^{(1)}| - t\alpha \leq \alpha \implies t + 1 \geq \frac{|a^{(1)}|}{\alpha}$.

Now consider the general dynamic $a^{(t+1)} = a^{(t)} - \alpha H(a^{(t)})$. If $a^{(1)} \neq 0$ (the case $a^{(1)} = 0$ is trivial), then the dynamic is $a^{(t+1)} = a^{(t)} - \alpha \text{sign}(a^{(t)})$ as long as $|a^{(t)}| \geq \alpha$, after which it will just oscillate in the ball $\{a, |a| \leq \alpha\}$ indefinitely. In fact, a fixed point $a$ must satisfy $a = a - \alpha H(a)$; i.e. $H(a) = 0$. The only case where $0 \in \partial |a|$ is $a = 0$ or when it lies in the interval where the subgradient can be 0. However, for any $a$ such that $|a| \leq \alpha$, it is possible to choose $H(a)$ (for instance, $H(a) = a/\alpha$) such that $a = a - \alpha H(a)$, making $a$ a fixed point. Conversely, if $|a| > \alpha$, then $|H(a)| = 1$ and $|a - \alpha H(a)| = ||a| - \alpha| > 0$, so $a$ is not a fixed point. $\qquad \square$

**Lemma C.31.** *Given $\alpha > 0$ and $\mathbf{a}^{(1)} \in \mathbb{R}^n$, let $\mathbf{a}^{(t+1)} = \mathbf{a}^{(t)} - \alpha H(\mathbf{a}^{(t)})$ for all $t \geq 1$, where $H(\mathbf{a}) \in \partial \|\mathbf{a}\|_1$.*

1. *A point $\mathbf{a}$ is stationary for this dynamical system if and only if $\|\mathbf{a}\|_\infty \leq \alpha$.*

2. *We have $\|\mathbf{a}^{(t)}\|_\infty \leq \alpha$ if and only if $t > \lfloor \frac{\|\mathbf{a}^{(1)}\|_\infty}{\alpha} \rfloor$.*

3. *In particular, for $h(\mathbf{a}) = \text{sign}(\mathbf{a}) \, \forall \mathbf{a} \in \mathbb{R}^n$, we have $\|\mathbf{a}^{(t)}\|_0 = \left| \left\{ i \mid \mathbf{a}_i^{(1)}/\alpha \in \mathbb{Z} \right\} \right|$ for all $t > \lfloor \frac{\|\mathbf{a}^{(1)}\|_\infty}{\alpha} \rfloor$.*

*Proof.* The proof is immediate by applying the Lemma C.30 coordinate-wise. $\qquad \square$

**Lemma C.32.** *Let $a, a^* \in \mathbb{R}$ with $a \neq 0$. If $|a| > |a^*|$, then $\text{sign}(a - a^*) = \text{sign}(a)$.*

*Proof.* We have $|a^*| < |a| \iff -|a| < a^* < |a| \iff -|a| - a^* < 0 < |a| - a^*$. This implies $a - a^* < 0$ for $a < 0$ and $0 < a - a^*$ for $0 < a$. $\qquad \square$

**Lemma C.33.** *Let $f(\theta) = g(\theta) + \beta h(\theta)$ be a function from $\mathbb{R}^n$ to $\mathbb{R}$, where $g$ is a differentiable function and $h$ is a sub-differentiable function. Consider the subgradient descent update $\theta^{(t+1)} = \theta^{(t)} - \alpha \left( \nabla g(\theta^{(t)}) + \beta H(\theta^{(t)}) \right)$ with a fixed small step size $\alpha > 0$, where $H(\theta^{(t)}) \in \partial h(\theta)$. If $\beta \|H(\theta^{(1)})\|_2 \gg \|\nabla g(\theta^{(1)})\|_2$ then the regularization term dominates the updates, causing the sequence $\{\theta^{(t)}\}_{t>1}$ to exhibit oscillatory behavior without convergence to a minimizer of $f$. The condition $\beta \|H(\theta^{(1)})\|_2 \gg \|\nabla g(\theta^{(1)})\|_2$ writes $\beta \sqrt{n} \gg \|\nabla g(\theta^{(1)})\|_2$ for $\ell_1$ regularization, $h(\theta) = \|\theta\|_1 \, \forall \theta \in \mathbb{R}^n$; and $\beta \sqrt{\min(n_1, n_2)} \gg \|\nabla g(\theta^{(1)})\|_2$ for $\ell_*$ regularization, $h(\theta) = \|\theta\|_* \, \forall \theta \in \mathbb{R}^{n_1 \times n_2}$.*

*Proof Sketch.* Given that $\|\nabla g(\theta^{(t)})\|_2 \approx \|\nabla g(\theta^{(1)})\|_2$ and $H(\theta^{(t)}) \approx H(\theta^{(1)})$ at the beginning of training, if $\beta \|H(\theta^{(1)})\|_2 \gg \|\nabla g(\theta^{(1)})\|_2$, then $\beta \|H(\theta^{(t)})\|_2 \gg \|\nabla g(\theta^{(t)})\|_2$. This inequality implies that the regularization term dominates the update, $\theta^{(t+1)} \approx \theta^{(t)} - \alpha\beta H(\theta^{(t)})$, with the influence of $\nabla g(\theta^{(t)})$ becoming negligible. Consequently, the iterates do not converge to a stable minimizer of $f$, and the training and test error metrics oscillate, remaining above some suboptimal value.

For $\ell_1$ regularization, $h(\theta) = \|\theta\|_1 \, \forall \theta \in \mathbb{R}^n$, given that $\|h(\theta^{(t)})\|_2 \approx \sqrt{n}$ at the beginning of training, if $\beta \sqrt{n} \gg \|\nabla g(\theta^{(1)})\|_2$, then that the update becomes dominated by the $\ell_1$-term. Because $H(\theta^{(t)})$ reflects the sign of $\theta^{(t)}$, the update effectively pushes the iterates in a direction that primarily depends on sign changes rather than the curvature of $g$ (Lemma C.31). This leads to overshooting and sign flipping in each coordinate, resulting in oscillations.

For $\ell_*$ regularization, $h(\theta) = \|\theta\|_* \, \forall \theta \in \mathbb{R}^{n_1 \times n_2}$, the subgradient $H(\theta^{(t)})$ of $\|\theta^{(t)}\|_*$ satisfy $\|H(\theta^{(t)})\|_* \approx \sqrt{\min(n_1, n_2)}$ at the beginning of training (full rank matrix), so $\|H(\theta^{(t)})\|_F \geq \|H(\theta^{(t)})\|_* / \text{rank}(H(\theta^{(t)})) \approx \sqrt{\min(n_1, n_2)} / \min(n_1, n_2) = \sqrt{\min(n_1, n_2)}$. If $\beta \sqrt{\min(n_1, n_2)} \gg \|\nabla g(\theta^{(1)})\|_2$, then the update is dominated by the $\ell_*$-term, making the iterates swing sharply depending on the current singular-vector configuration (Lemma C.42). $\qquad \square$

## C.5. Proof of Theorem 3.3

For a vector $\mathbf{u} \in \mathbb{R}^n$ and a set $S \subset [n]$, we let $\mathbf{u}_S = [\mathbf{u}_i]_{i \in S} \in \mathbb{R}^{|S|}$; and $\bar{S} = [n] \backslash S$.

**Definition C.34** (Null Space Property). A matrix $\mathbf{A} \in \mathbb{R}^{m \times n}$ is said to satisfy the null space property relative to a set $S \subset [n]$ if $\|\mathbf{u}_S\|_1 < \|\mathbf{u}_{\bar{S}}\|_1$ for all non zero vector $\mathbf{u} \in \mathbb{R}^n$ in $\ker \mathbf{A}$. It is said to satisfy the null space property of order $s \in \mathbb{N}$ if it satisfies the null space property relative to any set $S \subset [n]$ with $|S| \leq s$.

If we add $\|\mathbf{u}_S\|_1$ on both side of $\|\mathbf{u}_S\|_1 < \|\mathbf{u}_{\bar{S}}\|_1$ we obtain the equivalent formulation $2\|\mathbf{u}_S\|_1 < \|\mathbf{u}\|_1$. On the other hand, by choosing $S$ as an index set of $s$ largest (in absolute value) entries of $\mathbf{u}$ and this time by adding $\|\mathbf{u}_{\bar{S}}\|_1$ to both sides of the inequality, the null space property of order $s$ reads $\|\mathbf{u}\|_1 < 2\sigma_1(\mathbf{u})$, where $\sigma_p(\mathbf{u}) = \inf_{\|\mathbf{v}\|_0 \leq s} \|\mathbf{u} - \mathbf{v}\|_p$. In fact, $\sigma_p(\mathbf{u})$ is achieve (but not only) at $\mathbf{v} = H_s(\mathbf{u})$, with $H_s$ the hard thresholding operator (it keeps the $s$ largest entries of $\mathbf{u}$ in absolute value, and set the remaining to 0). We have $\|\mathbf{u} - H_s(\mathbf{u})\|_p = \|\mathbf{u}_{\bar{S}}\|_p$ with $S$ the support of $H_s(\mathbf{u})$.

In theory, sparse recovery methods are designed to recover exactly sparse vectors. However, in more realistic scenarios, the vectors we aim to recover are not exactly sparse but can be well-approximated by sparse vectors. In other words, while the true signal may not have strictly zero entries outside a small support, most of its energy or information content is concentrated on a small number of components. This motivates the need for recovery guarantees that extend beyond exact sparsity. In such settings, we no longer expect perfect recovery of the original vector. Instead, we aim to reconstruct a vector $\mathbf{a}$ such that the error between $\mathbf{a}$ and the true signal $\mathbf{a}^*$ is controlled by how well the true signal can be approximated by an $s$-sparse vector. In other words, the reconstruction error should scale with the sparsity defect $\sigma_s(\mathbf{a}^*)$, which is typically measured by the distance from the signal to the set of exactly $s$-sparse vectors. A reconstruction scheme that provides such guarantees is said to be stable with respect to the sparsity defect (Foucart & Rauhut, 2013).

**Definition C.35** (Stable Null Space Property). A matrix $\mathbf{A} \in \mathbb{R}^{m \times n}$ is said to satisfy the stable null space property with constant $\rho \in (0, 1)$ relative to a set $S \subset [n]$ if $\|\mathbf{u}_S\|_1 < \rho\|\mathbf{u}_{\bar{S}}\|_1$ for all $\mathbf{u} \in \ker \mathbf{A}$. It is said to satisfy the robust null space property of order $s \in \mathbb{N}$ with constant $\rho \in (0, 1)$ if it satisfies the robust null space property with constant $\rho$ relative to any set $S \subset [n]$ with $|S| \leq s$.

**Theorem C.36** (Theorem 4.14 (Foucart & Rauhut, 2013)). *The matrix $\mathbf{A} \in \mathbb{R}^{m \times n}$ satisfies the stable null space property with constant $\rho \in (0, 1)$ relative to a set $S \subset [n]$ if and only if $\|\mathbf{v} - \mathbf{u}\|_1 \leq \frac{1+\rho}{1-\rho}(\|\mathbf{v}\|_1 - \|\mathbf{u}\|_1 + 2\|\mathbf{u}_{\bar{S}}\|_1)$ for all vector $\mathbf{u}, \mathbf{v} \in \mathbb{R}^n$ with $\mathbf{A}\mathbf{u} = \mathbf{A}\mathbf{v}$.*

If we apply this to the noiseless version of the problem of minimizing $\|\mathbf{a}\|_1$ subject to $\|\mathbf{X}\mathbf{a} - \mathbf{y}^*\|_2 \leq \epsilon$, with $\mathbf{a}$ the solution return by, say basis pursuit, and $S$ the support of $H_s(\mathbf{a})$, the we get $\|\mathbf{a} - \mathbf{a}^*\|_1 \leq \frac{2(1+\rho)}{1-\rho}\sigma_s(\mathbf{a})$ since $\|\mathbf{a}^*\|_1 \leq \|\mathbf{a}\|_1$ and $\|\mathbf{a}_{\bar{S}}\|_1 = \sigma_s(\mathbf{a})$.

In practical applications, it is fundamentally unrealistic to assume that we can observe or measure a signal $\mathbf{a}^*$ with perfect, infinite precision. All real-world measurements are subject to various sources of error, such as sensor inaccuracies, quantization effects, environmental noise, or other imperfections in the acquisition process. As a result, the measurement vector $\mathbf{y}^*$ that we actually obtain is not exactly equal to the ideal linear measurement $\mathbf{X}\mathbf{a}^*$, but only an approximation of it. This deviation is typically modeled as an additive noise term so that we only know that the measurement error is bounded, $\|\mathbf{y}^* - \mathbf{X}\mathbf{a}^*\|_2 = \|\boldsymbol{\xi}\|_2 \leq \epsilon$ for some known or estimated noise level $\epsilon > 0$. In this noisy setting, an effective reconstruction scheme should not aim to recover $\mathbf{a}^*$ exactly since doing so is impossible, but instead produce an estimate $\mathbf{a}$ that is close to the true signal $\mathbf{a}^*$, with an error that is controlled by the magnitude of the measurement error $\epsilon$. That is, small changes or perturbations in the observed measurements should only lead to small changes in the reconstructed signal. This desirable behavior is known as the robustness of the reconstruction scheme with respect to measurement error.

**Definition C.37** (Robust Null Space Property). A matrix $\mathbf{A} \in \mathbb{R}^{m \times n}$ is said to satisfy the robust null space property (with respect to $\|\cdot\|$) with constant $\rho \in (0, 1)$ and $\tau > 0$ relative to a set $S \subset [n]$ if $\|\mathbf{u}_S\|_1 < \rho\|\mathbf{u}_{\bar{S}}\|_1 + \tau\|\mathbf{A}\mathbf{u}\|$ for all $\mathbf{u} \in \mathbb{R}^n$. It is said to satisfy the robust null space property of order $s \in \mathbb{N}$ with constant $\rho \in (0, 1)$ and $\tau > 0$ if it satisfies the robust null space property with constant $\rho$ and $\tau$ relative to any set $S \subset [n]$ with $|S| \leq s$.

**Theorem C.38** (Theorem 4.20 (Foucart & Rauhut, 2013)). *The matrix $\mathbf{A} \in \mathbb{R}^{m \times n}$ satisfies the robust null space property with constant $\rho \in (0, 1)$ and $\tau > 0$ relative to a set $S \subset [n]$ if and only if $\|\mathbf{v} - \mathbf{u}\|_1 \leq \frac{1+\rho}{1-\rho}(\|\mathbf{v}\|_1 - \|\mathbf{u}\|_1 + 2\|\mathbf{u}_{\bar{S}}\|_1) + \frac{2\tau}{1-\rho}\|\mathbf{A}(\mathbf{v} - \mathbf{u})\|$ for all vector $\mathbf{u}, \mathbf{v} \in \mathbb{R}^n$.*

**Theorem C.39.** *Assume the matrix $\mathbf{X} \in \mathbb{R}^{N \times n}$ satisfies the robust null space property with constant $\rho \in (0, 1)$ and $\tau > 0$ relative to the support of $\mathbf{a}^*$. Then, under the same condition as in Theorem C.29 on $\alpha$, $\beta$ and $\boldsymbol{\xi}$; i.e. $0 < \alpha\sigma_{\max}(\mathbf{X}^\top \mathbf{X}) < 2$,*

$0 < \beta\sqrt{n} < \sigma_{\max}(\mathbf{X}^\top\mathbf{X})$ *and* $\|\mathbf{X}^\top\boldsymbol{\xi}\|_2 \leq \sqrt{C\alpha}\beta$ *with* $C > 0$; *there exists* $C' > 0$ *such that for all* $\eta > 0$,

$$\min_{t_1 \leq t \leq t_2} \|\mathbf{a}^{(t)} - \mathbf{a}^*\|_1 \leq C_1\eta + C_2\alpha\beta + C_3\|\boldsymbol{\xi}\|_2 \iff t_2 \geq t_1 + \Delta t(\eta, t_1), \quad \Delta t(\eta, t_1) := \frac{\|\mathbf{a}^{(t_1)} - \mathbf{a}^*\|_2^2}{\alpha\beta\eta} \quad (94)$$

*with*

$$\begin{aligned}
\rho_2 &= \sigma_{\max}\left(\mathbb{1}_n - \alpha\mathbf{X}^\top\mathbf{X}\right) \\
C_1 &= \frac{1+\rho}{2(1-\rho)} \\
C_2 &= \frac{C + C'}{2}\frac{1+\rho}{1-\rho} + \frac{2\sqrt{n}\sigma_{\max}(\mathbf{X})}{1-\rho_2}\frac{2\tau}{1-\rho} = \frac{(C+C')(1-\rho_2)(1+\rho) + 8\sqrt{n}\sigma_{\max}(\mathbf{X})\tau}{2(1-\rho_2)(1-\rho)} \\
C_3 &= \frac{4\tau}{1-\rho}
\end{aligned} \quad (95)$$

*Proof.* Using Theorem C.38, we get $\|\mathbf{a} - \mathbf{a}^*\|_1 \leq \frac{1+\rho}{1-\rho}(\|\mathbf{a}\|_1 - \|\mathbf{a}^*\|_1) + \frac{2\tau}{1-\rho}\|\mathbf{X}(\mathbf{a} - \mathbf{a}^*)\|_2$. We also have $\min_{t_1 \leq t \leq t_2}\left(\|\mathbf{a}^{(t)}\|_1 - \|\mathbf{a}^*\|_1\right) \leq \frac{\eta + (C+C')\alpha\beta}{2}$ if and only if $t_2 \geq t_1 + \Delta t(\eta, t_1)$ (Theorem C.29). For $t \geq t_1$, we have

$$\begin{aligned}
\|\mathbf{X}(\mathbf{a}^{(t)} - \mathbf{a}^*)\|_2 &= \|\mathbf{X}\mathbf{a}^{(t)} - \mathbf{y}^* + \boldsymbol{\xi}\|_2 \\
&\leq \|\mathbf{X}\mathbf{a}^{(t)} - \mathbf{y}^*\|_2 + \|\boldsymbol{\xi}\|_2 \\
&\leq \frac{2\sqrt{n}\|\mathbf{X}\|_{2\to2}}{1-\rho_2}\alpha\beta + \|\mathbf{X}\hat{\mathbf{a}} - \mathbf{y}^*\|_2 + \|\boldsymbol{\xi}\|_2 \text{ (Equation (82))} \\
&= \frac{2\sqrt{n}\|\mathbf{X}\|_{2\to2}}{1-\rho_2}\alpha\beta + 2\|\boldsymbol{\xi}\|_2 \text{ since } \|\mathbf{X}\hat{\mathbf{a}} - \mathbf{y}^*\|_2^2 = \boldsymbol{\xi}^\top(\mathbb{1}_N - \mathbf{U}\mathbf{U}^\top)\boldsymbol{\xi} \leq \|\boldsymbol{\xi}\|_2^2 \text{ (Equation (72))}
\end{aligned} \quad (96)$$

So,

$$\begin{aligned}
\min_{t_1 \leq t \leq t_2}\|\mathbf{a}^{(t)} - \mathbf{a}^*\|_1 &\leq \frac{1+\rho}{1-\rho}\frac{\eta + (C+C')\alpha\beta}{2} + \frac{2\tau}{1-\rho}\left(\frac{2\sqrt{n}\sigma_{\max}(\mathbf{X})\alpha\beta}{1-\rho_2} + 2\|\boldsymbol{\xi}\|_2\right) \iff t_2 \geq t_1 + \Delta t(\eta, t_1) \\
&= C_1\eta + C_2\alpha\beta + C_3\|\boldsymbol{\xi}\|_2 \iff t_2 \geq t_1 + \Delta t(\eta, t_1)
\end{aligned} \quad (97)$$

$\square$

## C.6. Proof of Theorem 3.4

Let $\mathbf{y}(\mathbf{A}) = \mathbf{X}\operatorname{vec}(\mathbf{A})$ for $\mathbf{A} \in \mathbb{R}^{n_1 \times n_2}$. We have $\mathbf{y}^* = \mathbf{X}\operatorname{vec}(\mathbf{A}^*) + \boldsymbol{\xi}$, and want to minimize $f(\mathbf{A}) = g(\mathbf{A}) + \beta h(\mathbf{A})$ using gradient descent with learning rate $\alpha > 0$, where $h(\mathbf{A}) := \|\mathbf{A}\|_*$ and $g(\mathbf{A}) := \frac{1}{2}\|\mathbf{y}(\mathbf{A}) - \mathbf{y}^*\|_2^2 = \frac{1}{2}\mathbf{a}^\top\mathbf{X}^\top\mathbf{X}\mathbf{a} - \left(\mathbf{X}^\top\mathbf{X}\mathbf{a}^* + \mathbf{X}^\top\boldsymbol{\xi}\right)^\top\mathbf{a} + \frac{1}{2}\|\mathbf{X}\mathbf{a}^* + \boldsymbol{\xi}\|_2^2$. We write $F(\mathbf{A}) := G(\mathbf{A}) + \beta H(\mathbf{A})$ with $\operatorname{vec}G(\mathbf{A}) := \nabla_{\mathbf{a}}g(\mathbf{A}) = \mathbf{X}^\top\mathbf{X}\mathbf{a} - \left(\mathbf{X}^\top\mathbf{X}\mathbf{a}^* + \mathbf{X}^\top\boldsymbol{\xi}\right)$ and $H(\mathbf{A}) \in \partial\|\mathbf{A}\|_* = \{\mathbf{U}\mathbf{V}^\top + \mathbf{W}, \|\mathbf{W}\|_{2\to2} \leq 1, \mathbf{U}^\top\mathbf{W} = 0, \mathbf{W}\mathbf{V} = 0\}$ any subgradient of $\|\mathbf{A}\|_*$, with $\mathbf{A} = \mathbf{U}\boldsymbol{\Sigma}\mathbf{V}^\top$ under the compact SVD[7] [8]. Suppose we start at some $\mathbf{A}^{(1)}$. Using $\mathbf{F}^{(t)} := F(\mathbf{A}^{(t)})$, the subgradient update rule is

$$\mathbf{a}^{(t+1)} = \mathbf{a}^{(t)} - \alpha\mathbf{F}^{(t)} = \left(\mathbb{1}_n - \alpha\mathbf{X}^\top\mathbf{X}\right)\mathbf{a}^{(t)} + \alpha\left(\mathbf{X}^\top\mathbf{X}\mathbf{a}^* + \mathbf{X}^\top\boldsymbol{\xi}\right) - \alpha\beta\operatorname{vec}\left(H(\mathbf{A}^{(t)})\right) \quad \forall t > 1 \quad (98)$$

As in Section C.4, we let $\mathbf{X} = \mathbf{U}\boldsymbol{\Sigma}^{\frac{1}{2}}\mathbf{V}^\top$ under the compact SVD decomposition, with $\boldsymbol{\Sigma} = \operatorname{diag}(\sigma_k)_{k\in[r]}$, where $r = \operatorname{rank}(\mathbf{X})$ and $\sigma_{\max} = \sigma_1 \geq \cdots\sigma_k \geq \sigma_{k+1}\cdots \geq \sigma_{\min} = \sigma_r > \sigma_{r+1} = \cdots = 0$. We assume the step size $\alpha$ satisfies $0 < \alpha < \frac{2}{\sigma_{\max}}$. We define $\rho_p := \left\|\mathbb{1}_n - \alpha\mathbf{X}^\top\mathbf{X}\right\|_{p\to p}$ for all $p > 0$. We will show that for $\beta$ small enough, the update first moves near the least square solution of the problem, $\hat{\mathbf{a}} = \operatorname{vec}\hat{\mathbf{A}} = \left(\mathbf{X}^\top\mathbf{X}\right)^\dagger\mathbf{X}^\top\mathbf{y}^* = \mathbf{V}\left(\mathbf{V}^\top\mathbf{a}^* + \boldsymbol{\Sigma}^{-\frac{1}{2}}\mathbf{U}^\top\boldsymbol{\xi}\right)$. Later in training, $H(\mathbf{A})$ dominates the update, leading to $\|\mathbf{A}^{(t)}\|_* \approx \|\mathbf{A}^*\|_*$.

---

[7]The norm $\|\mathbf{A}\|_*$ is not differentiable everywhere because the singular values of $\mathbf{A}$ can be non-differentiable at points where they have multiplicities (e.g., when the singular values are not distinct)

[8]For the experiments, we used the polar factor $H(\mathbf{A}) = \mathbf{U}\mathbf{V}^\top$. This is the gradient provided by automatic differentiation in many optimization libraries, like Pytorch.

**Theorem C.40.** *Assume the learning rate, the regularization coefficient and the noise satisfy $0 < \alpha < \alpha_{\max}$, $0 < \beta < \frac{\sigma_{\max}(\mathbf{X}^\top \mathbf{X})}{\sqrt{\min(n_1, n_2)}}$ and $\|\mathbf{X}^\top \boldsymbol{\xi}\|_2 \leq \sqrt{C}\alpha\beta$, $C > 0$. Let $\rho_2 := \sigma_{\max}\left(\mathbb{I}_n - \alpha\mathbf{X}^\top\mathbf{X}\right)$. There exist $t_1 < \infty$ and a constant $C' > 0$ such that:*

$$\| \operatorname{vec}(\mathbf{A}^{(t)} - \hat{\mathbf{A}})\|_2 \leq \frac{2\alpha\beta n^{1/2}}{1 - \rho_2} \quad \text{and } g(\mathbf{A}^{(t)}) \leq g(\hat{\mathbf{A}}) + \frac{2n\alpha^2\beta^2\sigma_{\max}^2(\mathbf{X})}{(1 - \rho_2)^2} \quad \forall t \geq t_1, \text{ with } g(\hat{\mathbf{A}}) \leq \frac{1}{2}\|\boldsymbol{\xi}\|_2^2$$

$$\forall \eta > 0, \quad \min_{t_1 \leq t \leq t_2} \left( f(\mathbf{A}^{(t)}) - f(\mathbf{A}^*)\right) \leq \frac{(\eta + C'\alpha\beta)\,\beta}{2} \iff t_2 \geq t_1 + \Delta t(\eta, t_1), \quad \Delta t(\eta, t_1) := \frac{\|\mathbf{A}^{(t_1)} - \mathbf{A}^*\|_F^2}{\alpha\beta\eta} \quad (99)$$

$$\forall \eta > 0, \quad \min_{t_1 \leq t \leq t_2} \left( \|\mathbf{A}^{(t)}\|_* - \|\mathbf{A}^*\|_* \right) \leq \frac{\eta + (C + C')\alpha\beta}{2} \iff t_2 \geq t_1 + \Delta t(\eta, t_1)$$

*Proof.* First, we observe that if $\beta$ is too high, the subgradient term $H(\mathbf{A})$ dominates early, and there is no convergence, i.e., no memorization nor generalization. In fact, if $\beta > \frac{\sigma_{\max}}{\sqrt{\min(n_1, n_2)}}$ then the $\ell_*$-term dominates the updates, causing the sequence $\mathbf{A}^{(t)}$ to exhibit oscillatory behavior without convergence to a minimizer of $f(\mathbf{A}) = g(\mathbf{A}) + \beta\|\mathbf{A}\|_*$ (Lemma C.33). The memorization phase is similar to that of Theorem C.29, with $n = n_1 n_2$. For $t \geq t_1$, $g(\mathbf{A}^{(t)}) - g(\hat{\mathbf{A}}) \leq \frac{2n\alpha^2\beta^2\sigma_{\max}^2(\mathbf{X})}{(1-\rho_2)^2} = \mathcal{O}\left(2n\beta^2\sigma_{\max}^2(\mathbf{X})\right)$ and $\|\operatorname{vec} G(\mathbf{A}^{(t)})\|_2 = \Theta\left(2\beta\sqrt{n}\sigma_{\max}(\mathbf{X}^\top\mathbf{X})\right) = \Theta(\beta)$. So after time $t_1$, the contribution of the gradient $G$ to the update of $\mathbf{A}^{(t)}$ is dominated by the $\ell_*$–regularization term. Specifically for all $t \gg t_1$, the update rule approximates $\mathbf{A}^{(t+1)} \approx \mathbf{A}^{(t)} - \alpha\beta H(\mathbf{A}_i^{(t)}) = \mathbf{A}^{(t)} - \alpha\beta H(\mathbf{A}^{(t)} - \mathbf{A}^*)$ up to an error $\mathbf{E} \in \mathbb{R}^{n_1 \times n_2}$ of order $\|\mathbf{E}\|_F = \mathcal{O}\left(\frac{\alpha\beta\sqrt{\operatorname{rank}(\mathbf{A}^{(t_1)})}}{\sigma_{\min}(\mathbf{A}^{(t_1)})/\sigma_{\max}(\mathbf{A}^*) - 1}\right)$ (Lemma C.45). So, as soon as the singular–value gap widens (i.e. $\min_i |\Sigma_i^{(t_1)}| \gg \max_i |\Sigma_i^*|$, empirically typical after a warm–up phase), the approximation becomes tighter, even more so for small $\alpha\beta$. By Lemma C.42, this lead to $\|\mathbf{A}^{(t)} - \mathbf{A}^*\|_{2\to2} = \mathcal{O}(\alpha\beta)$ for (and only for) $t \geq t_1 + \left\lfloor \frac{\|\mathbf{A}^{(t_1)} - \mathbf{A}^*\|_{2\to2}}{\alpha\beta}\right\rfloor$. As the error $\|\mathbf{E}\|_F$ is of order $\mathcal{O}(\alpha\beta)$ for iterations directly following $t_1$, choosing a time $\Theta(1/\alpha\beta)$ also somehow counterbalances its cumulative effect. Equipped with this insight, we prove the exact delay below.

We have $g(\mathbf{A}^{(t)}) = \frac{1}{2}\|\mathbf{X}\operatorname{vec}\mathbf{A}^{(t)} - \mathbf{y}^*\|_2^2$, $h(\mathbf{A}^{(t)}) = \|\mathbf{A}^{(t)}\|_*$, $f(\mathbf{A}^{(t)}) = g(\mathbf{A}^{(t)}) + \beta h(\mathbf{A}^{(t)})$ and $f(\mathbf{A}^*) = \beta\|\mathbf{A}^*\|_* + \frac{1}{2}\|\boldsymbol{\xi}\|_2^2$, $\Theta_f = \{\mathbf{A}^*\}$ and $\Theta_g = \{\mathbf{A} \mid \mathbf{X}\operatorname{vec}(\mathbf{A} - \mathbf{A}^*) = \boldsymbol{\xi}\}$. Applying Lemma C.12, we get

$$\min_{t_1 \leq t \leq t_2}\left( f(\mathbf{A}^{(t)}) - f(\mathbf{A}^*)\right) \leq \frac{\|\operatorname{vec}\left(\mathbf{A}^{(t_1)} - \mathbf{A}^*\right)\|_2^2 + (t_2 - t_1)\alpha^2 \max_{t_1 \leq t \leq t_2}\|\operatorname{vec}\left(F(\mathbf{A}^{(t)})\right)\|_2^2}{2\alpha(t_2 - t_1)} \quad \forall t_2 \geq t_1 \quad (100)$$

Using $\|\operatorname{vec} F(\mathbf{A}^{(t)})\|_2 = \mathcal{O}(\beta)$ $\forall t \geq t_1$ we get from Theorem C.13 that there exists $C' > 0$,

$$\min_{t_1 \leq t \leq t_2}\left( f(\mathbf{A}^{(t)}) - f(\mathbf{A}^*)\right) \leq \frac{(\eta + C'\alpha\beta)\,\beta}{2} \iff t_2 \geq t_1 + \frac{\|\operatorname{vec}\left(\mathbf{A}^{(t_1)} - \mathbf{A}^*\right)\|_2^2}{\alpha\beta\eta} \quad (101)$$

and

$$\beta \min_{t_1 \leq t \leq t_2}\left( \|\mathbf{A}^{(t)}\|_* - \|\mathbf{A}^*\|_*\right) \leq \min_{t_1 \leq t \leq t_2}\left( f(\mathbf{A}^{(t)}) - f(\mathbf{A}^*)\right) - \left(\min_{t_1 \leq t \leq t_2} g(\mathbf{A}^{(t)}) - \frac{1}{2}\|\boldsymbol{\xi}\|_2^2\right) \quad \forall t_2 \geq t_1$$

$$\leq \frac{(\eta + C'\alpha\beta)\,\beta}{2} + \frac{1}{2}\|\mathbf{X}^\top\boldsymbol{\xi}\|_2^2 \iff t_2 \geq t_1 + \Delta t(\eta, t_1) \quad (102)$$

$$\leq \frac{(\eta + (C + C')\alpha\beta)\,\beta}{2} \iff t_2 \geq t_1 + \Delta t(\eta, t_1) \text{ since } \|\mathbf{X}^\top\boldsymbol{\xi}\|_2 \leq \sqrt{C}\alpha\beta$$

since $g(\mathbf{A}^{(t)}) - \frac{1}{2}\|\boldsymbol{\xi}\|_2^2 = \frac{1}{2}\|\mathbf{X}\operatorname{vec}(\mathbf{A}^{(t)} - \hat{\mathbf{A}})\|_2^2 - \frac{1}{2}\|\mathbf{U}^\top\boldsymbol{\xi}\|_2^2 \geq -\frac{1}{2}\|\mathbf{U}^\top\boldsymbol{\xi}\|_2^2$. $\qquad\square$

**Lemma C.41.** *Let $\mathbf{A} \in \mathbb{R}^{n_1 \times n_2}$. We have $\|\operatorname{vec}(\mathbf{H})\|_p \leq (n_1 n_2)^{1/p}$ for all $\mathbf{H} \in \partial\|\mathbf{A}\|_*$ and $p > 0$.*

*Proof.* Let $\mathbf{H} \in \partial\|\mathbf{A}\|_*$. Then $\|\mathbf{H}\|_{2\to2} \leq 1$. So by the definition of the spectral (operator) norm, we have $\|\mathbf{H}\|_{2\to2} = \sup_{\mathbf{x} \neq 0}\frac{\|\mathbf{H}\mathbf{x}\|_2}{\|\mathbf{x}\|_2} = \sigma_{\max}(\mathbf{H}) \leq 1$. Taking $\mathbf{x} = \mathbf{e}_j^{(n_2)}$, the $j$-th standard basis vector in $\mathbb{R}^{n_2}$, we obtain $\|\mathbf{H}_{:,j}\|_2 = \|\mathbf{H}\mathbf{e}_j^{(n_2)}\|_2 \leq 1$; which implied $\mathbf{H}_{ij} \leq \|\mathbf{H}_{:,j}\|_2 \leq 1$. So $\|\operatorname{vec}(\mathbf{H})\|_p = \left(\sum_{i=1}^{n_1}\sum_{j=1}^{n_2}|\mathbf{H}_{ij}|^p\right)^{1/p} \leq (n_1 n_2)^{1/p}$. $\qquad\square$

**Lemma C.42.** *Given $\alpha > 0$ and $\mathbf{A}^{(1)} \in \mathbb{R}^{n_1 \times n_2}$, let $\mathbf{A}^{(t+1)} = \mathbf{A}^{(t)} - \alpha H(\mathbf{A}^{(t)})$ for all $t \geq 1$, where $H(\mathbf{A}) \in \partial \|\mathbf{A}\|_*$.*

*1. A point $\mathbf{A}$ is stationary for this dynamical system if and only if $\|\mathbf{A}\|_{2 \to 2} = \sigma_{\max}(\mathbf{A}) < \alpha$.*

*2. $\|\mathbf{A}^{(t)}\|_{2 \to 2} < \alpha$ if and only if $t > \lfloor \frac{\|\mathbf{A}^{(1)}\|_{2 \to 2}}{\alpha} \rfloor$.*

*3. For all $t > \lfloor \frac{\|\mathbf{A}^{(1)}\|_{2 \to 2}}{\alpha} \rfloor$, $r_t := \mathrm{rank}(\mathbf{A}^{(t)}) = \left| \{ i \mid \sigma_i^{(1)} / \alpha \in \mathbb{Z} \} \right|$, with $\sigma_1^{(1)}, \ldots, \sigma_{r_1}^{(1)}$ the singular values of $\mathbf{A}^{(1)}$.*

*Proof.* We start with the subgradient $H(\mathbf{A}) = \mathbf{U}\mathbf{V}^\top$ for $\mathbf{A} = \mathbf{U}\Sigma\mathbf{V}^\top$, so that the update rule becomes

$$\mathbf{A}^{(t+1)} = \mathbf{A}^{(t)} - \alpha \mathbf{U}^{(t)}\mathbf{V}^{(t)\top} = \mathbf{U}^{(t)} \left( \Sigma^{(t)} - \alpha \mathbb{I}_{r_t} \right) \mathbf{V}^{(t)\top} \text{for all } t \geq 1 \tag{103}$$

This equation also writes

$$\begin{aligned}
\mathbf{A}^{(t+1)} &= \mathbf{U}^{(t+1)}\Sigma^{(t+1)}\mathbf{V}^{(t+1)\top} = \sum_{i=1}^{r_{t+1}} \sigma_i^{(t+1)} \mathbf{U}_{:,i}^{(t+1)} \mathbf{V}_{:,i}^{(t+1)\top} \\
&= \sum_{i=1}^{r_t} (\sigma_i^{(t)} - \alpha) \mathbf{U}_{:,i}^{(t)} \mathbf{V}_{:,i}^{(t)\top} = \sum_{i=1}^{r_t} |\sigma_i^{(t)} - \alpha| \cdot \mathrm{sign}(\sigma_i^{(t)} - \alpha) \mathbf{U}_{:,i}^{(t)} \mathbf{V}_{:,i}^{(t)\top}
\end{aligned} \tag{104}$$

This implies $\sigma_i^{(t+1)} = |\sigma_i^{(t)} - \alpha| \quad \forall i \in [r_1]$. So starting at $\sigma_i^{(1)}$, each $\sigma_i$ decay at each step by $\alpha$ until $\sigma_i^{(t)} =: \sigma_i^* \in [0, \alpha)$, and start oscillating between $\sigma_i^*$ and $\alpha - \sigma_i^*$. It starts doing so when $t > t_i := \lfloor \frac{\sigma_i^{(1)}}{\alpha} \rfloor$. We take $t = \max_i t_i$.

In general, we have $H(\mathbf{A}) = \mathbf{U}\mathbf{V}^\top + \mathbf{W}, \|\mathbf{W}\|_{2 \to 2} \leq 1, \mathbf{U}^\top \mathbf{W} = 0, \mathbf{W}\mathbf{V} = 0$. The dynamics becomes $\mathbf{A}^{(t+1)} = \mathbf{U}^{(t)} \left( \Sigma^{(t)} - \alpha \mathbb{I}_{r_t} \right) \mathbf{V}^{(t)\top} - \alpha \mathbf{W}^{(t)}$ with $\mathbf{U}^{(t)\top} \mathbf{W}^{(t)} = 0, \mathbf{W}^{(t)}\mathbf{V}^{(t)} = 0$. So $\mathbf{U}^{(t)\top} \mathbf{A}^{(t+1)} \mathbf{V}^{(t)} = \Sigma^{(t)} - \alpha \mathbb{I}_{r_t}$, leading again to $\sigma_i^{(t+1)} = |\sigma_i^{(t)} - \alpha| \quad \forall i \in [r_1]$. $\qquad\square$

We show in Lemma C.32 that for $a, a^* \in \mathbb{R}$, $|a| > |a^*| \implies \mathrm{sign}(a - a^*) = \mathrm{sign}(a)$, with $\mathrm{sign}(a) \in \partial |a|$. Now, considering the canonical nuclear–norm subgradient $H(\mathbf{A}) = \mathbf{U}\mathbf{V}^\top \in \partial \|\mathbf{A}\|_*$ for $\mathbf{A} = \mathbf{U}\Sigma\mathbf{V}^\top \in \mathbb{R}^{n_1 \times n_2}$, we ask if $\sigma_{\min}(\mathbf{A}) > \sigma_{\max}(\mathbf{A}^*) \implies H(\mathbf{A} - \mathbf{A}^*) = H(\mathbf{A})$, and if not, how does $H(\mathbf{A} - \mathbf{A}^*)$ deviate from $H(\mathbf{A})$. Throughout, let $\mathbf{A} = \mathbf{U}\Sigma\mathbf{V}^\top \in \mathbb{R}^{n_1 \times n_2}$ of rank $r \geq 1$, and $\mathbf{A} - \mathbf{A}^* = \widetilde{\mathbf{U}}\widetilde{\Sigma}\widetilde{\mathbf{V}}^\top \in \mathbb{R}^{n_1 \times n_2}$, with $\mathbf{U}, \widetilde{\mathbf{U}} \in \mathbb{R}^{n_1 \times r}$ and $\mathbf{V}, \widetilde{\mathbf{V}} \in \mathbb{R}^{n_2 \times r}$ having orthonormal columns. So $H(\mathbf{A}) = \mathbf{U}\mathbf{V}^\top$ and $H(\mathbf{A} - \mathbf{A}^*) = \widetilde{\mathbf{U}}\widetilde{\mathbf{V}}^\top$.

There exist matrices with $\sigma_{\min}(\mathbf{A}) > \sigma_{\max}(\mathbf{A}^*)$ for which $H(\mathbf{A} - \mathbf{A}^*) \neq H(\mathbf{A})$. E.g., $\mathbf{A} = \begin{bmatrix} 2 & 0 \\ 0 & 0 \end{bmatrix}, \mathbf{A}^* = \begin{bmatrix} 0 & 1 \\ 0 & 0 \end{bmatrix}$. But it easy to check that if $\sigma_{\min}(\mathbf{A}) > \sigma_{\max}(\mathbf{A}^*)$ and $\mathbf{A}$ and $\mathbf{A}^*$ have the same singular directions, then $H(\mathbf{A} - \mathbf{A}^*) = H(\mathbf{A})$.

**Lemma C.43.** *Let $\mathbf{A} = \mathbf{U}\Sigma\mathbf{V}^\top \in \mathbb{R}^{n_1 \times n_2}$. If $\mathbf{A}^* = \mathbf{U}\mathbf{B}\mathbf{V}^\top$ with $\sigma_{\max}(\mathbf{B}) < \sigma_{\min}(\mathbf{A})$, then $H(\mathbf{A} - \mathbf{A}^*) = H(\mathbf{A})$.*

Now we focus on bounding $\|H(\mathbf{A} - \mathbf{A}^*) - H(\mathbf{A})\|$ in a general setting. We introduce some notations, then Wedin's $\sin \Theta$ bound (Wedin, 1972), and our result. Let $S_L = \mathrm{span}(\mathbf{U}) \subset \mathbb{R}^{n_1}$ and $\widetilde{S}_L = \mathrm{span}(\widetilde{\mathbf{U}}) \subset \mathbb{R}^{n_1}$. The principal angles $0 \leq \theta_1 \leq \cdots \leq \theta_r \leq \pi/2$ between $S_L$ and $\widetilde{S}_L$ are defined by $\cos \theta_i := \sigma_i \left( \mathbf{U}^\top \widetilde{\mathbf{U}} \right) \forall i \in [r]$, where $\sigma_1 \geq \cdots \geq \sigma_r$ are the singular values. In fact, the classical (geometric) definition of the principal angles is given recursively by $\cos \theta_1 = \max_{\mathbf{u} \in S_L, \tilde{\mathbf{u}} \in \widetilde{S}_L} \mathbf{u}^\top \tilde{\mathbf{u}}$, $\cos \theta_2 = \max_{\mathbf{u} \in S_L, \tilde{\mathbf{u}} \in \widetilde{S}_L, \mathbf{u} \perp \mathbf{u}_1, \tilde{\mathbf{u}} \perp \tilde{\mathbf{u}}_1} \mathbf{u}^\top \tilde{\mathbf{u}}$... Each pair is taken orthogonal to the previous ones, so we end up with $r$ angles $\theta_1 \leq \cdots \leq \theta_r$, all in $[0, \pi/2]$. Write $\mathbf{u} = \mathbf{U}\mathbf{x}, \tilde{\mathbf{u}} = \widetilde{\mathbf{U}}\tilde{\mathbf{x}}$ with $\mathbf{x}, \tilde{\mathbf{x}} \in \mathbb{R}^r, \|\mathbf{x}\| = \|\tilde{\mathbf{x}}\| = 1$. The first maximisation therefore reads $\cos \theta_1 = \max_{\|\mathbf{x}\| = \|\tilde{\mathbf{x}}\| = 1} \mathbf{x}^\top \mathbf{U}^\top \widetilde{\mathbf{U}}\tilde{\mathbf{x}} = \sigma_1(\mathbf{U}^\top \widetilde{\mathbf{U}})$. The orthogonality constraints in the subsequent steps force $(\mathbf{x}_k, \tilde{\mathbf{x}}_k) = (\mathbf{U}^\top \mathbf{u}_k, \widetilde{\mathbf{U}}^\top \tilde{\mathbf{u}}_k)$ to lie in the left-over singular subspaces of $\mathbf{U}^\top \widetilde{\mathbf{U}}$, with $(\mathbf{u}_k, \tilde{\mathbf{u}}_k)$ the maximisers at step $k$. Inductively, one obtains $\cos \theta_i = \sigma_i(\mathbf{U}^\top \widetilde{\mathbf{U}}) \quad (i = 1, \ldots, r)$ as a standard consequence of the SVD and the min–max (Courant–Fischer/Ky Fan) principle.

It is customary to collect the angles into a diagonal matrix $\Theta_L := \mathrm{diag}(\theta_1, \ldots, \theta_r)$. Applying $\sin(\cdot)$ entry-wise gives $\sin \Theta_L := \mathrm{diag}(\sin \theta_1, \ldots, \sin \theta_r)$. One never needs to compute the angles explicitly because $\|\sin \Theta_L\|_{2 \to 2} = \sin \theta_{\max} = \|\mathbf{U}^\top \widetilde{\mathbf{U}}^\perp\|_{2 \to 2}$ where $\theta_{\max} = \theta_r$ is the largest principal angle, and $\widetilde{\mathbf{U}}^\perp$ is an orthonormal basis of $\widetilde{S}_L^\perp$. In fact, let $\mathbf{B} := \mathbf{U}^\top \widetilde{\mathbf{U}} \in \mathbb{R}^{r \times r}$ and $\mathbf{C} := \mathbf{U}^\top \widetilde{\mathbf{U}}^\perp \in \mathbb{R}^{r \times (n_1 - r)}$. We can write the SVD of $\mathbf{B}$ as $\mathbf{B} = \mathbf{D} \cos \Theta_L \mathbf{E}^\top$, with

$\mathbf{E} \in \mathbb{R}^{r \times r}$ orthogonal. This gives $\mathbf{BB}^\top = \mathbf{D}(\cos \Theta_L)^2 \mathbf{D}^\top$. Because of the orthogonality relation $\mathbf{BB}^\top + \mathbf{CC}^\top = \mathbf{U}^\top(\widetilde{\mathbf{U}}\widetilde{\mathbf{U}}^\top + \widetilde{\mathbf{U}}^\perp \widetilde{\mathbf{U}}^{\perp\top})\mathbf{U} = \mathbf{U}^\top \mathbf{U} = \mathbb{I}_r$, we have $\mathbf{CC}^\top = \mathbb{I}_r - \mathbf{BB}^\top = \mathbf{D}(\mathbb{I}_r - (\cos \Theta_L)^2)\mathbf{D}^\top = \mathbf{D}(\sin \Theta_L)^2 \mathbf{D}^\top$. So, under SVD, $\mathbf{C} = \mathbf{D} \sin \Theta_L \mathbf{F}^\top$ with $\mathbf{F} \in \mathbb{R}^{(n_1-r) \times (n_1-r)}$ orthogonal. This is related to the CS (cosine-sine) matrix decomposition of $\mathbf{U}^\top[\widetilde{\mathbf{U}} \ \widetilde{\mathbf{U}}^\perp]$. Therefore, $\|\mathbf{C}\|_{2 \to 2} = \|\sin \Theta_L\|_{2 \to 2} = \sin \theta_{\max}$.

Thus $\|\sin \Theta_L\|_{2 \to 2}$ measures the worst-case misalignment between the two subspaces; it equals the operator norm of the difference of their orthogonal projections. In the following, we use $\sin \Theta_L$ in place of $\|\sin \Theta_L\|_{2 \to 2}$ by abuse of notation. The following result can be derived from the Davis–Kahan $\sin \Theta$ theorem.

**Lemma C.44** (Wedin's $\sin \Theta$ bound). *Let $\mathbf{E} \in \mathbb{R}^{n_1 \times n_2}$ and set $\widetilde{\mathbf{A}} := \mathbf{A} + \mathbf{E}$. Let $\mathbf{U}_r, \widetilde{\mathbf{U}}_r \in \mathbb{R}^{n_1 \times r}$ span the leading $r$ left singular subspaces of $\mathbf{A}$ and $\widetilde{\mathbf{A}}$, respectively, and let $\Theta_L$ be the matrix of canonical angles between these subspaces. If $\|\mathbf{E}\|_{2 \to 2} < \sigma_r(\mathbf{A}) - \sigma_{r+1}(\mathbf{A})$ (with $\sigma_{r+1}(\mathbf{A}) = 0$ because $\mathrm{rank}(\mathbf{A}) = r$), then $\sin \Theta_L := \|\mathbf{U}_r^\top \widetilde{\mathbf{U}}_r^\perp\|_{2 \to 2} \leq \frac{\sigma_{\max}(\mathbf{E})}{\sigma_r(\mathbf{A}) - \sigma_{\max}(\mathbf{E})}$. An analogous bound on $\sin \Theta_R := \|\mathbf{V}_r^\top \widetilde{\mathbf{V}}_r^\perp\|_{2 \to 2}$ holds on the right singular side.*

**Lemma C.45.** *If $\sigma_{\min}(\mathbf{A}) > \sigma_{\max}(\mathbf{A}^*)$, then $\|H(\mathbf{A} - \mathbf{A}^*) - H(\mathbf{A})\|_F \leq \frac{2\sqrt{2 \, \mathrm{rank}(\mathbf{A})}}{\sigma_{\min}(\mathbf{A})/\sigma_{\max}(\mathbf{A}^*) - 1}$.*

*Proof.* Let $r = \mathrm{rank}(\mathbf{A})$. Write $H(\mathbf{A}) = \mathbf{UV}^\top$ and $H(\mathbf{A} - \mathbf{A}^*) = \widetilde{\mathbf{U}}\widetilde{\mathbf{V}}^\top$, with $\mathbf{U}, \mathbf{V}, \widetilde{\mathbf{U}}, \widetilde{\mathbf{V}}$ having orthonormal columns.

$$\|\mathbf{U} - \widetilde{\mathbf{U}}\|_F^2 = \|\mathbf{U}\|_F^2 + \|\widetilde{\mathbf{U}}\|_F^2 - 2 \, \mathrm{tr}\left(\widetilde{\mathbf{U}}^\top \mathbf{U}\right) = 2r - 2 \sum_{i=1}^r \cos \theta_i \text{ since } \|\mathbf{U}\|_F^2 = \|\widetilde{\mathbf{U}}\|_F^2 = r \text{ and } \sigma_i\left(\widetilde{\mathbf{U}}^\top \mathbf{U}\right) = \cos \theta_i$$

$$= 2 \sum_{i=1}^r (1 - \cos \theta_i) \leq 2 \sum_{i=1}^r \sin^2 \theta_i \text{ since } 1 - \cos \theta_i = \frac{\sin^2 \theta_i}{1 + \cos \theta_i} \leq \sin^2 \theta_i \text{ for } \theta_i \in [0, \pi/2]$$

$$\leq 2r(\sin \theta_{\max})^2 = 2r(\sin \Theta_L)^2$$

$$\leq 2r \left(\frac{\sigma_{\max}(\mathbf{A}^*)}{\sigma_{\min}(\mathbf{A}) - \sigma_{\max}(\mathbf{A}^*)}\right)^2 \text{ (Lemma C.44 with } \mathbf{E} = -\mathbf{A}^*)$$

$$(105)$$

Hence

$$\|\widetilde{\mathbf{U}}\widetilde{\mathbf{V}}^\top - \mathbf{UV}^\top\|_F \leq \|(\widetilde{\mathbf{U}} - \mathbf{U})\widetilde{\mathbf{V}}^\top\|_F + \|\mathbf{U}(\widetilde{\mathbf{V}} - \mathbf{V})^\top\|_F$$

$$\leq \|\widetilde{\mathbf{U}} - \mathbf{U}\|_F + \|\widetilde{\mathbf{V}} - \mathbf{V}\|_F \leq \frac{2\sqrt{2r}\sigma_{\max}(\mathbf{A}^*)}{\sigma_{\min}(\mathbf{A}) - \sigma_{\max}(\mathbf{A}^*)} \qquad (106)$$

$\square$

## C.7. Proof of Theorem 3.5

**Definition C.46** (Stable Rank Null Space Property). A linear measurement map $\mathcal{F} : \mathbb{R}^{n_1 \times n_2} \to \mathbb{R}^m$ is said to satisfy the stable rank null space property of order $r$ with constant $\rho \in (0, 1)$ if for all $\mathbf{A} \in \ker \mathcal{F} \setminus \{0\}$, the singular values of $\mathbf{U}$ satisfy $\sum_{i=1}^r \sigma_i(\mathbf{A}) \leq \rho \sum_{i=r+1}^{\min\{n_1, n_2\}} \sigma_i(\mathbf{A})$.

**Theorem C.47** (Exercises 4.19 in Foucart & Rauhut (2013)). *The linear measurement map $\mathcal{F} : \mathbb{R}^{n_1 \times n_2} \to \mathbb{R}^m$ satisfies the stable rank null space property of order $r$ with constant $\rho \in (0, 1)$ if and only if $\|\mathbf{B} - \mathbf{A}\|_* \leq \frac{1+\rho}{1-\rho}(\|\mathbf{B}\|_* - \|\mathbf{A}\|_* + 2 \sum_{i=r+1}^{\min\{n_1, n_2\}} \sigma_i(\mathbf{A}))$ for all vector $\mathbf{A}, \mathbf{B} \in \mathbb{R}^{n_1 \times n_2}$ with $\mathcal{F}(\mathbf{A}) = \mathcal{F}(\mathbf{B})$.*

**Definition C.48** (Robust Rank Null Space Property). A linear measurement map $\mathcal{F} : \mathbb{R}^{n_1 \times n_2} \to \mathbb{R}^m$ is said to satisfy the robust rank null space property of order $r$ (with respect to $\|\cdot\|$) with constants $\rho \in (0, 1)$ and $\tau > 0$ if, for all $\mathbf{A} \in \mathbb{R}^{n_1 \times n_2}$, the singular values of $\mathbf{A}$ satisfy $\sum_{\ell=1}^r \sigma_\ell(\mathbf{A}) \leq \rho \sum_{\ell=r+1}^{\min\{n_1, n_2\}} \sigma_\ell(\mathbf{A}) + \tau\|\mathcal{F}(\mathbf{A})\|$.

**Theorem C.49** (Exercises 4.19 in Foucart & Rauhut (2013)). *The linear measurement map $\mathcal{F} : \mathbb{R}^{n_1 \times n_2} \to \mathbb{R}^m$ satisfies the robust rank null space property of order $r$ (with respect to $\|\cdot\|$) with constants $\rho \in (0, 1)$ and $\tau > 0$ if and only if $\|\mathbf{B} - \mathbf{A}\|_* \leq \frac{1+\rho}{1-\rho}(\|\mathbf{B}\|_* - \|\mathbf{A}\|_* + 2 \sum_{\ell=r+1}^{\min\{n_1, n_2\}} \sigma_\ell(\mathbf{A})) + \frac{2\tau}{1-\rho}\|\mathcal{F}(\mathbf{B} - \mathbf{A})\|$ for all vector $\mathbf{A}, \mathbf{B} \in \mathbb{R}^{n_1 \times n_2}$.*

**Theorem C.50.** *Assume the linear measurement map $\mathcal{F}.(\mathbf{X})$ satisfies the robust rank null space property of order $r$ with constants $\rho \in (0, 1)$ and $\tau > 0$, i.e for all $\mathbf{A} \in \mathbb{R}^{n_1 \times n_2}$, $\sum_{\ell=1}^r \sigma_\ell(\mathbf{A}) \leq \rho \sum_{\ell=r+1}^{\min\{n_1, n_2\}} \sigma_\ell(\mathbf{A}) + \tau\|\mathcal{F}_{\mathrm{vec}(\mathbf{A})}(\mathbf{X})\|_2$. Then,*

*under the same condition as in Theorem C.40 on $\alpha$, $\beta$ and $\boldsymbol{\xi}$; i.e. $0 < \alpha\sigma_{\max}(\mathbf{X}^\top\mathbf{X}) < 2$, $0 < \beta\sqrt{\min(n_1, n_2)} < \sigma_{\max}(\mathbf{X}^\top\mathbf{X})$ and $\|\mathbf{X}^\top\boldsymbol{\xi}\|_2 \leq \sqrt{C}\alpha\beta$, $C > 0$; there exists $C' > 0$ such that for all $\eta > 0$,*

$$\min_{t_1 \leq t \leq t_2} \|\mathbf{A}^{(t)} - \mathbf{A}^*\|_* \leq C_1\eta + C_2\alpha\beta + C_3\|\boldsymbol{\xi}\|_2 \iff t_2 \geq t_1 + \Delta t(\eta, t_1), \quad \Delta t(\eta, t_1) := \frac{\|\mathbf{A}^{(t_1)} - \mathbf{A}^*\|_F^2}{\alpha\beta\eta} \quad (107)$$

*with $\rho_2 = \sigma_{\max}\left(\mathbb{I}_n - \alpha\mathbf{X}^\top\mathbf{X}\right)$,*

$$C_1 = \frac{1+\rho}{2(1-\rho)}, \quad C_2 = \frac{C+C'}{2}\frac{1+\rho}{1-\rho} + \frac{2\sqrt{n}\sigma_{\max}(\mathbf{X})}{1-\rho_2}\frac{2\tau}{1-\rho} \text{ and } C_3 = \frac{4\tau}{1-\rho} \quad (108)$$

*Proof.* We have $\|\mathbf{A} - \mathbf{A}^*\|_* \leq \frac{1+\rho}{1-\rho}(\|\mathbf{A}\|_1 - \|\mathbf{A}^*\|_*) + \frac{2\tau}{1-\rho}\|\mathbf{X}\,\mathrm{vec}(\mathbf{A} - \mathbf{A}^*)\|_2$ (Theorem C.49). We also have $\min_{t_1 \leq t \leq t_2}\left(\|\mathbf{A}^{(t)}\|_* - \|\mathbf{A}^*\|_*\right) \leq \frac{\eta+(C+C')\alpha\beta}{2}$ if and only if $t_2 \geq t_1 + \Delta t(\eta, t_1)$ (Theorem C.40). For $t \geq t_1$, we have

$$\begin{aligned}
\|\mathbf{X}\,\mathrm{vec}(\mathbf{A} - \mathbf{A}^*)\|_2 &\leq \|\mathbf{X}\,\mathrm{vec}\,\mathbf{A}^{(t)} - \mathbf{y}^*\|_2 + \|\boldsymbol{\xi}\|_2 \\
&\leq \frac{2\sqrt{n}\|\mathbf{X}\|_{2\to 2}}{1-\rho_2}\alpha\beta + \|\mathbf{X}\hat{\mathbf{a}} - \mathbf{y}^*\|_2 + \|\boldsymbol{\xi}\|_2 \text{ (Equation (82))} \\
&= \frac{2\sqrt{n}\|\mathbf{X}\|_{2\to 2}}{1-\rho_2}\alpha\beta + 2\|\boldsymbol{\xi}\|_2 \text{ (Equation (72))}
\end{aligned} \quad (109)$$

Combining these gives the desired result. $\qquad\square$

## C.8. Proof of Theorem 3.6

We want to minimize $f(\mathbf{a}) = g(\mathbf{a}) + \beta h(\mathbf{a})$ using gradient descent with a learning rate $\alpha$, where $\mathbf{g}(\mathbf{a}) = \frac{1}{2}\|\mathbf{X}\mathbf{a} - \mathbf{y}^*\|_2^2$ and $h(\mathbf{a}) = \frac{1}{2}\|\mathbf{a}\|_2^2$. Let $\mathbf{Q} := \mathbf{X}^\top\mathbf{X} + \beta\mathbb{I}_n$. We have

$$f(\mathbf{a}) := \frac{1}{2}\|\mathbf{y}(\mathbf{a}) - \mathbf{y}^*\|_2^2 + \frac{\beta}{2}\|\mathbf{a}\|_2^2 = \frac{1}{2}\mathbf{a}^\top\mathbf{Q}\mathbf{a} - \left(\mathbf{X}^\top\mathbf{X}\mathbf{a}^* + \mathbf{X}^\top\boldsymbol{\xi}\right)^\top\mathbf{a} + \frac{1}{2}\|\mathbf{X}\mathbf{a}^* + \boldsymbol{\xi}\|_2^2 \quad (110)$$

and

$$F(\mathbf{a}) := \nabla_\mathbf{a}f(\mathbf{a}) = \mathbf{X}^\top(\mathbf{y} - \mathbf{y}^*) + \beta\mathbf{a} = \mathbf{Q}\mathbf{a} - \left(\mathbf{X}^\top\mathbf{X}\mathbf{a}^* + \mathbf{X}^\top\boldsymbol{\xi}\right) \quad (111)$$

The subgradient update rule is

$$\mathbf{a}^{(t+1)} = \mathbf{a}^{(t)} - \alpha F(\mathbf{a}^{(t)}) = (\mathbb{I}_n - \alpha\mathbf{Q})\,\mathbf{a}^{(t)} + \alpha\left(\mathbf{X}^\top\mathbf{X}\mathbf{a}^* + \mathbf{X}^\top\boldsymbol{\xi}\right) \quad (112)$$

Let $\mathbf{X} = \mathbf{U}\Sigma^{\frac{1}{2}}\mathbf{V}^\top$ under the SVD decomposition, with $\Sigma = \mathrm{diag}(\sigma_k)_{k\in[r]}$, where $r = \mathrm{rank}(\mathbf{X})$ and $\sigma_{\max} = \sigma_1 \geq \cdots\sigma_k \geq \sigma_{k+1}\cdots \geq \sigma_{\min} = \sigma_r > \sigma_{r+1} = \cdots = 0$. If the step size $\alpha$ satisfies $0 < \alpha < \frac{2}{\sigma_{\max}+\beta}$. The update $\mathbf{a}^{(t)}$ converge to the least square solution $\hat{\mathbf{a}} := \mathbf{Q}^\dagger\mathbf{X}^\top\mathbf{y}^*$, but this solution can not give rise to generalization when $N < n$.

**Theorem C.51.** *For all $p > 0$, let define $\rho_p := \left\|\mathbb{I}_n - \alpha\left(\mathbf{X}^\top\mathbf{X} + \beta\mathbb{I}_n\right)\right\|_{p\to p}$. Assume the learning rate satisfies $0 < \alpha < \frac{2}{\sigma_{\max}(\mathbf{X}^\top\mathbf{X})+\beta}$. Then $\|\mathbf{a}^{(t)} - \hat{\mathbf{a}}\|_p \leq \rho_p^{t-1}\|\mathbf{a}^{(1)} - \hat{\mathbf{a}}\|_p$ for all $t \geq 1$. On the other hand, for $N < n$, $\|\hat{\mathbf{a}} - \mathbf{a}^*\|_2^2 \geq \|(\mathbb{I}_n - \mathbf{V}\mathbf{V}^\top)\mathbf{a}^*\|_2^2$. In particular, if $\mathbf{a}^*$ has a nonzero component orthogonal to the column space of $\mathbf{V}$, then $\hat{\mathbf{a}}$ cannot perfectly generalize to $\mathbf{a}^*$.*

*Proof.* In Lemma C.52, we show that $\|\mathbf{a}^{(t+1)} - \hat{\mathbf{a}}\|_p \leq \rho_p^t\|\mathbf{a}^{(1)} - \hat{\mathbf{a}}\|_p \quad \forall t \geq 0$. So for $t \geq \ln\left(\eta/\|\mathbf{a}^{(1)} - \hat{\mathbf{a}}\|_p\right)/\ln\rho_p$, we have $\|\mathbf{a}^{(t+1)} - \hat{\mathbf{a}}\|_p \leq \eta$. Consider the regularized least-squares estimator $\hat{\mathbf{a}} = \left(\mathbf{X}^\top\mathbf{X} + \beta\mathbb{I}_n\right)^\dagger\mathbf{X}^\top\mathbf{y}^* = \mathbf{V}\left(\Sigma + \beta\mathbb{I}\right)^{-1}\Sigma^{\frac{1}{2}}\mathbf{U}^\top\mathbf{y}^*$. We have $\mathbf{V}\mathbf{V}^\top\hat{\mathbf{a}} = \hat{\mathbf{a}}$, i.e. $\hat{\mathbf{a}} \in \mathrm{Col}(\mathbf{V})$. Let decompose $\mathbf{a}^*$ into two orthogonal components; $\mathbf{a}^* = \mathbf{a}_\| + \mathbf{a}_\perp$, where $\mathbf{a}_\| := \mathbf{V}\mathbf{V}^\top\mathbf{a}^* \in \mathrm{Col}(\mathbf{V})$ and $\mathbf{a}_\perp := (\mathbb{I}_n - \mathbf{V}\mathbf{V}^\top)\mathbf{a}^* \in \mathrm{Col}(\mathbf{V})^\perp$. Since $\hat{\mathbf{a}} \in \mathrm{Col}(\mathbf{V})$, $\mathbf{V}\mathbf{V}^\top(\hat{\mathbf{a}} - \mathbf{a}_\|) = \hat{\mathbf{a}} - \mathbf{a}_\|$ and $\mathbf{V}\mathbf{V}^\top\mathbf{a}_\perp = 0$ by orthogonality. Thus, we can express the error as $\hat{\mathbf{a}} - \mathbf{a}^* = \hat{\mathbf{a}} - (\mathbf{a}_\| + \mathbf{a}_\perp) = (\hat{\mathbf{a}} - \mathbf{a}_\|) - \mathbf{a}_\perp$. Because $\hat{\mathbf{a}} - \mathbf{a}_\| \in \mathrm{Col}(\mathbf{V})$ and $\mathbf{a}_\perp$ lies in the orthogonal complement of $\mathrm{Col}(\mathbf{V})$, these two vectors are orthogonal. Hence, $\|\hat{\mathbf{a}} - \mathbf{a}^*\|_2^2 = \|\hat{\mathbf{a}} - \mathbf{a}_\|\|_2^2 + \|\mathbf{a}_\perp\|_2^2 \geq \|\mathbf{a}_\perp\|_2^2 = \|(\mathbb{I}_n - \mathbf{V}\mathbf{V}^\top)\mathbf{a}^*\|_2^2. \quad\square$

**Lemma C.52.** *If* $\alpha \in (0, \frac{2}{\sigma_{\max}+\beta})$, *then* $F(\mathbf{a}^{(t)}) \to 0$ *as* $t \to \infty$; *where* $F(\mathbf{a}) = 0 \iff \mathbf{a} = \hat{\mathbf{a}} + \left(\mathbb{I}_n - \left(\mathbf{X}^\top\mathbf{X} + \beta\mathbb{I}_n\right)^\dagger \left(\mathbf{X}^\top\mathbf{X} + \beta\mathbb{I}_n\right)\right)\mathbf{c} = \hat{\mathbf{a}} + \left(\mathbb{I}_n - \mathbf{V}\mathbf{V}^\top\right)\mathbf{c} \quad \forall \mathbf{c} \in \mathbb{R}^n$. *Also,* $\|\mathbf{a}^{(t+1)} - \hat{\mathbf{a}}\|_p \leq \rho_p^t \|\mathbf{a}^{(1)} - \hat{\mathbf{a}}\|_p \quad \forall t$.

*Proof.* The solutions of $F(\mathbf{a}) = 0$ are

$$
\begin{aligned}
&\left(\mathbf{X}^\top\mathbf{X} + \beta\mathbb{I}_n\right)\mathbf{a} = \mathbf{X}^\top\mathbf{y}^* \\
\iff &\left(\mathbf{X}^\top\mathbf{X} + \beta\mathbb{I}_n\right)\mathbf{a} = \mathbf{X}^\top\mathbf{y}^* = \mathbf{X}^\top\mathbf{X}\mathbf{a}^* + \mathbf{X}^\top\boldsymbol{\xi} = \mathbf{V}\Sigma\mathbf{V}^\top\mathbf{a}^* + \mathbf{V}\Sigma^{\frac{1}{2}}\mathbf{U}^\top\boldsymbol{\xi} \\
\iff &\mathbf{a} = \left(\mathbf{X}^\top\mathbf{X} + \beta\mathbb{I}_n\right)^\dagger \mathbf{X}^\top\mathbf{y}^* + \left(\mathbb{I}_n - \left(\mathbf{X}^\top\mathbf{X} + \beta\mathbb{I}_n\right)^\dagger \left(\mathbf{X}^\top\mathbf{X} + \beta\mathbb{I}_n\right)\right)\mathbf{c} = \hat{\mathbf{a}} + \left(\mathbb{I}_n - \mathbf{V}\mathbf{V}^\top\right)\mathbf{c} \quad \forall \mathbf{c} \in \mathbb{R}^n
\end{aligned}
\tag{113}
$$

Let $\tilde{\Sigma} = \mathbb{I} - \alpha\left(\Sigma + \beta\mathbb{I}\right)$; $\mathbf{A} = \mathbb{I}_n - \alpha\left(\mathbf{X}^\top\mathbf{X} + \beta\mathbb{I}_n\right) = \mathbf{V}\tilde{\Sigma}\mathbf{V}^\top$ and $\mathbf{w} = \alpha\mathbf{X}^\top\mathbf{y}^* = \alpha\left(\mathbf{X}^\top\mathbf{X}\mathbf{a}^* + \mathbf{X}^\top\boldsymbol{\xi}\right) = \alpha\left(\mathbf{V}\Sigma\mathbf{V}^\top\mathbf{a}^* + \mathbf{V}\Sigma^{\frac{1}{2}}\mathbf{U}^\top\boldsymbol{\xi}\right)$; so that

$$
\mathbf{a}^{(t+1)} = \mathbf{A}\mathbf{a}^{(t)} + \mathbf{w} = \mathbf{A}^t\mathbf{a}^{(1)} + \left(\sum_{i=0}^{t-1}\mathbf{A}^i\right)\mathbf{w} = \mathbf{A}^t\mathbf{a}^{(1)} + (\mathbb{I} - \mathbf{A})^\dagger\left(\mathbb{I} - \mathbf{A}^t\right)\mathbf{w}
\tag{114}
$$

As $t \longrightarrow \infty$, $\tilde{\Sigma}^t \longrightarrow 0$, so $\mathbf{A}^t = \mathbf{V}\tilde{\Sigma}^t\mathbf{V}^\top \longrightarrow 0$. We have

$$
\begin{aligned}
\sum_{i=0}^{t-1}\mathbf{A}^i &= \mathbb{I}_n + \sum_{i=1}^{t-1}\mathbf{V}\tilde{\Sigma}^i\mathbf{V}^\top = \mathbb{I}_n - \mathbf{V}\mathbf{V}^\top + \sum_{i=0}^{t-1}\mathbf{V}\tilde{\Sigma}^i\mathbf{V}^\top \\
&= \mathbb{I}_n - \mathbf{V}\mathbf{V}^\top + \mathbf{V}\operatorname{diag}\left(\sum_{i=0}^{t-1}\tilde{\sigma}_k^i\right)_k\mathbf{V}^\top = \mathbb{I}_n - \mathbf{V}\mathbf{V}^\top + \mathbf{V}\operatorname{diag}\left(\frac{1-\tilde{\sigma}_k^t}{1-\tilde{\sigma}_k}\right)_k\mathbf{V}^\top \\
&= \mathbb{I}_n - \mathbf{V}\mathbf{V}^\top + \mathbf{V}\left(\mathbb{I} - \tilde{\Sigma}\right)^{-1}\left(\mathbb{I} - \tilde{\Sigma}^t\right)\mathbf{V}^\top \\
&\longrightarrow \mathbb{I}_n - \mathbf{V}\mathbf{V}^\top + \mathbf{V}\left(\mathbb{I} - \tilde{\Sigma}\right)^{-1}\mathbf{V}^\top = \mathbb{I}_n - \mathbf{V}\mathbf{V}^\top + \frac{1}{\alpha}\mathbf{V}\left(\Sigma + \beta\mathbb{I}_n\right)^{-1}\mathbf{V}^\top \text{ as } t \longrightarrow \infty
\end{aligned}
\tag{115}
$$

So, as $t \longrightarrow \infty$,

$$
\begin{aligned}
\mathbf{a}^{(t+1)} &= \left(\sum_{i=0}^{\infty}\mathbf{A}^i\right)\mathbf{w} = \alpha\left(\mathbb{I}_r - \mathbf{V}\mathbf{V}^\top + \frac{1}{\alpha}\mathbf{V}\left(\Sigma + \beta\mathbb{I}_r\right)^{-1}\mathbf{V}^\top\right)\left(\mathbf{V}\Sigma\mathbf{V}^\top\mathbf{a}^* + \mathbf{V}\Sigma^{\frac{1}{2}}\mathbf{U}^\top\boldsymbol{\xi}\right) \\
&= \mathbf{V}\left(\Sigma + \beta\mathbb{I}\right)^{-1}\mathbf{V}^\top\left(\mathbf{V}\Sigma\mathbf{V}^\top\mathbf{a}^* + \mathbf{V}\Sigma^{\frac{1}{2}}\mathbf{U}^\top\boldsymbol{\xi}\right) + \left(\mathbb{I}_n - \mathbf{V}\mathbf{V}^\top\right)\mathbf{c} \text{ with } \mathbf{c} = \mathbf{w} \\
&= \mathbf{V}\left(\Sigma + \beta\mathbb{I}\right)^{-1}\mathbf{V}^\top\left(\mathbf{V}\Sigma\mathbf{V}^\top\mathbf{a}^* + \mathbf{V}\Sigma^{\frac{1}{2}}\mathbf{U}^\top\boldsymbol{\xi}\right) = \hat{\mathbf{a}}
\end{aligned}
\tag{116}
$$

We have $\mathbf{A}\hat{\mathbf{a}}+\mathbf{c} = \hat{\mathbf{a}}$, so $\mathbf{a}^{(t+1)}-\hat{\mathbf{a}} = \mathbf{A}(\mathbf{a}^{(t)}-\hat{\mathbf{a}}) = \mathbf{A}^t(\mathbf{a}^{(1)}-\hat{\mathbf{a}})$, which implies $\|\mathbf{a}^{(t+1)}-\hat{\mathbf{a}}\|_p \leq \|\mathbf{A}^t\|_{p\to p}\|\mathbf{a}^{(1)}-\hat{\mathbf{a}}\|_2$. $\qquad\square$

## D. Other Iterative Method for $\ell_1$ and $\ell_*$ Minimization

### D.1. Sparse Recovery

For the experiments of this section, we use $(n, \zeta, \alpha, \beta) = (10^2, 10^{-6}, 10^{-1}, 10^{-5})$. We solve the sparse recovery problem using the projected subgradient (Figure 12) and the soft-thresholding algorithm (Figure 13). We observe a grokking-like pattern similar to the subgradient case. One training step is enough for the projected subgradient to get zero training error. This further shows that generalization is driven by $\alpha$ and, more importantly, $\beta$.

**Projected subgradient**   To ensure memorization, we can use the projected subgradient for problem of minimizing $\|\mathbf{a}\|_1$ subject to the constraint $\mathbf{X}\mathbf{a} = \mathbf{y}^*$, where at each step the update (using now just $\beta H(\mathbf{a})$ as gradient, not the whole $F(\mathbf{a})$) is projected onto the constraint set. In our case, the update write $\mathbf{a}^{(t+1)} = \Pi\left(\mathbf{a}^{(t)} - \alpha\beta H(\mathbf{a}^{(t)})\right)$ with $\Pi(\mathbf{a}) = \mathbf{a} - \mathbf{X}^\top\left(\mathbf{X}\mathbf{X}^\top\right)^\dagger\left(\mathbf{X}\mathbf{a} - \mathbf{y}^*\right)$ the projection of $\mathbf{a}$ on the set $\{\mathbf{a}, \mathbf{X}\mathbf{a} = \mathbf{y}^*\}$. We can show that the $\ell_1$ optimal

gap of this method enjoys the same bound $\mathcal{O}(\alpha\beta)$ given above (Theorem C.29) for the non-projected case. We have $\min_{t_1 \le t \le t_2} \left( \|\mathbf{a}^{(t)}\|_1 - \|\mathbf{a}^*\|_1 \right) \le \frac{\|\mathbf{a}^{(t_1)} - \mathbf{a}^*\|_2^2 + (\beta\alpha)^2(t_2-t_1)n}{2\beta\alpha(t_2-t_1)} \xrightarrow[t_2 \to 0]{} \frac{\alpha\beta}{2}$. The proof is the following:

$$
\begin{aligned}
0 \le \|\mathbf{a}^{(t_2+1)} - \mathbf{a}^*\|_2^2 &= \|\Pi\left(\mathbf{a}^{(t_2)} - \alpha\beta \cdot H(\mathbf{a}^{(t_2)})\right) - \mathbf{a}^*\|_2^2 \\
&\le \|\mathbf{a}^{(t_2)} - \mathbf{a}^* - \alpha\beta \cdot H(\mathbf{a}^{(t_2)})\|_2^2 \\
&= \|\mathbf{a}^{(t_2)} - \mathbf{a}^*\|_2^2 - 2\alpha\beta(\mathbf{a}^{(t_2)} - \mathbf{a}^*)^\top H(\mathbf{a}^{(t_2)}) + \beta^2\alpha^2\|H(\mathbf{a}^{(t_2)})\|_2^2 \\
&\le \|\mathbf{a}^{(t_2)} - \mathbf{a}^*\|_2^2 - 2\beta\alpha\left(\|\mathbf{a}^{(t_2)}\|_1 - \|\mathbf{a}^*\|_1\right) + \beta^2\alpha^2\|H(\mathbf{a}^{(t_2)})\|_2^2 \text{ (by the definition of } H) \\
&\le \|\mathbf{a}^{(t_1)} - \mathbf{a}^*\|_2^2 - 2\beta\alpha\sum_{t=t_1}^{t_2}\left(\|\mathbf{a}^{(t)}\|_1 - \|\mathbf{a}^*\|_1\right) + \beta^2\alpha^2\sum_{t=t_1}^{t_2}\|H(\mathbf{a}^{(t)})\|_2^2
\end{aligned}
\tag{117}
$$

This implies

$$
\begin{aligned}
&2\beta\left(\sum_{t=t_1}^{t_2}\alpha\right)\min_{t_1 \le t \le t_2}\left(\|\mathbf{a}^{(t)}\|_1 - \|\mathbf{a}^*\|_1\right) \le \|\mathbf{a}^{(t_1)} - \mathbf{a}^*\|_2^2 + \beta^2\alpha^2\sum_{t=t_1}^{t_2}\|H(\mathbf{a}^{(t)})\|_2^2 \\
&\iff \min_{t_1 \le t \le t_2}\left(\|\mathbf{a}^{(t)}\|_1 - \|\mathbf{a}^*\|_1\right) \le \frac{\|\mathbf{a}^{(t_1)} - \mathbf{a}^*\|_2^2 + \beta^2\alpha^2(t_2-t_1)\max_{t_1 \le t \le t_2}\|H(\mathbf{a}^{(t)})\|_2^2}{2\beta\alpha(t_2-t_1)}
\end{aligned}
\tag{118}
$$

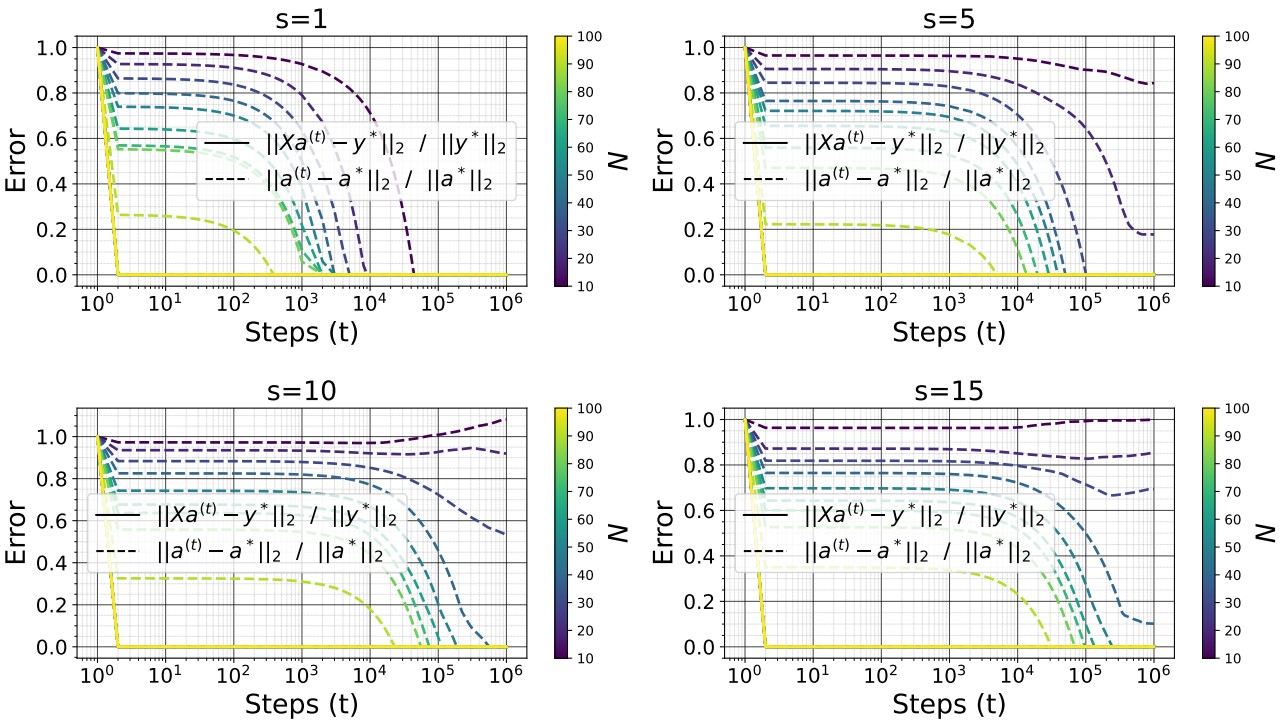

Figure 12: Sparse recovery with the projected subgradient descent method.

**Proximal Gradient Descent**  We have $\mathbf{a} - \alpha G(\mathbf{a}) = \arg\min_{\mathbf{c}} g(\mathbf{a}) + (\mathbf{c} - \mathbf{a})^\top G(\mathbf{a}) + \frac{1}{2\alpha}\|\mathbf{c} - \mathbf{a}\|_2^2$. So

$$
\begin{aligned}
\mathbf{a} - \alpha F(\mathbf{a}) &\approx \arg\min_{\mathbf{c}} g(\mathbf{a}) + (\mathbf{c} - \mathbf{a})^\top G(\mathbf{a}) + \frac{1}{2\alpha}\|\mathbf{c} - \mathbf{a}\|_2^2 + \beta\|\mathbf{c}\|_1 \\
&= \arg\min_{\mathbf{c}} \frac{1}{2\alpha}\|\mathbf{c} - (\mathbf{a} - \alpha G(\mathbf{a}))\|_2^2 + \beta\|\mathbf{c}\|_1 \\
&= \Pi_\alpha\left(\mathbf{a} - \alpha G(\mathbf{a})\right)
\end{aligned}
$$

with $\Pi_\alpha$ the proximal mapping for $\mathbf{c} \to \beta\|\mathbf{c}\|_1$, $\Pi_\alpha(\mathbf{a}) = \arg\min_{\mathbf{c}} \frac{1}{2\alpha}\|\mathbf{c}-\mathbf{a}\|_2^2 + \beta\|\mathbf{c}\|_1 = \arg\min_{\mathbf{c}} \frac{1}{2}\|\mathbf{c}-\mathbf{a}\|_2^2 + \alpha\beta\|\mathbf{c}\|_1 = S_{\alpha\beta}(\mathbf{a})$, where $S_\gamma(\mathbf{a}) = \text{sign}(\mathbf{a}) \odot \max(|\mathbf{a}| - \gamma, 0)$ the soft-thresholding operator[9],

$$
S_\gamma(\mathbf{a})_i = \begin{cases} \mathbf{a}_i - \gamma & \text{if } \mathbf{a}_i > \gamma \\ 0 & \text{if } -\gamma \leq \mathbf{a}_i \leq \gamma \\ \mathbf{a}_i + \gamma & \text{if } \mathbf{a}_i < -\gamma \end{cases}
$$

The final form of the update, known as the Iterative soft-thresholding algorithm (ISTA) (Daubechies et al., 2003), is then $\mathbf{a}^{(t+1)} = S_{\alpha\beta}\left(\mathbf{a}^{(t)} - \alpha G(\mathbf{a}^{(t)})\right) \quad \forall t > 1$. Let $L = \|\mathbf{X}^\top\mathbf{X}\|_{2\to2} = \sigma_{\max}(\mathbf{X}^\top\mathbf{X})$ be the Lipschitz constant for $G$. If $\alpha \leq 1/L$, then $\min_{t\leq T}(f(\mathbf{a}^{(t)}) - f(\mathbf{a}^*)) \leq \frac{\|\mathbf{a}^{(1)}-\mathbf{a}^*\|_2}{2\alpha T} \xrightarrow[T\to0]{} 0$ for the ISTA. We applied a standard bound on proximal gradient descent (Tibshirani, 2015) for a function of the form $f = g + h : \mathbb{R}^n \to \mathbb{R}$. Such result state that the proximal gradient descent with fixed step size $\alpha \leq 1/L$ satisfies $\min_{t\leq T}(f(\mathbf{a}^{(t)}) - f^*) \leq \frac{\|\mathbf{a}^{(1)}-\mathbf{a}^*\|_2^2}{2\alpha T}$ when $g$ is convex, differentiable, $\text{dom}(g) = \mathbb{R}^n$, $\nabla g$ is Lipschitz continuous with constant $L > 0$; and $h$ is convex and its proximal map $\Pi_\alpha$ can be evaluated.

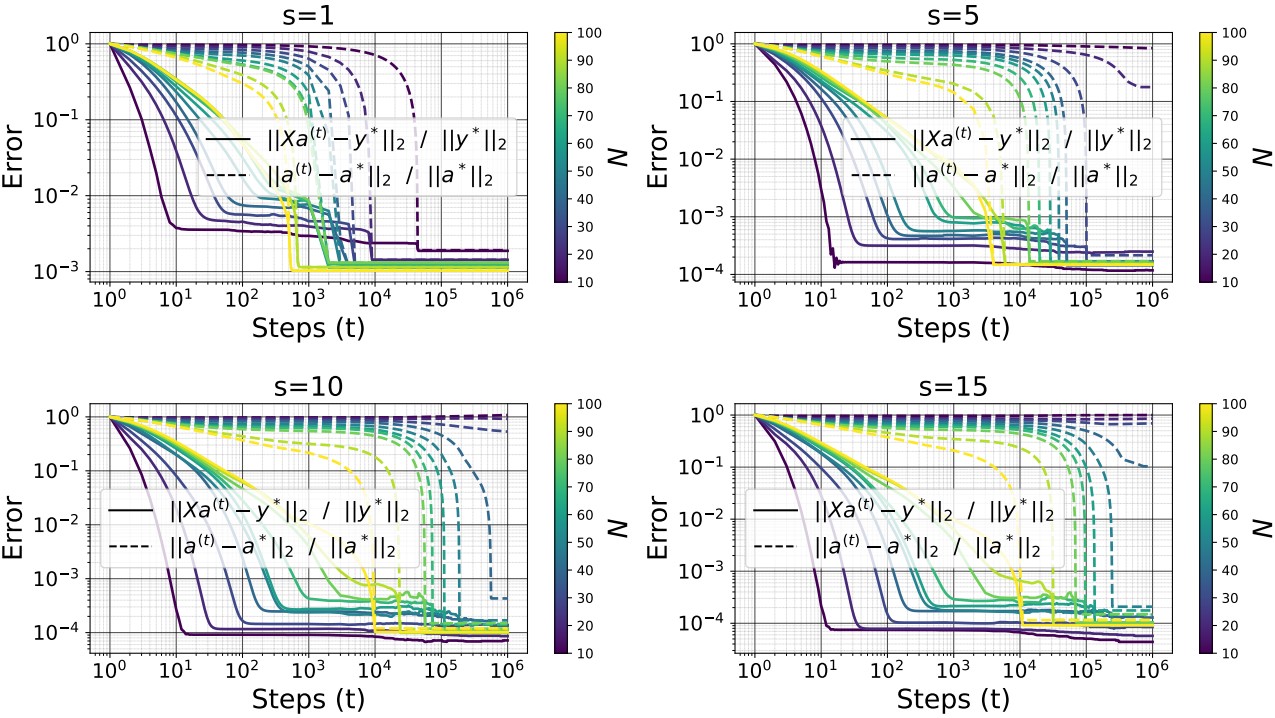

Figure 13: Sparse recovery with the soft-thresholding algorithm (ISTA)

### D.2. Low Rank Matrix Factorization

For the experiments of this section we use $(n_1, n_2, r, N, \zeta, \alpha, \beta) = (10, 10, 2, 70, 10^{-6}, 10^{-1}, 10^{-4})$ for a matrix sensing. Figure 14 shows the results for the projected subgradient and Figure 15 shows the results for the proximal gradient method.

**Projected subgradient**  At each step, the update (using now just $\beta H(\mathbf{A})$ as gradient) is projected onto the constraint set. In our case, the update write $\mathbf{A}^{(t+1)} = \Pi\left(\mathbf{A}^{(t)} - \alpha\beta H(\mathbf{A}^{(t)})\right)$ with $\Pi$ the projection on the set $\{\mathbf{A}, \mathbf{X} \text{vec}\, \mathbf{A} = \mathbf{y}^*\}$.

**Proximal Gradient Descent**  We have $\mathbf{A} - \alpha F(\mathbf{A}) = \Pi_\alpha(\mathbf{A} - \alpha G(\mathbf{A}))$ where $\Pi_\alpha$ is the proximal mapping for $\mathbf{B} \to \beta\|\mathbf{B}\|_*$, $\Pi_\alpha(\mathbf{A}) = \arg\min_{\mathbf{B}} \frac{1}{2\alpha}\|\mathbf{B} - \mathbf{A}\|_F^2 + \beta\|\mathbf{B}\| = S_{\alpha\beta}(\mathbf{A})$ with $S_\gamma(\mathbf{A}) = \mathbf{U}\max(\Sigma - \gamma, 0)\mathbf{V}^\top$ the soft-thresholding operator for $\mathbf{A} = \mathbf{U}\Sigma\mathbf{V}^\top$ under SVD, where $\max(\Sigma - \gamma, 0)_{ij} = \delta_{ij}\max(\Sigma_{ij} - \gamma, 0)$. The final form of the update is then $\mathbf{A}^{(t+1)} = S_{\alpha\beta}\left(\mathbf{A}^{(t)} - \alpha G(\mathbf{A}^{(t)})\right) \quad \forall t > 1$.

---

[9]On $\mathbb{C}$, the soft-thresholding operator $S_\gamma(\mathbf{a}) = \text{sign}(\mathbf{a}) \odot \max(|\mathbf{a}| - \gamma, 0)$ only shrinks the magnitude and keeps the phase fixed.

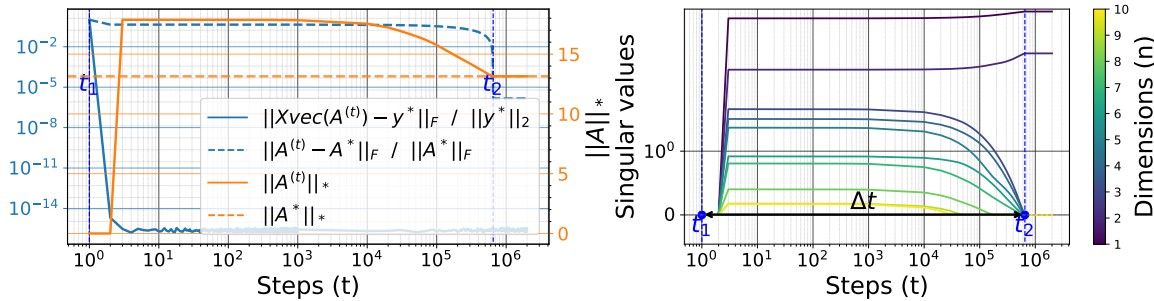

Figure 14: Matrix sensing with the projected subgradient method.

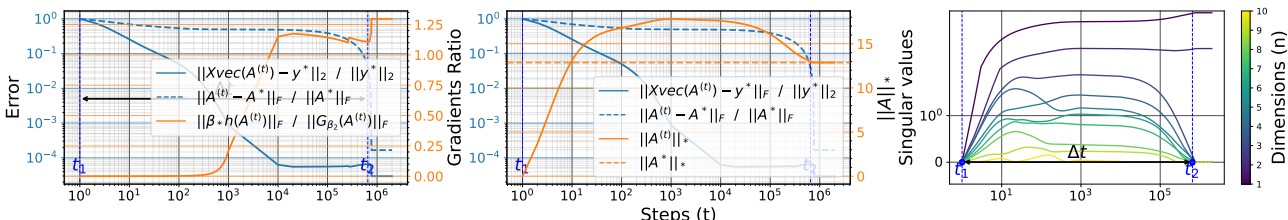

Figure 15: Matrix sensing with the Proximal Gradient Descent method.

# E. Grokking Without Understanding

Consider the sparse recovery problem (the explanation below also holds for matrix factorization). We start the optimization at $\mathbf{a}^{(1)} \overset{iid}{\sim} \zeta \mathcal{N}(0, 1/n)$ with $\zeta \geq 0$ the initialization scale. With a small initialization, $\ell_1$ regularization is sufficient for generalization to happen, provided $N$ is large enough and $\ell_2$ regularization is not very large. If the scale at initialization is large, then adding $\ell_2$ regularization is necessary to generalize, but is it sufficient? That is, can we generalize to the problem studied here with only $\ell_2$ regularization?

As shown in Section C.8, the answer to this question is no. But what we want to illustrate here is a phenomenon that contradicts previous art (Liu et al., 2023a; Lyu et al., 2023), namely that *in the over-parametrized regime ($N < n$ in our case), large initialization and non-zero weight decay do not always lead to grokking*. What happens is that, because of the large initialization, a more or less abrupt transition is observed in the generalization error during training, corresponding to a transition in the $\ell_2$ norm of the model parameters. But this can not be called grokking because the model only converges to a sub-

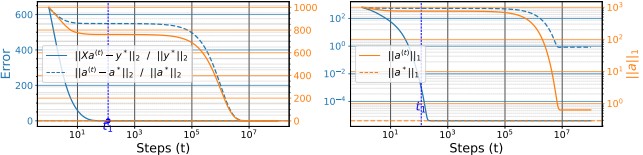

Figure 16: Relative errors and $\|\mathbf{a}^{(t)}\|_1$; with large initialization scale $\zeta = 10^1$ and small weights decay $\beta = 10^{-5}$. Without visualization of the error on a logarithmic scale (left), it looks like grokking has occurred, whereas this is not the case (right).

optimal solution. What is more, this transition appears even if the problem posed admits no solution, e.g., sparse recovery or matrix completion with a number $N$ of examples far below the theoretical limit required for the solution to the problem posed to be the optimal solution (by any method whatsoever). This transition appears abrupt just because the training error is large at the beginning of training, since the model's outputs are large. When its $\ell_2$ norm becomes small, its outputs also become small, leading to a transition in error. In Figure 16, without visualization of the error on a logarithmic scale, it looks like grokking has occurred, whereas this is not the case. For this figure we use $(n, s, N, \alpha) = (100, 5, 30, 10^{-1})$.

We call this phenomenon "grokking without understanding" like Levi et al. (2024) who illustrated it in the case of linear classification. They show that the sharp increase in generalization accuracy may often not imply a transition from "memorization" to "understanding" but can be an artifact of the accuracy measure. But in our case, we are not using any significant scale at initialization (we focus on $0 \leq \zeta \leq 10^{-5}$) and are not dealing with the generalization measure problem since our test error is directly the recovery error in the function space, not the accuracy.

We hypothesize that the interplay between large initialization and small non-zero weight decay that leads to grokking as predicted (provably) by Lyu et al. (2023) does not hold in our setting because our model violates they *Assumption 3.2*. Let $y_{\mathbf{a}}(\mathbf{x}) = \mathbf{a}^\top \mathbf{x}$ denote our model.

- *Assumption 3.1* (Lyu et al., 2023): For all $\mathbf{x} \in \mathbb{R}^n$, the function $\mathbf{a} \to y_{\mathbf{a}}(\mathbf{X})$ is 1-homogeneous since $y_{c\mathbf{a}}(\mathbf{x}) = c^1 y_{\mathbf{a}}(\mathbf{x})$ for all $c > 0$.

- *Assumption 3.2* (Lyu et al., 2023): for $\zeta = 0$, $y_{\mathbf{a}^{(1)}}(\mathbf{x}) = 0$ for all $\mathbf{x}$ (there is generalization in this case with $\ell_1$), but if $\zeta > 0$ (for instance $\zeta$ large), this is (almost surely) no longer true. So, this assumption is violated (with high probability).

- *Assumption 3.8* (Lyu et al., 2023): The NTK (Neural Tangent Kernel) features of training samples $\{\nabla_{\mathbf{a}} y_{\mathbf{a}}(\mathbf{X}_i)\}_{i \in [N]}$ are linearly independent (almost surely) at initialisation $\mathbf{a} = \mathbf{a}^{(1)}$. In fact, $\nabla_{\mathbf{a}} y_{\mathbf{a}}(\mathbf{x}) = \mathbf{x} \ \forall \mathbf{x}$. In the over-parametrized regime $N < n$, If $\mathbf{M} \in \mathbb{R}^{N \times n}$ has entries iid from a normal distribution, then the NTK features $\{\mathbf{X}_i\}_{i \in [N]}$ are linearly independent with high probability (because the rank of $\mathbf{X}$ is $N$ with high probability), so this assumption is verified.

# F. Implicit Bias of the Depth

### F.1. Deep Sparse Recovery

For sparse recovery, let now use the parameterization $\mathbf{a} = \odot_{k=1}^L \mathbf{A}_k \in \mathbb{R}^n$, with $\mathbf{A} \in \mathbb{R}^{L \times n}$. This corresponds to a linear network with $L$ layers, where each hidden layer has the parameter $\text{diag}(\mathbf{A}_k) \in \mathbb{R}^{n \times n}$—with this, increasing $L$ leads to overparameterization without altering the expressiveness of the function class $\mathbf{a} \to \mathcal{F}_{\mathbf{a}}(\mathbf{x}) = \mathbf{x}^\top \mathbf{a}$, since the model remains linear with respect to the input $\mathbf{x}$. Unlike the shallow case ($L = 1$), there is no need for $\ell_1$ regularization to generalize when $L \geq 2$ (and the initialization scale is small), as the experiments of this section suggest. With depth, the update for the whole iteration (which is now replaced by a product of matrices and a vector) is similar to the shallow case, but with a preconditioner in front of the gradient. This preconditioner makes it possible to recover the sparse signal without any regularization.

We let $h(\mathbf{A}_i) = \frac{1}{2}\|\mathbf{A}_i\|_2$ so that $H(\mathbf{A}_i) = \mathbf{A}_i$. Also $g(\mathbf{a}) = \frac{1}{2}\|\mathbf{y}(\mathbf{a}) - \mathbf{y}^*\|_2^2 = \frac{1}{2}\mathbf{a}^\top \mathbf{X}^\top \mathbf{X} \mathbf{a} - (\mathbf{X}^\top \mathbf{X} \mathbf{a}^* + \mathbf{X}^\top \boldsymbol{\xi})^\top \mathbf{a} + \frac{1}{2}\|\mathbf{X}\mathbf{a}^* + \boldsymbol{\xi}\|_2^2$. Let $G(\mathbf{a}) := \frac{\partial g(\mathbf{a})}{\partial \mathbf{a}} = \mathbf{X}^\top \mathbf{X}(\mathbf{a} - \mathbf{a}^*) - \mathbf{X}^\top \boldsymbol{\xi}$. The gradient for each $\mathbf{A}_i$ is $G(\mathbf{A}_i) := \frac{\partial g(\mathbf{a})}{\partial \mathbf{A}_i} = \frac{\partial \mathbf{a}}{\partial \mathbf{A}_i} \frac{\partial g(\mathbf{a})}{\partial \mathbf{a}} = \text{diag}(\odot_{k \neq i} \mathbf{A}_k) G(\mathbf{a})$. We start the optimization at $\mathbf{A}_i^{(1)} \overset{iid}{\sim} \zeta \mathcal{N}(0, 1/n)$ with $\zeta \geq 0$ the initialization scale. The update rule for each $\mathbf{A}_i$ is

$$\mathbf{A}_i^{(t+1)} = \mathbf{A}_i^{(t)} - \alpha G(\mathbf{A}_i^{(t)}) - \alpha\beta H(\mathbf{A}_i^{(t)}) = (1 - \alpha\beta)\mathbf{A}_i^{(t)} - \alpha \, \text{diag}(\odot_{k \neq i} \mathbf{A}_k^{(t)}) G(\mathbf{a}^{(t)}) \tag{119}$$

Without ovaparametrization ($L = 1$), the update is unconditioned and progresses uniformly in all directions. So without $\ell_1$-regularization, there is no mechanism to enforce sparsity, and perfect recovery of $\mathbf{a}^*$ is impossible. For $L = 2$, let $\mathbf{c} := \mathbf{A}_1 \odot \mathbf{A}_1 + \mathbf{A}_2 \odot \mathbf{A}_2$.

$$
\begin{aligned}
\mathbf{a}^{(t+1)} &= \mathbf{A}_1^{(t+1)} \odot \mathbf{A}_2^{(t+1)} \\
&= \left((1 - \alpha\beta)\mathbf{A}_1^{(t)} - \alpha \, \text{diag}(\mathbf{A}_2^{(t)}) G(\mathbf{a}^{(t)})\right) \odot \left((1 - \alpha\beta)\mathbf{A}_2^{(t)} - \alpha \, \text{diag}(\mathbf{A}_1^{(t)}) G(\mathbf{a}^{(t)})\right) \\
&= (1 - \alpha\beta)^2 \mathbf{a}^{(t)} - \alpha(1 - \alpha\beta) \, \text{diag}(\mathbf{A}_1^{(t)} \odot \mathbf{A}_1^{(t)} + \mathbf{A}_2^{(t)} \odot \mathbf{A}_2^{(t)}) G(\mathbf{a}^{(t)}) + \alpha^2 \, \text{diag}(\mathbf{a}^{(t)}) G(\mathbf{a}^{(t)})^{\odot 2} \\
&= (1 - \alpha\beta)^2 \mathbf{a}^{(t)} - \alpha(1 - \alpha\beta)\mathbf{c}^{(t)} \odot G(\mathbf{a}^{(t)}) + \alpha^2 \mathbf{a}^{(t)} \odot G(\mathbf{a}^{(t)})^{\odot 2} \\
&\approx (1 - 2\alpha\beta)\mathbf{a}^{(t)} - \alpha\mathbf{c}^{(t)} \odot G(\mathbf{a}^{(t)}) \text{ for } \alpha \to 0
\end{aligned}
\tag{120}
$$

and

$$
\begin{aligned}
\mathbf{c}^{(t+1)} &= \mathbf{A}_1^{(t+1)} \odot \mathbf{A}_1^{(t+1)} + \mathbf{A}_2^{(t+1)} \odot \mathbf{A}_2^{(t+1)} \\
&= (1 - \alpha\beta)^2 \mathbf{A}_1^{(t)} \odot \mathbf{A}_1^{(t)} - 2\alpha(1 - \alpha\beta) \, \text{diag}(\mathbf{A}_1^{(t)} \odot \mathbf{A}_2^{(t)}) G(\mathbf{a}^{(t)}) + \alpha^2 \, \text{diag}(\mathbf{A}_2^{(t)} \odot \mathbf{A}_2^{(t)}) G(\mathbf{a}^{(t)})^{\odot 2} \\
&\quad + (1 - \alpha\beta)^2 \mathbf{A}_2^{(t)} \odot \mathbf{A}_2^{(t)} - 2\alpha(1 - \alpha\beta) \, \text{diag}(\mathbf{A}_2^{(t)} \odot \mathbf{A}_1^{(t)}) G(\mathbf{a}^{(t)}) + \alpha^2 \, \text{diag}(\mathbf{A}_1^{(t)} \odot \mathbf{A}_1^{(t)}) G(\mathbf{a}^{(t)})^{\odot 2} \\
&= (1 - \alpha\beta)^2 \mathbf{c}^{(t)} - 4\alpha(1 - \alpha\beta)\mathbf{a}^{(t)} \odot G(\mathbf{a}^{(t)}) + \alpha^2 \mathbf{c}^{(t)} \odot G(\mathbf{a}^{(t)})^{\odot 2} \\
&\approx (1 - 2\alpha\beta)\mathbf{c}^{(t)} - 4\alpha\mathbf{a}^{(t)} \odot G(\mathbf{a}^{(t)}) \text{ for } \alpha \to 0
\end{aligned}
\tag{121}
$$

The depth adds the preconditioning $\mathbf{P}^{(t)} = (1 - \alpha\beta)\operatorname{diag}(\mathbf{c}^{(t)})$ in front of the update for $\mathbf{a}$. This preconditioning mechanism seems to implicitly favor sparsity and, thus, a perfect recovery after memorization. In fact, when $\mathbf{c}_i^{(t)}$ goes to zero (which is the case when $\mathbf{a}_i^{(t)}$ is also small), the update becomes $\mathbf{a}_i^{(t+1)} \approx (1 - 2\alpha\beta)\mathbf{a}_i^{(t)}$, and thus push $\mathbf{a}_i^{(t+1)}$ to 0 at a geometric rate of $\mathcal{O}(1 - 2\alpha\beta)$. Otherwise, $\mathbf{c}_i^{(t)}$ (large) will amplify the gradient so that $\mathbf{c}_i^{(t)} G(\mathbf{a}^{(t)})_i$ dominates the update, which pushes $\mathbf{a}^{(t)}$ towards $\mathbf{a}^*$ (as the gradient $G(\mathbf{a}^{(t)})$ points towards a small error $\mathbf{a}^{(t)} - \mathbf{a}^*$ direction, particularly for full rank $\mathbf{X}$ and high signal to ratio regime).

With the deep parametrization above, we solve the sparse recovery problem using the subgradient descent method with $(n, s, \alpha, \beta) = (10^2, 5, 10^{-1}, 0)$, for different values of $N$ and $L \in \{1, 2, 3, 4\}$ (Figure 17 and 18). We use a small initialization scale $\zeta = 10^{-6}$ for $L = 1$ and $\zeta = 10^{-2}$ for $L > 1$. Here, initializing $\mathbf{A}$ too close to the origin (initialization scale $\zeta \to 0$) leads $\mathbf{a}$ to not change during training. The model is able to recover the true signal $\mathbf{a}^*$, and the generalization delay becomes extremely small (compared to the shallow case with non-zero $\ell_1$ coefficient) for $L = 2$ and disappears (ungrokking) for $L > 2$. As $L$ becomes larger, the phase transition to generalization becomes extremely abrupt. The loss decreases in a staircase fashion, with more or less long plateaus of suboptimal generalization error during training. This type of behavior is generally observed in the optimization of *Soft Committee Machines* (Biehl & Schwarze, 1995; Saad & Solla, 1995b;a; 1996; Engel & Broeck, 2001; Aubin et al., 2018; Goldt et al., 2020), which are two-layer linear or non-linear teacher-student systems, with the output layer of the student fixed to that of the teacher during training.

Additionally, for a fixed number $N$ of measures, the test error decreases with $L$, indicating that depth aids in finding the signal with a smaller number of measures, albeit at the expense of a longer training time. So, the depth seems to have the same effect on generalization as $\ell_1$. This is in accord with the result of Arora et al. (2018) in the context of matrix factorization. They show that introducing depth effectively turns gradient descent into a shallow (single-layer) training process equipped with a built-in preconditioning mechanism. This mechanism biases updates toward directions already explored by the optimization, serving as an acceleration technique that fuses momentum with adaptive step sizes. Furthermore, they demonstrate that depth-based overparameterization can substantially speed up training, even in straightforward convex tasks like linear regression with $\ell_p$ loss, $p > 2$.

Note that for $L \geq 2$, using a large-scale initialization and a small but non-zero $\ell_2$ regularization $\beta$ results in grokking, unlike the case of $L = 1$ that gives the "grokking without understanding" phenomenon. In this regime, when $L$ increases, the number of steps required for the model to move from memorization to generalization is reduced (grokking acceleration), and the generalization error at the end of training is considerably lower. Lyu et al. (2023) used a similar setup to show that an interplay between large initialization and small nonzero weights decay gives rise to grokking with the diagonal linear network $y(\mathbf{x}) = \left(\mathbf{u}^{\odot L} - \mathbf{v}^{\odot L}\right)^{\top} \mathbf{x}$ in the context of binary classification, but there did not study the impact of $L$ on the generalization delay, and instead focus on characterizing how sharp is the transition from memorization to generalization as a function of the initialization scale and the weight decay coefficient, and how long it takes for this transition to occurs. This diagonal linear network is also often used for sparse recovery problems (Vavskevivcius et al., 2019), but the focus is generally on its ability to recover the optimal solution, not grokking.

### F.2. Deep Matrix Factorization

For matrix factorization, let use the parameterization $\mathbf{A} = \prod_{k=1}^{L} \mathcal{A}_k$, with $\mathcal{A}_1 \in \mathbb{R}^{n_1 \times d}$, $\mathcal{A}_L \in \mathbb{R}^{d \times n_2}$, and $\mathcal{A}_i \in \mathbb{R}^{d \times d}$ for all $i \in (1, L)$. This corresponds to a linear network with $L$ layers, where each hidden layer has the parameter $\mathcal{A}_k$—with this, increasing $L$ leads to overparameterization without altering the expressiveness of the function class $\mathbf{A} \to \mathcal{F}_{\mathbf{A}}(\mathbf{x}) = \mathbf{x}^{\top} \operatorname{vec} \mathbf{A}$, since the model remains linear with respect to the input $\mathbf{x}$. Like in compressed sensing, there is no need for $\ell_*$ (or any other form of regularization) to generalize when $L \geq 2$ (and the initialization scale is small), unlike the shallow case ($L = 1$). This is an observation already made and proven in previous art (Gunasekar et al., 2017; Arora et al., 2019; Gidel et al., 2019; Gissin et al., 2019; Razin & Cohen, 2020; Li et al., 2020). Gunasekar et al. (2017); Arora et al. (2019) show that increasing $L$ implicitly biases $\mathbf{A}$ toward a low-rank solution, which oftentimes leads to more accurate recovery for sufficiently large $N$. In fact, with depth, the update for the whole iterate is similar to the shallow case but with a preconditioner in front of the gradient (like in section F.1). This preconditioner makes it possible to recover the low-rank matrix without any regularization and with fewer samples than in the shallow case (Arora et al., 2018; 2019). It is also shown specifically for this problem that initializing the model very far from the origin and using a small (but non-zero) weight decay leads to grokking (Lyu et al., 2023), i.e., the model first memorizes the observed entries, then after a long training period, converges to the sought matrices provided the number of such observe entries is large enough.

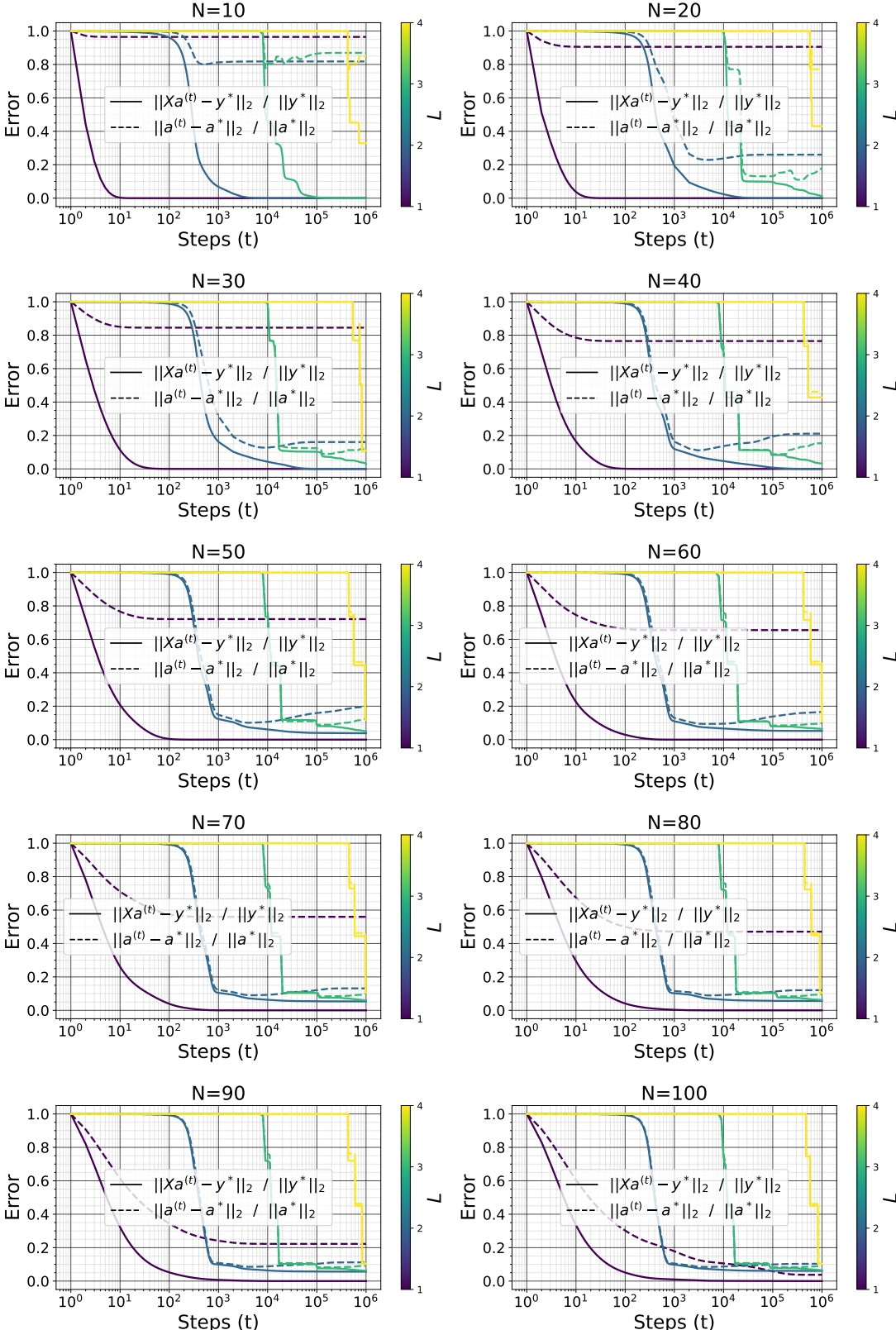

Figure 17: Training and recovery error as a function of the number of samples $N$ and the depth $L$.

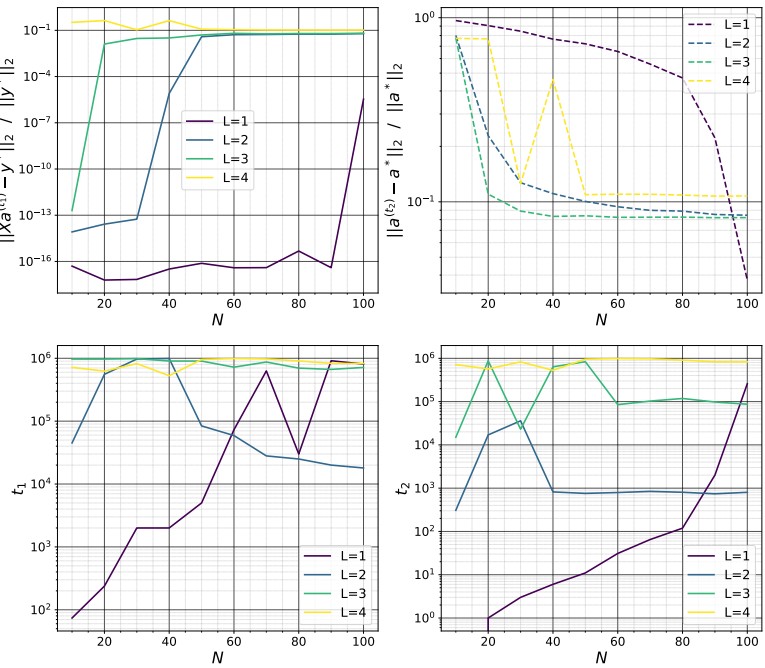

Figure 18: Training and error $\|\mathbf{X}\mathbf{a}^{(t_1)} - \mathbf{y}^*\|_2/\|\mathbf{y}^*\|_2$ and recovery error $\|\mathbf{a}^{(t_2)} - \mathbf{a}^*\|_2/\|\mathbf{a}^*\|_2$ (along with $t_1$ and $t_2$, the memorization and the generalization step) as a function of the number of sample $N$ and the depth $L$. The growth (as a function of $N$) in the test error for $L = 4$ is simply due to the fact that we did not optimize long enough for it to decrease.

# G. Amplifying Grokking through Data Selection

## G.1. Sparse Recovery

**Definition G.1** (Restricted Isometry Property (RIP) and Restricted Isometric Constant (RIC)). Let $\mathbf{A} \in \mathbb{R}^{m \times n}$ and $(s, \delta_s) \in [n] \times (0,1)$. The matrix $\mathbf{A}$ is said to satisfy the $(s, \delta_s)$-RIP if $(1 - \delta_s)\|\mathbf{x}\|_2^2 \leq \|\mathbf{A}\mathbf{x}\|_2^2 \leq (1 + \delta_s)\|\mathbf{x}\|_2^2$ for all $s$-sparse vector $\mathbf{x} \in \mathbb{R}^n$ (i.e. $\|\mathbf{x}\|_0 \leq s$). This is equivalent to saying that for every $J \subset [n]$ with $|J| = s$, $(1 - \delta_s)\|\mathbf{x}\|_2^2 \leq \|\mathbf{A}_{:,J}\mathbf{x}\|_2^2 \leq (1 + \delta_s)\|\mathbf{x}\|_2^2 \quad \forall \mathbf{x} \in \mathbb{R}^s$ where the submatrix $\mathbf{A}_{:,J} \in \mathbb{R}^{m \times s}$ of $\mathbf{A}$ is build by selecting the columns index in $J$. This condition is also equivalent to the statement $\|\mathbf{A}_{:,J}^\top \mathbf{A}_{:,J} - \mathbb{I}_s\|_{2 \to 2} \leq \delta_s$, which is finally equivalent to $\lambda\left(\mathbf{A}_{:,J}^\top \mathbf{A}_{:,J}\right) \in [1 - \delta_s, 1 + \delta_s]$ for all eigenvalues eigenvalue $\lambda\left(\mathbf{A}_{:,J}^\top \mathbf{A}_{:,J}\right)$ of $\mathbf{A}_{:,J}^\top \mathbf{A}_{:,J}$. We say that $\mathbf{A}$ satisfies $s$-RIP if it satisfies $(s, \delta_s)$-RIP with some $\delta_s \in (0,1)$. The $s$-RIC of $\mathbf{A}$ is defined as the infimum $\delta_s(\mathbf{A})$ of all possible $\delta_s$ such that $\mathbf{A} \in \mathbb{R}^{m \times n}$ satisfy the $(s, \delta_s)$-RIP. So, for all $\forall J \subset [n]$ with $|J| = s$, the condition number of $\mathbf{A}_{:,J}^\top \mathbf{A}_{:,J}$ is bounds from above by $\frac{1 + \delta_s(\mathbf{A})}{1 - \delta_s(\mathbf{A})}$, a the one of $\mathbf{A}_{:,J}$ by $\sqrt{\frac{1 + \delta_s(\mathbf{A})}{1 - \delta_s(\mathbf{A})}}$.

We say that a matrix $\mathbf{A}$ satisfies the RIP if $\delta_s(\mathbf{A})$ is small for reasonably large $s$. All the above definitions extend to any linear map $f : \mathbb{R}^n \to \mathbb{R}^m$. Note that $\delta_s(\mathbf{A}) \leq \delta_{s+1}(\mathbf{A})$ for all $\mathbf{A} \in \mathbb{R}^{m \times n}$ and $s \in [n]$.

Let $\mathcal{F} : \mathbb{R}^{m \times n} \to \mathbb{R}^q$ be a linear map and $(r, \delta_r) \in [n] \times (0,1)$. $f$ is said to satisfy $(r, \delta_r)$-RIP if for all rank-$r$ matrices $\mathbf{X} \in \mathbb{R}^{m \times n}$, $(1 - \delta_r)\|\mathbf{X}\|_F^2 \leq \|\mathcal{F}(\mathbf{X})\|_2^2 \leq (1 + \delta_r)\|\mathbf{X}\|_F^2$. We say that $\mathcal{F}$ satisfies $r$-RIP if $\mathcal{F}$ satisfies $(r, \delta_r)$-RIP with some $\delta_r \in (0,1)$, and the $r$-RIC of $\mathcal{F}$ is defined as the infimum $\delta_r(\mathcal{F})$ of all possible $\delta_r$ such that $\mathcal{F}$ satisfy the $(r, \delta_r)$-RIP.

**Definition G.2** (Coherence). The coherence between two matrices $\mathbf{A} \in \mathbb{R}^{q \times m}$ and $\mathbf{B} \in \mathbb{R}^{q \times n}$ is $\mu(\mathbf{A}, \mathbf{B}) = \max_{i \in [m], j \in [n]} \frac{|\langle \mathbf{A}_{:,i}, \mathbf{B}_{:,j} \rangle|}{\|\mathbf{A}_{:,i}\|\|\mathbf{B}_{:,j}\|} = \max_{i \in [m], j \in [n]} \frac{|[\mathbf{A}^\top \mathbf{B}]_{i,j}|}{\|\mathbf{A}_{:,i}\|\|\mathbf{B}_{:,j}\|}$. Coherence measures how similar or aligned two matrices or vectors are. Specifically, it measures how much overlap there is between the columns of $\mathbf{A}$ and $\mathbf{B}$. High coherence means they are similar or aligned, and low coherence (or incoherence) means they are very different. Incoherence is essentially the opposite of coherence. It refers to a low overlap or low similarity between the columns of $\mathbf{A}$ and $\mathbf{B}$.

The mutual coherence of a matrix $\mathbf{A} \in \mathbb{R}^{m \times n}$ is $\mu(\mathbf{A}) = \max_{(i,j) \in [m] \times [n], i \neq j} \frac{|\langle \mathbf{A}_{:,i}, \mathbf{A}_{:,j} \rangle|}{\|\mathbf{A}_{:,i}\|\|\mathbf{A}_{:,j}\|} = \max_{(i,j) \in [m] \times [n], i \neq j} \frac{[\mathbf{A}^\top \mathbf{A}]_{i,j}}{\|\mathbf{A}_{:,i}\|\|\mathbf{A}_{:,j}\|}$. If the coherence is small, then the columns of $\mathbf{A}$ are almost mutually orthogo-

nal. A small coherence is desired in order to have good sparse recovery properties. We also have the 1-coherence $\mu_1(\mathbf{A}, s) = \max_{i\in[n]} \max_{J\subseteq[n]\setminus i, |J|\leq s} \sum_{j\in J} \frac{|\langle \mathbf{A}_{:,i}, \mathbf{A}_{:,j}\rangle|}{\|\mathbf{A}_{:,i}\|\|\mathbf{A}_{:,j}\|} \leq s\mu(\mathbf{A})$.

For a matrix $\mathbf{A} \in \mathbb{R}^{m\times n}$ with unit norm columns, $\mu(\mathbf{A}) \geq \sqrt{\frac{n-m}{m(n-1)}}$ and $\mu_1(\mathbf{A}, s) \geq s\sqrt{\frac{n-m}{m(n-1)}}$ whenever $s \leq \sqrt{n-1}$ (Rauhut, 2010). In high dimensional space (large $n$), $\mu(\mathbf{A}) \geq \sqrt{\frac{n-m}{m(n-1)}}$ becomes $\mu(\mathbf{A}) \gtrsim \frac{1}{\sqrt{m}}$ : this lower bound is achieve when $\mathbf{A}$ is an equiangular tight frame.

**Proposition G.3** (Relation between $\mu$ and $\delta$). *For a matrix $\mathbf{A} \in \mathbb{R}^{m\times n}$ with unit norm columns, $\mu(\mathbf{A}) = \delta_2(\mathbf{A})$, $\mu_1(\mathbf{A}, s) = \max_{J\in[n], |J|\leq s+1} \|\mathbf{A}_{:,J}^\top \mathbf{A}_{:,J} - \mathbb{I}\|_{1\to 1}$, and $\delta_s(\mathbf{A}) \leq \mu_1(\mathbf{A}, s-1) \leq (s-1)\mu(\mathbf{A})$* (Rauhut, 2010).

### G.1.1. THE PROBLEM

Given the sparse basis (or dictionary) $\Phi \in \mathbb{R}^{n\times n}$, the measurement matrix $\mathbf{M} \in \mathbb{R}^{N\times n}$, and the measures $\mathbf{y}^* = \mathcal{F}_{\mathbf{b}^*}(\mathbf{M}) + \boldsymbol{\xi} = \mathbf{M}\mathbf{b}^* + \boldsymbol{\xi} \in \mathbb{R}^N$; we aim to solve $(P_0)$ Minimize $\|\mathbf{a}\|_0$ s.t. $\|\mathcal{F}_{\mathbf{a}}(\mathbf{M}\Phi) - \mathbf{y}^*\|_2 \leq \epsilon$, and more precisely, its convex relaxation $(P_1)$ Minimize $\|\mathbf{a}\|_1$ s.t. $\|\mathcal{F}_{\mathbf{a}}(\mathbf{M}\Phi) - \mathbf{y}^*\|_2 \leq \epsilon$. This problem has been well studied in the signal processing literature under the name *Basis Pursuit*. It is well known that under certain conditions on the measurement matrix $\mathbf{M}$ (e.g., coherence with respect to $\Phi$) and the sparsity of $\mathbf{b}^*$ in $\Phi$, sufficiently sparse solutions of $(P_1)$ are also solutions of $(P_0)$ (Donoho & Elad, 2003; Candes et al., 2006). Many lower bounds on the number of measures $N$ guaranteeing $\|\mathbf{a} - \mathbf{a}^*\|_2 \leq \epsilon$ with high probability have also been derived. Such lower bounds generally have the form $N = \Omega\left(\delta^{-C_1}\left(s\log^{C_2}(n/s) + \log 1/\eta\right)\right)$ (Rauhut, 2010), where $\delta$ capture the Restricted Isometry Property (Definition G.1) of $\mathbf{X} = \mathbf{M}\Phi$ ($\delta_{2s}(\mathbf{X}) \leq \delta$) and is also related to the coherence (Definition G.2) of $\mathbf{M}$ with respect to $\Phi$ (Proposition G.3), $\eta$ is the percentage of error (i.e. $N$ guaranteed a recovery with probability at least $1 - \eta$), $C_1 > 0$ and $C_2 > 0$ are constants. Observe that in the noiseless setting, we want $\mathbf{a}$ such that $\mathbf{X}\mathbf{a} = \mathbf{X}\mathbf{a}^*$, that is $\mathbf{a} \in \mathbf{a}^* + \text{Null}(\mathbf{X})$. Donoho (2006b;a) show that the nullspace $\{\mathbf{a}, \mathbf{X}\mathbf{a} = 0\}$ has a very special structure for certain $\mathbf{X}$ (e.g. incoherent with any orthonormal basis): when $\mathbf{a}^*$ is sparse, the only element in the affine subspace $\mathbf{a}^* + \text{Null}(\mathbf{X})$ that can have a small $\ell_1$ norm is $\mathbf{a}^*$ itself.

We will assume for simplicity that $\Phi$ is an orthonormal matrix, $\Phi^\top\Phi = \mathbb{I}_n$. It is common in sparse coding theory to consider $\Phi \in \mathbb{R}^{n\times m}$ as a dictionary with $m$ columns referred to as atoms (square, $n = m$; undercomplete, $n > m$; or overcomplete, $n < m$); and saying $\mathbf{b}^*$ is sparse means it can be written as a linear combination of a few of such atoms. But here, we assume for simplicity that we have $\mathbf{b}^* = \Phi\mathbf{a}^*$ with $\mathbf{a}^* \in \mathbb{R}^m$ and $\Phi \in \mathbb{R}^{n\times m}$ a set of $m \leq n$ linearly independent vectors (its column). Let $\Phi^\perp \in \mathbb{R}^{n\times(n-m)}$ be the orthogonal complement of $\Phi$ in $\mathbb{R}^n$, $\Psi := \begin{bmatrix}\Phi & \Phi^\perp\end{bmatrix} \in \mathbb{R}^{n\times n}$, $\tilde{\Phi} := \Psi\left(\Psi^\top\Psi\right)^{-1/2} \in \mathbb{R}^{n\times n}$ the orthonormal version of $\Psi$, and $\tilde{\mathbf{a}}^* := \left(\Psi^\top\Psi\right)^{1/2}\begin{bmatrix}\mathbf{a}^*\\0\end{bmatrix}$. We still have $\mathbf{b}^* = \tilde{\Phi}\tilde{\mathbf{a}}^*$, with $\|\tilde{\mathbf{a}}^*\|_0 = \|\mathbf{a}^*\|_0$ since $\Psi^\top\Psi$ is diagonal. So, assuming $\Phi$ orthonormal is without loss of generality.

**Example G.1.** For the Fourier basis $\sqrt{n}\Phi_{ji} = \mathbf{e}^{-2\pi\mathbf{i}\frac{ji}{n}}$, we have $\mu_1(\Phi, s) = s\mu(\Phi) = s/\sqrt{n}$ (Rauhut, 2010). Each column in this basis vector corresponds to a specific frequency. For a signal $\mathbf{b}^*$, if only a few frequency components contribute significantly to $\mathbf{b}^*$, then $\mathbf{a}^* = \Phi^{-1}\mathbf{b}^*$, the Fourier transform of $\mathbf{b}^*$, will be sparse.

### G.1.2. THE CONTROLS PARAMETERS

The incoherence between the measurement vectors (line of $\mathbf{M}$) and the sparse basis (column of $\Phi$) is crucial for successfully recovering $\mathbf{b}^*$ (or equivalently $\mathbf{a}^*$, the sparse representation). If $\mathbf{M}$ is incoherent with $\Phi$, each measurement captures a distinct "view" of $\mathbf{b}^*$, reducing redundancy. This diversity of information allows for the successful reconstruction of $\mathbf{a}^*$ even with fewer measurements (e.g., below the Nyquist rate for signals). Achieving low coherence (high incoherence) can be done by designing $\mathbf{M}$ to be a random matrix (e.g., Sub-Gaussian like Gaussian or Bernoulli matrices). Such random matrices are, with high probability, incoherent with any fixed orthonormal basis, as Theorems G.4 and G.5 show.

**Theorem G.4.** *Le $m \leq n$ and $\Phi \in \mathbb{R}^{n\times m}$ with $\Phi^\top\Phi = \mathbb{I}_m$. For any $N \geq 1$, $C_1 > 0$ and $C_2 \geq 1$; the matrix $\mathbf{M} \in \mathbb{R}^{N\times n}$ with $n^{C_1}\mathbf{M} \overset{iid}{\sim} \mathcal{N}(0,1)$ satisfies $\mu(\mathbf{M}^\top, \Phi) \leq 2C_2\frac{\sqrt{\ln(nN)}}{n^{C_1}}$ with probability at least $1 - 1/(nN)^{2C_2^2-1}$.*

*Proof.* Let $\sigma = n^{-C_1}$, $C_1 > 0$. We have $\mathbf{M} \overset{iid}{\sim} \mathcal{N}(0,\sigma^2)$, so $[\mathbf{M}\Phi]_{ij} \overset{iid}{\sim} \mathcal{N}(0,\sigma^2)$ since $\Phi$ has normal columns. This implies $\mathbb{P}\left[\left|[\mathbf{M}\Phi]_{ij}\right| \geq t\right] \leq \exp\left(-\frac{t^2}{2\sigma^2}\right)$, which in turn implies $\mathbb{P}\left[\max_{i,j}\left|[\mathbf{M}\Phi]_{ij}\right| \geq t\right] \leq \sum_{i,j}\mathbb{P}\left[\left|[\mathbf{M}\Phi]_{ij}\right| \geq t\right] \leq$

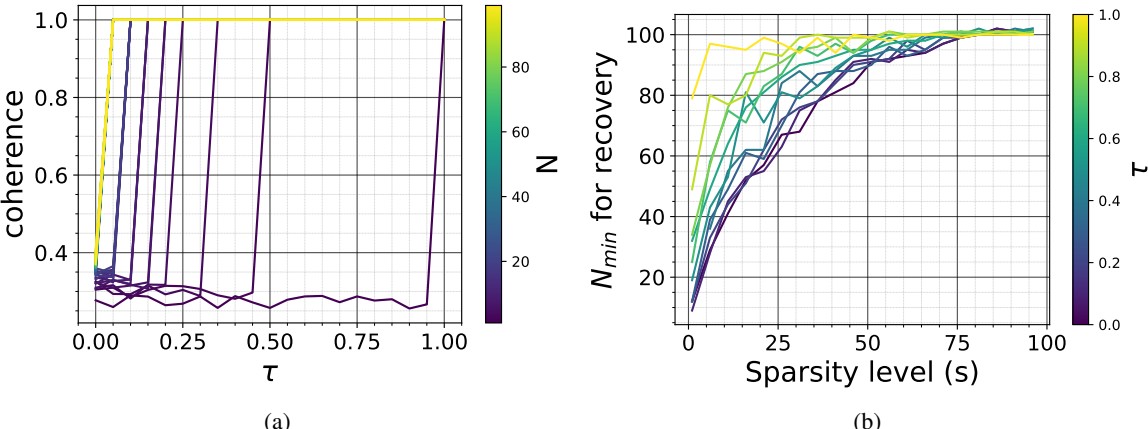

(a)                                    (b)

Figure 19: **(a)** Cohence $\mu(\mathbf{M}^\top, \Phi)$ as a function of $\tau \in (0,1)$ **(b)** Minimum number of samples for perfect recovery (relative recovery error $\leq 10^{-6}$) for $n = 10^2$ as a function of the sparsity level $s \in [n]$ and coherence parameter $\tau \in (0,1)$

$nN \exp\left(-\frac{t^2}{2\sigma^2}\right)$. Using $t = 2C_2 \frac{\sqrt{\ln(nN)}}{n^{C_1}}$ with $C_2 \geq 1$, we have $t^2 = 2\left(\frac{1}{n^{C_1}}\right)^2 \ln\left(\frac{nN}{\eta}\right)$ with $\eta = (nN)^{1-2C_2^2}$, so $nN \exp\left(-\frac{t^2}{2\sigma^2}\right) = \eta$. $\qquad\square$

We also have the following theorem from Rauhut (2010) about the RIP of such a matrix.

**Theorem G.5.** *Let $\mathbf{M} \in \mathbb{R}^{N \times n}$ be a Gaussian or Bernoulli random matrix. Let $\eta, \delta \in (0,1)$ and assume $N \geq C\delta^{-2}\left(s \ln(n/s) + \ln(1/\eta)\right)$ for a universal constant $C > 0$. Then, $\delta_s(\mathbf{M}) \leq \delta$ with probability at least $1 - \eta$.*

In the rest of this section,

- To control the incoherence, we generate $\mathbf{M}$ for a given $N$ by taking the first $N_1 = \min(\lfloor \tau N \rfloor, n)$ rows (with $0 \leq \tau \leq 1$, default to 0) from the first columns of $\Phi$ and the elements of the remaining $N_2 = N - N_1$ rows iid from $\mathcal{N}(0, 1/n)$ so that $\mathbf{X} = \mathbf{M}\Phi = \begin{bmatrix} \Phi_{:,:N_1}^\top \\ \mathbf{M}_{N_1:,:} \end{bmatrix} \Phi = \begin{bmatrix} \mathbb{I}_{N_1 \times n} \\ \mathbf{M}_{N_1:,:}\Phi \end{bmatrix}$ with $\mathbf{M}_{N_1:,:} \overset{iid}{\sim} \mathcal{N}(0, 1/n)$. The higher $\tau$ (and so $N_1$), the less is the incoherence between the measures (columns of $\mathbf{M}^\top$) and $\Phi$. $\tau = 0$ correspond to a full random gaussian $\mathbf{M}$, and correspond to the maximum incoherence, while $\tau = 1$ correspond to $\mathbf{M}_i \in \{\Phi_{:,j}\}_{j\in[n]}$ for all $i \in [N]$, and correspond minimum incoherence (coherence of 1).

- For a given $s$, we generate a random vector $\mathbf{a}^* \overset{iid}{\sim} \mathcal{N}(0, 1/n)$ such that $\|\mathbf{a}^*\|_0 \leq s$, and set $\mathbf{b}^* = \Phi \mathbf{a}^*$.

- We generated $\Phi$ by performing a QR decomposition on a random Gaussian matrix $n \times n$ and taking the Q part.

- For the noise $\xi \in \mathbb{R}^N$, we use $\xi \overset{iid}{\sim} \mathcal{N}(0, \sigma_\xi^2)$ with $\mathrm{SNR} = \frac{\mathbb{E}\|\mathbf{a}^*\|_2^2}{N\sigma_\xi^2} = 10^8$ (signal to noise ratio).

### G.1.3. IMPACT OF COHERENCE ON GROKKING

**Convex Optimization** We fix $n = 10^2$ and solve for different $(N, s, \tau)$ the problem $(P_1)$ using the `cvxpy` library. As $s$ and/or $\tau$ increases, $N_{\min}(s, \tau)$, the number of samples needs for perfect recovery increases (Figures 19 and 20). When $\tau$ converges to 1, $N_{\min}(s, \tau) \to n$ for all $s$. The error in those figures is the relative recovery error $\|\mathbf{a} - \mathbf{a}^*\|_2 / \|\mathbf{a}^*\|_2$. This error is usually of the order of $10^{-6}$, giving us a basis for comparison with other methods.

**Gradient Descent** Here, we also observe that the generalization time and the generalization delay increase with $\tau$ while the generalization error decreases with it (Figures 21 and 22). For $N < n$, when $\tau \to 1$, the generalization time $t_2 \to \infty$. This is because each measurement captures a single view (component) of $\mathbf{b}^* = \Phi \mathbf{a}^*$, and this makes it impossible to find the optimal $\mathbf{a}^*$ by solving the equation $\mathbf{M}\Phi \mathbf{a} = \mathbf{y}^*$ for $N < n$ (by any method whatsoever). On the other hand, as

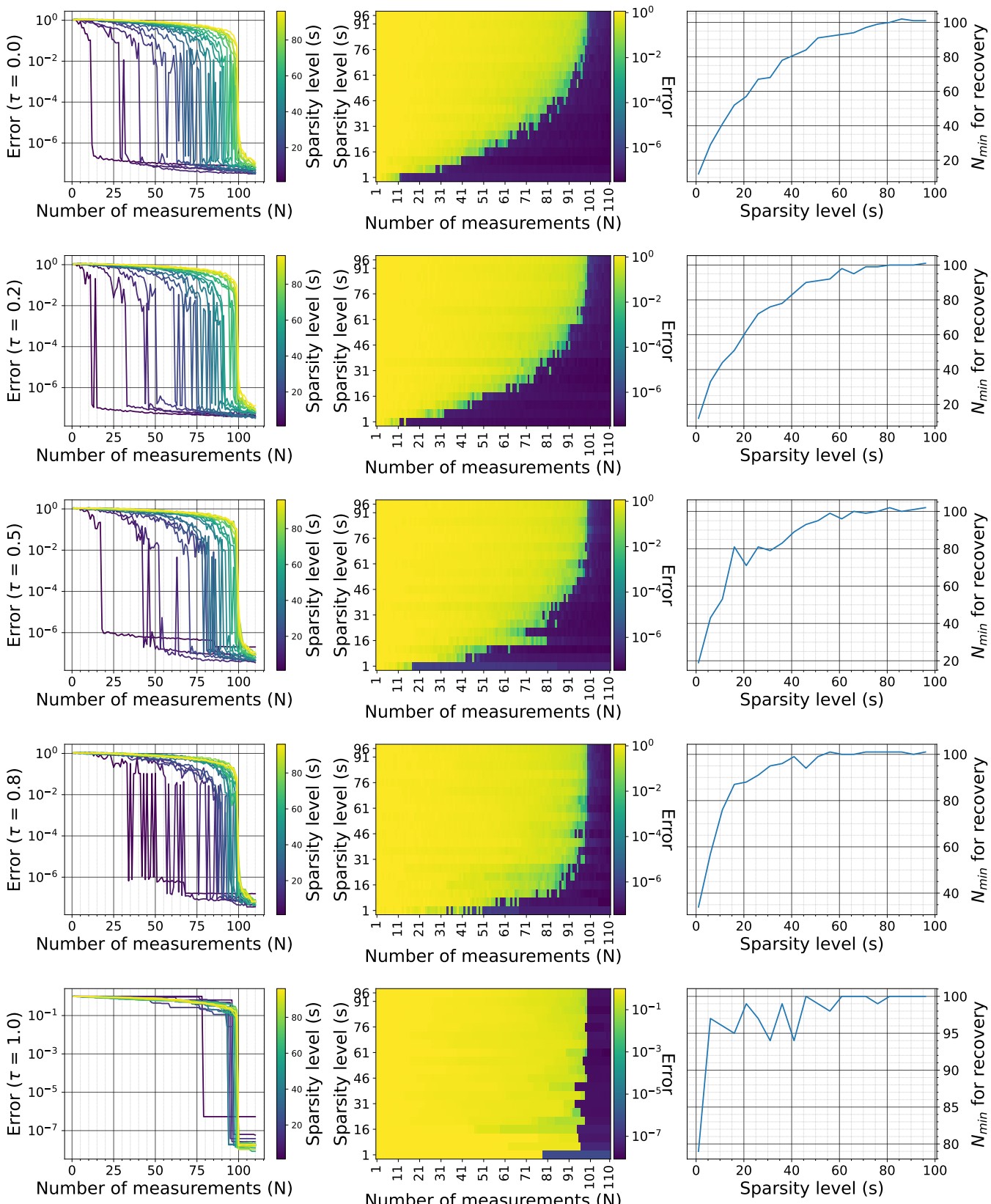

Figure 20: Relative error $\|\mathbf{a} - \mathbf{a}^*\|_2 / \|\mathbf{a}^*\|_2$ as a function of the number of measurements $N$, the sparsity level $s \in [n]$ and and coherence parameter $\tau \in (0, 1)$, for $n = 10^2$

$\tau \to 0$, $\mathbf{M}$ becomes completely random, and every measurement captures a distinct "view" of $\mathbf{b}^*$, giving the best possible generalization time for the data size considered. The error $\|\mathbf{a}^{(t_2)} - \mathbf{a}^*\|_2 / \|\mathbf{a}^*\|_2$ at generalization ($t_2$) as a function of $N$ and $\tau$ has the same shape as in the convex programming. We use $(n, s, \alpha, \beta, \zeta) = (10^2, 5, 10^{-1}, 10^{-5}, 10^{-6})$ in the experiments.

## G.2. Matrix Factorization

### G.2.1. THE PROBLEM

Given a low rank $r$ matrix $\mathbf{A}^* \in \mathbb{R}^{n_1 \times n_2}$, a measurement matrix $\mathbf{X} \in \mathbb{R}^{N \times n_1 n_2}$; we aim to solve the following problem for $\mathbf{A} \in \mathbb{R}^{n_1 \times n_2}$; $(P_3)$ Minimize $\operatorname{rank}(\mathbf{A})$ subject to $\|\mathcal{F}_{\operatorname{vec}(\mathbf{A})}(\mathbf{X}) - \mathbf{y}^*\|_2 \leq \epsilon$; and more precisely, its convex relaxation $(P_4)$ Minimize $\|\mathbf{A}\|_* = \sum_i \sigma_i(\mathbf{A})$ subject to $\|\mathcal{F}_{\operatorname{vec}(\mathbf{A})}(\mathbf{X}) - \mathbf{y}^*\|_2 \leq \epsilon$; where $\mathbf{y}^* = \mathcal{F}_{\operatorname{vec}(\mathbf{A}^*)}(\mathbf{X}) + \boldsymbol{\xi}$ are the measures and $\epsilon$ an upper bound on the size of the error term $\boldsymbol{\xi} \in \mathbb{R}^N$, $\|\boldsymbol{\xi}\|_2 \leq \epsilon$.

**Matrix Sensing**  Matrix sensing seeks to recover a low rank matrix $\mathbf{A}^* \in \mathbb{R}^{n_1 \times n_2}$ from $N$ measurement matrices $\{\mathbf{X}_i \in \mathbb{R}^{n_1 \times n_2}\}_{i \in [N]}$ and measures $\mathbf{y}^* = \left(\operatorname{tr}(\mathbf{X}_i^\top \mathbf{A}^*)\right)_{i \in [N]}$. We have $\mathbf{y}_i^* = \operatorname{tr}(\mathbf{X}_i^\top \mathbf{A}^*) = \operatorname{vec}(\mathbf{X}_i)^\top \operatorname{vec}(\mathbf{A}^*) = \mathcal{F}_{\operatorname{vec}(\mathbf{A}^*)}(\operatorname{vec}(\mathbf{X}_i))$. This gives us a compressed sensing problem, with the signal vector $\operatorname{vec}(\mathbf{A}^*) \in \mathbb{R}^{n_1 n_2}$ and the measurement matrix $\mathbf{X} = [\operatorname{vec}(\mathbf{X}_i)]_{i \in [N]} \in \mathbb{R}^{N \times n_1 n_2}$. In fact, under full SVD $\mathbf{A}^* = \mathbf{U}^* \Sigma^* \mathbf{V}^{*\top}$, we have $\mathbf{a}^* = \operatorname{vec}(\mathbf{A}^*) = \Phi \operatorname{vec}(\Sigma^*)$; where $\operatorname{vec}(\Sigma^*) \in \mathbb{R}^{n_1 n_2}$, is sparse since $\|\operatorname{vec}(\Sigma^*)\|_0 = \operatorname{rank}(\mathbf{A}^*) \leq \min(n_1, n_2) \ll n_1 n_2$; and $\Phi = \mathbf{V}^* \otimes \mathbf{U}^* \in \mathbb{R}^{n_1 n_2 \times n_1 n_2}$ has orthonormal column since $\Phi^\top \Phi = \left(\mathbf{V}^{*\top} \mathbf{V}^*\right) \otimes \left(\mathbf{U}^{*\top} \mathbf{U}^*\right) = \mathbb{I}_{n_1 n_2}$.

**Matrix Completion**  For a matrix completion problem with matrix $\mathbf{A}^* \in \mathbb{R}^{n_1 \times n_2}$, we have $N$ measurement vectors $\left(\mathbf{X}_i^{(1)}, \mathbf{X}_i^{(2)}\right) \in \mathbb{R}^{n_1} \times \mathbb{R}^{n_2}$ and measures $\mathbf{y}_i^* = \mathbf{X}_i^{(1)\top} \mathbf{A}^* \mathbf{X}_i^{(2)} = \left(\mathbf{X}_i^{(2)} \otimes \mathbf{X}_i^{(1)}\right)^\top \operatorname{vec}(\mathbf{A}^*) = \mathcal{F}_{\operatorname{vec}(\mathbf{A}^*)}\left(\mathbf{X}_i^{(2)} \otimes \mathbf{X}_i^{(1)}\right)$, i.e. $\mathbf{y}^* = \left(\mathbf{X}^{(2)} \bullet \mathbf{X}^{(1)}\right) \operatorname{vec}(\mathbf{A}^*) = \mathcal{F}_{\operatorname{vec}(\mathbf{A}^*)}\left(\mathbf{X}^{(2)} \bullet \mathbf{X}^{(1)}\right)$. This gives us a compressed sensing problem, with the signal vector $\operatorname{vec}(\mathbf{A}^*) \in \mathbb{R}^{n_1 n_2}$ and the measurement matrix $\mathbf{X} = \mathbf{X}^{(2)} \bullet \mathbf{X}^{(1)} \in \mathbb{R}^{N \times n_1 n_2}$. Standard matrix completion is usually defined as recovering missing elements of a matrix from its incomplete observation. This is equivalent to requiring $\mathbf{X}_i^{(k)}$ to be selection vectors for all $k \in [K]$ (with $K = 2$), i.e. $\mathbf{X}_i^{(k)}$ is the $s(i, k)^{\text{th}}$ vector of the canonical basis of $\mathbb{R}^{n_k}$ for a certain $s(i, k) \in [n_k]$. This make each $\mathbf{X}_i = \mathbf{X}_i^{(2)} \otimes \mathbf{X}_i^{(1)}$ a selection vector in $\mathbb{R}^n$, and $\mathbf{X} = \mathbf{X}^{(2)} \bullet \mathbf{X}^{(1)}$ a selection matrix in $\mathbb{R}^{N \times n}$, so that $\mathbf{y}_i^* = \mathbf{A}^*_{s(i,1), s(i,2)} \forall i \in [N]$. So, in this formulation, each $\mathbf{X}_i^{(k)}$ is a sample from the columns of $\mathbb{I}_{n_k}$. Note that under a change of basis $\tilde{\mathbf{X}}_i^{(k)} = \mathbf{P}^{(k)} \mathbf{X}_i^{(k)}$, we have $\tilde{\mathbf{y}}_i^* = \left(\otimes_{k=1}^K \mathbf{P}^{(k)}\right) \mathbf{y}_i^*$, that is $\tilde{\mathbf{y}}^* = \mathbf{y}^* \left(\otimes_{k=1}^K \mathbf{P}^{(k)}\right)^\top$. A less standard formulation of the matrix completion task requires each $\mathbf{X}_i^{(k)}$ to be a sample from an orthonormal basis, i.e., $\mathbf{X}_i^{(k)}$ is a sample from the columns of $\mathbf{V}^{(k)} \in \mathbb{R}^{n_k \times n_k}$ with $\mathbf{V}^{(k)\top} \mathbf{V}^{(k)} = \mathbb{I}_{n_k}$. We let $\mathbf{X}_i^{(k)}$ be the $s(i, k)^{\text{th}}$ column of $\mathbf{V}^{(k)}$ for a certain $s(i, k) \in [n_k]$. Then $\mathbf{y}_i^* = \tilde{\mathbf{A}}^*_{s(i,1), \cdots, s(i,K)}$ with $\tilde{\mathbf{A}}^* = \mathbf{A}^* \times_1 \mathbf{V}^{(1)} \times_2 \mathbf{V}^{(2)}$. So, any result state of $\mathbf{A}^*$ in the standard formulation where the measurement vectors are selection vectors is valid for the matrix $\tilde{\mathbf{A}}^*$.

Assume the target matrix $\mathbf{A}^*$ has rank $r$. Then it has $r(n_1 + n_2 - r)$ degree of freedom[10], and we need to observe at least $r(n_1 + n_2 - r)$ entries for perfect recovery. This bound can be improved by considering the structure of $\mathbf{A}^*$. Let $\mathbf{A}^* = \mathbf{U}^* \Sigma^* \mathbf{V}^{*\top}$ be the **full** SVD of $\mathbf{A}^*$. As observed above, we are dealing with a compressed sensing problem with the signal vector $\mathbf{a}^* = \operatorname{vec}(\mathbf{A}^*) = \Phi \operatorname{vec}(\Sigma^*)$; where $\operatorname{vec}(\Sigma^*) \in \mathbb{R}^{n_1 n_2}$ is sparse and $\Phi = \mathbf{V}^* \otimes \mathbf{U}^* \in \mathbb{R}^{n_1 n_2 \times n_1 n_2}$ has orthonormal column.

### G.2.2. THE CONTROL PARAMETERS

In this sub-section, we assume standard matrix completion. But the theories outlined here also apply to the general framework. The theory gives the minimal number of observations that guarantee $\mathbf{A}^*$ to be a unique solution to problem $(P_4)$ and allow perfect recovery of $\mathbf{A}^*$ with fewer samples (Candès & Tao, 2010; Candes & Recht, 2012; Chen et al., 2014). Generally, the lower bound on $N$ has a form $N \geq C \max(n_1, n_2)^{C_2} \left(r^{C_3} \log^{C_1}(\max(n_1, n_2)) + \log \frac{1}{\eta}\right)$ where $\eta$ is the percentage of error (i.e $N$ guaranteed perfect recovery with probability at least $1 - \eta$), $C_1 > 0, C_2 > 0, C_3 > 0$ are

---

[10]The first $r$ columns of $\mathbf{U}^*$ form an orthonormal basis for a $r$-dimensional subspace of $\mathbb{R}^{n_1}$ (the columns space of $\mathbf{A}^*$). Specifying this requires $r(n_1 - r)$ parameters. Similarly, the first $r$ columns of $\mathbf{V}^*$ form an orthonormal basis for a $r$-dimensional subspace of $\mathbb{R}^{n_2}$ (the rows space of $\mathbf{A}^*$), and specifying this requires $r(n_2 - r)$ parameters. The $r$ non-zero singular values are independent parameters. Thus, specifying them requires $r$ parameters.

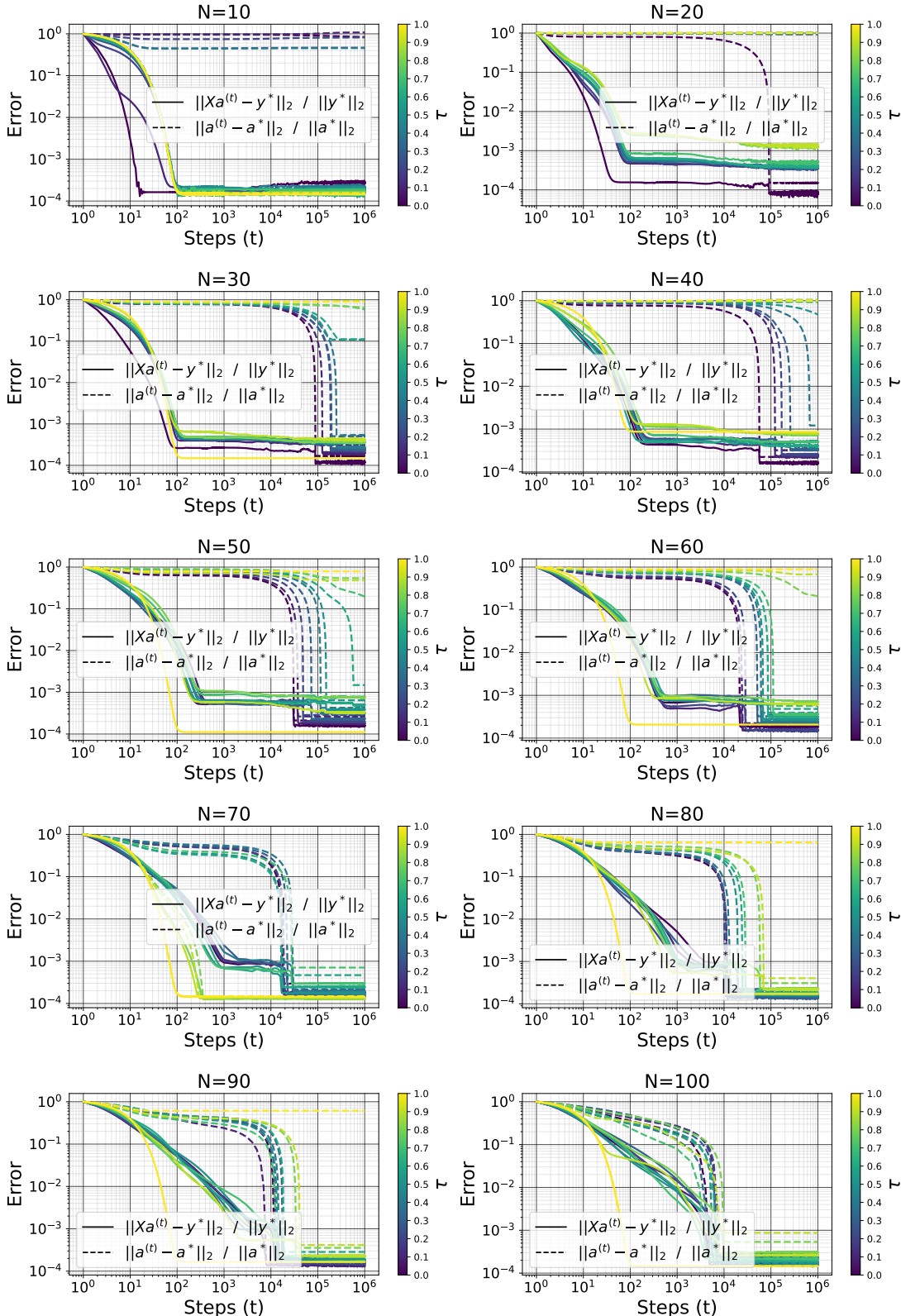

Figure 21: Training and recovery error as a function of the number of samples $N$ and the coherence parameter $\tau \in [0, 1]$.

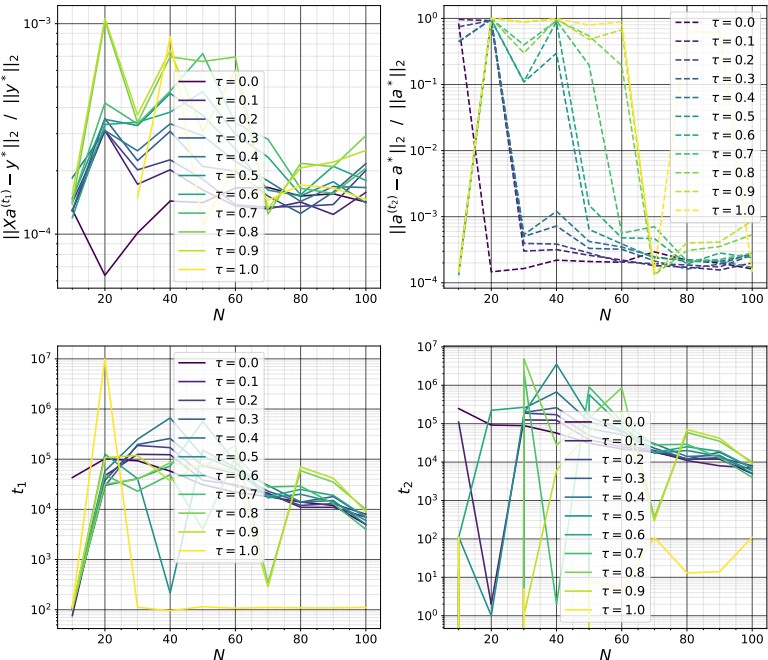

Figure 22: Training and error $\|\mathbf{X}\mathbf{a}^{(t_1)} - \mathbf{y}^*\|_2/\|\mathbf{y}^*\|_2$ and recovery error $\|\mathbf{a}^{(t_2)} - \mathbf{a}^*\|_2/\|\mathbf{a}^*\|_2$ (along with $t_1$ and $t_2$, the memorization and the generalization step) as a function of the number of sample $N$ and the coherence parameter $\tau \in [0, 1]$.

constant, and $C > 0$ a universal constant. For example, in Candes & Recht (2012), $(C_1, C_2, C_3) = (1, 1.2, 1)$ for small rank $r \leq \max(n_1, n_2)^{0.2}$, and $C_2 = 1.25$ for any rank. The term $\max(n_1, n_2) \log(\max(n_1, n_2))$ is due to the coupon collector effect since to recover an unknown matrix, one needs at least one observation per row and one observation per column (Candes & Recht, 2012).

**Definition G.6** (Random orthogonal model (Candes & Recht, 2012)). For a given $r$, we generate two orthonormal matrices $\mathbf{U}^* \in \mathbb{R}^{n_1 \times r}$ and $\mathbf{V}^* \in \mathbb{R}^{n_2 \times r}$ with columns selected uniformly at random among all families of $r$ orthonormal vectors; and a diagonal matrix $\Sigma^*$ with only the first $r$ diagonal element non-zero (with no assumptions about the singular values[11]), then set $\mathbf{A}^* = \mathbf{U}^* \Sigma^* \mathbf{V}^{*\top}$.

We have the following result about the standard formulation for such matrices in the absence of noise.

**Theorem G.7** (Theorem 1.1, Candes & Recht (2012)). *Let $\mathbf{A}^* \in \mathbb{R}^{n_1 \times n_2}$ be a matrix of rank $r$ sampled from the random orthogonal model, and put $n = \max(n_1, n_2)$. Suppose we observe $N$ entries of $\mathbf{A}^*$ with locations sampled uniformly at random. Then there are numerical constants $C$ and $c$ such that if $N \geq Cn^{5/4} r \log(n)$, the minimizer to the problem $(P_4)$ is unique and equal to $\mathbf{A}^*$ with probability at least $1 - c/n^3$; that is to say, the semidefinite program $(P_4)$ recovers all the entries of $\mathbf{A}^*$ with no error. In addition, if $r \leq n^{1/5}$, then the recovery is exact with probability at least $1 - c/n^3$ provided that $N \geq Cn^{6/5} r \log(n)$.*

Assume for example $\mathbf{A}^* = \mathbf{e}_k^{(n_1)} \mathbf{e}_\ell^{(n_2)}$ for $(k, \ell) \in [n_1] \times [n_2]$. Even if this matrix ranks at 1, it has only zeros everywhere except 1 at position $(k, \ell)$, so we have very little chance of reconstructing it in a high dimension by observing a portion of its inputs. The only way to guarantee observation of the input at position $(k, \ell)$ is to choose measurements coherently with its singular basis $\mathbf{e}_k^{(n_2)} \otimes \mathbf{e}_\ell^{(n_1)}$. This idea is formulated more generally below.

**Definition G.8.** Let $U$ be a subspace of $\mathbb{R}^n$ of dimension $r$ and $\mathbf{P}_U$ be the orthogonal projection onto $U$. Then, the coherence of $U$ vis-a-vis a basis $\{\mathbf{u}_i^{(n)}\}_{i \in [n]}$ is defined by $\mu(U) = \frac{n}{r} \max_i \|\mathbf{P}_U \mathbf{u}_i^{(n)}\|^2$. We have $1 \leq \mu(U) \leq n/r$ (Candes & Recht, 2012).

For a matrix $\mathbf{A} = \mathbf{U}\Sigma\mathbf{V}^\top \in \mathbb{R}^{n_1 \times n_2}$ under the **compact** SVD, the projection on the left singular value is $\mathbf{x} \to \mathbf{U}\mathbf{U}^\top \mathbf{x}$,

---

[11]Unless otherwise specified, we default the nonzero singular values to $1/\sqrt{r}$.

and $\|\mathbf{U}\mathbf{U}^\top\mathbf{x}\|_2^2 = \|\mathbf{U}^\top\mathbf{x}\|_2^2$ for all $\mathbf{x}$ (similarly for the right singular value). We have the following definition of coherence, which considers each matrix entry.

**Definition G.9** (Local coherence & Leverage score). Let $\mathbf{A} = \mathbf{U}\mathbf{\Sigma}\mathbf{V}^\top \in \mathbb{R}^{n_1 \times n_2}$ be the **compact** SVD of a matrix $\mathbf{A}$ of rank $r$. The local coherences of $\mathbf{A}$ are defined by

$$
\begin{aligned}
\mu_i(\mathbf{A}) &= \frac{n_1}{r}\|\mathbf{U}^\top \mathbf{e}_i^{(n_1)}\|^2 = \frac{n_1}{r}\|\mathbf{U}_{i,:}\|^2 \quad \forall i \in [n_1] \\
\nu_j(\mathbf{A}) &= \frac{n_2}{r}\|\mathbf{V}^\top \mathbf{e}_j^{(n_2)}\|^2 = \frac{n_2}{r}\|\mathbf{V}_{j,:}\|^2 \quad \forall j \in [n_2]
\end{aligned}
\tag{122}
$$

with $\mu_i$ for row $i$ and $\nu_j$ for row $j$. The quantities $\|\mathbf{U}^\top \mathbf{e}_i^{(n_1)}\|^2$ and $\|\mathbf{V}^\top \mathbf{e}_i^{(n_2)}\|^2$ are the leverage score of $\mathbf{A}$ (Chen et al., 2014), which indicate how "aligned" each row or column of the original data matrix is with the principal components (the columns of $\mathbf{U}$ or $\mathbf{V}$). For each row $i$, $\mu_i(\mathbf{A})$ measures how much this row vector projects onto the subspace spanned by the first $r$ left singular vectors in $\mathbf{U}$. Rows with high leverage scores contribute more to the low-rank structure of $\mathbf{A}$ and are more "influential" in representing $\mathbf{A}$. Similarly, $\nu_j(\mathbf{A})$ measures the coherence of each column $j$ in $\mathbf{A}$ with respect to the low-rank subspace formed by the right singular vectors in $\mathbf{V}$. High values indicate columns well-aligned with the principal directions of $\mathbf{A}$ and play a significant role in capturing its structure. Matrices with uniformly low coherence scores have rows and columns that are evenly influential. In contrast, matrices with high coherence scores for certain rows or columns have a few specific rows or columns that dominate the low-rank structure.

In the general formulation, this definition can be extended to the set from which the measures are chosen. But in general, it leads back to the standard formulation under the change of basis.

**Definition G.10** (Generalize local coherence & Leverage score). We generalize the notion of coherence to any arbitrary set of vectors $\mathbf{U}^{(n_1)} = \{\mathbf{u}_i^{(n_1)}\}_{i \in [N_1]} \in \mathbb{R}^{n_1 \times N_1}$ and $\mathbf{V}^{(n_2)} = \{\mathbf{v}_j^{(n_2)}\}_{j \in [N_2]} \in \mathbb{R}^{n_2 \times N_2}$, and defined the generalized local coherences as

$$
\mu_i(\mathbf{A}) = \frac{n_1}{r}\|\mathbf{U}^\top \mathbf{u}_i^{(n_1)}\|^2 \quad \forall i \in [N_1], \qquad \nu_j(\mathbf{A}) = \frac{n_2}{r}\|\mathbf{V}^\top \mathbf{v}_j^{(n_2)}\|^2 \quad \forall j \in [N_2]
\tag{123}
$$

Suppose the sets $\mathbf{U}^{(n_1)}$ and $\mathbf{V}^{(n_2)}$ are be orthonormal basis (i.e. $(N_1, N_2) = (n_1, n_2)$, $\mathbf{u}_i^{(n_2)\top}\mathbf{u}_k^{(n_2)} = \delta_{ik}$ and $\mathbf{v}_j^{(n_1)\top}\mathbf{v}_l^{(n_1)} = \delta_{jl}$). We can write $\mathbf{u}_i^{(n_1)} = \mathbf{P}^{(1)}\mathbf{e}_i^{(n_1)}$ and $\mathbf{v}_j^{(n_2)} = \mathbf{P}^{(2)}\mathbf{e}_j^{(n_2)}$ with $\mathbf{P}^{(k)} \in \mathbb{R}^{n_k \times n_k}$ the base change matrix from the canonical basis to $\mathbf{U}^{(n_1)}$ and $\mathbf{V}^{(n_2)}$ respectively. So

$$
\begin{aligned}
\mu_i(\mathbf{A}) &= \frac{n_1}{r}\|\mathbf{U}^\top \mathbf{P}^{(1)}\mathbf{e}_i^{(n_1)}\|^2 = \frac{n_1}{r}\|\tilde{\mathbf{U}}^\top \mathbf{e}_i^{(n_1)}\|^2 = \mu_i(\tilde{\mathbf{A}}) \quad \forall i \in [N_1] \\
\nu_j(\mathbf{A}) &= \frac{n_2}{r}\|\mathbf{V}^\top \mathbf{P}^{(2)}\mathbf{e}_j^{(n_2)}\|^2 = \frac{n_2}{r}\|\tilde{\mathbf{V}}^\top \mathbf{e}_j^{(n_2)}\|^2 = \nu_i(\tilde{\mathbf{A}}) \quad \forall j \in [N_2]
\end{aligned}
\tag{124}
$$

with $\tilde{\mathbf{A}} = \mathbf{A} \times_1 \mathbf{P}^{(1)} \times_2 \mathbf{P}^{(2)} = \mathbf{P}^{(1)\top}\mathbf{A}\mathbf{P}^{(2)} = \mathbf{P}^{(1)\top}\mathbf{U}\mathbf{\Sigma}\left(\mathbf{P}^{(2)\top}\mathbf{V}\right)^\top = \tilde{\mathbf{U}}\mathbf{\Sigma}\tilde{\mathbf{V}}^\top$. That said, any result stated in the standard formulation for $\mathbf{A}$ is valid for $\tilde{\mathbf{A}}$ under the general orthonormal formulation.

Candès & Tao (2010) and Candes & Recht (2012) used mainly an upper bound $\mu_0$ on $\mu_i$ and $\nu_i$; $\mu_0 \geq \max\left(\max_{i \in [n_1]} \mu_i(\mathbf{A}^*), \max_{i \in [n_2]} \nu_i(\mathbf{A}^*)\right)$, and define a constant $\mu_1$ such that the $\max_{i,j}[\mathbf{U}^*\mathbf{V}^{*\top}]_{ij} = \max_{i,j} \sum_k \mathbf{U}_{i,k}^* \mathbf{V}_{j,k}^* \leq \mu_1 \sqrt{\frac{r}{n_1 n_2}}$. Since $\left|\sum_k \mathbf{U}_{i,k}^* \mathbf{V}_{j,k}^*\right| \leq \sqrt{\sum_k \mathbf{U}_{i,k}^{*2}}\sqrt{\sum_k \mathbf{V}_{j,k}^{*2}} = \|\mathbf{U}_{i,:}^*\|_2\|\mathbf{V}_{j,:}^*\|_2 = \frac{r}{\sqrt{n_1 n_2}}\sqrt{\mu_i(\mathbf{A}^*)\nu_j(\mathbf{A}^*)} \leq \frac{r}{\sqrt{n_1 n_2}}\mu_0$ for all $i, j$; we can just take $\mu_1 \geq \mu_0\sqrt{r}$. From this, Candes & Recht (2012) show that if the coherence $\mu_0$ is low, few samples are required to recover $\mathbf{A}^*$.

**Theorem G.11** (Theorem 1.3, Candes & Recht (2012))**.** *Let $\mathbf{A}^* \in \mathbb{R}^{n_1 \times n_2}$ be a matrix of rank $r$ sampled from the random orthogonal model, and put $n = \max(n_1, n_2)$. Suppose we observe $N$ entries of $\mathbf{A}^*$ with locations sampled uniformly at random. Then there are numerical constants $C$ and $c$ such that if $N \geq C \max\left(\mu_1^2, \mu_0^{\frac{1}{2}}\mu_1, \mu_0 n^{\frac{1}{4}}\right)nr\beta \log(n)$ for some $\beta > 2$, the minimizer to the problem $(P_4)$ is unique and equal to $\mathbf{A}^*$ with probability at least $1 - c/n^3$. In addition, if $r \leq n^{1/5}/\mu_0$, then the recovery is exact with probability at least $1 - c/n^3$ provided that $N \geq C\mu_0 n^{6/5}r\beta \log(n)$.*

Chen et al. (2014) show that sampling the element at position $(i, j)$ with probability $p_{ij} = \Omega(\mu_i + \nu_j)$ allows perfect recovery of $\mathbf{A}^*$ with fewer samples, and called such sampling strategies *local coherence sampling*.

**Theorem G.12** (Theorem 3.2 and Corollary 3.3, Chen et al. (2014)). *Let $\mathbf{A}^* \in \mathbb{R}^{n_1 \times n_2}$ be a matrix of rank $r$ with local coherence $\{\mu_i, \nu_j\}_{i \in [n_1], j \in [n_2]}$. There are universal constant $c_0, c_1, c_2 > 0$ such that if each element $(i,j)$ is independently observed with probability $p_{ij} \geq \max\left\{\min\left\{c_0 \frac{(\mu_i + \nu_j) r \log^2(n_1 + n_2)}{\min(n_1, n_2)}, 1\right\}, \frac{1}{\min(n_1, n_2)^{10}}\right\}$, then $\mathbf{A}^*$ is the unique optimal solution of the nuclear minimization problem $(P_4)$ with probability at least $1 - c_1/(n_1 + n_2)^{c_2}$, for a number of sample $N \in \mathcal{O}\left(\max(n_1, n_2) r \log^2(n_1 + n_2)\right)$.*

Given $N$ and $\tau \in [0, 1]$, to control the coherence,

- For matrix factorization, we select the first $N_1 = \tau N$ examples with the highest values of $\mu_i(\mathbf{A}^*) + \nu_j(\mathbf{A}^*)$, and select the remaining $(1 - \tau)N$ examples uniformly among the rest. The positions selected are one-hot encoded in dimensions $n_1$ (for row positions) and $n_2$ (for column positions) to have $\mathbf{X}^{(1)}$ and $\mathbf{X}^{(2)}$, respectively.

- For matrix sensing, we generate $\mathbf{X}^{(1)}$ (resp. $\mathbf{X}^{(2)}$) by taking the first $N_1 = \min(\lfloor \tau N \rfloor, n_1)$ (resp. $N_1 = \min(\lfloor \tau N \rfloor, n_2)$) rows from the first columns of $\mathbf{U}^*$ (resp. $\mathbf{V}^*$) and the elements of the remaining $N_2 = N - N_1$ rows iid from the Gaussian distribution $\mathcal{N}(0, \sigma^2)$ with $\sigma = 1/n_1$ (resp. $\sigma = 1/n_2$).

The higher $\tau$ (and so $N_1$), the less incoherence between the measures (rows of $\mathbf{X} = \mathbf{X}^{(2)} \bullet \mathbf{X}^{(1)}$) and $\Phi = \mathbf{V}^* \otimes \mathbf{U}^*$.

### G.2.3. IMPACT OF COHERENCE ON GROKKING

**Linear programming**  We fix $n_1 = n_2 = 10^2$ and $\boldsymbol{\xi} = 0$ (no noise) and solve for different $(N, r, \tau)$ the matrix factorization problem presented above using standard linear programming (we use the `cvxpy` library). As $r$ and/or $\tau$ increases, the number of samples needed for perfect recovery **decreases**. The relative recovery error $\|\mathbf{A} - \mathbf{A}^*\|_2 / \|\mathbf{A}^*\|_2$ obtained is usually of the order of $10^{-6}$ and gives us a basis for comparison with other methods. We do not include figures to save space.

**Gradient Descent**  Unlike compressed sensing where large values of $\tau$ are detrimental to generalization, here, as $\tau \to 1$, performance improves, and the number of examples required to generalize decreases exponentially, as does the time it takes the models to do so. See Figure 23 for $(n_1, n_2, r, \alpha, \beta, \zeta) = (10, 10, 2, 10^{-1}, 10^{-5}, 10^{-6})$. Note that here, for matrix completion, for a fixed $\tau$, we select the first $\tau N$ examples with the highest values of $\mu_i(\mathbf{A}^*) + \nu_j(\mathbf{A}^*)$, and select the remaining $(1 - \tau)N$ examples at random, uniformly.

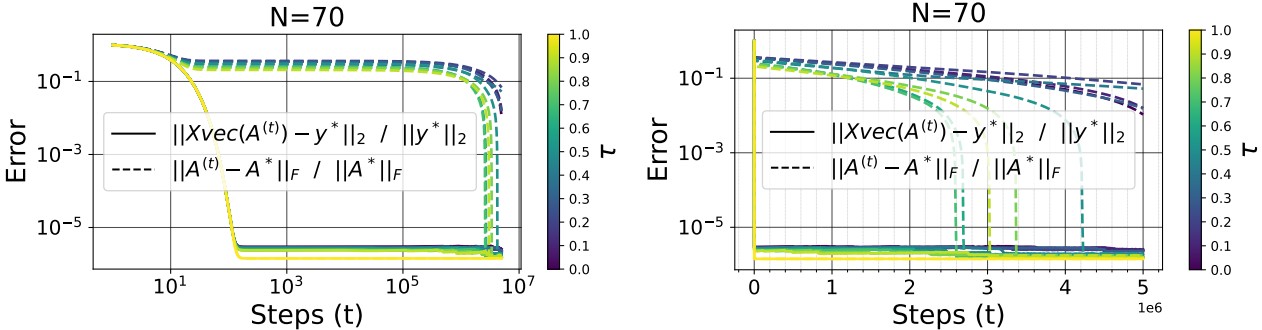

Figure 23: Training error $\|\mathbf{X} \text{vec} \mathbf{A}^{(t)} - \mathbf{y}^*\|_2 / \|\mathbf{y}^*\|_2$ and recovery error $\|\mathbf{A}^{(t)} - \mathbf{A}^*\|_2 / \|\mathbf{A}^*\|_F$ as a function of the number of sample $N$ and the coherence parameter $\tau \in [0, 1]$

## H. Additional Experiments

### H.1. Sparse Recovery

**Optimization landscape**  We look at the landscape of the solution in the context of sparse recovery. Let $I := \{i \in [n] \mid \mathbf{a}_i^* \neq 0\}$ be the support of the ground truth signal $\mathbf{a}^*$; $u(t) = \|\mathbf{a}_I^{(t)}\|_2$ and $v(t) = \|\mathbf{a}_{[n] \setminus I}^{(t)}\|_2$ be the norms of $\mathbf{a}^{(t)}$ restraint on its indexes in $I$ and outside $I$, respectively. Figure 24 shows how $\mathbf{a}^{(t)}$ first converges to the least square

solution (memorization), and from the least square solution to $\mathbf{a}^*$ ($N$ large enough) or a suboptimal solution ($N$ too small). After memorization, when $N$ is large enough, $v(t)$ converges to zero while $u(t)$ converges to the norm of $\mathbf{a}^*$. This is because the components of $\mathbf{a}^{(t)}$ that are not in $I$ are shrunk at each training step until they all reach 0 (Figure 25). This convergence is impossible if the $\ell_1$ regularization strength is 0 (even if $\ell_2$ is used). For the experiments of this section we use $N \in \{20, 30, 40, 50, 60, 70\}$, for $(n, s) = (100, 5)$ and $(\alpha, \beta) = (10^{-1}, 10^{-5})$.

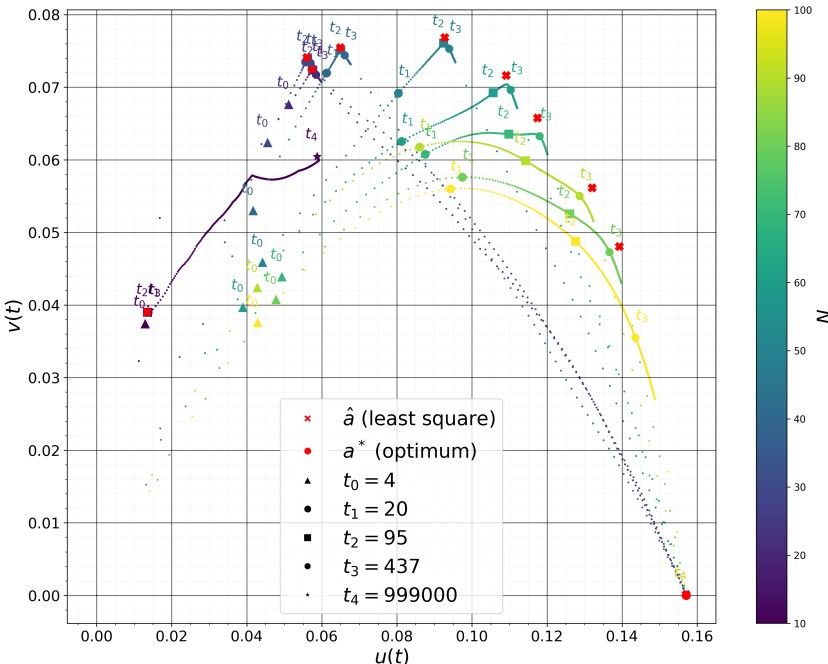

Figure 24: From initialization to least square solution $\hat{\mathbf{a}}$ (memorization), and from least square solution to $\mathbf{a}^*$ ($N$ large enough) or a suboptimal solution ($N$ too small). The steps $t_1$ and $t_2$ are different from those introduced above to measure memorization and generalization (respectively). They are just a means of tracing the evolution of training here.

**Scaling the Learning Rate $\alpha$ and the Regularization Strength $\beta$**   We solve the sparse recovery problem using the subgradient descent method with $(n, s, N, \zeta) = (10^2, 5, 30, 10^{-6})$ for different values of $\alpha$ and $\beta$. As expected, larger $\alpha$ and/or $\beta$ lead to fast convergence and do so at a suboptimal value of the test error (Figure 26). However, small values require longer training time to plateau and generally do so at a lower value of recovery error (grokking). These are the experiments used to produce Figure 4.

**Scaling the Data Size $N$ and the Sparsity Parameter $s$**   We solve the sparse recovery problem using the subgradient descent method with $(n, \zeta, \alpha, \beta) = (10^2, 10^{-6}, 10^{-1}, 10^{-5})$, for $s \in \{1, 5, 10, 15\}$ and $N \in \{10, 20, \ldots 100\}$. See Figures 27 and 28 for the results. Smaller $s$ requires a smaller $N$ for generalization.

### H.2. Matrix Factorization

We optimize the noiseless matrix completion problem using the subgradient descent method with $(n_1, n_2, r, N, \zeta) = (10, 10, 2, 70, 10^{-6})$ for different values of $\alpha$ and $\beta$. As expected, larger $\alpha$ and/or $\beta$ lead to fast convergence and do so at a suboptimal value of the test error (Figure 29). These are the experiments used to produce Figure 6.

### H.3. General Setting

In this section we optimize functions of the form $f(\theta) = g(\theta) + \beta h(\theta)$, where $g$ is the square loss or cross-entropy loss function of the considered model on the training data, $\theta$ the set of model parameters, and $h$ a regularizer applied to $\theta$. It can be the standard $\ell_p$ norm or quasi-norm of $\theta$, the sum of the nuclear norms of each matrix in $\theta$ (i.e. $\ell_*$), etc. By normal initialization for a parameter $\mathbf{A} \in \mathbb{R}^{n_1 \times n_2}$, we mean $\mathbf{A}^{(1)} \overset{iid}{\sim} \mathcal{N}(0, 1/n_1)$. For the experiments of this section only, we used

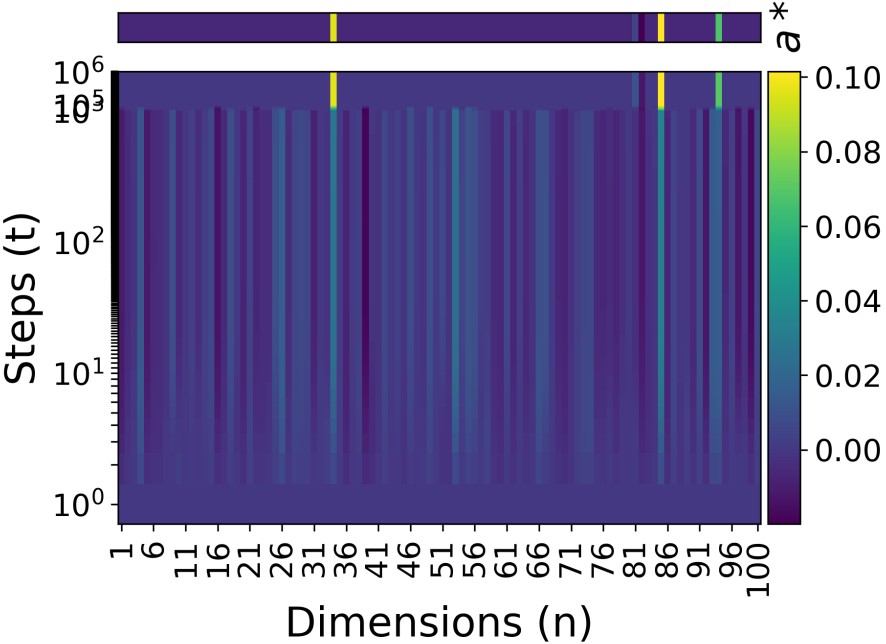

Figure 25: Convergence of $\mathbf{a}_i^{(t)}$ to $\mathbf{a}_i^*$ for each $i \in [n]$. Here $(n, s, N) = (100, 5, 30)$ and $(\alpha, \beta) = (10^{-1}, 10^{-5})$.

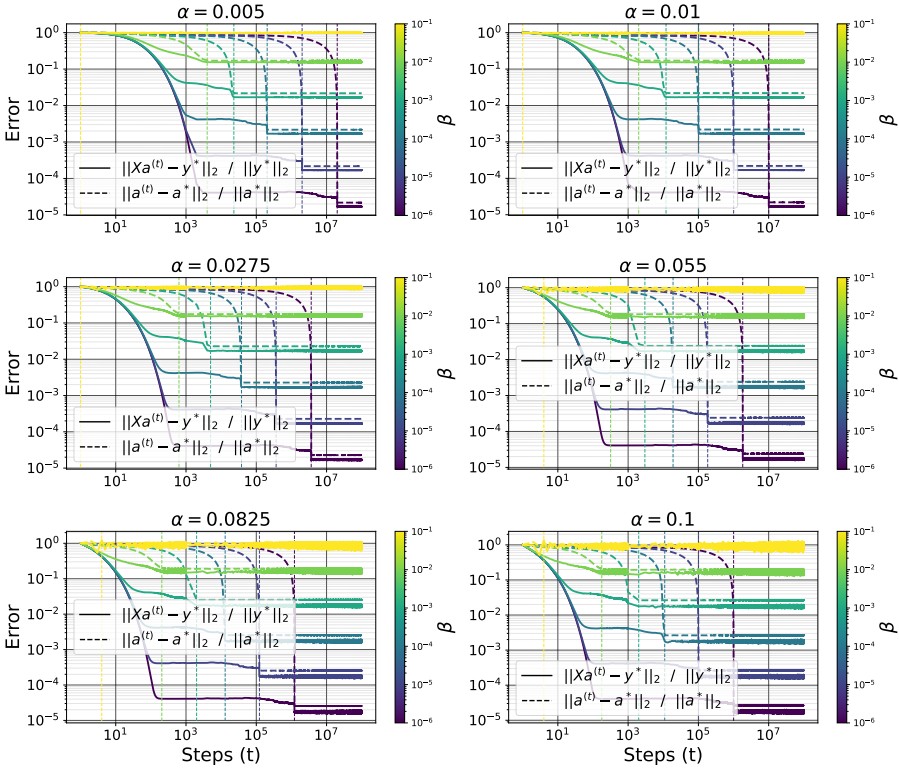

Figure 26: Training and recovery error as a function of the learning rate $\alpha$ and the $\ell_1$-regularization coefficient $\beta$.

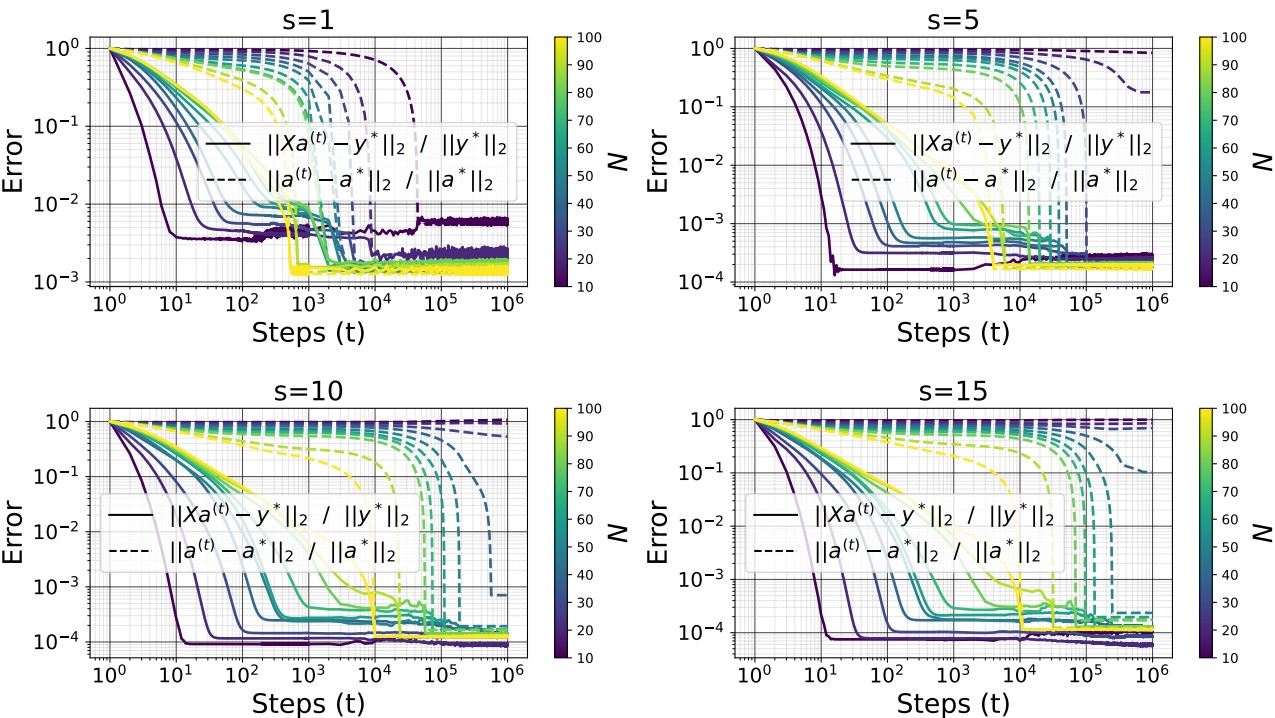

Figure 27: Training and recovery error as a function of the sparsity level $s$ and the number of measurements $N$.

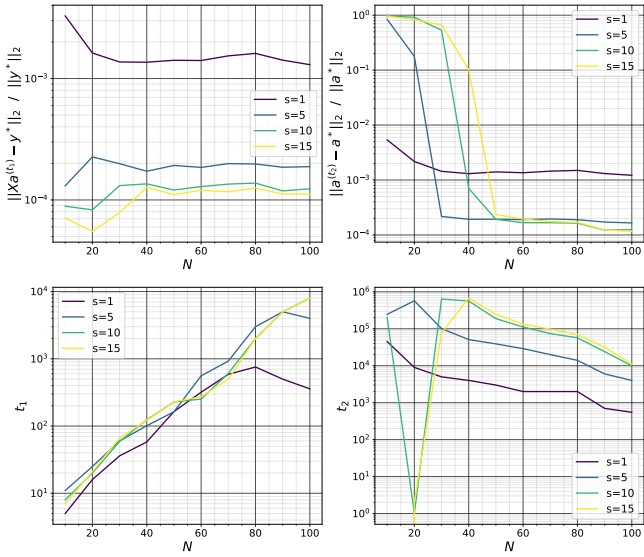

Figure 28: Training error $\|\mathbf{X}\mathbf{a}^{(t_1)} - \mathbf{y}^*\|_2/\|\mathbf{y}^*\|_2$ at memorization, recovery error $\|\mathbf{a}^{(t_2)} - \mathbf{a}^*\|_2/\|\mathbf{a}^*\|_2$ at generalization, memorization step $t_1$ (smaller $t$ such that $\|\mathbf{X}\mathbf{a}^{(t)} - \mathbf{y}^*\|_2/\|\mathbf{y}^*\|_2 \leq 10^{-4}$), and generalization step (smaller $t$ such that $\|\mathbf{a}^{(t)} - \mathbf{a}^*\|_2/\|\mathbf{a}^*\|_2 \leq 10^{-4}$ or the maximum training step).

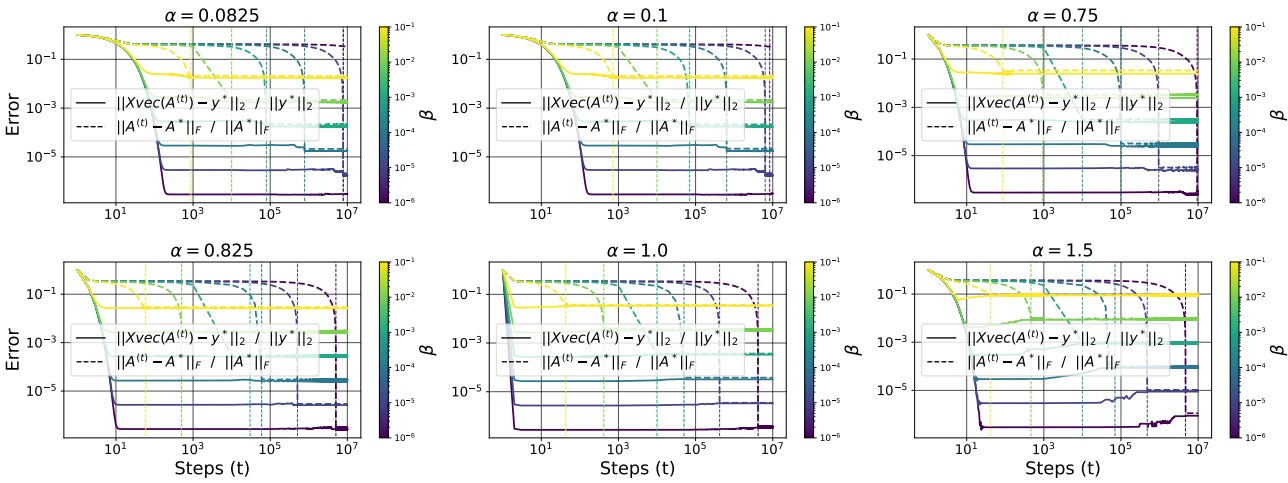

Figure 29: Training and recovery error as a function of the learning rate $\alpha$ and the $\ell_*$-regularization coefficient $\beta$.

Adam as the optimizer, with its default parameters (as specified in PyTorch), except for the learning rate.

### H.3.1. ALGORITHMIC DATASET

We consider addition modulo $p = 97$ as described in Section 2.1 with $r_{\text{train}} = 40\%$. For MLP, $\ell_1$ and $\ell_*$ have the same effect on grokking as $\ell_2$, i.e., large $\alpha\beta$ values lead to faster grokking. See Figure 2 for $\ell_*$, and 30 for $\ell_1$ and $\ell_2$. Note that for the MLP, the logits are given by $\mathbf{y}(\mathbf{x}) = \mathbf{b}^{(2)} + \mathbf{W}^{(2)}\phi\left(\mathbf{b}^{(1)} + \mathbf{W}^{(1)}\left(\mathbf{E}_{\langle x_1 \rangle} \circ \mathbf{E}_{\langle x_2 \rangle}\right)\right)$, with $\phi(z) = \max(z, 0)$, $\mathbf{E} \in \mathbb{R}^{p \times d_1}$, $\mathbf{W}^{(1)} \in \mathbb{R}^{d_2 \times d_1}$, $\mathbf{b}^{(1)} \in \mathbb{R}^{d_2}$, $\mathbf{W}^{(2)} \in \mathbb{R}^{p \times d_2}$, and $\mathbf{b}^{(2)} \in \mathbb{R}^p$, where $d_1$ the embedding dimension. We use $d_1 = d_2 = 2^6$.

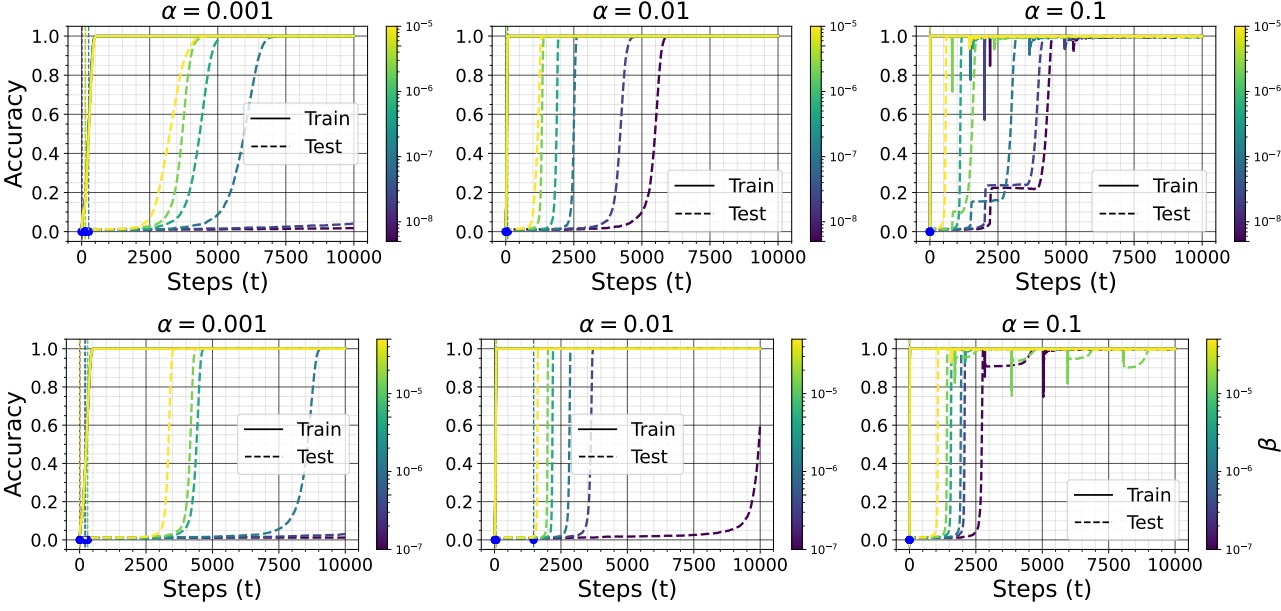

Figure 30: Training and test accuracy of a MLP trained on modular addition with $\ell_1$ (top) and $\ell_2$ (bottom) regularization for different values of the learning rate $\alpha$ and the $\ell_1$ coefficient $\beta$.

### H.3.2. NON LINEAR TEACHER-STUDENT

We consider a teacher $\mathbf{y}^*(\mathbf{x}) = \mathbf{B}^*\phi(\mathbf{A}^*\mathbf{x})$ from $\mathbb{R}^d$ to $\mathbb{R}^c$ with $r$ hidden neurons ($\mathbf{A}^* \in \mathbb{R}^{r\times d}$ and $\mathbf{B}^* \in \mathbb{R}^{c\times r}$); where $\phi(z) = \max(z,0)$ and $\mathbf{x}, \mathbf{A}^*, r\mathbf{B}^* \stackrel{iid}{\sim} \mathcal{N}(0,1)$. We i.i.d sample $N$ inputs output pair $\mathcal{D}_{\text{train}} = \{(\mathbf{x}_i, \mathbf{y}^*(\mathbf{x}_i))\}_{i=1}^N$ and optimize the parameters $\theta = (\mathbf{A},\mathbf{B})$ of a student $\mathbf{y}_\theta(\mathbf{x}) = \mathbf{B}\phi(\mathbf{A}\mathbf{x})$ on them, starting from normal initialization, with the loss function $g(\theta) = \frac{1}{2N}\sum_{i=1}^N \|\mathbf{y}_\theta(\mathbf{x}_i) - \mathbf{y}^*(\mathbf{x}_i)\|_2^2$ and different regularizer $h(\theta)$, $\ell_p$ for $p \in \{1,2,*\}$. For all these regularizers, the smaller $\alpha\beta$ is, the longer the delay between memorization and generalization. See Figures 31, 32 and 33 for an experiment with $(d, r, c, N) = (100, 500, 2, 10^2)$.

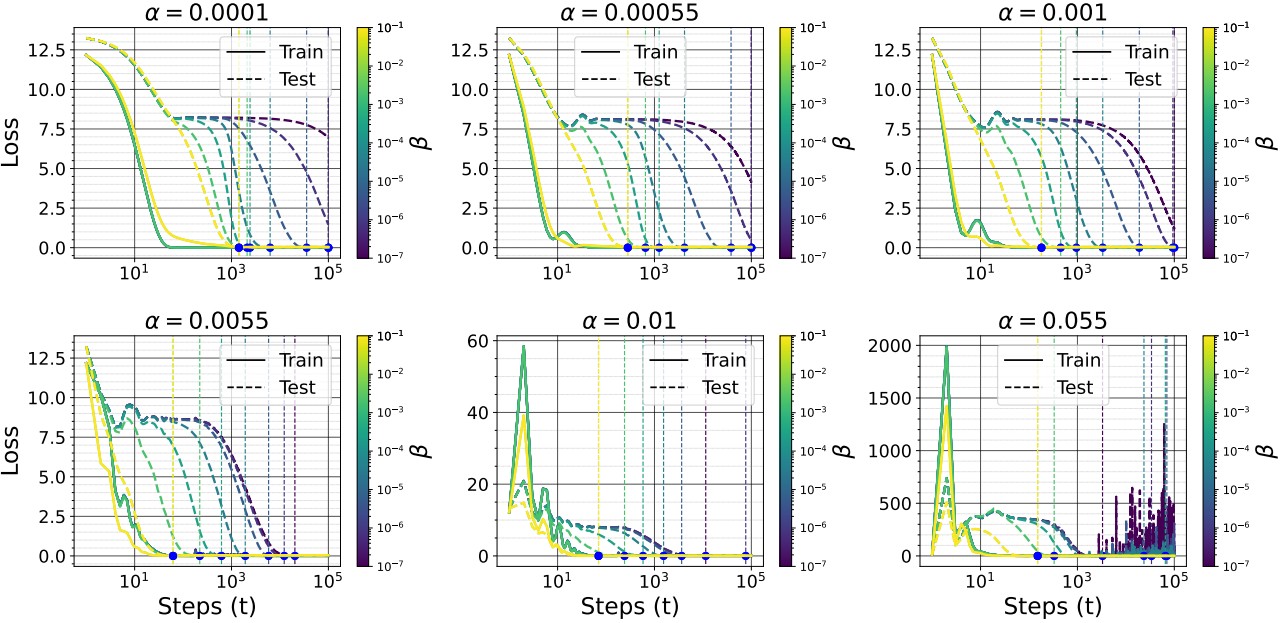

Figure 31: Training and test error two layers `ReLU` teacher-student with $\ell_1$ regularization, for different values of the learning rate $\alpha$ and the $\ell_1$ coefficient $\beta$.

### H.3.3. DOMAIN SPECIFIC REGULARIZATION

Physics-Informed Neural Networks (Raissi et al., 2019) leverage prior knowledge from differential equations by incorporating their residuals into the loss function, ensuring that solutions remain consistent with physical laws. Sobolev training (Czarnecki et al., 2017) generalizes this idea by incorporating not only input-output pairs but also derivatives of the target function. More precisely, given input-output pairs $\{(\mathbf{x}_i, \mathbf{y}^*(\mathbf{x}_i)\}_{i\in[N]}$ along with known derivatives $\left\{\left.\frac{\partial^k \mathbf{y}^*(\mathbf{x})}{\partial \mathbf{x}^k}\right|_{\mathbf{x}=\mathbf{x}_i}\right\}_{i\in[N]}$ for $k \in [K]$, the goal is to train a neural network $\mathbf{y}_\theta(\mathbf{x})$ that approximates both the output and its derivatives. The loss function extends the standard mean squared error (MSE) to include Sobolev penalties:

$$f(\theta) = \underbrace{\frac{1}{N}\sum_{i=1}^N \|\mathbf{y}_\theta(\mathbf{x}_i) - \mathbf{y}^*(\mathbf{x}_i)\|^2}_{\text{data loss}} + \underbrace{\frac{\beta}{N}\sum_{k=1}^K\sum_{i=1}^N \left\|\frac{\partial^k \mathbf{y}_\theta}{\partial \mathbf{x}^k}(\mathbf{x}_i) - \frac{\partial^k \mathbf{y}^*}{\partial \mathbf{x}^k}(\mathbf{x}_i)\right\|_{\text{F}}^2}_{\text{Sobolev penalty}} \qquad (125)$$

The hyperparameter $\beta$ controls the contribution of the derivative alignment term. This penalty ensures that the model not only fits the data but also respects known smoothness constraints or differential structure, which is crucial in physics-based applications (Lu et al., 2021). We consider the two layers feed forward teacher of Section H.3.2, and optimize the parameters $\theta = (\mathbf{A}, \mathbf{B})$ of a student using the sobolev objectify for $K = 1$, $\frac{\partial \mathbf{y}^*(\mathbf{x})}{\partial \mathbf{x}} = \mathbf{B}^*\text{diag}\left(\phi'(\mathbf{A}^*\mathbf{x})\right)\mathbf{A}^*$. The smaller is $\alpha\beta$, the longer is the delay between memorization and generalization. See Figure 34 for an experiment with $(d, r, c, N) = (100, 500, 2, 10^2)$.

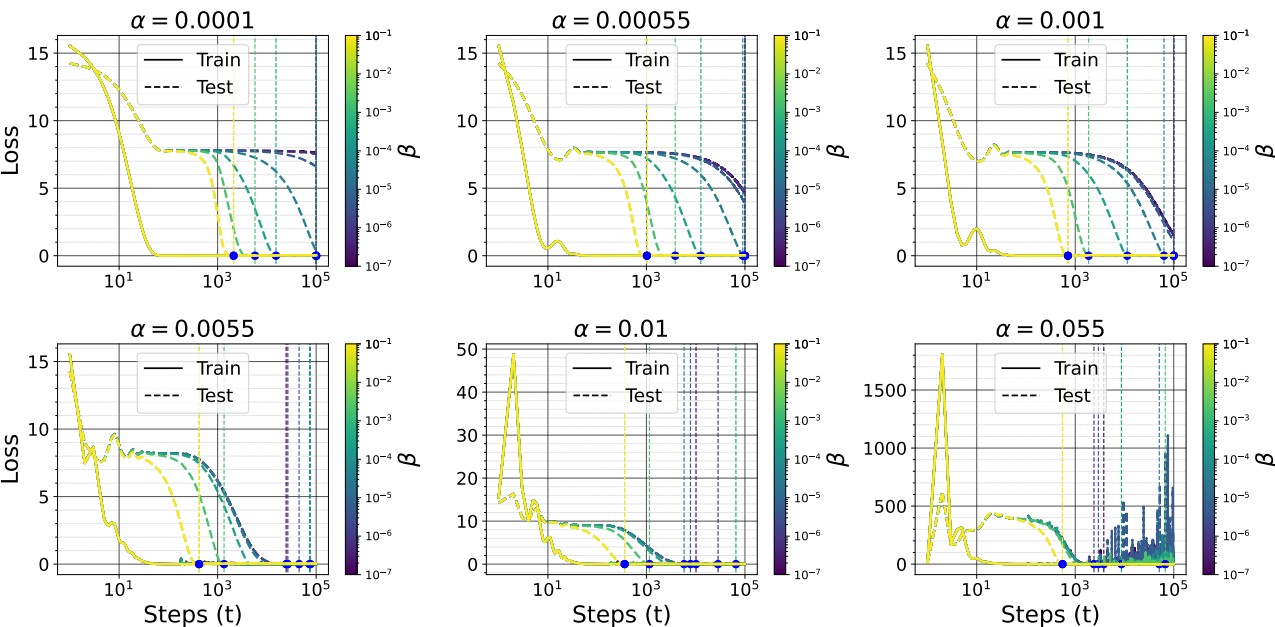

Figure 32: Training and test error two layers `ReLU` teacher-student with $\ell_2$ regularization, for different values of the learning rate $\alpha$ and the $\ell_2$ coefficient $\beta$.

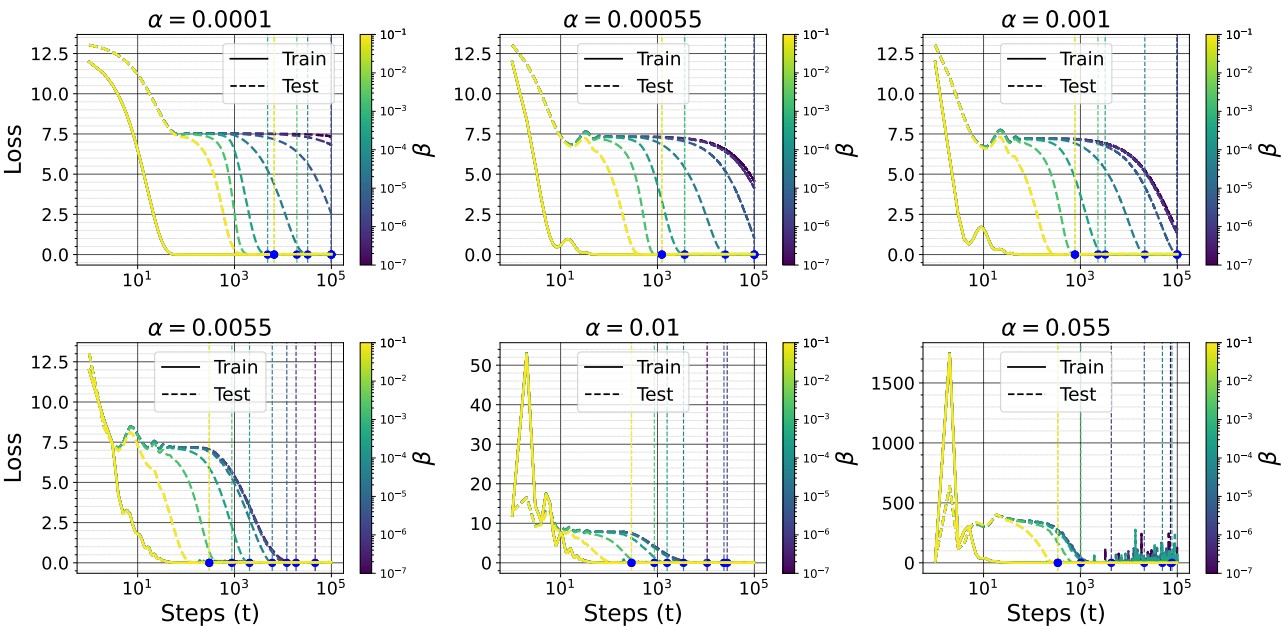

Figure 33: Training and test error two layers `ReLU` teacher-student with $\ell_*$ regularization, for different values of the learning rate $\alpha$ and the $\ell_*$ coefficient $\beta$.

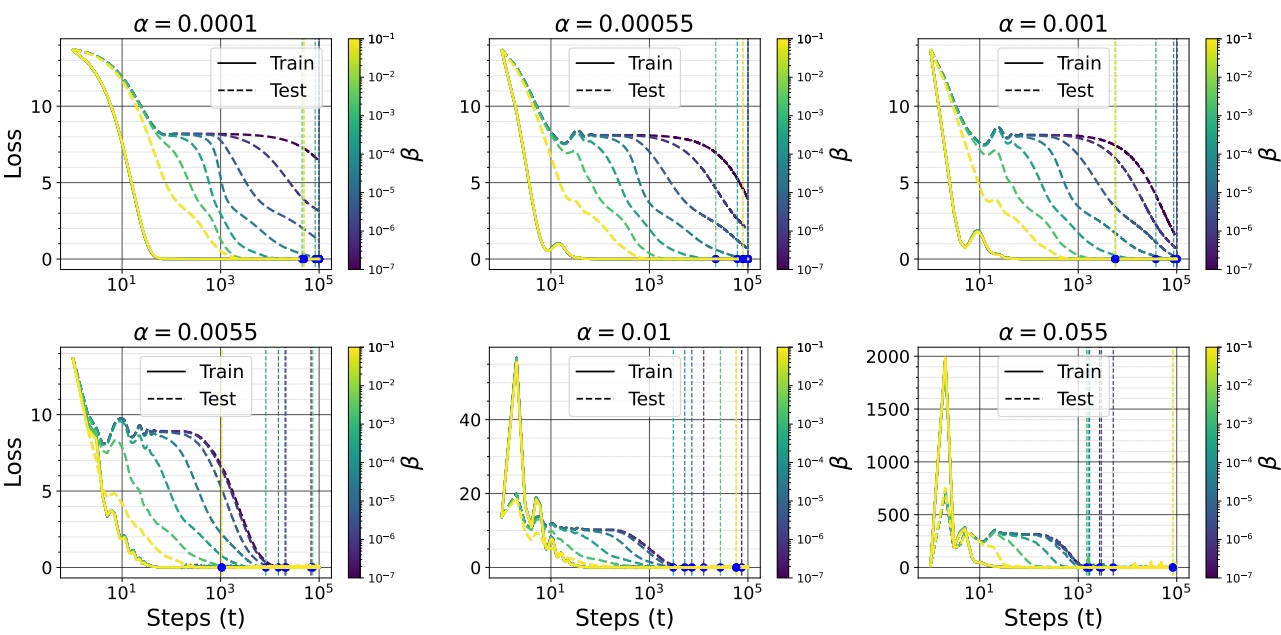

Figure 34: Training and test error two layers `ReLU` teacher-student with Sobolev training, for different values of the learning rate $\alpha$ and the $\ell_1$ coefficient $\beta$.

### H.3.4. IMAGE CLASSIFICATION

We optimize the parameters $\theta = (\mathbf{A}, \mathbf{B})$ of a model $\mathbf{y}_\theta(\mathbf{x}) = \mathbf{B}\phi(\mathbf{A}\mathbf{x})$ on $N = 1000$ samples of the MNIST dataset. Figure 35 show the results for $\ell_1$ : the result for $\ell_2$ and $\ell_*$ are similar.

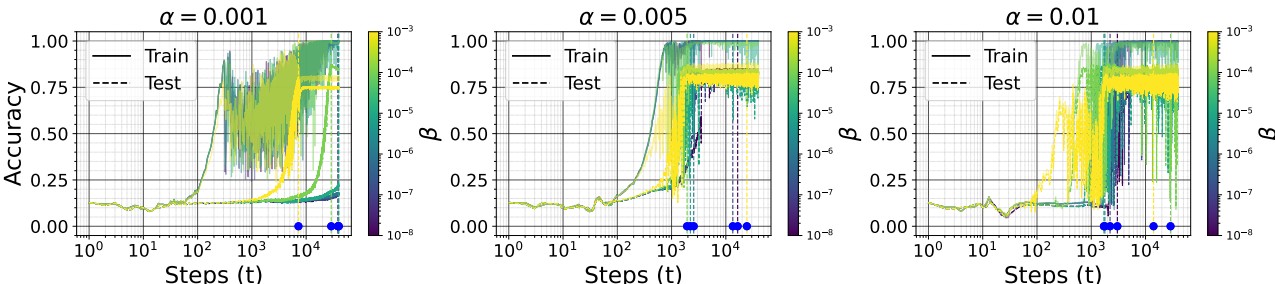

Figure 35: Training and test accuracy of a MLP trained on MNIST with $\ell_1$ regularization for different values of the learning rate $\alpha$ and the $\ell_1$ coefficient $\beta$

