# OpenReview forum: "Grokking Beyond the Euclidean Norm of Model Parameters"
_ICML.cc/2025/Conference — ICML 2025 poster_

### Official Review · Reviewer_zx76 · 2025-03-08

**Overall Recommendation:** 3

**Summary:**

The paper studies grokking i.e., delayed generalization analytically in sparse recovery and matrix factorization settings. The main finding i in the paper provides settings that question the  accepted wisdom in grokking - a small low $L_{2}$ regularization (weight decay) is necessary for grokking and generalization. In other words, $L_{2}$ norm cannot be used as an indicator of delayed generalization. The paper provides extensive analysis and some empirical results to support the main claim

## update after rebuttal

My current rating is 3. I like the paper but am unwilling to move the score up after reading the rebuttal and other reviews. I have started a discussion with the AC and other reviewers to obtain clarifications but yet to hear from them. Score is unchanged but I have already marked it as a 3.

**Claims And Evidence:**

The main claim is that if a problem as a certain property $P$, then appropriate regularization $L_{p}$ is required to reach the optimal solution.
- Analytical results appear to support the claim
- Experiments on synthetic data supports this claim as well in Section 2 and Section 3
- Non-linear teacher student model also appears to support the claim

The support provided with PINNs and classification are not as clear as the above primarily because the nature of the optimal solution is not obvious (at least to this reader)

**Essential References Not Discussed:**

None

**Experimental Designs Or Analyses:**

The paper is theoretical work, so the experimental implementation is acceptable

**Methods And Evaluation Criteria:**

The paper is theoretical work, so the datasets used with nonlinear models are acceptable

**Other Comments Or Suggestions:**

Grokking is commonly thought of as delayed generalization. Typically papers include a "training" vs "validation" plot to show delayed generalization.

Could the authors simplify Figure 1 and perhaps Figure 3 to relate these terms to the terms plotted in the figure?
Could the authors simplify Figure 2 and Figure 4 as there are too many terms plotted in a single plot?

**Other Strengths And Weaknesses:**

# Strengths

- The paper uses a setup that's simple enough for analysis and is able to show that accepted wisdom in grokking, i.e., L_2 norm of parameters may not suffice. Note that this reminds me of a string of papers that questions "deep learning may not be explained by norms". Happy to see these papers cited in the work as well

# Weaknesses

- The paper is very dense and some of the sections, if rewritten, would be accessible to a broader audience

**Questions For Authors:**

Please see my comments on including proof sketches for main theorems as well as improving figures in the paper made above.

**Relation To Broader Scientific Literature:**

The paper does an excellent job relating their work to broader scientific literature

**Theoretical Claims:**

The paper provides a theorem in Section 2 and one in Section 3 for sparse recovery and matrix factorization respectively. The theorems actually summarize several theorems proposed and proved in the appendix. The issue is its easy for the reader to get lost between the claims in Theorem 2.1 (and 3.1) as the text indicates they summarize several claims and proofs in the appendix. It would be better if the Theorem in the main text has a proof sketch in the main paper so that the interested reader can fill in the details from proof(s) in the appendix.

---

> ### Author Rebuttal · Authors · 2025-04-01
>
> We are grateful to the reviewer for their time and thoughtful feedback. We sincerely appreciate the effort put into evaluating our work and offering constructive suggestions. Below, we respond to each of the points raised.
>
> **Theoretical Claims**
>
> We've rewritten the proofs of the two main theorems (2.1 and 3.1) more simply, this time with a proof sketch, which we can summarize as follows. The idea is to first show that, given a matrix $\tilde{\mathbf{X}} \in \mathbb{R}^{N \times n}$ and a vector $\mathbf{b}^* \in \mathbb{R}^{n}$ of sparsity $s$ ($\|\mathbf{b}^*\|_0 \le s$), we can write, for any vector $\mathbf{b} \in \mathbb{R}^{n}$, $
> \|\mathbf{b} - \mathbf{b}^*\|_2 \le C_1 \| \tilde{\mathbf{X}} \left(\mathbf{b} - \mathbf{b}^*\right)\|_2 + C_2 | \|\mathbf{b}\|_1 - \|\mathbf{b}^*\|_1|$ where $C_1$ and $C_2$ are constants that depend only on $s$ and $\tilde{\mathbf{X}}$ (notably on the restricted isometry constant of $\tilde{\mathbf{X}}$). With this decomposition in mind, we then show that :
> * there is a memorization phase, after which the term $\| \tilde{\mathbf{X}} \left(\mathbf{b}^{(t)} - \mathbf{b}^*\right)\|_2$ vanish (or becomes proportional to the noise if there is any), since we have $ \tilde{\mathbf{X}} \mathbf{b}^{(t)} \approx \mathbf{y}^* = \tilde{\mathbf{X}} \mathbf{b}^* + \xi$.
> * and a second phase, during which $\|\mathbf{b}^{(t)}\|_1$ converges to $\|\mathbf{b}^*\|_1$: we show that this phase takes a time inversely proportional to $\alpha$ (the learning step) and $\beta_1$ (the regularization strength).
>
> We arrive at the final result by combining these two steps. The same type of decomposition applies to matrix factorization, with a slight difficulty because not only singular values but also singular vectors are taken into account.
>
> **Weaknesses: Rewriting**
>
> The rewriting of the proofs allows us to considerably reduce the number of pages in the appendix and to move many results from the appendix to the main text.
>
> **Validation plot**
>
> In our case, for an iterate $\mathbf{b}$, the training error is $\|\tilde{\mathbf{X}} \mathbf{b} - \mathbf{y}^*\|_2^2$, and the test (or validation) error is $\|\mathbf{b} - \mathbf{b}^*\|_2^2$. Normally, the generalization error should be $\mathcal{E}(\mathbf{b}) = \mathbb{E}\left(\mathbf{x}^\top \mathbf{b} - \mathbf{y}^*(\mathbf{x})\right)^2>$ (expectation with respect to $\mathbf{x}$ and $\xi$), where $\mathbf{y}^*(\mathbf{x}) = \mathbf{x}^\top \mathbf{b}^* + \xi$. Assuming $\mathbb{E}\xi = 0$, and using $\Sigma = \mathbb{E}[\mathbf{x} \mathbf{x}^\top]$, we get $\mathcal{E}(\mathbf{b}) = \left(\mathbf{b} - \mathbf{b}^* \right)^\top \Sigma \left(\mathbf{b} - \mathbf{b}^* \right) + \mathbb{E}\xi^2$.
>
> As we illustrate in the appendix, random matrices, notably Gaussian and Bernoulli with independent entries, allow for the lowest restricted isometry constant and, thus, better recovery of sparse vectors. Under this iid assumption, we have $\Sigma = \sigma^2\mathbf{I}_n$ for a certain $\sigma > 0$, which implies $\mathcal{E}(\mathbf{b}) = \sigma^2 \|\mathbf{b} - \mathbf{b}^*\|_2^2 + \mathbb{E}\xi^2$. So $\|\mathbf{b} - \mathbf{b}^*\|_2^2$ captures well the notion of test (or validation) error commonly used in the context of grokking, while $\mathbb{E}\xi^2$ capture the irreducible part of that error (we exclude it in the figure for simplicity).
>
> **Could the authors simplify Figure 1 and perhaps Figure 3 to relate these terms to the terms plotted in the figure?**
>
> Thanks for the suggestion, we will simplify the figures as best we can.

---

> > ### Comment · Reviewer_zx76 · 2025-04-01
> >
> > Thank you. If I understand your rebuttal,
> >
> > (1) the proof-sketch shows the two phase nature of error reduction for sparse recovery.
> >
> > (2) Validation plot -> shows the terms for train and test error. I agree with these definitions.
> >
> > My last point was to gently push the authors to make the paper accessible to more folks but my inputs are optional. In general, I'd like the authors to do what they believe is correct and treat my suggestions as being offered in the spirit of being helpful.

---

### Official Review · Reviewer_Cifa · 2025-03-11

**Overall Recommendation:** 3

**Summary:**

This paper studies grokking phenomenons in the setting where the model has a certain special property $P$, and reveals that by using GD with a small but non-zero regularization of $P$ it is possible to observe grokking. In addition, it shows that modifying model depth or performing data selection can also amplify grokking.

### Update after rebuttal
My original concern still remains while the authors did not provide further reply. Hence I keep my score unchanged.

**Claims And Evidence:**

Claims are supported by both theoretical and empirical evidence.

**Essential References Not Discussed:**

The discussion of the essential references are sufficient.

**Experimental Designs Or Analyses:**

The experimental designs are sound.

**Methods And Evaluation Criteria:**

Not applicable, as this paper does not propose new methods.

**Other Comments Or Suggestions:**

Please see Weaknesses.

**Other Strengths And Weaknesses:**

### Strengths

1. The notations and introduction to the related work (e.g., those about the backgrounds of the problem setting) are satisfying.
2. Overall the writing is clear. And the proofs are clear and complete.
3. The findings for the grokking phenomenons in these new settings are also interesting.


### Weaknesses

1. This paper gives me a very mixed feeling: I can see that the authors have made significant efforts to try to derive a general result to understand the grokking phenomenon for a certain kind of problem (GD with small regularization), while the authors decided to have two separate but very repetitive sections: the results for Theorem 2.1 and Theorem 3.1 are in fact very similar even though they are studying two different problems.

    This strongly suggests that an underlying mechanism is behind somewhere: it seems that the grokking phenomenons in these two settings are both caused by the different generalization properties (implicit bias) of the early solution and that of the late solution. This is basically the idea proposed by Lyu et al., 2023, although Lyu et al., 2023 focused on classification problem.

2. In my view, the formulations of the regularization are nothing but the types of tendency towards specific late solution, especially when you are using small regularization coefficient, thus can be covered by the core idea of Lyu et al., 2023. In this way, I think the authors should sharpen their message from a higher point of view, clearly summarize their insights compared to those in Lyu et al., 2023, discuss their additional insights at the start of the main result, and reduce the repetitive discussion for the sparse recovery problem and matrix factorization problem.

Reference

Lyu et al. Dichotomy of early and late phase implicit biases can provably induce grokking.

**Questions For Authors:**

What are the difference between the core idea in this paper and that of Lyu et al., 2023? By saying core idea, I mean the grokking phenomenon is formed by the transition from the early solution with poor generalization to the late solution solution that generalizes better.

**Relation To Broader Scientific Literature:**

The contributions are an addition to the theoretical understanding of the grokking phenomenon.

**Theoretical Claims:**

I checked the proofs for Section 2 roughly, which appear to be correct.

---

> ### Author Rebuttal · Authors · 2025-04-01
>
> We thank the reviewer for their thoughtful feedback and the time spent reviewing our work. Our responses to the comments are provided below.
>
> **Weaknesses**
> 1) We have rewritten both sections to emphasize the fundamental difference between the two main theorems (2.1 about sparse recovery and 3.1 about matrix factorization) with proof sketches. We first show (using standard results from compressed sensing literature) that, given a matrix $\tilde{\mathbf{X}} \in \mathbb{R}^{N \times n}$ and a vector $\mathbf{b}^* \in \mathbb{R}^{n}$ of sparsity $s$ ($\|\mathbf{b}^*\|_0 \le s$), we can write, for any vector $\mathbf{b} \in \mathbb{R}^{n}$, $\|\mathbf{b} - \mathbf{b}^*\|_2 \le C_1 \| \tilde{\mathbf{X}} \left(\mathbf{b} - \mathbf{b}^*\right)\|_2 + C_2 | \|\mathbf{b}\|_1 - \|\mathbf{b}^*\|_1|$ where $C_1$ and $C_2$ are constants that depend only on $s$ and $\tilde{\mathbf{X}}$. After a short amortization phase,  the term $\| \tilde{\mathbf{X}} \left(\mathbf{b}^{(t)} - \mathbf{b}^*\right)\|_2$ vanish (or becomes proportional to the noise if there is any), since we have $ \tilde{\mathbf{X}} \mathbf{b}^{(t)} \approx \mathbf{y}^* = \tilde{\mathbf{X}} \mathbf{b}^* + \xi$. Then comes a second phase, during which $\|\mathbf{b}^{(t)}\|_1$ converges to $\|\mathbf{b}^*\|_1$: our contribution is to show that this phase takes a time inversely proportional to $\alpha$ (the learning step) and $\beta_1$ (the regularization strength).
>
>   The same type of decomposition applies to matrix factorization, with a slight difficulty because not only singular values but also singular vectors are taken into account.  In fact, although the two results (sparse recovery and matrix factorization) are similar, the distinction lies at several levels:
>  * in sparse recovery problems, we take into account only the iterate $\mathbf{b}^{(t)}$ (especially its $\ell_1$ norm), whereas, in matrix factorization, we take into account not only the singular value matrix $\Sigma^{(t)}$ (especially its $\ell_1$ norm) of the iterate $\mathbf{A}^{(t)} =  \mathbf{U}^{(t)} \Sigma^{(t)} \mathbf{V}^{(t)\top}$, but also its singular vectors $\mathbf{U}^{(t)}$ (left) and $\mathbf{V}^{(t)}$ (right), which makes proofs in the second case slightly more difficult.
>   * if we take a matrix factorization problem and just optimize it with $\ell_1$, there's no grokking unless the matrix is extremely sparse so that the notion of sparsity prevails over the notion of rank, which shows the point of studying $\ell_*$ (nuclear norm regularization) separately.
>    * models (linear or not) trained with $\ell_2$, $\ell_1$ and $\ell_*$ have very different properties: the aim of our work is to show that in all these cases, if we want to have a model that generalizes, and which has the desired property (weights of low $\ell_2$ norm, sparse, or of low rank), then it's better to use a property-specific, weak regularization, and to train the model long enough.
>
> This extends Luy et al.'s work, which focuses solely on $\ell_2$. Indeed, there will be no grokking by using only $\ell_2$ in the problems we study theoretically in the paper. This also shows that the mechanism behind grokking goes beyond the simple $\ell_2$ norm (we show that with other regularization, we can change the delay between memorization and generalization).
>
> 2) Our results cannot be obtained directly from Luy et al.'s work. They show that large-scale initialization and non-zero weight decay lead to grokking provided the model is homogenous. However, as we show, using only $\ell_2$ (with large-scale initialization or not) in the problems we study theoretically in the paper, there will be no grokking. With large-scale initialization and $\ell_2$ only, there is an abrupt transition in the generalization error during training, driven by changes in the $\ell_2$-norm of the model parameters. This transition, however, does not result in convergence to an optimal solution. We call this phenomenon "grokking without understanding," and we attribute it to the fact that the assumptions underlying the theoretical predictions of Luy et al. are violated in our setting.
>
> **Questions: Difference between the core idea of the paper and that of Lyu et al. (2023)**
>
> Luy et al.'s focus on $\ell_2$ and large initialization. But in our case, there will be no grokking by using only $\ell_2$ in the problems we study theoretically. So, we extend their work to other regularizers, and there is no need for large-scale initialization in our setting. In addition, we show how grokking can be necessary in specific contexts. Suppose we want a model with low-rank weights and opt to achieve that through $\ell_*$ regularization. In that case, our work suggests using the lowest possible regularization coefficient (the one the computational resources allow) and training the model as long as possible, far beyond the overfitting point (grokking). We also show that grokking can be extremely amplified/reduced by selecting the data appropriately.

---

> > ### Comment · Reviewer_Cifa · 2025-04-04
> >
> > I thank the authors for the response.
> >
> > First of all, I would like to point out that the cited work should be "Lyu et al.", rather than "Luy et al.".
> >
> > Based on the rebuttal, I think that the authors did not fully understand my main question, and I must make it clear that I did not claim that your results can be obtained directly from Lyu et al, 2022. My point lies in that whether the grokking phenomenon manifests as a dichotomy of early implicit bias and later implicit bias of the learning dynamics, as shown by Lyu et al., no matter whether the type of the implicit bias is $\ell_2$ or not. If this is the case, then the authors only need to first present this general result and then discuss the application of this general for sparse recovery and matrix factorization by revealing their early and implicit bias, rather than presenting two separate sections as if there is no inherent connection between these two settings.

---

### Official Review · Reviewer_8Ky8 · 2025-03-13

**Overall Recommendation:** 2

**Summary:**

The paper mainly focuses on the theoretical approach to debunk the necessity of L2 norm for exhibiting grokking phenomenon. As an alternative, the authors suggest that sparsity of the solution space can be alternative condition to have grokking phenomenon in deeper layer (proven practically). The paper includes rich theoretical justification on a linear matrix fitting problem and demonstrates with 2-layer to 3-layer MLP with ReLU activation, PINN, and LSTM to justify their claim.

**Claims And Evidence:**

The paper has laid some theoretical works to disentangle the necessity of L2-type regularization and the phenomenon of grokking. Moreover, this leads to that L2 weight norm may not be a good indicator for grokking progression measure. I find potential significant issues in the manuscript:

- Although suggesting that the “solution sparsity” behaves as alternative potential source of grokking is interesting and meaningful, there should be more detailed definition with rich examples what solution sparsity is. For example, is the modular arithmetic problem to recover the removed half of the binary relationships “sparse” (this is the first problem that grokking is reported on)?
- There is a clear logic gap between the theoretical justification and the actual claim. I agree that the amount of theoretical works of the authors are respectable; the theory seems to be rich enough, but for its own. The theory deals with the problem with deep linear layers which is not commonly used in practice nor it is dealt in the typical literature on grokking. The way the claim that made on the linear layer connects to nonlinear experiment seems a large logical gap to me. And to address this, I would like to suggest a theoretical work on nonlinear layers.
- It is already known that other factors than L2 norm can reduce grokking. For example, papers like Grokfast (arXiv 2024) have shown that enhancing a momentum-like term of the gradient descent orthogonally reduce the grokking delay. Therefore, the “connection between L2 norm and grokking” may not be as significant as the authors has initially claimed.

Unless these issues are addressed in the rebuttal, I am afraid I cannot agree with the significance of the author’s claim.

**Essential References Not Discussed:**

- The discussion between grokking and double descent has been quite significant, and the authors might get better insight in the joint works, e.g., "Unifying Grokking and Double Descent"
- Groktransfer (ICLR 2025) and Grokfast (arXiv) both addresses different acceleration strategies other than exploiting sparsity and l2 norm.
- Papers like “Grokked Transformers are Implicit Reasoners” discusses large scale experiments, too. I believe that grokking effect can have better meanings in this large scale, highly nonlinear experiments. And the concept of solution sparsity should be discussed in this large domains, on how it should be “extended”.

**Experimental Designs Or Analyses:**

If the Transformer-based experiments (the one like in the first report on Grokking) are presented along with the other experiments, it would be more nicely looking. This is not a cumbersome experiment.

**Methods And Evaluation Criteria:**

The paper tries to justify its claim theoretically especially on matrix factorization and least squares problem, and then tries to relate its conclusions on the deeper layer neural networks. Although the grokking analysis on “deep linear problems” is interesting enough, I do not agree that the conclusion for this problem trivially relates to the nonlinear network case. I have checked multiple times throughout the theoretical section, but I have not found a theory that focuses on at least two-layer nonlinear networks. There is a missing gap between the theory and the evaluation and I believe that these two phenomenon can still be regarded as separate phenomena unless a proper discussion encloses the gap.

**Other Comments Or Suggestions:**

- In line 432, LSMT→LSTM.

**Other Strengths And Weaknesses:**

Please refer to other sections of this review. I respect the author’s effort and work for providing theoretical justification in this work. However, I am not still convinced on the logical flow of this manuscript.

**Questions For Authors:**

- Since grokking phenomenon is firstly observed in Transformer network and imposes significance the most in these types of network, why the authors only have chosen to demonstrate for MLP and LSTM networks?

**Relation To Broader Scientific Literature:**

Deep linear layers are in fact has a domain specific usage, e.g., in kernel estimation problem (e.g., in KernelGAN) and other problems that require redundancy in parameter space for smoother training. However, it is not well known that grokking is significant in these model domains. There this paper might get its significance.

**Theoretical Claims:**

I have tried my best to check the correctness of the proof, but I must say I have missed many theorems especially those appear in the supplementary material. Although I agree that the presented mathematical justification indeed supports the author’s claim that Euclidean norm is not necessary for grokking, I do not believe this can be trivially extended to more complicated architecture, e.g., Transformers, where the grokking phenomenon is first observed. Moreover, I am still not convinced that the theory on linear problem is extendable to nonlinear experiments.

---

> ### Author Rebuttal · Authors · 2025-04-01
>
> We appreciate the reviewer’s time and insightful comments. Please find our responses to the points below.
>
> **Claims And Evidence**
>
> * We worked in a setting where the notion of sparsity (resp. low rank) is well defined, $\mathbb{R}^n$ (resp. $\mathbb{R}^{n_1 \times n_2}$). This is the number of non-zero elements of a vector (resp. the number of non-zero singular values of a matrix). If we decide to work, say, in $\mathbb{Z}/p\mathbb{Z}$ for some prime number $p$ (like in the first problem grokking was first reported on, modular arithmetic), then these definitions are not trivially valid, and the theory we developed no longer applies (indeed, the theory of sparse recovery and matrix factorization over finite fields is very complex and uses completely different tools from those used in $\mathbb{C}$). This is why we can not answer the question, “Is the modular arithmetic problem to recover the removed half of the binary relationships sparse?” using our current theory and leave it as a future direction (see section F of the Appendix). Although for tasks such as the sparse parity task (on which grokking is observed [1, 2]) one can construct sparse MLP that fits the data [1], we are not aware of any work that has done this for modular arithmetic. [3] constructs a two-layer MLP with square activations that does modular addition, but the model weights constructed are dense.
>
> * The most crucial point of our work is to show that not only does the grokking mechanism go beyond the $\ell_2$ norm, but also why it is necessary to do so. Indeed, we show that the time to generalize is inversely proportional to the regularization strength used and that the generalization error after convergence decreases with it. Moreover, we show that grokking can also be highly accelerated by simply choosing appropriate data samples. To the best of our knowledge, no previous work on grokking has done all these.
>
>     Although our theory relies on simple frameworks that allow us to extract the laws justifying our results, we also show this empirically for different types of models and datasets, including the basic setup on which grokking was first observed, the algorithmic dataset. We acknowledge that developing a theory in a simplified framework and then testing it experimentally in a more complicated setup has its flaws. The point of our paper is not to say that the mechanisms behind linear models extend to non-linear models but to illustrate that grokking is nevertheless observed if we use the insights obtained theoretically on linear models. We'll try to make this point clearer in the final version of the paper.
>
> * We acknowledge that grokking can be caused by a number of factors. This is the point of our work, even if we focus on regularization. Other work (Grokfast, etc) focuses on other factors to amplify/reduce the Grokking delay, and we will discuss this in detail in the related works section.
>
> **Methods And Evaluation Criteria, Theoretical Claims**
>
> The point of our work is not that our theoretical results on linear models **trivially** extend to non-linear models. We simply show that the mechanism behind grokking (and regularization) goes beyond the $\ell_2$ norm: we show it theoretically and empirically in specific settings and just empirically in others (and leave the theory in such settings as future work).
>
> **Essential References Not Discussed**
>
> Thanks for the references, we will discuss this in the related work section. These works differ from ours in that the first focuses on double-descent (which is not the point of our work) and the other on other strategies to accelerate grokking (we focus on different types of regularization and data selection).
>
> **Questions For Authors**
>
> We excluded Transformer results on modular arithmetic as the paper was already too long. On Transformer, $\ell_1$ and $\ell_*$ still affect the delay between memorization and generalization.
>
> **References**
>
> [1] Hidden Progress in Deep Learning: SGD Learns Parities Near the Computational Limit (https://arxiv.org/abs/2207.08799)
>
> [2] A Tale of Two Circuits: Grokking as Competition of Sparse and Dense Subnetworks (https://arxiv.org/abs/2303.11873)
>
> [3] Grokking modular arithmetic (https://arxiv.org/abs/2301.02679)

---

> > ### Comment · Reviewer_8Ky8 · 2025-04-08
> >
> > I appreciate the responses.
> >
> > My original concern was on the missing gap between the author's claim on grokking in a general sense and the theoretical justification done for linear layers. To my understanding from this rebuttal, the theoretical part and the empirical part of this work still has unresolved logical gap, and one conclusion does not necessarily connects to the other conclusion, as the authors have acknowledged as:
> >
> > >This is why we can not answer the question, “Is the modular arithmetic problem to recover the removed half of the binary relationships sparse?” using our current theory and leave it as a future direction.
> >
> > Even if we disregard this gap, the theory established on linear models carry little significance in general applications of grokking phenomenon on deeper, more complex models. Therefore, I am sorry I cannot give higher score based on this rebuttal. However, I believe the author's claim of disentangling the necessity of $\ell_2$ norm and the grokking phenomenon is indeed interesting, and I would like to recommend the authors to further develop on this work.

---

### Official Review · Reviewer_WymT · 2025-03-14

**Overall Recommendation:** 2

**Summary:**

The authors study delayed generalization (grokking) in sparse/low rank recovery tasks. The authors focus on the transition from an overfitting solution to a generalizing solution, both in linear sparse recovery and in low rank matrix factorization. For the linear cases, the authors derive the scaling of the grokking time difference. They show that for sparse tasks, grokking can be tracked and controlled by the $L_1$ or the trace norm of the weights or dictionary vectors.

**Claims And Evidence:**

The claims are both proven and supported by empirical evidence.

**Essential References Not Discussed:**

No essential references clearly come to mind for the specific context of this paper, certainly not all grokking papers are cited but since the focus is not on feature learning but just on the transition between possible solutions. Perhaps the authors could contrast their work more with the feature learning works such as [1,2].


Refs:

[1] - GROKKING AS A FIRST ORDER PHASE TRANSITION IN
TWO LAYER NETWORKS, Rubin et al., ICLR 2024

[2] - GROKKING AS THE TRANSITION FROM LAZY TO RICH
TRAINING DYNAMICS, Kumar et al., ICLR 2024

**Experimental Designs Or Analyses:**

The main experiments are simple and done on convex problems, and are therefore reproducible with the parameters given in the paper.

**Methods And Evaluation Criteria:**

The problem is well set up and the evaluation criteria make sense fo this problem.

**Other Comments Or Suggestions:**

**Partial list of typos and grammatical errors:**

- The paper has no page numbers at the bottom, not sure how this happened.
- L20 (right) - "we always need an ℓ2 regularization term."
- L23 (right) - "the dynamics of"
- L37 (right) - "previous art"
- L62 - “with an mlp”
- L125 - "number of samples needs" should be "needed"
- L163 - "since for problem of interest", should be "for the..."
- L171 - "If β2 is choose such.."
- L216 - "number of measures N...", should be "samples/measurements"

- $\tau$ is not defined before it is used in the text, and it is unclear if the error term $\xi$ is always meant to be a constant vector or a random vector which is different per sample, etc.

There are more which I have not included here.

**Suggestions:**

My main suggestion given the weaknesses and strengths of the paper is simple: divide this paper into at least 2 different works, one which simply focuses on sparse recovery/deep sparse recovery, and explains the related results in depth in the main text, including sketches of proofs, intuition, and dependence on the relevant parameters. Then a separate work that focuses on low rank estimation, or putting the low rank estimation in the appendix of the first work, since it doesn't seem to provide a much deeper insight except for replacement of the lasso norm with the nuclear norm. Then I would suggest a concrete rewriting of the paper to ensure that the grammatical mistakes do not repeat and improving the structure. I would also suggest that the authors consider describing the problem in terms of its invariant quantities that actually dictate the scaling, for instance $\alpha \beta_1$ is one of the main scaling quantities, not $\alpha,\beta_1$ separately. It would be smart to simply define products and ratios of the original parameters and study the problem in terms of these quantities, to avoid cumbersome notations and clearer takeaway messages.

**Other Strengths And Weaknesses:**

**Strengths:**

1) The problem setup and topic are very interesting in my opinion, and are of interest to the DL community.  I have not seen works that focus on the relation between grokking and sparsity in this way, and I believe it can be a key contributor to this phenomenon in many real world settings.

2) The results seem robust and correct, and provide a new way to look at progress measures of grokking, possibly in problems where the problem is not tractable analytically.

**Weaknesses:**

The main and overwhelming weakness of this paper is its extreme breadth and length. The authors attempt to cover too many setups and end up compromising quality for quantity. By attempting to tackle sparse recovery, deep sparse recovery, data selection, low rank recovery, matrix completion, and extending to nonlinear cases, the main text cannot provide more than a shallow discussion for many of these cases, which leads to the following problems:

- The paper is poorly written, both in terms of the number of typos/grammatically incorrect sentences.
- When theorems are given, there are no sketches of the proofs, which should provide the intuition and deep insight as to why these results work and make sense, but instead just given in terms of full proofs in the appendices, which are $O(70)$ pages. This is not the correct format for a conference paper.
- Many results and figures do not appear in the main text, but are referred to the appendices. This would be fine if not for comments such as *"While grokking has been studied extensively, the impact of data selection on grokking remains largely unexplored, making this one of the first works to address this critical aspect."*  Either a topic is important enough to appear fully in the main text and discussed seriously, or it is not truly critical, it cannot be both. Results in appendices should be either trivial extensions that do not add any deep insights, or indeed proofs, when the sketch is given in the main text. Another example is the Deep Sparse Recover paragraph, which directly points to the appendix, as well as the Realistic Signals paragraph. All of these should either be removed from this work or discussed seriously in the main text.
 - There is no mention of the second effect that typically arises in grokking which is the non-monotonicity of the test loss while the training loss monotonically decreases. This is a minor point but is never discussed.


This weakness is the reason for my inclination to reject this paper at the current version.

**Questions For Authors:**

**Questions:**

1) What is the overlap between $b(t_1)$ and $b(t_2)$? What changes more, the norm or the direction?
2) Can you comment on the fact that the test loss is always monotonic in your settings? in general it is possible to have a non monotonic test loss even for a convex problem if the transition from overfitting to generalizing solution involves a big change in the parameters (either alignment or norm), so I'm curious if you can explain why it doesn't happen in your case. In some figures (fig 5 for example) you do have it, but only when it is accompanied by a previous non-monotonic effect in the training loss, which is non generic.

**Relation To Broader Scientific Literature:**

The contributions presented in this work are related to the broader scientific literature, in particular they relate to grokking in linear estimators, where delayed generalization can occur even in convex problems depending on certain parameters of the setup. They extend these ideas to more complicated settings where a clearly defined overfitting and generalizing solutions exist.

**Theoretical Claims:**

I only checked the main theorem 2.1, while the rest are somewhat demonstrated directly by the experiments.

I would state that the paper contains 81 pages, and it would be unreasonable to check in its entirety in the allotted time for review.

---

> ### Author Rebuttal · Authors · 2025-04-01
>
> We thank the reviewer for their time and insightful comments. We appreciate the effort taken to evaluate our work and provide constructive feedback. Below, we address each of the points raised.
>
> **Weaknesses**
>
> We understand that the paper is poorly organized. That is why, after submission, we completely rewrote it as follows:
> * All the proofs have been simplified, and we've included the proof sketch for each main theorem. For example, here's the main idea behind Theorem 2.1. We first show (using standard results from compressed sensing literature) that, given a matrix $\tilde{\mathbf{X}} \in \mathbb{R}^{N \times n}$ and a vector $\mathbf{b}^* \in \mathbb{R}^{n}$ of sparsity $s$ ($\|\mathbf{b}^*\|_0 \le s$), we can write, for any vector $\mathbf{b} \in \mathbb{R}^{n}$, $\|\mathbf{b} - \mathbf{b}^*\|_2 \le C_1 \| \tilde{\mathbf{X}} \left(\mathbf{b} - \mathbf{b}^*\right)\|_2 + C_2 | \|\mathbf{b}\|_1 - \|\mathbf{b}^*\|_1|$ where $C_1$ and $C_2$ are constants that depend only on $s$ and $\tilde{\mathbf{X}}$. After a short amortization phase ($t \le t_1$), the term $\| \tilde{\mathbf{X}} \left(\mathbf{b}^{(t)} - \mathbf{b}^*\right)\|_2$ vanish (or becomes proportional to the noise if there is any), since we have $ \tilde{\mathbf{X}} \mathbf{b}^{(t)} \approx \mathbf{y}^* = \tilde{\mathbf{X}} \mathbf{b}^* + \xi$. Then comes a second phase, during which $\|\mathbf{b}^{(t)}\|_1$ converges to $\|\mathbf{b}^*\|_1$ at $t_2$: our contribution is to show that $t_2 - t_1$ is inversely proportional to $\alpha$ (the learning step) and $\beta_1$ (the regularization strength).
> * We made the body of the paper self-contained insofar as any contribution mentioned in the abstract and introduction is proven in the body of the paper theoretically and/or empirically (grokking time, effect of data selection on grokking, limits of $\ell_2$, and grokking without understanding, ...). So, the results in the appendix this time are just extensions of those in the main text.
> * With the rewrite, the appendix has also been reduced from $\mathcal{O}(70)$ to less than $50$ pages.
> * We have left the part on matrix factorization in the main text since the mechanics behind the generalization phase, in this case, differs from sparse recovery. In sparse recovery problems, we take into account only the iterate $\mathbf{b}^{(t)}$ (especially its $\ell_1$ norm), whereas, in matrix factorization, we take into account not only the singular value matrix $\Sigma^{(t)}$ (especially its $\ell_1$ norm) of the iterate $\mathbf{A}^{(t)}$, but also its singular vectors $\mathbf{U}^{(t)}$ (left) and $\mathbf{V}^{(t)}$ (right), which makes the dynamic in the second case completely different from the first case. Also, in a matrix factorization problem, replacing $\ell_*$ (the nuclear norm regularization) by $\ell_{*/2}$ does not induce grokking (at least in the shallow case used in the theory).
>
> Thanks for the suggestion about figures and parameters of interest. We will define a parameter $\tilde{\alpha} = \alpha \beta$, which is the main parameter in the theory ($\alpha$ is the learning rate and $\beta$ is the regularization).
>
> **What is the overlap between $\mathbf{b}^{(t_1)}$ and $\mathbf{b}^{(t_2)}$? What changes more, the norm or the direction?** Once memorization occurs ($\mathbf{X}\mathbf{b}^{(t_1)}=\mathbf{X}\mathbf{b}^*+ \xi$), the direction of $\mathbf{b}^{(t)}$ that explains the data does not need to change drastically.  The main change from $t_1$ to $t_2$ is the magnitude of $\mathbf{b}^{(t)}$ in $\ell_1$—i.e., it transitions from something closer to the least‐square solution’s norm $\|\hat{\mathbf{b}}\|_1$ down to the teacher’s norm $\|\mathbf{b}^*\|_1$ (while still maintaining some form of memorization). So, $\mathbf{b}^{(t_1)}$ and $\mathbf{b}^{(t_2)}$ are closely aligned in direction; the main difference is that the $\ell_1$ norm shrinks from $\|\hat{\mathbf{b}}\|_1$ to $\|\mathbf{b}^*\|_1$. Hence, the norm changes more than the direction.
>
> **Can you comment on the fact that the test loss is always monotonic in your settings?**
>
> The monotonic decrease in test loss is due to how we choose the hyperparameters, mainly the learning rate and the regularization strengths. In the case of sparse recovery, for example, the test loss is controlled mainly by $\|\textbf{b}^{(t)}\|_1$, even more so after memorization, which takes only a few steps. After this memorization, $\|\textbf{b}^{(t)}\|_1$ converge very slowly towards $\|\textbf{b}^*\|_1$. When $\alpha$ and/or $\beta_1$ are larger than the values used in the paper, there are indeed a lot of oscillations in test loss. The same reasoning applies to experiments outside the linear scope (e.g., figure 5). We chose the learning rate and regularization strength to illustrate grokking. Larger values show oscillations (e.g., the Slingshot Mechanism [1]).
>
> [1] The Slingshot Mechanism: An Empirical Study of Adaptive Optimizers and the Grokking Phenomenon (https://arxiv.org/abs/2206.04817)

---

> > ### Comment · Reviewer_WymT · 2025-04-06
> >
> > I thank the authors, and appreciate their willingness to accept the criticism. I believe that the paper definitely has merit and the results are sound. However, due to the format of ICML, it is impossible for me to review a revised version, and since the revised version includes a full rewriting, I cannot raise my score in good faith. I hope the authors understand, and should the paper not be accepted in this iteration I highly recommend they resubmit the fully revised version to another venue.

---

### Decision · Program_Chairs · 2025-05-01

**Decision:**

Accept (poster)

**Comment:**

The authors convincingly demonstrate—both analytically in sparse recovery / matrix‑factorization settings and empirically in deeper non‑linear networks—that delayed generalization (grokking) is not tied to the Euclidean (L2) norm but rather to whichever structural property R is truly capacity‑controlling for the task (sparsity, low rank, etc.).  Reviewers agreed the theoretical results are sound, the revised draft now contains clear proof sketches and cleaner figures, and extensive synthetic‐and‑real experiments—including teacher–student MLPs, PINNs, and LSTMs—validate the predicted scaling and show that depth‑induced implicit bias or targeted data selection can amplify or suppress the effect.

Although the paper remains dense, the authors’ rewrite addresses presentation concerns: each main theorem is now better motivated in the body, repetitive material has been pruned, and the connection between the linear theory and non‑linear evidence is explicitly framed as an empirical test of the theory’s insight rather than a claimed formal extension.

In addition, even though the reviewers were concerned that the results are limited to linear models, I believe this is acceptable for two reasons. The goal of the paper is not to make claims about grokking in all model families, but rather to isolate and analyze the underlying mechanism in a setting that is amenable to precise analysis. Second, the empirical evidence presented across a range of non-linear architectures (MLPs, PINNs, LSTMs) suggests that the insights obtained in the linear case continue to hold qualitatively.